

# Topological characterization of Lieb-Schultz-Mattis constraints and applications to symmetry-enriched quantum criticality

Weicheng Ye[1,2], Meng Guo[1,3], Yin-Chen He[1], Chong Wang[1] and Liujun Zou[1]

**1** Perimeter Institute for Theoretical Physics, Waterloo, Ontario, Canada N2L 2Y5
**2** Department of Physics and Astronomy, University of Waterloo,
Waterloo, Ontario, Canada N2L 3G1
**3** Department of Mathematics, University of Toronto, Toronto, Ontario, Canada M5S 2E4

## Abstract

Lieb-Schultz-Mattis (LSM) theorems provide powerful constraints on the *emergibility* problem, i.e. whether a quantum phase or phase transition can emerge in a many-body system. We derive the topological partition functions that characterize the LSM constraints in spin systems with $G_s \times G_{int}$ symmetry, where $G_s$ is an arbitrary space group in one or two spatial dimensions, and $G_{int}$ is any internal symmetry whose projective representations are classified by $\mathbb{Z}_2^k$ with $k$ an integer. We then apply these results to study the emergibility of a class of exotic quantum critical states, including the well-known deconfined quantum critical point (DQCP), $U(1)$ Dirac spin liquid (DSL), and the recently proposed non-Lagrangian Stiefel liquid. These states can emerge as a consequence of the competition between a magnetic state and a non-magnetic state. We identify all possible realizations of these states on systems with $SO(3) \times \mathbb{Z}_2^T$ internal symmetry and either $p6m$ or $p4m$ lattice symmetry. Many interesting examples are discovered, including a DQCP adjacent to a ferromagnet, stable DSLs on square and honeycomb lattices, and a class of quantum critical spin-quadrupolar liquids of which the most relevant spinful fluctuations carry spin-2. In particular, there is a realization of spin-quadrupolar DSL that is beyond the usual parton construction. We further use our formalism to analyze the stability of these states under symmetry-breaking perturbations, such as spin-orbit coupling. As a concrete example, we find that a DSL can be stable in a recently proposed candidate material, NaYbO$_2$.

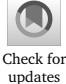

# 1  Introduction

An important task of condensed matter physics is to understand the quantum phase or phase transition that emerges from a many-body system. However, this is often challenging in strongly correlated systems, both theoretically and experimentally, due to the lack of i) theoretical tools to exactly solve the many-body ground state in the generic setting, and ii) experimentally accessible signatures that can unambiguously diagnose the nature of the phase or phase transition.

In light of this, Lieb-Shultz-Mattis (LSM) type constraints are especially valuable [1–3]. Given some general symmetry-related properties of a system, which are often relatively easy to determine, the LSM constraints constrain the *emergibility* of a phase or phase transition, i.e., whether this phase or phase transition can possibly emerge from this system. Such constraints have been widely applied to the search for exotic states beyond the symmetry-breaking paradigm, e.g., quantum spin liquid phases and exotic phase transitions. For instance, a simple example of LSM constraints states that in a $(d + 1)$-d lattice spin system with $SO(3)$ spin rotation and lattice translation symmetries that are not explicitly or spontaneously broken, if each unit cell hosts an odd number of spin-1/2 moments, then the ground state must be exotic (i.e., topologically ordered or gapless). Since symmetry breaking is often relatively easy to detect experimentally and numerically, its absence is often taken as the first evidence of an exotic state in such systems.

There has been great progress in understanding LSM constraints in recent years [4–10]. In particular, it was realized that LSM constraints can be captured by *LSM anomalies*, the quantum anomalies carried by the boundaries of some higher-dimensional topological crystalline phases. Such relations between LSM constraints and anomalies can be very powerful in constraining the emergibility of a phase or phase transition, because the quantum anomaly of this phase or phase transition, which we refer to as its *IR anomaly*, must match with the LSM anomaly (in a sense to be sharpened later).

In order to utilize these constraints, we need to know how to compare an LSM anomaly and an IR anomaly. The latter can be derived from the effective field theory of the corresponding phase or phase transition, and it is often characterized by a topological partition function (TPF). However, to date the TPFs corresponding to the LSM anomalies are unknown in the general setting, so the full power of the LSM constraints has not been uncovered; although these constraints have been applied to various systems and shed important insights in the emergibility of some states [11–14], most previous analyses were performed in a case-by-case manner and/or did not take the full symmetry constraint into account, and a systematic framework is lacking.

The first major goal of this paper is to fill this gap. Motivated by the studies of quantum magnetism, we consider $(2 + 1)$-d spin systems with $G_s \times G_{int}$ symmetry, where the lattice symmetry $G_s$ is any of the 17 wallpaper groups, and $G_{int}$ is any internal symmetry whose projective representations are classified by $\mathbb{Z}_2^k$ with $k$ some integer, e.g., $G_{int} = SO(3) \times \mathbb{Z}_2^T$, the combination of $SO(3)$ spin rotational symmetry and time reversal symmetry. Given $G_s \times G_{int}$, there are still topologically distinct LSM constraints, specified by the projective representation

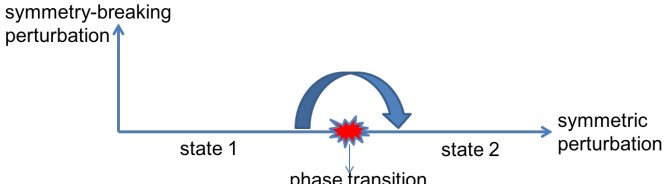

Figure 1: If two states have the same emergent order and exact microscopic symmetry, and if they can be smoothly connected when symmetry-breaking perturbations are allowed, but are necessarily separated by a phase transition when the relevant symmetries are preserved, then these states are said to have symmetry-protected distinction.

(PR) under $G_{int}$ carried by the degrees of freedom (DOF) of the system, and the spatial distribution of these DOF. For *all* cases, we derive the TPFs of the LSM anomalies. Similar analysis is also performed for $(1+1)$-d lattice spin systems. This topological characterization of the LSM constraints is the basis of a systematic framework that uses the LSM constraints to understand the emergibility of quantum phases and phase transitions in a many-body system.

The second major goal of this paper is to apply the obtained topological characterization of the LSM constraints to study the emergibility of exotic states. Here we focus on exotic quantum criticality, rather than other classes of exotic states, e.g., topological phases, which may be more commonly done in the literature. Our choice is motivated by the following reasons. First, quantum critical states may have many elegant structures that are worth studying, such as emergent conformal invariance at low energies. Second, many quantum critical states can serve as the parent states of other phases (including topological phases), which can emerge through perturbing the quantum critical states. So a thorough understanding of the quantum criticality may provide a unified understanding of not only the critical state itself, but also the nearby phases. However, compared to topological phases, quantum criticality is much less understood, especially in two and three spatial dimensions. So it is interesting and important to further explore them.

A useful notion here is symmetry-enriched quantum criticality. This notion is actually rather familiar, but let us discuss it in a more modern perspective. By now, it is well appreciated that the universal long-distance and low-energy physics of most (if not all) quantum many-body systems are specified by two levels of data. The first level is characterized by what we refer to as the *emergent order*. In the language of renormalization group (RG), the emergent order is described by properties of the RG fixed point corresponding to this system, which are independent of the exact microscopic symmetry. For example, the RG fixed point corresponding to gapped states are described by certain topological quantum field theory (TQFT), or variants of it. Short-range entangled (SRE) states, i.e., states smoothly connected to a product state without quantum entanglement, are related to a trivial TQFT. In contrast, long-range entangled gapped states, which cannot be smoothly connected to product states, correspond to some nontrivial TQFT. On the other hand, gapless states have different emergent orders, and many of their RG fixed points are described by a conformal field theory (CFT). States described by different RG fixed points are distinct at the level of their emergent orders.

Even if two states have the same emergent order (RG fixed point), their exact microscopic symmetries provide a second level of data that may distinguish them. Two states with the same emergent order but different exact microscopic symmetries are considered distinct. If they have the same emergent order and same exact microscopic symmetries, they may still have *symmetry-protected distinction*: they are not smoothly connected if certain symmetries are imposed, while they are if these symmetries are broken (see Fig. 1). Two SRE states

with symmetry-protected distinction are referred to as different SPTs, two topological orders with symmetry-protected distinction are referred to as different symmetry-enriched topological states (SETs), and two quantum critical states with symmetry-protected distinction are referred to as different symmetry-enriched criticality. There has been great progress in understanding SPTs and SETs in the past years, but a systematic understanding of symmetry-enriched criticality is lacking.

In this paper, we focus on the emergibility of a family of quantum critical states dubbed Stiefel liquids (SLs), each of which is labeled by an integer $N \geqslant 5$ and denoted by SL$^{(N)}$ [15]. The well-known deconfined quantum critical point (DQCP) [16–19] and $U(1)$ Dirac spin liquid (DSL) [20–22] are unified as the two simplest SLs, with $N = 5$ and $N = 6$, respectively. SL$^{(N \geqslant 7)}$ are conjectured to be *non-Lagrangian*, i.e., they are so strongly interacting, such that they cannot be described by any weakly-coupled continuum Lagrangian at any energy scale. We would like to understand whether the SLs can emerge in lattice spin systems, and if they can, which different types of symmetry-enriched SLs can emerge.

Here we characterize each realization of SL$^{(N)}$ by its symmetry embedding pattern (SEP), i.e., how the microscopic symmetries act on its *local*, *low-energy* DOF. This characterization has a number of advantages. First and most fundamentally, it captures the symmetry actions in an intrinsic and direct way. This is in contrast to the more common treatment of emergent gauge theories in condensed matter physics (e.g., for DQCP and DSL), where one first considers the symmetry actions on gauge non-invariant operators (such as spinons) and then converts them into actions on local operators, which is indirect and sometimes complicated, especially when there is a $(2 + 1)$-d $U(1)$ gauge field where the quantum numbers of the local monopole operators cannot be identified with those of any gauge-invariant composite of the matter fields, and when some symmetries act as duality between different gauge-theoretic formulations of the same critical state. Second, using this characterization we can easily read off the symmetry-breaking patterns of the ordered phases adjacent to the exotic quantum criticality. This information provides valuable guidance on where to look for these quantum critical states: if the corresponding ordered phases are found in a material or model, then exploring the vicinity of the phase diagram may result in the critical state. Third, using this characterization it is easy to check the stability of the critical state under various perturbations, e.g., spin-orbit couplings (SOC). Recently, $NaYbO_2$ and related materials emerge as candidates for DSL [23–32]. These systems have strong SOC, so it is important to ask if DSL remains stable under SOC. We showcase how to use our approach to argue that the DSL can be stable in $NaYbO_2$.

To check the emergibility of a Stiefel liquid with a given symmetry embedding pattern, we rely on the *hypothesis of emergibility* [15]: a state is emergible if and only if its IR anomaly matches with that of the LSM-like anomaly of the microscopic system. The necessity of this condition has been established, while its sufficiency is hypothetical, but supported by many nontrivial examples. Using the symmetry embedding pattern, we can match the IR anomaly of an SL with the LSM anomaly of a lattice spin system characterized by the TPF we derive[1], and we search for *all* realizations of SL$^{(N=5,6,7)}$ that can emerge due to the competition between a magnetic state and a non-magnetic state on lattice systems with $G_s \times G_{int}$ symmetry, where $G_s$ is either the $p4m$ or $p6m$ wallpaper group, and $G_{int} = SO(3) \times \mathbb{Z}_2^T$. We discover many interesting realizations of these states. For example, we find that the DSL can be realized as a quantum critical spin-quadrupolar liquid, i.e., a critical state whose most relevant spinful excitations have spin-2. So far, the construction of the DSL is often based on a type of parton

---

[1]In Ref. [15], the TPFs for some of the LSM anomalies are listed, and anomaly-matching is performed to check the emergibility of various SLs. However, both those TPFs and the anomaly-matching calculations therein are problematic, and the current paper presents the correct TPFs and anomaly-matching calculations. All specific examples studied in Ref. [15] are treated with care in this paper (see Sec. 3.3 and Appendix I), and it is found that all final physical results regarding the emergibility of these examples are correctly obtained in Ref. [15].

mean field. However, we show that our spin-quadrupolar realization of the DSL is beyond that parton mean field. With all realizations at hand, we will see that, given an $SL^{(N)}$ and its microscopic symmetries, different symmetry embedding patterns typically correspond to different symmetry-enriched SLs (we discuss the subtle cases where this may not be true at the end of Sec. 4).

We highlight that this exhaustive search of realizations of SLs is possible because we have obtained our topological characterization of the LSM contraints, without which we cannot examine the emergibility of states systematically. We also remark that even if the hypothesis of emergibility turns out to be false, i.e., it is just a necessary but insufficient condition for emergibility, the result of our search is still useful, because all SLs that are emergible must belong to the ones we identify.

The organization of the rest of the paper and a brief summary of the main results are as follows.

1. In Sec. 2, we derive the topological partition functions of the LSM constraints of the lattice spin systems of our interest. The structure of these topological partition functions is given in Eq. (1), where $\eta$ is determined by the projective representation carried by the local degrees of freedom under the internal symmetry, and $\lambda$ is determined by the locations of the local degrees of freedom. The characterization of $\lambda$ for different space groups can be found in Secs. 2.2.1, 2.2.2 and 2.2.3, and Appendix F. Some further arguments leading to these topological partition functions are presented in Appendices B, C, D and G.

2. In Sec. 3, we sketch how to use anomaly-matching to understand the emergibility of various Stiefel liquids. Detailed examples of caculations are presented in this section and also in Appendix I.

3. In Secs. 4 and 5, we present some interesting realizations of SLs, while the complete results are summarized in the attached codes, which can be read with the instruction in Appendix J. Table 1 records the total numbers of realizations in different cases, and Table 2 records the numbers of realizations that are adjacent to classical regular magnetic orders. The stability of each realization is also analyzed, which is recorded in the attached codes. In Appendix K, we present all stable realizations on various familiar lattice systems. The highlighted examples in the main text include i) a deconfined quantum critical point between a ferromagnet and a valence bond solid, ii) stable $U(1)$ Dirac spin liquids in spin-1/2 square and honeycomb lattices, iii) various realizations of the non-Lagrangian Stiefel liquid, and iv) realizations of SLs where the most relevant spinful excitations carry spin-2, which, in particular, include a $U(1)$ Dirac spin liquid that cannot be desribed by the usual parton approach.

4. We demonstrate how to use our formalism to study the stability of these states under symmetry-breaking perturbations in Sec. 6, where we argue that the DSL can be stable in $NaYbO_2$. More analysis regarding $NaYbO_2$ and twisted bilayer $WSe_2$ is presented in Appendix M.

5. We conclude in Sec. 7.

6. Various appendices include further details, some of which may be of general interest. For example, Appendix A is a review of the basic mathematical tools we use. Appendix E contains descriptions of all 17 wallpaper groups, as well as information about their $\mathbb{Z}_2$ cohomology, including their $\mathbb{Z}_2$ cohomology rings and all representative cochains at degree 1 and 2. Appendix H contains more details of the Stiefel liquids, including some

that do not appear in Ref. [15]. Appendix L presents the configurations of spins of all classical regular magnetic orders in triangular, honeycomb, kagome and square lattices.

# 2 Topological characterization of LSM constraints

In this section, we develop a topological characterization of the LSM constraints applicable to a $(2+1)$-d lattice spin system, whose Hilbert space is a tensor product of local bosonic Hilbert spaces, and whose Hamiltonian is also local. We assume that the system has a symmetry group $G = G_s \times G_{int}$, where $G_s$ is one of the 17 wallpaper groups and $G_{int}$ is an internal symmetry group. Throughout this paper, we consider $G_{int}$ whose projective representations (PR) are classified by $\mathbb{Z}_2^k$ with some $k \in \mathbb{N}^+$, i.e., $H^2(G_{int}, U(1)_\rho) = \mathbb{Z}_2^k$ [2], with the subscript $\rho$ indicating the complex conjugation action of any spacetime orientation reversal symmetry on the $U(1)$ coefficient. Typical examples of such $G_{int}$ include $SO(3)$, $SO(3) \times \mathbb{Z}_2^T$, $\mathbb{Z}_2^T$, $O(2)$, $\mathbb{Z}_2 \times \mathbb{Z}_2$, etc. These choices of $G$ and $G_{int}$ are motivated by the systems and models relevant to quantum magnetism. We will also perform a similar analysis for $(1+1)$-d lattice spin systems.

Some $G_{int}$ may have multiple types of PR. For example, for $G_{int} = SO(3) \times \mathbb{Z}_2^T$, $H^2(SO(3) \times \mathbb{Z}_2^T, U(1)_\rho) = \mathbb{Z}_2^2$, so there are 3 different types of nontrivial PR, corresponding to spinor under $SO(3)$ while Kramers singlet under $\mathbb{Z}_2^T$, singlet under $SO(3)$ while Kramers doublet under $\mathbb{Z}_2^T$, and spinor under $SO(3)$ while Kramers doublet under $\mathbb{Z}_2^T$. In this paper, we will mainly consider systems with at most one type of nontrivial PR, i.e., there may be some DOF carrying trivial PR under $G_{int}$, but all DOF with nontrivial PR carry the same type of nontrivial PR which we refer to as the *PR type of the system*. If all DOF carry trivial PR, then the PR type of the system is said to be the trivial type. Many of our results can be straightforwardly generalized to the case where the system has DOF with different types of nontrivial PR, on which we sometimes explicitly comment.

## 2.1 Review of lattice homotopy and the connection to SPT

To be self-contained, we begin by reviewing lattice homotopy [5], in a way that will lead to our topological characterization of the LSM constraints most easily.

All LSM constraints should be fully determined by the spatial distribution of the DOF in the system. The key idea of lattice homotopy is that, to characterize the LSM constraints for a given lattice system, one can always first smoothly deform the system so that all DOF are moved to the high-symmetry points of the corresponding wallpaper symmetry group, while preserving the $G = G_s \times G_{int}$ symmetry during the process. These high-symmetry points are called the irreducible Wyckoff positions (IWP); their precise definition can be found in Ref. [5] and they are well documented for each space group in the standard crystallographic literature. All distributions of DOF that can be smoothly deformed into each other are referred to be in the same *lattice homotopy class*. Below we always assume that a smooth deformation has been performed, such that all DOF are located at some IWP. Then to determine the presence or absence of an LSM constraint, one can invoke one or multiple of the following 3 types of basic no-go theorems that preclude symmetric SRE (sym-SRE) ground states in various cases [5,33]:

1. Define a fundamental domain to be a region that tiles the 2D space under the actions of translation and glide symmetries. When the total PR within a fundamental domain is nontrivial, a sym-SRE ground state is forbidden.

---

[2] In this paper, a few different objects have the structure of $\mathbb{Z}_2^k$ with some $k \in \mathbb{N}$. These $k$'s are independent unless explicitly claimed, and we will abuse the notation to use the same $k$ when we make a statement about this $\mathbb{Z}_2^k$ structure.

2. When there is a translation symmetry along a mirror axis, and the total PR within a translation unit along this mirror axis is nontrivial, a sym-SRE ground state is forbidden.

3. In our case, the PR of $G_{int}$ are classified by $\mathbb{Z}_2^k$. Then if the total PR at a $C_2$ rotation center is nontrivial, a sym-SRE ground state is forbidden. However, PR at a $C_n$ rotation center for odd $n$ does not forbid a sym-SRE ground state.

Note that these no-go theorems do not require the full wallpaper symmetry to be applicable. In particular, the first applies whenever there are translation or glide symmetries, the second applies whenever there are commuting translation and mirror symmetries, and the last applies whenever there is a rotation symmetry. When a full wallpaper symmetry is present, there are often multiple translations, glide reflections, mirror and rotation symmetries, so a given distribution of DOF may trigger multiple of these basic no-go theorems. It is straightforward to check that knowing which no-go theorems are triggered is actually also sufficient to know which lattice homotopy class this distribution of DOF is in.

One can see that, for a given wallpaper group $G_s$ and a PR type of the system, the spatial distributions of DOF form an Abelian group, denoted by $A_{\mathrm{LH}}$. Each group element in $A_{\mathrm{LH}}$ corresponds to a lattice homotopy class of distributions of DOF, the multiplication between two group elements corresponds to physically stacking two such distributions of DOF together, and the trivial group element corresponds to a distribution of DOF that is free of all 3 basic no-go theorems above (i.e., a distribution of DOF with no net nontrivial PR in any fundamental domain, any translation unit on any mirror axis, or any $C_2$ rotation center). Due to the $\mathbb{Z}_2$ nature of the PR, the inverse of each group element is itself, so $A_{\mathrm{LH}} = \mathbb{Z}_2^k$ with $k \in \mathbb{N}$.

It turns out that elements in $A_{\mathrm{LH}}$ are in one-to-one correspondence with different LSM constraints [5], i.e., the ground states emergible in systems with distributions of DOF corresponding to different group elements of $A_{\mathrm{LH}}$ must be different, in the sense that they have different emergent order or symmetry-protected distinction. As an example, the trivial element represents the absence of any LSM constraint, i.e., a sym-SRE ground state is allowed if the microscopic DOF of the system are arranged in a configuration corresponding to the trivial element. Therefore, the intuitive geometric picture based on lattice homotopy gives an elegant characterization and classification of LSM constraints. An important observation that will be very useful later is that the structure of $A_{\mathrm{LH}}$ only depends on $G_s$ and the fact that all PR of $G_{int}$ has a $\mathbb{Z}_2$ nature, but not on other details of $G_{int}$.

When PR of $G_{int}$ are $\mathbb{Z}_2^k$-classified with $k > 1$, the above discussion applies to the case where at most one type of nontrivial PR is present in the system. If all nontrivial PR are allowed to be present, all LSM constraints are classified by $A_{\mathrm{LH}}^k$, i.e., each nontrivial PR can result in LSM constraints classified by $A_{\mathrm{LH}}$, and nontrivial LSM constraints from different nontrivial PR are all different.

To make this discussion more concrete, below we consider two specific examples that will be relevant for the later part of the paper.

### 2.1.1 $G_s = p6m$

We start with the example where $G_s = p6m$, which is the symmetry group of triangular, kagome and honeycomb lattices. The generators, a translation unit cell and IWP of $p6m$ are shown in Fig. 2. The translation vectors of $T_1$ and $T_2$ have the same length, and their angle is $2\pi/3$. There is also a 6-fold rotational symmetry, denoted by $C_6$. Finally, there is a mirror symmetry $M$, whose mirror axis passes through the $C_6$-center and bisects the translation vectors of $T_1$ and $T_2$.

We wish to understand how to identify the distributions of DOF with the elements in $A_{\mathrm{LH}}$ in this example. First consider the case where all DOF in the system are in the trivial PR. This

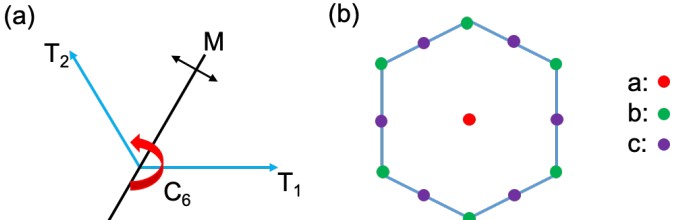

Figure 2: Panel (a) shows the generators of the wallpaper group $p6m$. In panel (b), the hexagon is a translation unit cell of the wallpaper group $p6m$. It has three IWP, usually labelled by $a$, $b$ and $c$ in crystallography, and they form the sites of the triangular, honeycomb and kagome lattices, respectively. The $C_6$ rotation center is at the type-$a$ IWP.

distribution of DOF is free of all the 3 basic no-go theorems, so it corresponds to the trivial element of $A_{LH}$, which physically implies that there is no LSM constraint associated with this distribution of DOF, and sym-SRE ground states are allowed. This is indeed the common belief.

Next, consider putting DOF with nontrivial PR on any of the three types of IWP. First, imagine putting DOF with nontrivial PR on the type-$b$ IWP. One can check that none of the 3 basic no-go theorems is triggered, so this distribution of DOF also corresponds to the trivial group element, and there should be no associated LSM constraint. Indeed, this configuration is where the DOF are on a honeycomb lattice, and it is known that sym-SRE ground states are allowed in this case [34–37], consistent with the absence of any LSM constraint. Second, imagine putting DOF with nontrivial PR on the type-$a$ IWP. One can check that all 3 basic no-go theorems are triggered, so this configuration should correspond to a nontrivial element in $A_{LH}$, and such a system has a nontrivial LSM constraint that precludes any sym-SRE ground state. The same is true if DOF with nontrivial PR are put on the type-$c$ IWP. Moreover, one can also check that the distributions of DOF on type-$a$ and type-$c$ IWP are in different lattice homotopy classes, i.e., they cannot be smoothly deformed into each other. So they correspond to different group elements in $A_{LH}$, which indicates different LSM constraints. These two types of IWP form a triangular and kagome lattice, respectively, and there is indeed no known example of symmetric states that can emerge in both triangular and kagome lattices, without showing any difference in emergent order or symmetry-protected distinction.[3]

Finally, one can also consider putting DOF with nontrivial PR on multiple of the three types of IWP. For instance, putting these DOF on both type-$a$ and type-$c$ IWP is equivalent to stacking systems with DOF arranged on a triangular lattice and kagome lattice together, which corresponds to multiplying the two nontrivial group elements in the last paragraph.

Taken together, the above analysis indicates that the LSM constraints on a lattice with $p6m$ symmetry are classified by $A_{LH} = \mathbb{Z}_2^2$, and the two generators can be taken to correspond to distributions of DOF on triangular and kagome lattices, respectively.

In the above, we have worked out $A_{LH}$ by examining whether any of the basic no-go theorems is triggered by a distribution of DOF. To finish the discussion of this case with $G_s = p6m$, we demonstrate how the information about which basic no-go theorems are triggered can uniquely determine the lattice homotopy class. In this case, we just need to consider the third type of the basic no-go theorems. For this type of no-go theorems, there are two independent ones, triggered by putting DOF with nontrivial PR on the type-$a$ and type-$c$ IWP, respectively. So if we know which of the no-go theorems are triggered, we also know whether there are

---

[3]In fact, even spontaneously-symmetry-breaking states (such as ferromagnetic states) realized on these two lattices should be distinct, because they have different anomalies. However, to the best of our knowledge, it is still an open problem to explicitly calculate the complete anomalies for these spontaneously-symmetry-breaking states, which is an interesting problem beyond the scope of the current paper.

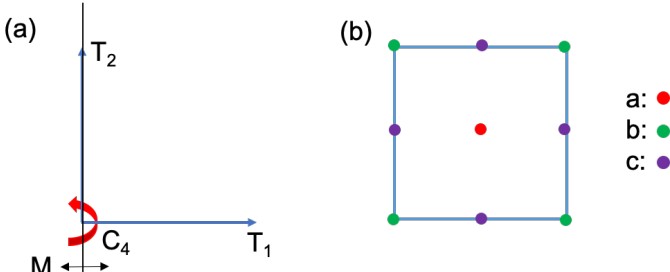

Figure 3: Panel (a) shows the generators of the wallpaper group $p4m$. In panel (b), the square is a translation unit cell of the wallpaper group $p4m$. It has three IWP, usually labelled by $a$, $b$ and $c$ in crystallography. Type-$a$ and type-$b$ both form a square lattice. The $C_4$ rotation center in panel (a) is taken to be at the type-$a$ IWP.

nontrivial PR carried by type-$a$ and type-$c$ IWP. From the previous discussion, this can uniquely determine the lattice homotopy class. This observation will be very useful when we construct a topological characterization of the LSM constraints later.

### 2.1.2 $G_s = p4m$

Warmed up with the example where $G_s = p6m$, now we can easily apply the similar analysis to the other 16 wallpaper groups. Here, we examine the case where $G_s = p4m$, which will be relevant to our later discussion.

The $p4m$ group describes the symmetry of square and checkerboard lattices. The generators, a translation unit cell and IWP of $p4m$ are shown in Fig. 3. The translation vectors of $T_1$ and $T_2$ have the same length and are perpendicular. There is also a 4-fold rotational symmetry, denoted by $C_4$. Finally, there is a mirror symmetry $M$, whose mirror axis passes through the $C_4$-center and is parallel to the translation vector of $T_2$. There are 3 types of IWP. The type-$a$ IWP is the 2-fold rotation centers of $C_4^2$, the type-$b$ IWP is the 2-fold rotation centers of $T_1 T_2 C_4^2$, and the type-$c$ IWP includes the 2-fold rotation centers of both $T_1 C_4^2$ and $T_2 C_4^2$. Note that the type-$a$ and type-$b$ are actually also 4-fold rotation centers, and all three IWP lie on some mirror axes.

Below, we enumerate some distributions of DOF that correspond to different elements in $A_{\text{LH}}$ in this case:

1. All DOF have trivial PR: trivial element in $A_{\text{LH}}$.

2. DOF with nontrivial PR at one of the three types of IWP: three different elements in $A_{\text{LH}}$.

3. DOF with nontrivial PR at multiple of the IWP: product of elements in the previous case.

This analysis implies that the LSM constraints on a lattice with $p4m$ symmetry are classified by $A_{\text{LH}} = \mathbb{Z}_2^3$, and the three generators can be taken to correspond to distributions of DOF on the three types of IWP. Note that both the type-$a$ and type-$b$ IWP form a square lattice, and type-$c$ IWP form a checkerboard lattice. Again, it is easy to see that knowing which of the basic no-go theorems are triggered can uniquely determine the lattice homotopy class.

Before finishing the review, we note that it has also been realized that LSM constraints are intimately related to anomalies and higher dimensional SPTs [4, 6–9]. In the present context, our $(2+1)$-d system with DOF carrying PR can be viewed as a boundary of a $(3+1)$-d system made of stacked $(1+1)$-d SPTs protected by $G_{int}$, which are also classified by $H^2(G_{int}, U(1)_\rho) = \mathbb{Z}_2^k$. The spatial extension of these $(1+1)$-d SPTs is along the extra dimension. The boundaries of these SPTs carry the PR, whose types and locations precisely

match with the DOF of the original $(2+1)$-d system, which have been moved to the IWP using lattice homotopy. Furthermore, the wallpaper symmetry $G_s$ can be naturally extended into a symmetry of the $(3+1)$-d system. Then the $(3+1)$-d system is an SPT protected by $G_s \times G_{int}$, and a sym-SRE boundary of such a nontrivial SPT is forbidden due to the nontrivial quantum anomaly, which implies the LSM constraints. Moreover, different SPTs have different anomalies on the boundary, so their corresponding LSM constraints must be different, such that ground states emergible in systems with different LSM constraints must have distinction in their emergent order or symmetry-protected distinction. For these reasons, in the following we will view an LSM constraint and the $(3+1)$-d $G_s \times G_{int}$ SPT corresponding to this LSM constraint on equal footing.

## 2.2 Topological characterization of the LSM constraints

The above picture of lattice homotopy and higher dimensional SPTs allows us to derive a topological characterization of the LSM constraints. In particular, we will identify the topological partition function (TPF) of the $(3+1)$-d SPT corresponding to each nontrivial LSM constraint for a given $G_s$ and $G_{int}$.

To do it, we use the fact that the SPT of interest can be constructed by stacking the nontrivial $(1+1)$-d $G_{int}$ SPT at various IWP. Suppose, in the language of Dijkgraff-Witten theories [38–40], the TPF of this $(1+1)$-d SPT is encoded in a nontrivial cocycle in $H^2(G_{int}, U(1)_\rho) \cong \mathbb{Z}_2^k$, which can be represented by $\exp(i\pi\eta(a_1, a_2))$, where $a_{1,2} \in G_{int}$ and $\eta$ takes values in $\{0, 1\}$ (taking $\eta \in \{0, 1\}$ is valid since such SPTs are $\mathbb{Z}_2^k$-classified). To write down the TPF of the relevant $(3+1)$-d $G_s \times G_{int}$ SPT, we view $G_s$ on equal footing with $G_{int}$, keeping in mind that any orientation-reversal element in $G_s$ should also complex conjugate the $U(1)$ coefficient, in accordance with the crystalline equivalence principle [41]. Then the TPF can be encoded in a cocycle $\Omega(g_1, g_2, g_3, g_4)$ in $H^4(G_s \times G_{int}, U(1)_\rho)$, where $g_{1,2,3,4} \in G_s \times G_{int}$. The picture based on lattice homotopy and stacks of $(1+1)$-d $G_{int}$ SPT strongly suggests that $\Omega(g_1, g_2, g_3, g_4)$ takes the form

$$\Omega(g_1, g_2, g_3, g_4) = e^{i\pi\lambda(l_1, l_2)\eta(a_3, a_4)}, \tag{1}$$

where $g_i \in G_s \times G_{int}$ is written as $g_i = l_i \otimes a_i$, with $l_i \in G_s$ and $a_i \in G_{int}$, and $\lambda$ also takes values in $\{0, 1\}$. Physically, $\lambda$ encodes the information of which IWP host the $(1+1)$-d $G_{int}$ SPT. The lattice homotopy picture further suggests that $\lambda$ is completely determined by $G_s$ and the lattice homotopy class corresponding to the particular LSM constraint, and should be the same for all $G_{int}$ with $\mathbb{Z}_2^k$-classified PR and all PR types of the system. Such a cocycle implies that the TPF, in terms of lattice gauge theory on a triangulated manifold, takes the form

$$\mathcal{Z} = e^{i\pi \int_{\mathcal{M}_4} \lambda[A_s] \cup \eta[A_{int}]}, \tag{2}$$

where $\mathcal{M}_4$ is the 4 dimensional spacetime manifold of the SPT, $A_s$ and $A_{int}$ are the (1-form) gauge fields resulting from gauging $G_s$ and $G_{int}$, respectively, and $\exp(i\pi \int \eta[A_{int}])$ gives the TPF of the $(1+1)$-d $G_{int}$ SPT. Note that although the TPF is constructed from a cup product of $\lambda$ and $\eta$, generically $\lambda$ (or $\eta$) itself cannot be written as a cup product of $A_s$ (or $A_{int}$).

In Appendix B, we show that the above expectation is indeed correct. Furthermore, $\lambda(l_1, l_2)$ can be viewed as a representative cochain in $H^2(G_s, \mathbb{Z}_2)$. Assuming that the $(1+1)$-d $G_{int}$ SPT is already understood (i.e., the $\eta$ corresponding to the PR type of the system is known), the task to identify the TPF for the $(3+1)$-d $G_s \times G_{int}$ SPT corresponding to the LSM constraints becomes identifying $\lambda(l_1, l_2)$ for a given $G_s$ and lattice homotopy class.

Before proceeding, let us pause to clarify what it means to identify $\lambda(l_1, l_2)$. After all, as reviewed in Appendix A.1, $\lambda(l_1, l_2)$ changes under coboundary transformations, so it is not an invariant characterization of the LSM constraints. However, inequivalent $\lambda$'s can be

diagnosed by quantities related to it that are invariant under coboundary transformations. So identifying $\lambda(l_1, l_2)$ really means identifying these *topological invariants*. To relate to some known results of such topological invariants, we define $\omega(l_1, l_2) \equiv e^{i\pi\lambda(l_1, l_2)}$, which encodes the same information as $\lambda(l_1, l_2)$. Then a topological invariant takes the form of $\alpha[\omega]$, a functional of $\omega$.

Now we proceed to derive these topological invariants. Because $\lambda(l_1, l_2)$ or $\omega(l_1, l_2)$ is the same for all $G_{int}$, it suffices to derive it in a particularly simple and illuminating case, i.e., $G_{int} = SO(3)$. According to Sec. 2.1, in this case the $(3+1)$-d $G_s \times G_{int}$ SPTs related to the LSM constraints are fully characterized by the spatial distribution of Haldane chains, i.e., $(1+1)$-d SPT protected by the $SO(3)$ symmetry. Therefore, to characterize the LSM constraints, all we have to do is to identify topological invariants for $H^2(G_s, \mathbb{Z}_2)$ that can tell us which IWP host Haldane chains. To this end, we utilize the fact that, for a given spatial distribution of Haldane chains, which IWP host Haldane chains is fully encoded in which of the 3 basic no-go theorems are triggered. So if we can characterize the 3 basic no-go theorems using some topological invariants, we can further get the topological invariants corresponding to the LSM constraints.

To obtain the topological invariants corresponding to the 3 basic no-go theorems, it is useful to consider coupling the system to a probe gauge field of the $SO(3)$ symmetry and examine the monopoles of this $SO(3)$ gauge field, which is a method proven to be extremely powerful [14, 42–47]. Because the wave function of the system acquires a $-1$ topological phase factor when an $SO(3)$ monopole circles around a Haldane chain[4], we will see below that if any of the 3 basic no-go theorems is triggered, the $G_s$ symmetry will fractionalize on the $SO(3)$ monopole in a specific way, i.e., the $SO(3)$ monopole will carry a specific projective representation of $G_s$. The symmetry fractionalization pattern of $G_s$ on the $SO(3)$ monopole will thus completely encode the LSM constraint. Since the fusion rule of the $SO(3)$ monopole is determined by $\pi_1(SO(3)) = \mathbb{Z}_2$ [48], the symmetry fractionalization patterns of $G_s$ on the $SO(3)$ monopole are classified by $H^2(G_s, \mathbb{Z}_2)$ [12, 13, 49, 50]. So the LSM constraints can be characterized by elements in $H^2(G_s, \mathbb{Z}_2)$, consistent with the previous general discussion. This also implies that when $G_{int} = SO(3)$, for a given $G_s$ and lattice homotopy class, the $\lambda(l_1, l_2)$ in Eq. (1) should be precisely the element in $H^2(G_s, \mathbb{Z}_2)$ that describes the symmetry fractionalization pattern of $G_s$ on the $SO(3)$ monopole in the corresponding SPT. However, one should not expect that all symmetry fractionalization patterns captured by $H^2(G_s, \mathbb{Z}_2)$ are related to LSM constraints. To see it, consider breaking the $SO(3)$ symmetry to $U(1)$. Then the original LSM-related $G_s \times SO(3)$ SPT will become a trivial $G_s \times U(1)$ SPT, since the Haldane chain is trivialized upon this symmetry breaking. Therefore, the $U(1)$ monopole, which is the descendent of the $SO(3)$ monopole after symmetry breaking, should carry no nontrivial symmetry fractionalization pattern. It implies that certain nontrivial symmetry fractionalization pattern on the $SO(3)$ monopoles, or certain elements in $H^2(G_s, \mathbb{Z}_2)$, may be unrelated to LSM constraints. We will see this explicitly below.

We start with the first no-go theorem, and focus on the case where only translation symmetry is important, and defer a similar discussion where the glide reflection is also important to Appendix C. Denote the two translation generators by $T_1$ and $T_2$, and apply the operation $T_2^{-1} T_1^{-1} T_2 T_1$ to an $SO(3)$ monopole, which moves it around a translation unit cell. If each translation unit cell constains an odd (even) number of Haldane chains, this process results in a $-1$ $(1)$ phase factor, which precisely characterizes how the translation symmetry fraction-

---

[4]Consider moving a Haldane chain around an $SO(3)$ monopole. The topological phase factor generated in this process is given by the topological partition function of the Haldane chain, calculated on the manifold defined by the spacetime trajectory it moves along, with a background $SO(3)$ gauge bundle exerted by the $SO(3)$ monopole. It is known that the topological partition function of a Haldane chain is $e^{i\pi \int_{\mathcal{M}} w_2^{SO(3)}}$, where $w_2^{SO(3)}$ is the second Stiefel-Whitney class of the $SO(3)$ gauge bundle. Furthermore, $\int_{\mathcal{M}} w_2^{SO(3)} = 1$ around an $SO(3)$ monopole. Therefore, there is a $-1$ phase factor generated in this process, which also implies that moving an $SO(3)$ monopole around a Haldane chain results in a $-1$ topological phase factor.

alizes on the $SO(3)$ monopole. By slightly abusing the notation, we write the subgroup of $G_s$ generated by $T_1$ and $T_2$ as $T_1 \times T_2$. The fractionalization patterns of the $T_1 \times T_2$ symmetry should be classified by $H^2(T_1 \times T_2, \mathbb{Z}_2) = \mathbb{Z}_2$, so the aforementioned phase factor must be given by the unique nontrivial topological invariant in $H^2(T_1 \times T_2, \mathbb{Z}_2)$, i.e., $\alpha_1[\omega] = \frac{\omega(T_1, T_2)}{\omega(T_2, T_1)}$. Denote two elements in this subgroup by $l_1 = T_1^{x_1} T_2^{y_1}$ and $l_2 = T_1^{x_2} T_2^{y_2}$, with $x_{1,2}, y_{1,2} \in \mathbb{Z}$, a representative cochain that triggers this topological invariant is $\omega(l_1, l_2) = (-1)^{y_1 x_2}$.

Next, consider the second no-go theorem. Denote the generator of the relevant mirror symmetry by $M$, and suppose $T$ generates a translation symmetry on the mirror plane. Note that this implies $TM = MT$. Apply the operation $MT^{-1}MT$ to an $SO(3)$ monopole, which moves it along a trajectory that encloses a translation unit along the mirror plane. Suppose there is an odd (even) number of Haldane chains in this translation unit, this process results in a $-1$ (1) phase factor, which precisely characterizes how the symmetry group generated by $M$ and $T$ fractionalizes on the $SO(3)$ monopole. Write the subgroup of $G_s$ generated by $M$ and $T$ as $M \times T$, the fractionalization patterns of the $M \times T$ symmetry are classified by $H^2(M \times T, \mathbb{Z}_2) = \mathbb{Z}_2^2$. So there are two nontrivial topological invariants in $H^2(M \times T, \mathbb{Z}_2)$, and they can be written as $\alpha_2[\omega] = \frac{\omega(T,M)}{\omega(M,T)}$ and $\alpha_{\text{non-LSM}} = \frac{\omega(M,M)}{\omega(1,1)}$, where in the denominator 1 stands for the trivial group element in $M \times T$. Note that $\alpha_{\text{non-LSM}} = -1$ would imply when the $SO(3)$ symmetry is broken to $U(1)$, the resulting $G_s \times U(1)$ state is a nontrivial SPT, since this represents a nontrivial symmetry fractionalization pattern of a $U(1)$ monopole [45, 46]. According to the previous general discussion, $\alpha_{\text{non-LSM}}$ should be unrelated to LSM constraints of interest.

We can also directly see that $\alpha_{\text{non-LSM}}$ is unrelated to the LSM constraints without considering breaking the $SO(3)$ symmetry. Denote two elements in $M \times T$ by $l_1 = T^{x_1} M^{m_1}$ and $l_2 = T^{x_2} M^{m_2}$, with $x_{1,2} \in \mathbb{Z}$ and $m_{1,2} \in \{0, 1\}$, representative cochains that trigger these two topological invariants are $\omega(l_1, l_2) = (-1)^{m_1 x_2}$ and $\omega(l_1, l_2) = (-1)^{m_1 m_2}$, respectively. Suppose $\lambda$ in Eq. (1) constains a piece $\lambda(l_1, l_2) = m_1 m_2$, such that $\alpha_{\text{non-LSM}} = -1$, from Eq. (2), we see the TPF of the $(3+1)$-d SPT contains a part given by $\exp(i\pi \int (w_1^{TM})^2 w_2^{SO(3)})$, where $w_1^{TM}$ is the first Stiefel-Whitney class of the tangent bundle of the spacetime manifold, and $w_2^{SO(3)}$ is the second Stiefel-Whitney class of the $SO(3)$ gauge bundle. In writing this down, we have used that $M$ is an orientation reversal symmetry and that the TPF of a Haldane chain is $\exp(i\pi \int w_2^{SO(3)})$. This means that when the symmetry is broken down to $M \times SO(3)$, the system is still a nontrivial SPT.[5] However, all SPTs corresponding to LSM constraints become trivial when the lattice symmetry contains only a mirror symmetry (as can be seen from lattice homotopy, or simply from the lack of basic no-go theorem that only requires a mirror symmetry as the lattice symmetry), and hence a contradiction. This again means $\alpha_{\text{non-LSM}} = 1$ for SPTs corresponding to LSM constraints (see Appendix D for the physics of the SPTs that trigger $\alpha_{\text{non-LSM}}$). Therefore, the phase factor resulted from acting $MT^{-1}MT$ to an $SO(3)$ monopole must be given by $\alpha_2$.

Finally, consider the third no-go theorem. Denote the generator of the relevant 2-fold rotational symmetry by $C_2$, and apply $C_2$ to an $SO(3)$ monopole twice, which moves it around a $C_2$ rotation axis. Suppose there is an odd (even) number of Haldane chains in this $C_2$ rotation axis, this process results in a $-1$ (1) phase factor, which precisely characterizes how the $C_2$ rotational symmetry fractionalizes on the $SO(3)$ monopole. Write the subgroup of $G_s$ generated by $C_2$ also as $C_2$. The fractionalization patterns of the $C_2$ symmetry are classified by $H^2(C_2, \mathbb{Z}_2) = \mathbb{Z}_2$, so the aforementioned phase factor must be given by the unique topolog-

---

[5]In this SPT, the symmetry $M$ fractionalizes on the $SO(3)$ monopole, i.e., acting $M$ twice on an $SO(3)$ monopole yields a $-1$ phase factor. This symmetry fractionalization pattern is captured by $H^2(M, \mathbb{Z}_2) = \mathbb{Z}_2$, whose unique topological invariant is $\alpha_{\text{non-LSM}}$. So $\alpha_{\text{non-LSM}}$ should be identified as this phase factor. In Appendix D, we further show that such an SPT can be constructed by putting on its $M$ mirror plane a $(2+1)$-d $\mathbb{Z}_2 \times SO(3)$ SPT, whose $\mathbb{Z}_2$ domain walls are decorated with Haldane chains.

ical invariant in $H^2(C_2, \mathbb{Z}_2)$, i.e., $\alpha_3[\omega] = \frac{\omega(C_2, C_2)}{\omega(1,1)}$. Denote two elements in this subgroup by $l_1 = C_2^{c_1}$ and $l_2 = C_2^{c_2}$, with $c_{1,2} \in \{0, 1\}$, a representative cochain that triggers this topological invariant is $\omega(l_1, l_2) = (-1)^{c_1 c_2}$.

In summary, we have found 4 basic types of symmetry fractionalization patterns of $G_s$, characterized by the above 4 types of topological invariants, $\alpha_{1,2,3}$ and $\alpha_{\text{non-LSM}}$. The first three are related to the 3 basic no-go theorems, thus to the LSM constraints, while the last is a non-LSM symmetry fractionalization pattern. As mentioned before, if $G_{int} = SO(3)$ is broken to $U(1)$, $\alpha_{\text{non-LSM}}$ detects a nontrivial symmetry fractionalization pattern of a $U(1)$ monopole, captured by a nontrivial element in $H^2(G_s, U_\rho(1))$ [6]. One can also see that $\alpha_{1,2,3} = -1$ does not imply that the descendent $G_s \times U(1)$ SPT is nontrivial, since they correspond to trivial elements in $H^2(G_s, U_\rho(1))$. These 4 basic fractionalization patterns are clearly independent of each other, as they correspond to completely different $(3 + 1)$-d SPTs. Furthermore, for all 17 wallpaper groups $G_s$, these 4 types of fractionalization patterns give a complete set of topological invariants that can distinguish all elements of $H^2(G_s, \mathbb{Z}_2)$, as explicitly checked in Appendix F. These actually mean that

$$A_{\text{LH}} = \ker[\tilde{i} : H^2(G_s, \mathbb{Z}_2) \to H^2(G_s, U(1)_\rho)], \tag{3}$$

where $\tilde{i}$ is the map defined in Eq. (57).

With this in mind, to further derive the topological invariants corresponding to an LSM constraint, we just need to write down the complete set of independent topological invariants of $H^2(G_s, \mathbb{Z}_2)$, and bridge the combinations of these topological invariants with distributions of DOF. Then each combination is a topological invariant for an LSM constraint, which determines $\lambda$ in Eq. (1). Combined with $\eta$ corresponding to the PR type of the system, Eq. (1) or (2) gives the TPF of the $(3 + 1)$-d $G_s \times G_{int}$ SPT corresponding to this LSM constraint. An advantage of this approach is its intuitive nature, i.e., everything can be done by simply inspecting the IWP. Below we perform this analysis in detail for the cases with $G_s = p6m$ and $G_s = p4m$, which will be relevant to the discussion of symmetry-enriched criticality later in the paper. In Appendix F, we present all topological invariants that characterize $H^2(G_s, \mathbb{Z}_2)$, with $G_s$ being any of the 17 wallpaper groups.

Before moving on, we stress again that the topological characterization of the LSM constraints obtained here applies to all $G_{int}$ with $\mathbb{Z}_2^k$-classified PR and all PR types of the system, although it is derived in a special case with $G_{int} = SO(3)$.

### 2.2.1 $G_s = p6m$

All fractionalization patterns of $p6m$ are classified by $H^2(p6m, \mathbb{Z}_2) = \mathbb{Z}_2^4$. As discussed in Sec. 2.1, $p6m$ has two IWP related to LSM constraints, type-$a$ and type-$c$. The former is the 2-fold rotation center of $C_6^3$, and the latter includes the 2-fold rotation centers of $T_1 C_6^3$, $T_2 C_6^3$ and $T_1 T_2 C_6^3$. In addition, $p6m$ also has two independent mirror symmetries, $M$ and $C_6^3 M$. Using the 4 types of basic topological invariants discussed above, we can immediately write down the complete set of independent topological invariants which can distinguish all elements in

---

[6]More precisely, $H^2_{\text{Borel}}(G_s, U(1)_\rho)$. Especially, $H^2_{\text{Borel}}(p1, U(1)_\rho) \cong H^3(p1, \mathbb{Z}_\rho) = 0$.

$H^2(p6m, \mathbb{Z}_2)$:

$$\alpha_1^{p6m}[\omega] = \frac{\omega(C_6^3, C_6^3)}{\omega(1,1)},$$

$$\alpha_2^{p6m}[\omega] = \frac{\omega(T_1 C_6^3, T_1 C_6^3)}{\omega(1,1)},$$

$$\alpha_3^{p6m}[\omega] = \frac{\omega(M, M)}{\omega(1,1)},$$

$$\alpha_4^{p6m}[\omega] = \frac{\omega(C_6^3 M, C_6^3 M)}{\omega(1,1)}. \tag{4}$$

Physically, $\alpha_1^{p6m}$ and $\alpha_2^{p6m}$ measure the PR at the type-$a$ and type-$c$ IWP, respectively, while $\alpha_3^{p6m}$ and $\alpha_4^{p6m}$ determine whether the $(3+1)$-d $G_s \times G_{int}$ SPT contains a non-LSM component. Mathematically, the correctness, completeness and independence of these topological invariants can be checked using the representative cochains in Appendix E.

Therefore, when $\alpha_3^{p6m} = \alpha_4^{p6m} = 1$, the combinations $(\alpha_1^{p6m}, \alpha_2^{p6m})$ are the sought-for topological invariants that characterize the LSM constraints in a lattice with $G_s = p6m$. In particular, $(\alpha_1^{p6m}, \alpha_2^{p6m}) = (-1, 1)$ and $(\alpha_1^{p6m}, \alpha_2^{p6m}) = (1, -1)$ imply that there are DOF with nontrivial PR at the type-$a$ and type-$c$ IWP, respectively, which are the generators of $A_{\mathrm{LH}}$, as discussed in Sec. 2.1. When at least one of $\alpha_3^{p6m}$ and $\alpha_4^{p6m}$ is $-1$, this combination does not correspond to any LSM constraint.

We remark that the choice of topological invariants is not unique. For example, the expression of $\alpha_2^{p6m}[\omega]$ can be replaced by either $\frac{\omega(T_2 C_6^3, T_2 C_6^3)}{\omega(1,1)}$ or $\frac{\omega(T_1 T_2 C_6^3, T_1 T_2 C_6^3)}{\omega(1,1)}$, because $\frac{\omega(T_1 C_6^3, T_1 C_6^3)}{\omega(1,1)} = \frac{\omega(T_2 C_6^3, T_2 C_6^3)}{\omega(1,1)} = \frac{\omega(T_1 T_2 C_6^3, T_1 T_2 C_6^3)}{\omega(1,1)}$ for a cocycle $\omega(g_1, g_2)$ in $H^2(p6m, \mathbb{Z}_2)$, as can be checked by using the representative cochains in Appendix E. Physically, this just means that the 2-fold rotation centers of $T_1 C_6^3$, $T_2 C_6^3$ and $T_1 T_2 C_6^3$ are related by symmetry, so the PR at these three rotation centers should be the same. We can also replace the expression of $\alpha_2^{p6m}[\omega]$ by $\frac{\omega(T_1, T_2)}{\omega(T_2, T_1)}$, which tells us whether there is a net nontrivial PR in a translation unit cell and equals $\alpha_1^{p6m}[\omega] \cdot \alpha_2^{p6m}[\omega]$. This information combined with $\alpha_1^{p6m}[\omega]$ also completely specifies which IWP host Haldane chains.

It is useful to notice some interesting relations between LSM constraints with $G_s = p6m$ and those with $G_s$ being a subgroup of $p6m$. In particular, consider the case where $G_s = cmm$, which is a subgroup of $p6m$ generated by $T_1$, $T_2$, $C_2 \equiv C_6^3$ and $M$. That is, the 3-fold rotational symmetry generated by $C_6^2$ is absent. This wallpaper group has 3 IWP, where the first is the 2-fold rotation center of $C_2$, the second is the 2-fold rotation center of $T_1 T_2 C_2$, and the last includes the 2-fold rotation centers of both $T_1 C_2$ and $T_2 C_2$. Furthermore, there are two independent mirror symmetries, generated by $M$ and $C_2 M$. Similar analysis as before indicates that, for $cmm$, the 3 LSM fractionalization patterns and 2 non-LSM fractionalization patterns

are detected by topological invariants

$$\alpha_1^{cmm}[\omega] = \frac{\omega(C_2, C_2)}{\omega(1,1)},$$

$$\alpha_2^{cmm}[\omega] = \frac{\omega(T_1 T_2 C_2, T_1 T_2 C_2)}{\omega(1,1)},$$

$$\alpha_3^{cmm}[\omega] = \frac{\omega(T_1 C_2, T_1 C_2)}{\omega(1,1)}, \tag{5}$$

$$\alpha_4^{cmm}[\omega] = \frac{\omega(M, M)}{\omega(1,1)},$$

$$\alpha_5^{cmm}[\omega] = \frac{\omega(C_2 M, C_2 M)}{\omega(1,1)}.$$

So when $\alpha_4^{cmm} = \alpha_5^{cmm} = 1$, the combinations $(\alpha_1^{cmm}, \alpha_2^{cmm}, \alpha_3^{cmm})$ are the topological invariants that characterize the LSM constraints in a lattice with $G_s = cmm$.

It is easy to see that the first IWP of $cmm$ is just the descendent of the type-$a$ IWP of $p6m$, and the second and third IWP of $cmm$ are descendent of the type-$c$ IWP of $p6m$. Moreover, the mirror symmetries of $cmm$ are also the descendent mirror symmetries of $p6m$. This means that the fractionalization pattern of $p6m$ can be completely specified by that of its $cmm$ subgroup. More precisely, for a $cmm$ subgroup of $p6m$, we have

$$\alpha_1^{p6m} = \alpha_1^{cmm}, \quad \alpha_2^{p6m} = \alpha_2^{cmm} = \alpha_3^{cmm},$$

$$\alpha_3^{p6m} = \alpha_4^{cmm}, \quad \alpha_4^{p6m} = \alpha_5^{cmm}. \tag{6}$$

These relations allow us to focus on the $cmm$ subgroup of a $p6m$ group when we consider its fractionalization classes, which sometimes simplifies the analysis.

### 2.2.2 $G_s = p4m$

Using similar analysis as before, it is easy to see that the LSM constraints for the case with $G_s = p4m$ are classified by $\mathbb{Z}_2^3$, generated by distributions of DOF with nontrivial PR on the 3 IWP. The 3 root LSM constraints can be detected by topological invariants

$$\alpha_1^{p4m} = \frac{\omega(C_4^2, C_4^2)}{\omega(1,1)},$$

$$\alpha_2^{p4m} = \frac{\omega(T_1 T_2 C_4^2, T_1 T_2 C_4^2)}{\omega(1,1)}, \tag{7}$$

$$\alpha_3^{p4m} = \frac{\omega(T_1 C_4^2, T_1 C_4^2)}{\omega(1,1)}.$$

Again, the $p4m$ symmetry requires that $\frac{\omega(T_1 C_4^2, T_1 C_4^2)}{\omega(1,1)} = \frac{\omega(T_2 C_4^2, T_2 C_4^2)}{\omega(1,1)}$. There are also non-LSM fractionalization patterns classified by $\mathbb{Z}_2^3$, with the following topological invariants for the corresponding generators:

$$\alpha_4^{p4m}[\omega] = \frac{\omega(M, M)}{\omega(1,1)},$$

$$\alpha_5^{p4m}[\omega] = \frac{\omega(T_1 M, T_1 M)}{\omega(1,1)}, \tag{8}$$

$$\alpha_6^{p4m}[\omega] = \frac{\omega(C_4 M, C_4 M)}{\omega(1,1)}.$$

In the case with $G_{int} = SO(3)$, these three topological invariants imply that acting $M$, $T_1 M$ and $C_4 M$ on an $SO(3)$ monopole twice yields a $-1$ phase factor, respectively.

Therefore, when $\alpha_4^{p4m} = \alpha_5^{p4m} = \alpha_6^{p4m} = 1$, the combinations $(\alpha_1^{p4m}, \alpha_2^{p4m}, \alpha_3^{p4m})$ are the topological invariants that characterize the LSM constraints in a lattice with $G_s = p4m$.

Again, it is useful to note the relation between the LSM constraints for $G_s = p4m$ and those for its subgroups. Let us consider the $pmm$ subgroup of $p4m$, generated by $T_1$, $T_2$, $M$ and $C_2 \equiv C_4^2$. That is, the 4-fold rotation is absent while the 2-fold rotation is retained in $pmm$. By inspecting the IWP of $pmm$, we can immediately write down the topological invariants corresponding to the LSM constraints

$$
\begin{aligned}
\alpha_1^{pmm}[\omega] &= \frac{\omega(C_2, C_2)}{\omega(1,1)}, \\
\alpha_2^{pmm}[\omega] &= \frac{\omega(T_1 T_2 C_2, T_1 T_2 C_2)}{\omega(1,1)}, \\
\alpha_3^{pmm}[\omega] &= \frac{\omega(T_1 C_2, T_1 C_2)}{\omega(1,1)}, \\
\alpha_4^{pmm}[\omega] &= \frac{\omega(T_2 C_2, T_2 C_2)}{\omega(1,1)},
\end{aligned}
\tag{9}
$$

and the topological invariants for the non-LSM fractionalization patterns

$$
\begin{aligned}
\alpha_5^{pmm}[\omega] &= \frac{\omega(M, M)}{\omega(1,1)}, \\
\alpha_6^{pmm}[\omega] &= \frac{\omega(C_2 M, C_2 M)}{\omega(1,1)}, \\
\alpha_7^{pmm}[\omega] &= \frac{\omega(T_1 M, T_1 M)}{\omega(1,1)}, \\
\alpha_8^{pmm}[\omega] &= \frac{\omega(T_2 C_2 M, T_2 C_2 M)}{\omega(1,1)}.
\end{aligned}
\tag{10}
$$

So when all $\alpha_5^{pmm} = \alpha_6^{pmm} = \alpha_7^{pmm} = \alpha_8^{pmm} = 1$, the combinations $(\alpha_1^{pmm}, \alpha_2^{pmm}, \alpha_3^{pmm}, \alpha_4^{pmm})$ are the topological invariants that characterize the LSM constraints in a lattice with $G_s = pmm$.

Furthermore, by examining the relation between IWP of $p4m$ and the IWP of its $pmm$ subgroup, we get

$$
\begin{aligned}
\alpha_1^{p4m} &= \alpha_1^{pmm}, \quad \alpha_2^{p4m} = \alpha_2^{pmm}, \\
\alpha_3^{p4m} &= \alpha_3^{pmm} = \alpha_4^{pmm}, \\
\alpha_4^{p4m} &= \alpha_5^{pmm} = \alpha_6^{pmm}, \\
\alpha_5^{p4m} &= \alpha_7^{pmm} = \alpha_8^{pmm}.
\end{aligned}
\tag{11}
$$

So 5 of the 6 topological invariants for $p4m$ can be determined by examining its $pmm$ subgroup. To further determine the last topological invariant, $\alpha_6^{p4m}$, one can simply examine the subgroup generated by $C_4 M$. This observation will also simplify some analysis.

### 2.2.3 Topological characterization of the LSM constraints in $(1+1)$-d

In the above we have derived the topological characterization of LSM constraints in $(2+1)$-d systems. Similar derivation can be carried out for $(1+1)$-d systems with $G_s \times G_{int}$ symmetry, where $G_s$ is one of the two line groups, and $G_{int}$ is an internal symmetry group whose PR are $\mathbb{Z}_2^k$-classified. In this case, the lattice homotopy picture still applies in an analogous way, and

there are $(2+1)$-d $G_s \times G_{int}$ SPTs corresponding to each LSM constraint. Here we present the cocycle and TPF of these SPTs, and leave the details of derivation to Appendix G.

When $G_s = p1$, the line group that contains only translation generated by $T$, the classification of LSM constraints is $\mathbb{Z}_2$, detected by the total PR inside each translation unit cell. The cocycle describing the $(2+1)$-d $p1 \times G_{int}$ SPT related to the nontrivial LSM constraint is

$$\Omega(g_1, g_2, g_3) = e^{i\pi x_1 \eta(a_2, a_3)}, \tag{12}$$

where $g_i = T^{x_i} \otimes a_i$, with $x_i \in \mathbb{Z}$ and $a_i \in G_{int}$, for $i = 1, 2, 3$. The corresponding TPF can be written as

$$\mathcal{Z} = e^{i\pi \int_{\mathcal{M}_3} x \cup \eta[A_{int}]}, \tag{13}$$

where $\mathcal{M}_3$ is the $(2+1)$-d spacetime manifold the SPT lives in, and $x$ is the gauge field corresponding to the translation symmetry.

When $G_s = p1m$, the line group that contains both translation $T$ and mirror $M$, the classification of LSM constraints is $\mathbb{Z}_2^2$, detected by the total PR at the mirror centers of $M$ and $TM$. For the case where only the mirror center of $M$ has a net nontrivial PR, the corresponding cocycle is

$$\Omega(g_1, g_2, g_3) = e^{i\pi(x_1 + m_1)\eta(a_2, a_3)}, \tag{14}$$

where $g_i = T^{x_i} M^{m_i} \otimes a_i$, with $x_i \in \mathbb{Z}$, $m_i \in \{0, 1\}$ and $a_i \in G_{int}$, for $i = 1, 2, 3$. The corresponding TPF can be written as

$$\mathcal{Z} = e^{i\pi \int_{\mathcal{M}_3} (x+m) \cup \eta[A_{int}]}, \tag{15}$$

where $x$ is still the gauge field of translation, and $m$ is the gauge field of mirror symmetry. For the case where only the mirror center of $TM$ has a net nontrivial PR, using similar notations, the corresponding cocycle and TPF are respectively

$$\Omega(g_1, g_2, g_3) = e^{i\pi x_1 \eta(a_2, a_3)}, \tag{16}$$

and

$$\mathcal{Z} = e^{i\pi \int_{\mathcal{M}_3} x \cup \eta[A_{int}]}. \tag{17}$$

## 3 Applications to symmetry-enriched quantum criticality

The above topological characterization of the LSM constraints is not only conceptually important, but also of practical relevance. A crucial question in condensed matter physics is what we call the question of *emergibility*: given an IR effective theory, can it emerge at low energies in a lattice system described by a local Hamiltonian? This question is generically rather challenging, and we will utilize the *hypothesis of emergibility*: given a $(d+1)$-dimensional IR effective theory with symmetry $G_{IR}$, a necessary and sufficient condition for it to emerge from a lattice system with symmetry $G_{UV}$ is that there is a symmetry embedding pattern (SEP), i.e., a homomorphism $\varphi$

$$\varphi : G_{UV} \to G_{IR}, \tag{18}$$

such that the anomaly of this IR effective theory matches with the anomaly of the lattice system coming from the LSM-like constraint, in the sense that

$$\Omega_{UV} = \varphi^*(\Omega_{IR}), \tag{19}$$

where $\Omega_{\mathrm{UV}}$ describes the LSM-like anomaly of the lattice system, $\Omega_{\mathrm{IR}}$ is the anomaly of the IR effective theory, and $\varphi^*$ is the pullback induced by $\varphi$ (see Appendix A.2 for a review). In fact, the necessity of this condition has been established (i.e., 't Hooft anomaly-matching condition), and only the sufficiency of it is hypothetical. Although this hypothesis has not been proved so far, it is supported by many nontrivial examples. In the following we will *assume* the correctness of the hypothesis of emergibility.

The hypothesis of emergibility provides an intrinsic characterization of the emergibility of an IR effective theory, without relying on any of its specific constructions. It is especially useful when there is no known lattice construction of this IR effective theory, but its anomaly is known. A class of such IR theories is the *non-Lagrangian* Stiefel liquids (SLs) proposed in Ref. [15]. A theory is Lagrangian if it can be described by a *weakly-coupled* Lagrangian at high energies, which at low energies may flow to a strongly-coupled fixed point under RG. A non-Lagrangian theory is one that is so strongly interacting, such that it cannot be described by any weakly-coupled Lagrangian at any energy scale. The SLs are proposed to be an infinite family of quantum critical states, where its simplest members are the celebrated deconfined quantum critical point (DQCP) and $U(1)$ Dirac spin liquid (DSL), while other members are conjectured to be non-Lagrangian. Due to the intrinsic absence of a weakly-coupled description, it is difficult to construct these non-Lagrangian states on a lattice system by usual means. However, the anomalies of these SLs are derived in Ref. [15]. With $\Omega_{\mathrm{UV}}$ derived in Sec. 2 (given by Eqs. (1) or (2)), we can check the emergibility of these states in various lattice spin systems, by checking the existence of SEP that can match the anomalies. Based on this approach, some interesting realizations of the non-Lagrangian SLs on triangular and kagome lattices are proposed [15]. Here we will explore this problem more systematically.

Motivated by their relevance in the study of quantum magnetism, in the following we will focus on lattice systems with $G_{\mathrm{UV}} = G_s \times G_{int}$ symmetry, where $G_s$ is $p4m$ or $p6m$, and $G_{int} = O(3)^T \equiv SO(3) \times \mathbb{Z}_2^T$, the product of spin rotation and time reversal symmetries. We further demand that the PR type of the system correspond to half-integer spin, i.e., spinor under $SO(3)$ while Kramers doublet under $\mathbb{Z}_2^T$, which implies that the $(1+1)$-d SPT related to the LSM constraints has a TPF $\exp\left(i\pi \int_{M_2}(w_2^{SO(3)} + t^2)\right) = \exp\left(i\pi \int_{M_2} w_2^{O(3)^T}\right)$. For the IR effective theory, we focus on DQCP, DSL, and the simplest non-Lagrangian SL, denoted by SL$^{(7)}$. We will exhaustively search SEP that can match the anomalies of these IR theories with the LSM anomalies on these lattices, assuming that the IR theories can emerge as a consequence of the competition between a magnetic state (a state that breaks the $SO(3)$ spin rotational symmetry, e.g., a Neel state) and a non-magnetic state (an $SO(3)$ symmetric state, e.g., a valance bond solid (VBS)).

## 3.1 Review of Stiefel liquids

First, we briefly review the physics of SLs [15] (see Appendix H for more details, including useful information absent in Ref. [15]). In Ref. [15], the proposed non-Lagrangian SLs are defined in terms of $2+1$ dimensional non-linear sigma models with target spaces being a Stiefel manifold. Although this sigma-model description is very effective in capturing the kinematic properties of the SLs, these non-renormalizable sigma models have infinite-dimensional parameter spaces, and they do not fully specify the universal low-energy physics of the system until all their infinitely many parameters are specified. So it is desirable to have a definition of these SLs without explicitly referring to any Lagrangian. In the following, we will review the symmetries, anomalies and some dynamical aspects of the SLs, and these proporties can also be viewed as an intrinsic definition of SLs without relying on any Lagrangian.

A SL is labelled by an integer $N \geqslant 5$, and we denote this state by SL$^{(N)}$. The DQCP and DSL correspond to SL$^{(5)}$ and SL$^{(6)}$, respectively, and SL$^{(N>6)}$ are conjectured to be non-Lagrangian.

The DOF of SL$^{(N)}$ is represented by an $N \times (N-4)$ matrix $n$ with orthonormal columns. The symmetry $G_{\text{IR}}$ of SL$^{(N)}$ includes Poincaré symmetry and

$$\frac{O(N)^T \times O(N-4)^T}{\mathbb{Z}_2}. \tag{20}$$

The $O(N)$ acts as $n \to Ln$ with $L \in O(N)$, and the $O(N-4)$ acts as $n \to nR$ with $R \in O(N-4)$. The superscript "T" represents a locking condition: an improper rotation of either the $O(N)$ or $O(N-4)$ is a symmetry if and only if it is combined with a spacetime orientation reversal symmetry. This locking condition is one of the reasons why SL$^{(N \geqslant 7)}$ may be non-Lagrangian (see Appendix H). The modding of $\mathbb{Z}_2$ is because the operation with $L = -I_N$ and $R = -I_{N-4}$ has no action on $n$. For $N = 5$, $n$ reduces to a 5-component vector, and $G_{\text{IR}}$ includes Poincaré symmetry and an $O(5)^T$ symmetry that acts by left multiplication on $n$, such that the improper rotation is combined with a spacetime orientation reversal symmetry.

The anomaly of SL$^{(N)}$ is captured by $\Omega_{\text{IR}}$, an element in $H^4(G_{\text{IR}}, U(1)_\rho)$. It is useful to consider the projection from $\tilde{G}_{\text{IR}} \equiv O(N)^T \times O(N-4)^T$ to $G_{\text{IR}}$, and the pullback of $\Omega_{\text{IR}}$ induced by the projection is given by $\tilde{\Omega}_{\text{IR}} = e^{i\pi\tilde{L}_{\text{IR}}} \in H^4(\tilde{G}_{\text{IR}}, U(1)_\rho)$, where

$$\tilde{L}_{\text{IR}} = w_4^{O(N)} + w_4^{O(N-4)} + \left[ w_2^{O(N-4)} + \left( w_1^{O(N-4)} \right)^2 \right] \left( w_2^{O(N)} + w_2^{O(N-4)} \right) + \left( w_1^{O(N-4)} \right)^4, \tag{21}$$

supplemented with a constraint $w_1^{TM} + w_1^{O(N)} + w_1^{O(N-4)} = 0$ (mod 2), which originates from the locking between the spacetime orientation reversals and the improper rotations of $O(N)$ and $O(N-4)$. Here $w_i^{O(N)}$, $w_i^{O(N-4)}$ and $w_i^{TM}$ are the $i$-th Stiefel-Whitney classes of the $O(N)$, $O(N-4)$ gauge bundles and the tangent bundle of the spacetime manifold, respectively. For odd $N$, $\tilde{\Omega}_{\text{IR}}$ completely characterizes $\Omega_{\text{IR}}$ (see Appendix H for more details). However, for even $N$, $\tilde{\Omega}_{\text{IR}}$ misses some important information. Fortunately, it turns out that $\tilde{\Omega}_{\text{IR}}$ is still adequate for the following discussion, even for the case with $N = 6$ (see Appendix I.2 and discussions below Eq. (229)). Below we will view $\tilde{\Omega}_{\text{IR}}$ as the IR anomaly of SLs and omit the tilde symbol, i.e., we rewrite $\tilde{\Omega}_{\text{IR}}$ and $\tilde{L}_{\text{IR}}$ as $\Omega_{\text{IR}}$ and $L_{\text{IR}}$ for simplicity.

The low-energy dynamics of SLs is not fully understood so far. There is evidence that DQCP is a pseudo-critical state [19,51–54], which can be approximated by a CFT, whose relevant operators are the conserved current, the $SO(5)$ vector and symmetric traceless rank-2 tensor, and possibly time-reversal breaking $SO(5)$ singlet. There is also evidence that SL$^{(N \geqslant 6)}$ are genuine CFTs with no $G_{\text{IR}}$-symmetric relevant operator. Furthermore, various numerical studies (e.g. see a recent conformal bootstrap study [55] and references therein) give (indirect) support that the only relevant operators in these states are either conserved currents, or time-reversal-breaking operators, or Lorentz scalar operators in the representations $(V_L, V_R)$ and $(A_L, A_R)$, where $V_L$ ($V_R$) and $A_L$ ($A_R$) represent the vector and anti-symmetric rank-2 tensor of $SO(N)$ ($SO(N-4)$), respectively. The effects of these relevant operators are complicated: some of them change the emergent order of the state, but others do not (see Appendix H for more details). We are interested in the stability of these states in a specific lattice realization, whose symmetry is $G_{\text{UV}}$. Some perturbations that are not $G_{\text{IR}}$-symmetric can be $G_{\text{UV}}$-symmetric and drive the states unstable. For a realization to be stable, we demand that $G_{\text{UV}}$ forbid all the aforementioned relevant perturbations that change the emergent order of the state.

It is also sometimes useful to refer to the gauge-theoretic description of the DSL (SL$^{(6)}$). This description is in terms of 4 flavors of gapless Dirac fermions coupled to a dynamical $U(1)$ gauge field. The global symmetry of this theory is given by Eq. (20) with $N = 6$. The fundamental operator in this theory is the monopole operators of the $U(1)$ gauge field, which transform as a bi-vector under the $(SO(6) \times SO(2))/\mathbb{Z}_2$ symmetry [56–58]. These monopoles are represented by the $6 \times 2$ matrix $n$ in the language of SLs, which are also the $(V_L, V_R)$ relevant

operator mentioned above. In terms of the gauge theory, other relevant perturbations listed above are as follows. The conserved currents are the flavor currents of the Dirac fermions and the electromagnetic field strengths of the $U(1)$ gauge field, the time-reversal-breaking operator is the fermion mass that is a singlet under the flavor symmetry, and the $(A_L, A_R)$ operator is the fermion mass that is in the adjoint representation of the flavor symmetry.

## 3.2 Method for anomaly-matching

Now we sketch a streamlined method to check the emergibility condition Eq. (19), for a given symmetry embedding pattern (SEP) $\varphi$. This method crucially relies on the fact that $H^4(G_{\mathrm{UV}}, U(1)_\rho) = \mathbb{Z}_2^k$ with some $k \in \mathbb{N}$, which always holds if $G_{\mathrm{UV}} = G' \times \mathbb{Z}_2^T$ with some group $G'$. In our case, $G' = G_s \times SO(3)$. More generally, as long as $G_{\mathrm{UV}} = G' \times \mathbb{Z}_2^T$ for any $G'$, we expect this method to be useful in matching the LSM anomaly of a lattice system with the anomaly of any IR effective theories. This subsection is relatively formal and abstract, and readers more interested in the physical results can skip to the next section.

To motivate this method, first note that $\Omega_{\mathrm{UV}}$ and $\varphi^*(\Omega_{\mathrm{IR}})$ are elements in $H^4(G_{\mathrm{UV}}, U(1)_\rho)$. To compare two elements in $H^4(G_{\mathrm{UV}}, U(1)_\rho)$, generically we need a complete set of topological invariants (or some equivalents) for $H^4(G_{\mathrm{UV}}, U(1)_\rho)$, which is often difficult to obtain. This difficulty comes from the fact that we are considering cohomology with $U(1)$ coefficients.

Nevertheless, simplification occurs when $G_{\mathrm{UV}} = G' \times \mathbb{Z}_2^T$ and hence $H^4(G_{\mathrm{UV}}, U(1)_\rho) = \mathbb{Z}_2^k$ with some $k \in \mathbb{N}$. This enables us to connect $\Omega_{\mathrm{UV}}$ and $\varphi^*(\Omega_{\mathrm{IR}})$ to elements in $H^*(G_{\mathrm{UV}}, \mathbb{Z}_2)$, which simplifies the analysis due to the salient features of cohomologies with $\mathbb{Z}_2$ coefficients.

To see the connection to $H^*(G_{\mathrm{UV}}, \mathbb{Z}_2)$, first recall that $\Omega_{\mathrm{UV}}$ takes the form of Eq. (1). We can view $\lambda$ and $\eta$ as elements in $H^2(G_s, \mathbb{Z}_2)$ and $H^2(G_{int}, \mathbb{Z}_2)$, respectively. Then $\lambda(l_1, l_2)\eta(a_3, a_4)$ is in fact the cup product $\lambda \cup \eta$ [7], which is an element in $H^4(G_{\mathrm{UV}}, \mathbb{Z}_2)$ that we denote by $L_{\mathrm{UV}}$. As a group, here the group operation of two elements in $H^4(G_{\mathrm{UV}}, \mathbb{Z}_2)$ is realized as the mod 2 addition of the representative cochains of these elements, which take values in $\mathbb{Z}_2 = \{0, 1\}$. Then $\Omega_{\mathrm{UV}}$ can be written as $e^{i\pi L_{\mathrm{UV}}}$, or more formally as $\tilde{i}(L_{\mathrm{UV}})$, where $\tilde{i}$ is a map induced by the inclusion $i : \mathbb{Z}_2 \to U(1)$ introduced in Eq. (57). That is, the LSM anomaly $\Omega_{\mathrm{UV}}$ can be expressed as an image of an element $L_{\mathrm{UV}} \in H^4(G_{\mathrm{UV}}, \mathbb{Z}_2)$ under $\tilde{i}$. [8]

Furthermore, there is an *injective* map from $H^4(G_{\mathrm{UV}}, U(1)_\rho)$ to $H^5(G_{\mathrm{UV}}, \mathbb{Z}_2)$, given by $\tilde{p} \circ \beta$, i.e., the combination of the Bockstein homorphism $\beta : H^4(G_{\mathrm{UV}}, U(1)_\rho) \to H^5(G_{\mathrm{UV}}, \mathbb{Z}_\rho)$ and an injective map $\tilde{p} : H^5(G_{\mathrm{UV}}, \mathbb{Z}_\rho) \to H^5(G_{\mathrm{UV}}, \mathbb{Z}_2)$ (see Appendix A.2 for a brief introduction of these maps). Here the fact that $\tilde{p}$ is injective is again guaranteed by $H^4(G_{\mathrm{UV}}, U(1)_\rho) = \mathbb{Z}_2^k$, which is crucial for this method. This means that checking Eq. (19) is equivalent to checking

$$(\tilde{p} \circ \beta)\Omega_{\mathrm{UV}} = (\tilde{p} \circ \beta)\varphi^*(\Omega_{\mathrm{IR}}), \tag{22}$$

where both sides are elements in $H^5(G_{\mathrm{UV}}, \mathbb{Z}_2)$.

Now we discuss the relevant simplifying features of cohomology with $\mathbb{Z}_2$ coefficient. First, for any group $G$, $H^*(G, \mathbb{Z}_2)$ has a *ring structure*, where the addition is the mod 2 addition as above, and the multiplication between two elements is realized as their cup product. The entire cohomology ring $H^*(G, \mathbb{Z}_2)$ can be presented by generators and relations, such that any of its elements can be written as sum of cup products of these generators, while the relations dictate that some sums are in fact the trivial element.

Moreover, $H^*(G_s \times G_{int}, \mathbb{Z}_2) \cong H^*(G_s, \mathbb{Z}_2) \otimes H^*(G_{int}, \mathbb{Z}_2)$ for any $G_s$ and $G_{int}$, which allows us to understand $H^*(G_s \times G_{int}, \mathbb{Z}_2)$ by understanding $H^*(G_s, \mathbb{Z}_2)$ and $H^*(G_{int}, \mathbb{Z}_2)$ separately.

---

[7]More specifically, the cross product $\lambda \times \eta$, defined in Eq. (67) in Appendix A.3.

[8]In fact, since $G_{\mathrm{UV}} = G' \times \mathbb{Z}_2^T$, which implies that $H^n(G_{\mathrm{UV}}, U(1)_\rho) = \mathbb{Z}_2^k$ for any $n \in \mathbb{N}$, any element in $H^n(G_{\mathrm{UV}}, U(1)_\rho)$ can be written as the image of an element in $H^n(G_{\mathrm{UV}}, \mathbb{Z}_2)$ under $\tilde{i}$.

We are interested in the case with $G_{int} = O(3)^T \equiv SO(3) \times \mathbb{Z}_2^T$. The cohomology ring $H^*(O(3)^T, \mathbb{Z}_2)$ is generated by the Stiefel-Whitney classes of $O(3)^T$, i.e., $w_1^{O(3)^T} \in H^1(O(3)^T, \mathbb{Z}_2)$, $w_2^{O(3)^T} \in H^2(O(3)^T, \mathbb{Z}_2)$ and $w_3^{O(3)^T} \in H^3(O(3)^T, \mathbb{Z}_2)$, with no relation among the generators. Sometimes we also need to write $H^*(O(3)^T, \mathbb{Z}_2)$ as $H^*(SO(3), \mathbb{Z}_2) \otimes H^*(\mathbb{Z}_2^T, \mathbb{Z}_2)$, where $H^*(SO(3), \mathbb{Z}_2)$ is generated by the Stiefel-Whitney classes $w_2^{SO(3)}$ and $w_3^{SO(3)}$ of $SO(3)$, and $H^*(\mathbb{Z}_2^T, \mathbb{Z}_2)$ is generated by $t \in H^1(\mathbb{Z}_2^T, \mathbb{Z}_2)$. These two sets of generators are related by

$$
\begin{aligned}
w_1^{O(3)^T} &= t\,, \\
w_2^{O(3)^T} &= w_2^{SO(3)} + t^2\,, \\
w_3^{O(3)^T} &= w_3^{SO(3)} + t w_2^{SO(3)} + t^3\,.
\end{aligned}
\tag{23}
$$

As for $H^*(G_s, \mathbb{Z}_2)$, we have calculated the $\mathbb{Z}_2$ cohomology ring, i.e., the generators and relations, of all 17 wallpaper groups (see Appendix E). It turns out that for all wallpaper groups $G_s$ except $p4g$, all generators belong to $H^1(G_s, \mathbb{Z}_2)$ and $H^2(G_s, \mathbb{Z}_2)$. For $p4g$, besides elements in $H^1(p4g, \mathbb{Z}_2)$ and $H^2(p4g, \mathbb{Z}_2)$, another element in $H^3(p4g, \mathbb{Z}_2)$ is also needed to form a complete set of generators.

The above observations motivate us to consider the following diagram, where each rectangular sub-diagram is commuting [9]:

$$
\begin{array}{ccccccc}
H^4(G_{\mathrm{IR}}, \mathbb{Z}_2) & \dashrightarrow^{\tilde{i}} & H^4(G_{\mathrm{IR}}, U(1)_\rho) & \xrightarrow{\beta} & H^5(G_{\mathrm{IR}}, \mathbb{Z}_\rho) & \xrightarrow{\tilde{p}} & H^5(G_{\mathrm{IR}}, \mathbb{Z}_2) \\
\downarrow{\varphi^*} & & \downarrow{\varphi^*} & & \downarrow{\varphi^*} & & \downarrow{\varphi^*} \\
H^4(G_{\mathrm{UV}}, \mathbb{Z}_2) & \xrightarrow{\tilde{i}} & H^4(G_{\mathrm{UV}}, U(1)_\rho) & \xrightarrow{\beta} & H^5(G_{\mathrm{UV}}, \mathbb{Z}_\rho) & \xrightarrow{\tilde{p}} & H^5(G_{\mathrm{UV}}, \mathbb{Z}_2)
\end{array}
\tag{24}
$$

From the commutativity of the diagram, checking Eq. (22) is equivalent to checking

$$
\mathcal{SQ}^1(L_{\mathrm{UV}}) = \varphi^*(\tilde{p} \circ \beta)(\Omega_{\mathrm{IR}})
\tag{25}
$$

in $H^5(G_{\mathrm{UV}}, \mathbb{Z}_2)$, where

$$
\mathcal{SQ}^1 \equiv \tilde{p} \circ \beta \circ \tilde{i}\,.
\tag{26}
$$

Some important properties and calculations of $\mathcal{SQ}^1$ are given in Appendix A.4. Because of the salient features of cohomologies with $\mathbb{Z}_2$ coefficients, checking Eq. (25) is expected to be simpler than directly checking Eq. (19) for a generic IR effective theory.

For SLs, a further simplification takes place since $\Omega_{\mathrm{IR}} = e^{i\pi L_{\mathrm{IR}}} \in H^4(G_{\mathrm{IR}}, U(1)_\rho)$ for SLs. Here $L_{\mathrm{IR}}$ can also be viewed as an element in $H^4(G_{\mathrm{IR}}, \mathbb{Z}_2)$, in a way similar to $L_{\mathrm{UV}} \in H^4(G_{\mathrm{UV}}, \mathbb{Z}_2)$. Then $\Omega_{\mathrm{IR}}$ is the image of $L_{\mathrm{IR}}$ under the map $\tilde{i} : H^4(G_{\mathrm{IR}}, \mathbb{Z}_2) \to H^4(G_{\mathrm{IR}}, U(1)_\rho)$. Therefore, Eq. (25) becomes

$$
\mathcal{SQ}^1(L_{\mathrm{UV}}) = \varphi^*(\mathcal{SQ}^1(L_{\mathrm{IR}}))\,.
\tag{27}
$$

Below we will use this equation to check the emergibility of various SLs. We remark that to check Eq. (19), one may attempt to check if $L_{\mathrm{UV}} = \varphi^*(L_{\mathrm{IR}})$. However, since $\tilde{i}$ is not injective, this is just a sufficient but unnecessary condition of Eq. (19). As we have checked, $L_{\mathrm{UV}} \neq \varphi^*(L_{\mathrm{IR}})$ in many examples where Eq. (27) holds.

---

[9]The reason to use dashed lines to connect the left corner of the diagram to the rest is because $H^4(G_{\mathrm{IR}}, \mathbb{Z}_2)$ is relevant in this analysis only for theories like SLs, where $\Omega_{\mathrm{IR}}$ is the image of an element in $H^4(G_{\mathrm{IR}}, \mathbb{Z}_2)$ under $\tilde{i}$. For a generic IR effective theory, the left corner is irrelevant to the analysis of anomaly-matching. See Appendix I.1 for an IR effective theory (the $SU(2)_1$ CFT) where this is the case.

### 3.3 Example: anomaly matching for DQCP

To make this discussion more concrete, we showcase this method in a concrete example in detail (see Appendix I for more examples, including an example in $(1+1)$-d).

Consider the classic realization of DQCP (SL$^{(5)}$) on a square lattice [16, 17, 19]. For DQCP, $G_{\text{IR}} = O(5)^T$ and Eq. (21) becomes

$$\Omega_{\text{IR}} \equiv \exp(i\pi L_{\text{IR}}) = \exp\left(i\pi w_4^{O(5)}\right). \tag{28}$$

In this realization, $G_{\text{UV}} = p4m \times O(3)^T$ and the SEP $\varphi$ reads [15, 19],

$$
T_1 \rightarrow \begin{pmatrix} -I_3 & & \\ & -1 & \\ & & 1 \end{pmatrix}, \quad
T_2 \rightarrow \begin{pmatrix} -I_3 & & \\ & 1 & \\ & & -1 \end{pmatrix},
$$
$$
C_4 \rightarrow \begin{pmatrix} I_3 & & \\ & & 1 \\ & -1 & \end{pmatrix}, \quad
M \rightarrow \begin{pmatrix} I_3 & & \\ & -1 & \\ & & 1 \end{pmatrix}, \tag{29}
$$
$$
O(3)^T \rightarrow \begin{pmatrix} O(3)^T & \\ & I_2 \end{pmatrix},
$$

where $I_k$ denotes the $k \times k$ identity matrix. Note that the locking between the spacetime orientation reversals and improper rotations of $O(5)$ is satisfied above. The LSM anomaly Eqs. (1) or (2) in this case can be written as

$$\Omega_{\text{UV}} \equiv \exp(i\pi L_{\text{UV}}) = \exp\left(i\lambda_1 w_2^{O(3)^T}\right), \tag{30}$$

where $\lambda_1 \in H^2(p4m, \mathbb{Z}_2)$ triggers $\alpha_1^{p4m}$ in Eq. (7), i.e., define $\omega_1 \equiv \tilde{i}(\lambda_1) = e^{i\pi\lambda_1}$, then $\alpha_1^{p4m}[\omega_1] = -1$ while $\alpha_i^{p4m}[\omega_1] = 1$ for $i = 2, \ldots, 6$. As a concrete realization of the DQCP, Eq. (19) must hold. Below we check it by checking its equivalent form, Eq. (27).

According to Appendix A.2,

$$\mathcal{SQ}^1(L_{\text{IR}}) = w_5^{O(5)} \quad \text{and} \quad \mathcal{SQ}^1(L_{\text{UV}}) = \lambda_1 w_3^{O(3)^T}, \tag{31}$$

where $w_3^{O(3)^T} = w_3^{SO(3)} + t w_2^{SO(3)} + t^3$ and $t \in H^1(\mathbb{Z}_2^T, \mathbb{Z}_2)$ corresponds to the gauge field of time reversal symmetry $\mathbb{Z}_2^T$, when pulled back to the spacetime manifold $\mathcal{M}_4$.

It remains to calculate the pullback $\varphi^*(\mathcal{SQ}^1(L_{\text{IR}}))$. Since the embedding $\varphi$ is block-diagonal with a $3 \times 3$ block and a $2 \times 2$ block, invoking the Whitney product formula, $w_5^{O(5)} = w_3^{O(3)} w_2^{O(2)}$ [10], we get

$$\varphi^*\left(\mathcal{SQ}^1(L_{\text{IR}})\right) = \varphi^*\left(w_3^{O(3)}\right) \cup \varphi^*\left(w_2^{O(2)}\right). \tag{32}$$

Hence we just need to calculate $\varphi^*(w_3^{O(3)})$ and $\varphi^*(w_2^{O(2)})$. The calculation of $\varphi^*(w_3^{O(3)})$ is straightforward,

$$\varphi^*\left(w_3^{O(3)}\right) = w_3^{SO(3)} + (t + A_{x+y}) w_2^{SO(3)} + (t + A_{x+y})^3, \tag{33}$$

where $A_{x+y} \in H^1(p4m, \mathbb{Z}_2)$ corresponds to the sum of gauge fields of $T_1$ and $T_2$, when pulled back to the spacetime manifold $\mathcal{M}_4$ (see Appendix E).

---

[10] Technically speaking, what we are doing is factorizing $\varphi$ into an embedding $\tilde{\varphi} : G_{\text{UV}} \rightarrow O(3) \times O(2)$ composed with an embedding $\varphi_0 : O(3) \times O(2) \rightarrow O(5)$. Then this equation should be thought of as the pullback of $w_5^{O(5)}$ under $\varphi_0$, which can be proven by considering the diagonal $\mathbb{Z}_2^5$ symmetry. In this paper we will omit this fine detail for simplicity.

The pullback of $w_2^{O(2)}$ needs more consideration. As $\varphi^*\left(w_2^{O(2)}\right) \in H^2(p4m, \mathbb{Z}_2)$, it is completely determined by its action on the 6 topological invariants identified in Eqs. (7) and (8), i.e., $\alpha_i^{p4m}[\omega]$ with $\omega = \tilde{i}(\varphi^*(w_2^{O(2)}))$, for $i = 1, \cdots 6$. To obtain $\alpha_i^{p4m}[\omega]$, consider the six $\mathbb{Z}_2$ subgroups, denoted by $\mathbb{Z}_2^{(i)}$ with $i = 1, \cdots, 6$, generated by $C_2$, $T_1 C_2$, $T_1 T_2 C_2$, $M$, $T_1 M$ and $C_4 M$, respectively. Their embedding into $O(2)$ reads:

$$C_2 \to \begin{pmatrix} -1 & \\ & -1 \end{pmatrix}, \quad T_1 T_2 C_2 \to \begin{pmatrix} 1 & \\ & 1 \end{pmatrix},$$

$$T_1 C_2 \to \begin{pmatrix} 1 & \\ & -1 \end{pmatrix}, \quad M \to \begin{pmatrix} -1 & \\ & 1 \end{pmatrix},$$

$$T_1 M \to \begin{pmatrix} 1 & \\ & 1 \end{pmatrix}, \quad C_4 M \to \begin{pmatrix} & 1 \\ 1 & \end{pmatrix}.$$

The pullback under the embedding $\mathbb{Z}_2^{(i)} \to O(2)$ results in an element in $H^2(\mathbb{Z}_2^{(i)}, \mathbb{Z}_2) = \mathbb{Z}_2$, which is precisely detected by the topological invariant $\alpha_i^{p4m}[\omega]$. Calculating these six pullbacks via the Whitney product formula, we find $\alpha_1^{p4m}[\omega] = -1$, while other topological invariants are $+1$. Hence, we establish that [11]

$$\varphi^*(w_2^{O(2)}) = \lambda_1. \tag{34}$$

Finally, combining Eqs. (31-34) and $\lambda_1 A_{x+y} = 0$, a relation among the cohomology generators in $H^*(p4m, \mathbb{Z}_2)$ (see Appendix E), we establish that Eq. (27) indeed holds, as expected.

We mention that some previous works have performed anomaly-matching for this example, but some of them only did it by restricting both $G_{\mathrm{UV}}$ and $G_{\mathrm{IR}}$ to a few subgroups [8], and some used non-rigorous method [15]. To the best of our knowledge, the analysis above is the first that performs this anomaly-matching via a rigorous method, while keeping track of the full $G_{\mathrm{UV}}$ and $G_{\mathrm{IR}}$. When checking emergibility below, we always maintain such completeness and rigor.

# 4 Deconfined quantum critical point and quantum critical spin liquids

With the formalism developed in the previous sections, we perform an exhaustive search of realizations of $\mathrm{SL}^{(N=5,6,7)}$ that can match certain LSM constraints on lattice spin systems with $p6m \times O(3)^T$ or $p4m \times O(3)^T$ symmetry, if this realization is adjacent to a magnetic state and a non-magnetic state (this means that the $SO(3)$ symmetry acts on some but not all entries of $n$, the $N \times (N-4)$ matrix representing the DOF of $\mathrm{SL}^{(N)}$). This search can be efficiently done using a computer, and the complete results can be found in the attached codes [59] with the help of Appendix J. The numbers of different types of realizations are in Table 1, where each row represents a distinct LSM constraint, or lattice homotopy class, labeled by the IWP that hosts half-integer spins (see Figs. 2 and 3 for the symbols of IWP), and 0 means there is no nontrivial LSM constraint, which applies to systems with integer-spin moments or honeycomb lattice half-integer spin systems. Note that for $p4m$, situations $a$ and $b$ always have the same number of realizations in each case, since they both correspond to square lattice half-integer spin systems and they are related to each other via a redefinition of the $C_4$ center. However, these two situations should still be viewed as distinct, because they cannot be smoothly deformed into each other once the $p4m$ symmetry is specified, which means, in particular, the

---

[11] In practice, to obtain this result, it suffices to only consider the $pmm$ subgroup and $\mathbb{Z}_2$ subgroup generated by $C_4 M$, as argued in Section 2.2.2. Since the embedding of $pmm$ is also in the diagonal form, the calculation is as straightforward.

$C_4$ centers are fixed. The same holds for situations $a$&$c$ and $b$&$c$. In terms of symmetry-enriched quantum criticality, we have found 12 different $p6m \times O(3)^T$ symmetry-enriched DQCP, $105 + 1 = 106$ different $p6m \times O(3)^T$ symmetry-enriched DSL, $705 + 14 = 719$ different $p6m \times O(3)^T$ symmetry-enriched SL$^{(7)}$, 26 different $p4m \times O(3)^T$ symmetry-enriched DQCP, $372 + 1 = 373$ different $p4m \times O(3)^T$ symmetry-enriched DSL, and $3819 - 27 + 29 = 3821$ different $p4m \times O(3)^T$ symmetry-enriched SL$^{(7)}$. The reason for subtracting 27 in the last case is explained at the end of this section. Many of these realizations are unstable, in the sense that they require fine-tuning due to the existence of one or more microscopic symmetry allowed relevant operators (see Appendix K for all stable realizations on various systems).

Below we present some interesting examples. To the best of our knowledge, none of these examples has been discussed before. When we discuss a realization of a SL, we will also comment on its nearby phases, which are often (but not always) some simple ordered states and relatively easy to detect. This provides useful guide for the search of an SL, since if such an ordered state can be found in a material or model, perturbing this ordered state may result in an SL. A smoking-gun signature of the SLs is their large emergent symmetries, which can manifest themselves in a set of singular correlation functions with the same critical exponent. Moreover, for all classical regular magnetic orders [60], i.e., classical magnetic orders in which any broken lattice symmetry can be compensated by a spin operation (see Appendix L for their spin configurations), we identify the numbers of realizations of SLs adjacent to them (see Table 2).

In this section, we focus on realizations where the most relevant spinful excitations have spin-1. In particular, we describe examples of realizations of DQCP as a (pseudo-)critical point, which has a single relevant perturbation allowed by the microscopic symmetries, and stable realizations of DSL, which has no relevant perturbation allowed by the microscopic symmetries. For SL$^{(7)}$, we discuss a realization without symmetry-allowed relevant perturbation, and another example with a single symmetry-allowed relevant perturbation that nevertheless does not change the state. We view both realizations of SL$^{(7)}$ as stable.

## 4.1 DQCP

It is known that there are two types of DQCPs proximate to classical regular magnetic orders [17], both are transitions from an anti-ferromagnetic state to a VBS state, i.e., the columnar VBS for square spin-1/2 systems [61, 62] and the Kekule VBS for honeycomb spin-1/2 systems [63]. Interestingly, we find another realization of DQCP on a honeycomb lattice spin-1/2 system, as a transition between a ferromagnetic state and a staggered VBS state. [12] The symmetries are realized as

$$
\begin{aligned}
T_{1,2} &: n \to n, \\
C_6 &: n \to \begin{pmatrix} I_3 & & \\ & -\frac{1}{2} & \frac{\sqrt{3}}{2} \\ & -\frac{\sqrt{3}}{2} & -\frac{1}{2} \end{pmatrix} n, \\
M &: n \to \begin{pmatrix} I_3 & & \\ & -1 & \\ & & 1 \end{pmatrix} n, \\
O(3)^T &: n \to \begin{pmatrix} O(3)^T & \\ & I_2 \end{pmatrix} n.
\end{aligned}
\tag{35}
$$

---

[12]Due to the fact that a fully polarized ferromagnetic state is always an exact eigenstate of any $SO(3)$ symmetric Hamiltonian, the ferromagnetic state immediately adjacent to this DQCP, which is partially polarized, must be separated from a fully polarized one by a level crossing, i.e., first-order transition.

Table 1: Numbers of realizations for DQCP, DSL and SL$^{(7)}$ in spin systems with a *p*6*m* (upper) or *p*4*m* (lower) lattice symmetry. Two realizations with symmetry actions related by a similarity transformation are considered as a single realization. The columns without (with) subscript "quad" represent realizations where the most relevant spinful excitations, i.e., the *n* modes that transform nontrivially under the $SO(3)$ spin rotational symmetry, carry spin-1 (spin-2). No realization of DQCP has the *n* modes carrying spin-2. The numbers in parenthesis are the numbers of *stable* realizations. Here a stable DQCP means a realization that has a single relevant perturbation allowed by the microscopic symmetry, and a stable DSL, SL$^{(7)}$, DSL$_{\text{quad}}$ and SL$^{(7)}_{\text{quad}}$ means a realization that has no relevant perturbation allowed by the microscopic symmetry. For all columns except SL$^{(7)}_{\text{incom}}$, the *n* modes are at high-symmetry momenta in the Brillouin zone. For SL$^{(7)}$ realized on *p*4*m* symmetric lattices, there are realizations with some *n* modes at incommensurate momenta, and the column SL$^{(7)}_{\text{incom}}$ documents the numbers of families of these realizations, where each family includes infinitely many realizations labeled by a momentum, which continuously interpolate between two realizations in the column SL$^{(7)}$. Two continuous families of realizations may share a common high-symmetry momentum, at which these two realizations turn out to be always distinct, in that symmetries other than translation are implemented distinctly. $(23, 2)$ means that there are 23 families of realizations, such that as long as a given realization is in the "interiors" of the family (i.e., not all *n* modes are at high-symmetry momenta), the only symmetric relevant perturbation is the one that shifts the momenta of *n* modes, and there are 2 other families, such that this is still the case except at two exceptional points in the interior, where there is an additional symmetric relevant perturbation that changes the emergent order. The symmetry actions of the stable realizations are explicitly listed in *ReadMe.nb*.

$$G_s = p6m$$

| spin-1/2 position | DQCP | DSL | SL$^{(7)}$ | DSL$_{\text{quad}}$ | SL$^{(7)}_{\text{quad}}$ |
|---|---|---|---|---|---|
| 0 | 10(2) | 76(1) | 453(0) | 1(1) | 12(2) |
| *a* | 0 | 3(3) | 41(8) | 0 | 0 |
| *c* | 0 | 3(3) | 35(9) | 0 | 0 |
| *a*&*c* | 2(1) | 23(5) | 176(2) | 0 | 2(0) |
| total | 12 (3) | 105 (12) | 705 (19) | 1 (1) | 14 (2) |

$$G_s = p4m$$

| spin-1/2 position | DQCP | DSL | SL$^{(7)}$ | SL$^{(7)}_{\text{incom}}$ | DSL$_{\text{quad}}$ | SL$^{(7)}_{\text{quad}}$ |
|---|---|---|---|---|---|---|
| 0 | 19(0) | 217(0) | 1849(0) | 2(0) | 1(1) | 22(4) |
| *a* | 1(1) | 23(3) | 299(2) | 3(2,1) | 0 | 1(1) |
| *b* | 1(1) | 23(3) | 299(2) | 3(2,1) | 0 | 1(1) |
| *c* | 3(0) | 56(4) | 632(0) | 2(2) | 0 | 3(1) |
| *a*&*b* | 1(1) | 22(0) | 279(0) | 11(11) | 0 | 1(0) |
| *a*&*c* | 0 | 6(6) | 117(6) | 0 | 0 | 0 |
| *b*&*c* | 0 | 6(6) | 117(6) | 0 | 0 | 0 |
| *a*&*b*&*c* | 1(1) | 19(2) | 227(0) | 6(6) | 0 | 1(0) |
| total | 26 (4) | 372 (24) | 3819 (16) | 27 (23,2) | 1 (1) | 29 (7) |

Table 2: Numbers of realizations for DQCP (top), DSL (middle) and SL$^{(7)}$ (bottom) adjacent to some colinear, coplanar and non-coplanar magnetic orders, respectively, of triangular, kagome, honeycomb and square lattice spin-1/2 (or general half-integer-spin) systems (third column) and spin-1 (or general integer-spin) systems (fourth column). The numbers in parenthesis are the numbers of stable realizations (defined in the same way as in Table 1). F stands for Ferromagnetic while AF stands for Anti-ferromagnetic. "1 Incom" means that realizations of SL$^{(7)}$ adjacent to tetrahedral umbrella order on the square lattice spin-1/2 systems belong to a continuous family of realizations, where the non-magnetic components of $n$ can have continuously changing momenta. See Appendix L for the spin configurations of these magnetic orders, and the attached code *ReadMe.nb* for the explicit symmetry actions.

| Lattice | Colinear order | spin-1/2 | spin-1 |
|---|---|---|---|
| Triangular | F | 0 | 2(1) |
| Kagome | F | 0 | 2(1) |
| Honeycomb | F | 2(1) | 2(1) |
| | AF | 2(1) | 2(1) |
| Square | F | 0 | 2(0) |
| | AF (Neel) | 1(1) | 2(0) |

| Lattice | Coplanar order | spin-1/2 | spin-1 |
|---|---|---|---|
| Triangular | 120° | 1(1) | 3(0) |
| Kagome | $q = 0$ | 1(1) | 2(0) |
| | $\sqrt{3} \times \sqrt{3}$ | 0 | 3(0) |
| Honeycomb | V | 3(0) | 3(0) |
| Square | V | 1(0) | 3(0) |
| | Orthogonal | 1(0) | 1(0) |

| Lattice | Non-Coplanar order | spin-1/2 | spin-1 |
|---|---|---|---|
| Triangular | Tetrahedral | 8(4) | 2(0) |
| | F umbrella | 1(0) | 4(0) |
| Kagome | Octahedral | 0 | 2(0) |
| | Cuboc1 | 3(2) | 1(0) |
| | Cuboc2 | 4(3) | 1(0) |
| | $q = 0$ umbrella | 2(1) | 3(0) |
| | $\sqrt{3} \times \sqrt{3}$ umbrella | 1(1) | 4(0) |
| Honeycomb | Tetrahedral | 2(0) | 2(0) |
| | Cubic | 1(0) | 1(0) |
| Square | Tetrahedral umbrella | 1 Incom | 2(0) |
| | F umbrella | 2(0) | 2(0) |

The components of $n$ can be identified with microscopic operators that transform identically under the above symmetries. Denote the microscopic spin-1/2 operator on the $A$ and $B$ sublattices as $\mathbf{S}_A(\mathbf{r}) \equiv \mathbf{S}\left(\mathbf{r} + \frac{2T_1 + T_2}{3}\right)$ and $\mathbf{S}_B(\mathbf{r}) \equiv \mathbf{S}\left(\mathbf{r} + \frac{T_1 - T_2}{3}\right)$, respectively, where $\mathbf{r}$ is the position of the $C_6$ center of each unit cell, and $\mathbf{T}_{1,2}$ is the translation vector of $T_{1,2}$. Then

$S_{A,i}(\boldsymbol{r}) = S_{B,i}(\boldsymbol{r}) \sim n_i$ for $i = 1,2,3$. Denote the dimer operators as $D_x(\boldsymbol{r}) \equiv S(\boldsymbol{r} + \frac{-T_1+T_2}{3}) \cdot S\left(\boldsymbol{r} + \frac{T_1+2T_2}{3}\right)$, $D_y(\boldsymbol{r}) \equiv S\left(\boldsymbol{r} + \frac{T_1+2T_2}{3}\right) \cdot S(\boldsymbol{r} + \frac{2T_1+T_2}{3})$, and $D_z(\boldsymbol{r}) \equiv S\left(\boldsymbol{r} + \frac{2T_1+T_2}{3}\right) \cdot S\left(\boldsymbol{r} + \frac{T_1-T_2}{3}\right)$. Then $D_x(\boldsymbol{r}) + e^{i\frac{2\pi}{3}} D_y(\boldsymbol{r}) + e^{i\frac{4\pi}{3}} D_z(\boldsymbol{r}) \sim e^{-i\frac{5\pi}{6}}(n_4 - in_5)$. So $n_{1,2,3}$ and $n_{4,5}$ can be identified as the order parameters of a ferromagnet and a stacked VBS, respectively. For examples below, one can perform similar analysis to identify components of $n$ with microscopic operators, but we will not explicitly showcase them.

Since this ferromagnetic DQCP is the simplest example of new states discovered using our approach, it will be reassuring to also have a traditional parton-based construction [64]. Indeed this DQCP can be constructed using Schwinger bosons $S = \frac{1}{2}b_\alpha^\dagger \boldsymbol{\sigma}_{\alpha\beta} b_\beta$, where the bosonic spinons $b_\alpha$ couple to a dynamicsl $U(1)$ gauge field $a_\mu$. To realize the staggered VBS, we put the Schwinger bosons into the "featureless Mott insulator" discussed in Ref. [34] – effectively this state is constructed by putting a spin-singlet, gauge-charge $Q = 2$ spinon "Cooper pair" at each $C_6$ center. When coupled to the dynamical $U(1)$ gauge field, the monopole operator acquires nontrivial lattice symmetry quantum numbers due to the charged insulating background. For example, the gauge charge $Q = 2$ at each $C_6$ rotation center gives the monopole a $C_6$ angular momentum $e^{i2\pi/3}$ from the Aharanov-Bohm effect. Other lattice symmetry quantum-numbers can be analyzed in a similar fashion, following methods develped in Ref. [57]. It turns out that the monopole carries exactly the symmetry quantum numbers of the staggered VBS. At low energies the monopole will spontaneously condense and confine the gauge theory, resulting in the staggered VBS phase. To access the magnetically ordered phase, we drive the spinons $b_\alpha$ through an insulator-superfluid transition and Higgs the $U(1)$ gauge field. The fact that $b_\alpha$ do not carry any nontrival projective representation in this construction means that they can be condensed without breaking any lattice symmetry, which means that the magnetically ordered phase obtained this way is a ferromagnet. The effective field theory at the phase transition is the standard (non-compact) CP$^1$ theory for DQCP [16], described by an $SU(2)$-fundamental complex Wilson-Fisher boson coupled to a dynamical $U(1)$ gauge field $a_\mu$.

We remark that, compared to the standard DQCP realization where the magnetic side is anti-ferromagnetic, in this realization there is one more perturbation that is likely irrelevant at the transition, but relevant in the ferromagnetic phase and responsible for making the dispersion of the magnon quadratic. In the CP$^1$ formulation of the DQCP with Schwinger bosons $b$ [16,17], this operator is $(ib^\dagger \boldsymbol{\sigma} \partial_t b) \cdot (b^\dagger \boldsymbol{\sigma} b)$. In the CP$^N$ generalization of this theory, this operator is indeed dangerously irrelevant in the large-$N$ limit.

The simple nature of the magnetic and VBS phases here suggests that this DQCP may be realizable in relatively simple spin models. It will be interesting to find a sign-problem-free lattice model and simulate this transition with the quantum Monte Carlo approach.

## 4.2 DSL

DSLs have been constructed for various lattices using the parton construction. There are two widely studied DSLs: one is on the kagome lattice spin-1/2 system proximate to the $\boldsymbol{q} = 0$ coplanar magnetic order [22,65–68]; the other is on the triangular spin-1/2 lattice proximate to the 120° coplanar order [57,58,69–73]. On the other hand, the previously constructed DSLs on the honeycomb and square lattices are unstable due to the presence of $G_{\text{UV}}$-symmetric monopole (i.e., the $n$ modes) [57,58,74,75]. Interestingly, we find new stable DSLs on square and honeycomb lattices. Our complete classification also shows that there is no DSL proximate to the $\sqrt{3} \times \sqrt{3}$ coplanar order on the kagome spin-1/2 system.

For the honeycomb lattice spin-1/2 system, the symmetries act as:

$$T_{1,2} : n \rightarrow \begin{pmatrix} I_3 & & \\ & -\frac{1}{2} & \frac{\sqrt{3}}{2} \\ & -\frac{\sqrt{3}}{2} & -\frac{1}{2} \\ & & & 1 \end{pmatrix} n \,,$$

$$C_6 : n \rightarrow \begin{pmatrix} I_3 & & \\ & 1 & \\ & & -1 \\ & & & -1 \end{pmatrix} n \begin{pmatrix} \frac{1}{2} & -\frac{\sqrt{3}}{2} \\ \frac{\sqrt{3}}{2} & \frac{1}{2} \end{pmatrix} \,,$$

$$M : n \rightarrow \begin{pmatrix} I_3 & & \\ & -1 & \\ & & -1 \\ & & & 1 \end{pmatrix} n \begin{pmatrix} -1 & \\ & 1 \end{pmatrix} \,,$$

$$O(3)^T : n \rightarrow \begin{pmatrix} O(3)^T & \\ & I_3 \end{pmatrix} n \,.$$

(36)

The magnetic order adjacent to this DSL is a regular magnetic order, i.e., a magnetic order in which any broken lattice symmetry can be compensated by a spin operation [60]. However, the magnetic order here appears missing in the classification in Ref. [60], which is possibly because all magnetic orders in Ref. [60] are assumed to be realizable by product states. It is known that some SRE states in a honeycomb lattice spin-1/2 system cannot be realized by product states, so we do not make this assumption [34–37]. This DSL should also be emergible in a triangular or kagome lattice integer-spin system. In these cases, the adjacent magnetic orders are also regular but not realizable by product states. See Ref. [76] for a recent study of these *entanglement-enabled symmetry-breaking orders*.

For the square lattice spin-1/2 system, the symmetries act as:

$$T_1 : n \rightarrow \begin{pmatrix} I_3 & & \\ & -1 & \\ & & 1 \\ & & & -1 \end{pmatrix} n \begin{pmatrix} -1 & \\ & -1 \end{pmatrix} \,,$$

$$T_2 : n \rightarrow \begin{pmatrix} I_3 & & \\ & 1 & \\ & & -1 \\ & & & -1 \end{pmatrix} n \begin{pmatrix} -1 & \\ & -1 \end{pmatrix} \,,$$

$$C_4 : n \rightarrow \begin{pmatrix} I_3 & & \\ & & 1 \\ & -1 & \\ & & & 1 \end{pmatrix} n \begin{pmatrix} -1 & \\ & -1 \end{pmatrix} \,,$$

(37)

$$M : n \rightarrow \begin{pmatrix} I_3 & & \\ & 1 & \\ & & -1 \\ & & & -1 \end{pmatrix} n \begin{pmatrix} -1 & \\ & 1 \end{pmatrix} \,,$$

$$O(3)^T : n \rightarrow \begin{pmatrix} O(3)^T & \\ & I_3 \end{pmatrix} n \,.$$

The magnetic order adjacent to this DSL is also an entanglement-enabled regular magnetic order.

One interesting aspect of these realizations is that all perturbations proportional to the entries of $n$ are forbidden by symmetries, and the lack of this property is the reason why the previous constructions on these systems are unstable [57, 58, 74, 75]. This property implies that these realizations cannot be obtained as a descendent state of an $SU(2)$ DSL [57], which is described by 2 flavors of Dirac fermions coupled to an emergent $SU(2)$ gauge field (including the 2 colors, there are in total 4 Dirac fermions). To see it, note that the emergent symmetry of the $SU(2)$ DSL is just $O(5)^T$, so if a $U(1)$ DSL is its descendent, all microscopic symmetries will be embedded into the $O(5)^T$ symmetry, which necessarily leaves some components of $n$ symmetry-allowed. The previous constructions of the $U(1)$ DSL on a square and honeycomb lattice spin-1/2 systems are indeed descendents of an $SU(2)$ DSL, and it would be interesting to find a parton construction of our new realizations.

### 4.3 SL$^{(7)}$

Two realizations of the conjectured non-Lagrangian state SL$^{(7)}$ are given in Ref. [15]. Here we describe some other interesting realizations.

On a kagome lattice spin-1/2 system, there is a realization with the following symmetry actions:

$$
\begin{aligned}
T_1 : n &\to \begin{pmatrix} I_3 & & & & \\ & -\frac{1}{2} & \frac{\sqrt{3}}{2} & & \\ & -\frac{\sqrt{3}}{2} & -\frac{1}{2} & & \\ & & & 1 & \\ & & & & 1 \end{pmatrix} n \begin{pmatrix} 1 & & \\ & -1 & \\ & & -1 \end{pmatrix}, \\[6pt]
T_2 : n &\to \begin{pmatrix} I_3 & & & & \\ & -\frac{1}{2} & \frac{\sqrt{3}}{2} & & \\ & -\frac{\sqrt{3}}{2} & -\frac{1}{2} & & \\ & & & 1 & \\ & & & & 1 \end{pmatrix} n \begin{pmatrix} -1 & & \\ & 1 & \\ & & -1 \end{pmatrix}, \\[6pt]
C_6 : n &\to \begin{pmatrix} I_3 & & & & \\ & 1 & & & \\ & & -1 & & \\ & & & -1 & \\ & & & & -1 \end{pmatrix} n \begin{pmatrix} & & -1 \\ 1 & & \\ & 1 & \end{pmatrix}, \\[6pt]
M : n &\to \begin{pmatrix} I_3 & & & & \\ & -1 & & & \\ & & -1 & & \\ & & & 1 & \\ & & & & 1 \end{pmatrix} n \begin{pmatrix} & & -1 \\ -1 & & \\ & 1 & \end{pmatrix}, \\[6pt]
O(3)^T : n &\to \begin{pmatrix} O(3)^T & \\ & I_4 \end{pmatrix} n .
\end{aligned}
\tag{38}
$$

The magnetic order adjacent to this SL$^{(7)}$ is the cuboc1 order, a good classical ground state for Heisenberg like models [60], and was found in a $J_1$-$J_2$-$J_3$ model [77].

We also note that, in contrast to the DQCP and DSL, where all realizations are proximate to some commensurate states, i.e., $n$ have commensurate momenta in those realizations, SL$^{(7)}$ can have realizations with $n$ at incommensurate momenta. For example, on a square lattice

spin-1/2 system, there is a family of realization with the following symmetry actions:

$$T_1 : n \rightarrow \begin{pmatrix} I_3 & & \\ & \exp(-i\sigma_y k) & \\ & & -I_2 \end{pmatrix} n \begin{pmatrix} -1 & & \\ & 1 & \\ & & -1 \end{pmatrix},$$

$$T_2 : n \rightarrow \begin{pmatrix} I_3 & & \\ & -I_2 & \\ & & \exp(i\sigma_y k) \end{pmatrix} n \begin{pmatrix} 1 & & \\ & -1 & \\ & & -1 \end{pmatrix},$$

$$C_4 : n \rightarrow \begin{pmatrix} I_3 & & & & \\ & & 1 & & \\ & & & 1 & \\ & & 1 & & \\ & 1 & & & \end{pmatrix} n \begin{pmatrix} & 1 & \\ 1 & & \\ & & 1 \end{pmatrix},$$

$$M : n \rightarrow \begin{pmatrix} I_3 & & & & \\ & 1 & & & \\ & 1 & & & \\ & & & 1 & \\ & & & & 1 \end{pmatrix} n,$$

$$O(3)^T : n \rightarrow \begin{pmatrix} O(3)^T & \\ & I_4 \end{pmatrix} n,$$

$$(39)$$

where $k \in [-\pi, \pi)$ is a generic momentum. The magnetic order adjacent to this realization is the tetrahedral umbrella order [60].

The above represents an infinite family of realizations, where the momenta of some $n$ modes continuously change in the Brillouin zone. Among the relevant operators discussed in Sec. 3.1, there is only a single one allowed by the microscopic symmetries in this family of realizations, i.e., the $SO(7)$ current $\sim (n_{4i}\partial_x n_{5i} + n_{6i}\partial_y n_{7i})$. We believe all these realizations can actually be smoothly connected without encountering a phase transition, so they all represent the same symmetry-enriched SL. This also imposes some constraints on the low-energy dynamics of $SL^{(7)}$, i.e., although the above $SO(7)$ conserved current is relevant, it can merely shift the "zero momentum", but not really change the state (see Appendix H for more discussions).

# 5 Quantum critical spin-quadrupolar liquids

Besides the previous case, we also find realizations where the most relevant spinful excitations carry spin-2. We dub these states quantum critical spin-quadrupolar liquids.

We have identified an interesting realization of the DSL as a quantum critical spin-quadrupolar liquid. This realization can actually be realized on any lattice that has no nontrivial LSM constraint, including spin-1 systems on any lattice, spin-1/2 systems on honeycomb lattice, etc. If the lattice has a $p6m$ or $p4m$ symmetry, this is the only spin-quadrupolar realization of DSL. The lattice translation and rotation symmetries leave $n$ invariant, and $SO(3)$, time reversal $\mathcal{T}$ and lattice reflection $M$ (if any) act as

$$SO(3) : n \rightarrow \begin{pmatrix} \varphi_5(SO(3)) & \\ & 1 \end{pmatrix} n,$$

$$\mathcal{T} : n \rightarrow \begin{pmatrix} I_5 & \\ & -1 \end{pmatrix} n,$$

$$M : n \rightarrow \begin{pmatrix} I_5 & \\ & -1 \end{pmatrix} n,$$

$$(40)$$

where $\varphi_5(SO(3))$ represents the spin-2 representation of $SO(3)$. For this realization, if $SO(3) \times \mathbb{Z}_2^T$ and an arbitrary lattice rotational symmetry are preserved, all relevant perturbations listed in Sec. 3.1 are forbidden. Even if only $SO(3) \times \mathbb{Z}_2^T$ is preserved while all lattice symmetries are broken, the only symmetry-allowed relevant perturbations are the spatial components of the conserved current associated with the $SO(2)$ emergent symmetry, which are expected to retain the emergent order (see Appendix H). So this realization represents a rare example of quantum critical liquid that requires only internal symmetry (but not lattice symmetry) to be stable. The magnetic state adjacent to this DSL is a spin-quadrupolar order where the Goldstond modes are at the $\Gamma$ point of the Brillouin zone. For the non-magnetic state, it is possible to have $\langle n_{61} \rangle \neq 0$ while all other entries of $n$ have zero expectation value. This is a spin-quadrupolar realization of the DQCP, and the only possible relevant perturbation is an $SO(5)$ singlet that breaks time reversal, which may drive the system to forming a chiral spin liquid.

Usually, a DSL is constructed by fermionic partons that have a non-interacting mean field with 4 Dirac cones, which are coupled to an emergent $U(1)$ gauge field. Below we show that the realization above cannot be constructed in this way, which may be its most interesting property.

To see it, let us consider how the Dirac fermions transform under the $SO(3)$ spin rotational symmetry. Denote the Dirac fermion operator as $\psi_i$ with $i = 1, \cdots 4$, which transforms in the fundamental representation of the emergent $SU(4)$ flavor symmetry. It is known that $\bar{\psi}_i \psi_j - \frac{1}{4} \bar{\psi} \psi \delta_{ij}$, which is the fermion mass in the $SU(4)$ adjoint representation, is identified with $A_{i_1 i_2} \epsilon_{j_1 j_2} n_{i_1 j_1} n_{i_2 j_2}$, with $A$ and $\epsilon$ an anti-symmetric $6 \times 6$ and $2 \times 2$ real matrix, respectively [15,57,58]. Because under $SO(3)$ spin rotational symmetry, part of the latter operators transforms in the spin-3 representation, the former operator must also contain components in the spin-3 representation, which implies that the Dirac fermions must transform in the spin-3/2 representation of the $SO(3)$ symmetry, i.e., all 4 flavors of Dirac fermions together form this spin-3/2 object.

Now suppose this state can be realized by a *non-interacting* parton mean field with 4 Dirac cones (coupled to an emergent $U(1)$ gauge field), the mean-field Hamiltonian of the partons must have an *on-site* $U(4)$ symmetry. In the presence of this $U(4)$ and time reversal symmetries, there must be at least 8 Dirac cones in the mean field. To see it, it suffices to consider one of the 4 flavors, whose mean field has on-site $U(1)$ and time reversal symmetries. To avoid the parity anomaly, there are necessarily an even number of Dirac cones. So taken 4 flavors together, there are at least 8 Dirac cones, which contradicts our starting point, i.e., the mean field has only 4 Dirac cones.

The above argument shows that this realization is beyond the simplest parton mean fields. However, it is still possible to realize it if the partons are strongly interacting (even without considering their coupling to the emergent gauge field), so that at low energies 4 flavors of Dirac fermions emerge out of the strong interactions. This might be theoretically described, say, by a further parton decomposition of the partons themselves. This is possible because if besides time reversal the on-site symmetry is only $SO(3)$ but not $U(4)$, there is no anomaly, and hence no contradiction with having 4 Dirac cones while realizing these symmetries in an on-site fashion.[13] It is an interesting challenge to find such a concrete construction in the future. This situation is similar to the Standard Model in particle physics: the Standard Model cannot be realized through lattice free fermions coupled to gauge fields due to fermion doubling, but it is believed to be realizable using strongly interacting fermions since all the quantum anomalies vanish [78–85].

Finally, we give an interesting realization of $SL^{(7)}$ as a quantum critical spin-quadrupolar

---

[13]One can in principle also try to implement some of these symmetries on the partons in a non-on-site fashion, but then it is challenging to have all on-site symmetries acting on the physical operators in an on-site fashion.

liquid, on a honeycomb lattice half-integer-spin system or any integer-spin system with *p6m* symmetry. The symmetries act as follows:

$$
\begin{aligned}
SO(3) : n &\to \begin{pmatrix} \varphi_5(SO(3)) & \\ & I_2 \end{pmatrix} n, \\
\mathcal{T} : n &\to \begin{pmatrix} I_5 & & \\ & -1 & \\ & & 1 \end{pmatrix} n, \\
T_1 : n &\to n \begin{pmatrix} -\frac{1}{2} & -\frac{\sqrt{3}}{2} & \\ \frac{\sqrt{3}}{2} & -\frac{1}{2} & \\ & & 1 \end{pmatrix}, \\
T_2 : n &\to n \begin{pmatrix} -\frac{1}{2} & -\frac{\sqrt{3}}{2} & \\ \frac{\sqrt{3}}{2} & -\frac{1}{2} & \\ & & 1 \end{pmatrix}, \\
C_6 : n &\to \begin{pmatrix} I_5 & & \\ & 1 & \\ & & -1 \end{pmatrix} n \begin{pmatrix} 1 & & \\ & -1 & \\ & & 1 \end{pmatrix}, \\
M : n &\to \begin{pmatrix} I_5 & & \\ & -1 & \\ & & 1 \end{pmatrix} n.
\end{aligned}
\tag{41}
$$

The nearby phases of this $SL^{(7)}$ can be very interesting. It is possible to have $\langle n_{73} \rangle \neq 0$ while all other entries of $n$ have zero expectation value. This results in the spin-quadrupolar DSL (see Eq. (40)), except that the $C_2 \equiv C_6^3$ symmetry is broken, while all other symmetries (including $C_3 \equiv C_6^2$) are intact. We can also view the above realization of $SL^{(7)}$ as an unnecessary phase transition in a $p31m \times O(3)^T$ symmetric DSL phase. This DSL is still stable, but the $n$ modes are at the $\pm K$ points. It is also possible to have $\langle n_{13} \rangle \neq 0$ while all other entries of $n$ have zero expectation value, where our choice of basis is such that this condensation pattern breaks the $SO(3)$ symmetry to $U(1)$. This results in a stable $p6m \times \mathbb{Z}_2^T \times U(1)$ symmetric DSL that simultaneously has a spin-quadrupolar order. Again, the above realization of $SL^{(7)}$ can be regarded as an unnecessary phase transition in a $p6m \times \mathbb{Z}_2^T \times U(1)$ symmetric DSL phase.[14]

# 6 Stability under symmetry breaking

In this section we demonstrate how to use the SEP to analyse the stability of these realizations under symmetry-breaking perturbations. As a concrete example, we focus on a realization of DSL on a triangular lattice spin-1/2 system that is perturbed by spin-orbit coupling (SOC), which may be relevant to NaYbO$_2$. In Appendix M, we give a few other examples of such analysis, which may be relevant to twisted bilayer WSe$_2$, a recently realized quantum simulator for triangular lattice spin-1/2 models [86–88].

---

[14]Strictly speaking, the DSL states on the two sides of this $SL^{(7)}$ are slightly different, since they have different quantum anomalies if the entire emergent symmetry is taken into account. In Ref. [15], these two DSLs are denoted by $SL^{(6,1)}$ and $SL^{(6,-1)}$, respectively. However, if we only look at the remaining exact symmetries, there is no difference between them. Furthermore, even if the entire emergent symmetry is considered, all correlation functions in these two cases are simply related by a unitary transformation (which is not a symmetry of the DSL), so practically the DSL in the two sides can be viewed as in the same phase [15]. The same is true for the $p31m \times O(3)^T$ symmetric DSL.

Without considering the SOC, the triangular lattice spin-1/2 system has a $p6m \times O(3)^T$ symmetry, which acts on this DSL as:

$$T_1 : n \to \begin{pmatrix} I_3 & & & \\ & 1 & & \\ & & -1 & \\ & & & -1 \end{pmatrix} n \begin{pmatrix} -\frac{1}{2} & -\frac{\sqrt{3}}{2} \\ \frac{\sqrt{3}}{2} & -\frac{1}{2} \end{pmatrix},$$

$$T_2 : n \to \begin{pmatrix} I_3 & & & \\ & -1 & & \\ & & 1 & \\ & & & -1 \end{pmatrix} n \begin{pmatrix} -\frac{1}{2} & -\frac{\sqrt{3}}{2} \\ \frac{\sqrt{3}}{2} & -\frac{1}{2} \end{pmatrix},$$

$$C_6 : n \to \begin{pmatrix} I_3 & & & \\ & & 1 & \\ & & & 1 \\ & -1 & & \end{pmatrix} n \begin{pmatrix} 1 & \\ & -1 \end{pmatrix},$$  \hfill (42)

$$M : n \to \begin{pmatrix} I_3 & & & \\ & & -1 & \\ & -1 & & \\ & & & 1 \end{pmatrix} n,$$

$$O(3)^T : n \to \begin{pmatrix} O(3)^T & \\ & I_3 \end{pmatrix} n.$$

This realization was discussed in Refs. [15, 57, 58, 69–73], and it is shown in Appendix I.2 that the anomaly-matching condition Eq. (19) is indeed satisfied. From this symmetry action, it is straightforward to check that all the relevant operators listed in Sec. 3.1 are symmetry-forbidden, so this realization is expected to be stable if the full $p6m \times O(3)^T$ symmetry is preserved.

Recently, a quantum disordered liquid was reported in $NaYbO_2$ [23–27] (similar phenomena were reported in related materials including $NaYbS_2$ and $NaYbSe_2$ [28–32]). In particular, there is evidence that this state is gapless with a low-temperature specific heat scaling as temperature squared, and that it has a critical mode located at the $\pm K$ points in the Brillouin zone, which are consistent with the above DSL realization. So it was proposed that a DSL may be realized in $NaYbO_2$. However, due to SOC, the symmetry of $NaYbO_2$ is smaller than $p6m \times O(3)^T$, and an important question is whether there is symmetry-allowed relevant perturbation that would destabilize a DSL in $NaYbO_2$.

$NaYbO_2$ is a layered material with space group symmetry $R\bar{3}m$. Restricted to a single layer, the remaining symmetries are [89]

$$T_{1,2}, \ C_6^* \equiv S_3 \cdot C_6, \ M^* \equiv S_M \cdot M, \ \mathcal{T}, \hfill (43)$$

where $S_3$ and $S_M$ act in the spin space:

$$S_3 : \begin{pmatrix} S_x \\ S_y \\ S_z \end{pmatrix} \to \begin{pmatrix} -\frac{1}{2} & \frac{\sqrt{3}}{2} & \\ -\frac{\sqrt{3}}{2} & -\frac{1}{2} & \\ & & 1 \end{pmatrix} \begin{pmatrix} S_x \\ S_y \\ S_z \end{pmatrix},$$  \hfill (44)

$$S_M : \begin{pmatrix} S_x \\ S_y \\ S_z \end{pmatrix} \to \begin{pmatrix} -\frac{1}{2} & \frac{\sqrt{3}}{2} & \\ \frac{\sqrt{3}}{2} & \frac{1}{2} & \\ & & -1 \end{pmatrix} \begin{pmatrix} S_x \\ S_y \\ S_z \end{pmatrix},$$

with $S_{x,y,z}$ the microscopic (effective) spin-1/2 operators.

Using Eq. (42), it is straightforward to extract the actions of the remaining symmetry Eq. (43), from which one can see that all relevant operators in Sec. 3.1 are still symmetry-forbidden. This means that the DSL can be stably realized on NaYbO$_2$. Of course, whether NaYbO$_2$ actually realizes a DSL requires futher investigation.

# 7 Discussion

In this paper we have achieved two major goals: i) deriving the topological partition functions corresponding to the LSM constraints in a large class of systems relevant to the study of quantum magnetism, and ii) studying the emergibility of various Stiefel liquids (SLs) in lattice spin systems. The former has wide applicability and can be applied to constrain the emergibility of any state on the relevant lattice spin systems, and the latter paves the way to further understand the elusive strongly-interacting quantum critical states.

The SLs discussed in this paper are the simplest members of their entire family, i.e., SL$^{(N)}$ with $N = 5, 6, 7$. In fact, our results have indications on the emergibility of more complicated SLs, i.e., SL$^{(N,m)}$ with $N = 5, 6, 7$ and $m > 1$ [15]. Just like SL$^{(N)}$, the degrees of freedom of SL$^{(N,m>1)}$ are also characterized by an $N \times (N-4)$ matrix with orthonormal columns, and this state can be obtained by coupling together $m$ copies of SL$^{(N)}$ in a specifc way. Interestingly, SL$^{(5,m)}$ can be viewed as a $USp(2m)$ gauge theory with 2 flavors of gapless Dirac fermions, and SL$^{(6,m)}$ can be viewed as 4 flavors of gapless Dirac fermions coupled to a $U(m)$ gauge field. It is expected that for a given $N$, there is an $m_c(N)$ such that SL$^{(N,m)}$ is a CFT if and only if $m < m_c(N)$, and $m_c(N \geqslant 6) > 1$. To discuss the emergibility of SL$^{(N,m>1)}$ in lattice spin systems with $p6m \times O(3)^T$ or $p4m \times O(3)^T$ symmetry, we can think that all SL$^{(N,m)}$ have the same IR anomaly as SL$^{(N)}$ if $m$ is odd, and all SL$^{(N,m)}$ have no IR anomaly if $m$ is even. Furthermore, because the degrees of freedom of SL$^{(N,m>1)}$ are represented in the same way as those in SL$^{(N)}$, a given symmetry embedding pattern for SL$^{(N)}$ is also a valid one for SL$^{(N,m>1)}$, and vice versa. So our results imply: i) For SL$^{(N,m>1)}$ with an odd $m$ and a given symmetry embedding pattern, it can emerge in a lattice spin system if and only if SL$^{(N)}$ with the same symmetry embedding pattern can emerge in this system. ii) SL$^{(N,m)}$ with an even $m$ can only emerge in lattice spin systems with a vanishing LSM anomaly, and in such a system any symmetry embedding pattern defined by Eq. (18) satisfies the emergibility condition Eq. (19), and is expected to describe a physical realization of SL$^{(N,m)}$ in this system.

We remark that our philosophy to study the emergibility of a quantum phase or phase transition is different from the conventional one. Our strategy is based on anomaly-matching, while the conventional one is based on explicit constructions of this phase or phase transition, often in terms of a mean field (including parton gauge mean field) or a wave function. We believe that the anomaly-based strategy captures the intrinsic essence of emergibility. After all, any mean-field construction is also a way of doing anomaly-matching in disguise, and such a construction by itself cannot rigorously prove the emergibility. On the other hand, although a microscopic wave function can guarantee the emergibility, it is generically difficult to read off the universal physics encoded in a wave function, and there is no guarantee that a proposed wave function indeed describes the quantum phase or phase transition of interest - in fact, in general there is no guarantee that such a wave function could be realized as the ground state of any local Hamiltonian. So the significance of this work is not only reflected by the specific results, but also by the fact that it demonstrates the feasibility of the anomaly-based framework of emergibility, and the fact that this framework can yield interesting results not envisioned before.

This anomaly-based framework of emergibility is established for lattice spin systems in this paper. An interesting and important open problem is to generalize the topological character-

izations of LSM constraints to other systems, and apply the results to study the emergibility of other quantum phases and phase transitions. Systems of particular relevance are those in (3+1)-d, those with spin-orbit coupling, those with a filling constraint due to a $U(1)$ symmetry, those with long-range interactions, those with a constrained Hilbert space, fermionic systems, etc. We leave these for future work.

We have assumed that the hypothesis of emergibility is a necessary and sufficient condition of emergibility. As mentioned before, its necessity has been established, while the sufficiency is a reasonable conjecture. It is important to further justify or disprove (the sufficiency of) this hypothesis. If it is disproved, it will be extremely interesting and valuable to identify a correct necessary and sufficient condition of emergibility.

The realizations of symmetry-enriched SLs discussed here give useful guidance for the search of these states in real materials and models. Because the ordering patterns of the nearby phases of the SLs can be read off from the implementations of the microsopic symmetries, a practical strategy is to identify materials and models that host these ordered states, and to explore the vicinity of the phase diagram in order to find SLs. A smoking-gun signature of the SLs is their large emergent symmetries, which can manifest themselves in a set of singular correlation functions with the same critical exponent.

Our results rule out many realizations of symmetry-enriched SLs because their IR anomalies do not match with the LSM anomalies. However, variants of these realizations are still possible if there is a sector of anomalous topological order in the system (in additional to the gapless degrees of freedom from the SLs), whose anomaly precisely compensates the mismatch between the IR anomaly of the SL and the LSM anomaly. Although it may be unnatural in a realistic material or model, this is a valid theoretical possibility. It may be interesting to study such realizations in the future.

Finally, we further comment on our characterization of the symmetry enrichment pattern of a quantum critical state with a given emergent order. Our characterization is based on how the microscopic symmetries act on the the local, low-energy degrees of freedom. As reviewed in Introduction, in the literature the symmetry enrichment pattern of an emergent gauge theory is usually specified by how the symmetries act on various "fractionalized degrees of freedom", represented by gauge non-invariant operators [64]. This usual approach is appropriate for emergent gauge theories with well-defined fractionalized quasi-particles, where symmetry fractionalization on these fractionalized quasi-particles can be sharply defined based on symmetry localization [90]. However, the quantum critical states discussed here are not expected to have any well-defined quasiparticle, and all degrees of freedom are strongly coupled, which makes the notion of symmetry localization ill-defined. So it is more appropriate to directly characterize these states using symmetry actions on local operators. More formally, the former type of theories have emergent higher-form symmetries, and symmetry actions on fractionalized quasi-particles can be viewed as the interplay between the ordinary symmetries and higher-form symmetries, captured by, e.g., topological terms involving both types of symmetries (e.g., see Ref. [47]). The critical states discussed here are believed to have no emergent higher-form symmetry, and no such topological term exists. So it is appropriate to directly discuss the symmetry actions on local operators.

On the other hand, we also note that even for a quantum critical state with a given emergent order and given symmetry actions on local, low-energy degrees of freedom, there may be multiple different symmetry-enriched quantum critical states that are distinguished by the symmetry actions on some *non-local* and/or *gapped* degrees of freedom, which may manifest themselves by distinct boundary critical behavior [91–98]. A detailed study of this phenomenon is left for future work.

# Acknowledgements

We thank Chenjie Wang for helpful discussion. WY acknowledges supports from the Natural Sciences and Engineering Research Council of Canada(NSERC) through Discovery Grants. Research at Perimeter Institute is supported in part by the Government of Canada through the Department of Innovation, Science and Industry Canada and by the Province of Ontario through the Ministry of Colleges and Universities.

# A  Review of mathematical background

In this appendix, we briefly review various mathematical concepts used in this paper. We also define some new concepts that will be useful in the paper.

## A.1  Group cohomology

In this sub-appendix, we provide a brief review of the fundamentals of group cohomology. See Refs. [39, 99, 100] for more details.

Given a (discrete) group $G$, let $X$ be an Abelian group equipped with a $G$ action $\rho : G \times X \to X$, which is compatible with group multiplication, i.e., for any $g, h \in G$, $e$ the identity element in $G$ and $a, b \in X$, we have

$$\text{Identity of Group Action}: \rho_e(a) = a\,,$$
$$\text{Compatibility of Group Action}: \rho_g(\rho_h(a)) = \rho_{gh}(a)\,, \qquad (45)$$
$$\text{Compatibility of Module}: \rho_g(ab) = \rho_g(a)\rho_g(b)\,.$$

We leave the group multiplication symbols implicit in the above. Such an Abelian group $X$ with $G$ action $\rho$ is called a $G$-module, denoted by $X_\rho$. In this paper, we will mainly consider three different cases of $X$, i.e., $\mathbb{Z}_2$, $U(1)$ and $\mathbb{Z}$. In particular, when $X = \mathbb{Z}_2$, the action $\rho_g$ is always trivial for any $g \in G$. When $X = U(1)$ ($X = \mathbb{Z}$), the action $\rho_g$ is either trivial or complex conjugation (multiplication by $-1$), i.e., a $\mathbb{Z}_2$ action. Therefore, $\rho$ can be defined by a homomorphism $\tilde{\rho} : G \to \mathbb{Z}_2$, and whether $\tilde{\rho}(g)$ equals $+1$ or $-1$ determines whether the action of $\rho_g$ on $U(1)$ and $\mathbb{Z}$ is trivial or non-trivial.

Let $\omega(g_1, \ldots, g_n) \in X$ be a function of $n$ group elements with $g_i \in G$ for $i = 1, \ldots, n$. Such a function is called an $n$-cochain, and the set of all $n$-cochains is denoted by $C^n(G, X_\rho)$. They naturally form an Abelian group under multiplication,

$$(\omega \cdot \omega')(g_1, \ldots, g_n) = \omega(g_1, \ldots, g_n)\omega'(g_1, \ldots, g_n)\,, \qquad (46)$$

and the identity element is the trivial cochain $\omega(g_1, \ldots, g_n) = 1$ for every $(g_1, \ldots, g_n)$, where $1$ is the identity element in $X$.

We now define the coboundary map $d : C^n(G, X_\rho) \to C^{n+1}(G, X_\rho)$ acting on cochains to be

$$(d\omega)(g_1, \ldots, g_{n+1}) = \rho_{g_1}(\omega(g_2, \ldots, g_{n+1})) \times$$
$$\times \prod_{j=1}^{n} \left(\omega(g_1, \ldots, g_{j-1}, g_j g_{j+1}, g_{j+2}, \ldots, g_{n+1})\right)^{(-1)^j} \left(\omega(g_1, \ldots, g_n)\right)^{(-1)^{n+1}}\,.$$
$$(47)$$

One can directly verify that $d(d\omega) = 1$ for any $\omega \in C^n(G, X_\rho)$, where $1$ denotes the trivial cochain in $C^{n+2}(G, X_\rho)$. With the coboundary map, we next define $\omega \in C^n(G, X_\rho)$ to be an $n$-cocycle if it satisfies the condition $d\omega = 1$, and all $n$-cocycles naturally form an Abelian group

$$Z^n(G, X_\rho) = \ker[d : C^n(G, X_\rho) \to C^{n+1}(G, X_\rho)] = \{\, \omega \in C^n(G, X_\rho) \mid d\omega = 1 \,\}. \tag{48}$$

We also define $\omega \in C^n(G, X_\rho)$ to be an $n$-coboundary if it satisfies the condition $\omega = d\mu$ for some $(n-1)$-cochain $\mu \in C^{n-1}(G, X_\rho)$, and all $n$-coboundaries naturally form an Abelian group

$$
\begin{aligned}
B^n(G, X_\rho) &= \mathrm{im}[d : C^{n-1}(G, X_\rho) \to C^n(G, X_\rho)] \\
&= \{\, \omega \in C^n(G, X_\rho) \mid \exists \mu \in C^{n-1}(G, X_\rho) : \omega = d\mu \,\}.
\end{aligned}
\tag{49}
$$

Clearly, $B^n(G, X_\rho) \subseteq Z^n(G, X_\rho) \subseteq C^n(G, X_\rho)$, and we define the $n$-th group cohomology of $G$ to be the quotient group

$$H^n(G, X_\rho) = \frac{Z^n(G, X_\rho)}{B^n(G, X_\rho)}. \tag{50}$$

In other words, $H^n(G, X_\rho)$ collects the equivalence classes of $n$-cocycles, where two $n$-cocycles are considered equivalent if they differ by an $n$-coboundary.

It is instructive to look at the lowest cohomology groups. Let us first consider $H^1(G, X_\rho)$:

$$
\begin{aligned}
Z^1(G, X_\rho) &= \{\, \omega \mid \omega(g_1) \rho_{g_1}(\omega(g_2)) = \omega(g_1 g_2) \,\}, \\
B^1(G, X_\rho) &= \{\, \omega \mid \omega(g) = \rho_g(\mu)\mu^{-1} \,\}.
\end{aligned}
\tag{51}
$$

If the $G$-action on $X$ is trivial, then $B^1(G, X_\rho) = \{1\}$ and $Z^1(G, X_\rho)$ consists of group homomorphisms from $G$ to $X$, which, in particular, map elements in the same conjugacy class to the same image, i.e.,

$$\omega(g_2^{-1} g_1 g_2) = \omega(g_1), \tag{52}$$

for any $g_{1,2} \in G$.

For the second cohomology, we have

$$
\begin{aligned}
Z^2(G, X_\rho) &= \{\, \omega \mid \rho_{g_1}(\omega(g_2, g_3)) \omega(g_1, g_2 g_3) = \omega(g_1, g_2) \omega(g_1 g_2, g_3) \,\}, \\
B^2(G, X_\rho) &= \{\, \omega \mid \omega(g_1, g_2) = \rho_{g_1}(\mu(g_2)) (\mu(g_1 g_2))^{-1} \mu(g_1) \,\}.
\end{aligned}
\tag{53}
$$

In particular, $H^2(G, U(1)_\rho)$ classifies all inequivalent complex projective representations of $G$, while $H^2(G, \mathbb{Z}_2)$ classifies all inequivalent real orthogonal projective representations of $G$, which will be most useful throughout the paper.

## A.2 Maps of group Cohomology

In this sub-appendix, we review various maps of group cohomology, which will be used throughout the paper.

The first map we consider is the pullback of group cohomology. Consider a map between two groups $\varphi : G \to H$ compatible with their respective group action $\rho_G$ and $\rho_H$ on $X$, in the sense that $\rho_{\varphi(g)}(a) = \rho_g(a)$ for any $a \in X$ and any $g \in G$ or, in the case of $X = U(1), \mathbb{Z}$, $\tilde{\rho}_H \circ \varphi = \tilde{\rho}_G$. Given such a map, we can define the pullback from $H^n(H, X_\rho)$ to $H^n(G, X_\rho)$, which can be defined on the representative cochain $\omega \in C^n(H, X_\rho)$ as follows

$$(\varphi^*(\omega))(g_1, \ldots, g_n) \equiv \omega(\varphi(g_1), \ldots, \varphi(g_n)). \tag{54}$$

It is straightforward to check that it maps cocycles to cocycles, and coboundaries to coboundaries, so it gives a well-defined map from $H^n(H, X_\rho)$ to $H^n(G, X_\rho)$,

$$\varphi^* : H^n(H, X_\rho) \to H^n(G, X_\rho). \tag{55}$$

The second map we consider is the map of group cohomology induced by a map of $G$-modules $i : X \to Y$. Here $i$ is any map from $G$-module $X$ to $G$-module $Y$ that preserves the action of $G$, i.e., for any $a \in X$ and $g \in G$ we have $\rho_g(i(a)) = i(\rho_g(a))$. Then for any $n$-cochain $\omega(g_1, \ldots, g_n) \in C^n(G, X_\rho)$, we can map it to another $n$-cochain $\tilde{i}(\omega)$ such that

$$(\tilde{i}(\omega))(g_1, \ldots, g_n) \equiv i(\omega(g_1, \ldots, g_n)). \tag{56}$$

It is straightforward to check that it maps cocycles to cocycles, and coboundaries to coboundaries, so it gives a well-defined map from $H^n(G, X_\rho)$ to $H^n(G, Y_\rho)$,

$$\tilde{i} : H^n(G, X_\rho) \to H^n(G, Y_\rho). \tag{57}$$

We will frequently use this map to convert cohomology elements in $H^n(G, \mathbb{Z}_2)$ to elements in $H^n(G, U(1)_\rho)$, induced by the inclusion $i$ of $\mathbb{Z}_2 = \{\pm 1\}$ into $U(1)$. Note that the representative cochains $\omega$ and $\tilde{i}(\omega)$ as a function from $G^n$ to $\mathbb{Z}_2$ and $U(1)$ are manifestly the same, but a function representing a nontrivial element in $H^n(G, \mathbb{Z}_2)$ can represent a trivial element in $H^n(G, U(1)_\rho)$, because the module $U(1)_\rho$ in general yields more coboundaries compared to the module $\mathbb{Z}_2$. We also consider the map of group cohomology $\tilde{p}$ induced by the projection $p$ of $\mathbb{Z}$ onto $\mathbb{Z}_2 = \{0, 1\}$

The third map which will be useful in the analysis of anomaly/anomaly-matching is the Bockstein homomorphism [101, 102]. Consider a short exact sequence of $G$-modules,

$$1 \longrightarrow X \xrightarrow{i} Z \xrightarrow{p} Y \longrightarrow 1 \;, \tag{58}$$

with the map $i : X \to Z$ injective, the map $p : Z \to Y$ surjective and $\ker[p] = \text{im}[i]$. There is a long exact sequence of the cohomology of $G$ associated to this short exact sequence, such that $\ker = \text{im}$ at any place of the following chain of maps,

$$\ldots \longrightarrow H^n(G, X_\rho) \xrightarrow{\tilde{i}} H^n(G, Z_\rho) \xrightarrow{\tilde{p}} H^n(G, Y_\rho)$$

$$\xrightarrow{\beta} H^{n+1}(G, X_\rho) \xrightarrow{\tilde{i}} \ldots \tag{59}$$

The map $\beta$, called the Bockstein homomorphism, is defined as follows. For $[\omega] \in H^n(G, Y_\rho)$ and a representative cochain $\omega$, choose a function $\tilde{\omega}$ from $G^n$ to $Z_\rho$ such that

$$p((\tilde{\omega})(g_1, \ldots, g_n)) = \omega(g_1, \ldots, g_n). \tag{60}$$

Because $p$ is surjective, $\tilde{\omega}$ always exists. For any choice of $\tilde{\omega}$, it is straightforward to see that $p((d\tilde{\omega})(g_1, \ldots, g_n)) = 0$ and as a result $(d\tilde{\omega})(g_1, \ldots, g_n)$ is in the image of $i$. Then we define this (unique) preimage to be the image of $\omega$ under the Bockstein homomorphism, i.e., we have

$$\beta(\omega) \equiv \tilde{i}^{-1}(d\tilde{\omega}). \tag{61}$$

There are several short exact sequences that we should pay special attention to. The first one is

$$1 \longrightarrow \mathbb{Z} \xrightarrow{\times 2\pi} \mathbb{R} \xrightarrow{\text{mod } 2\pi} U(1) \longrightarrow 1 \;. \tag{62}$$

When $H^n(G, \mathbb{Z}_\rho)$ and $H^{n+1}(G, \mathbb{Z}_\rho)$ contain torsion elements only, $H^n(G, \mathbb{R}_\rho) = H^{n+1}(G, \mathbb{R}_\rho) = 0$, and from Eq. (59) we see that the associated Bockstein homomorphism $\beta : H^n(G, U(1)_\rho) \to H^{n+1}(G, \mathbb{Z}_\rho)$ is an isomorphism. For most discussions in

this paper, especially when $G$ is a finite group (and $n > 0$), this Bockstein homomorphism is indeed an isomorphism, and only in the example in Appendix I.1 it is not, on which we will comment explicitly.

The second short exact sequence that is important to us is

$$1 \longrightarrow \mathbb{Z} \xrightarrow{\times 2} \mathbb{Z} \xrightarrow{\mathrm{mod}\ 2} \mathbb{Z}_2 \longrightarrow 1 \ . \tag{63}$$

For $x \in H^n(G, \mathbb{Z}_2)$, the Bockstein homomorphism $\beta_2$ is sometimes written as

$$\beta_2(x) = \frac{1}{2} dx \,. \tag{64}$$

When $H^n(G, \mathbb{Z}_\rho) = (\mathbb{Z}_2)^k$ with some non-negative integer $k$, $\tilde{i}$ maps $H^n(G, \mathbb{Z}_\rho)$ to 0 in $H^n(G, \mathbb{Z}_\rho)$. Therefore, from Eq. (59), we see that $\tilde{p}$ is injective while $\beta_2$ is surjective.

We can also consider the natural map from Eq. (63) to Eq. (62), which is inclusion for every factor as follows,

$$\begin{array}{ccccccccc}
1 & \longrightarrow & \mathbb{Z} & \xrightarrow{\times 2\pi} & \mathbb{R} & \xrightarrow{\mathrm{mod}\ 2\pi} & U(1) & \longrightarrow & 1 \\
& & \cong \big\uparrow & & \times \pi \big\uparrow & & i \big\uparrow & & \\
1 & \longrightarrow & \mathbb{Z} & \xrightarrow{\times 2} & \mathbb{Z} & \xrightarrow{\mathrm{mod}\ 2} & \mathbb{Z}_2 & \longrightarrow & 1
\end{array} \,, \tag{65}$$

where $i$ is again the inclusion of $\mathbb{Z}_2 = \{\pm 1\}$ into $U(1)$. As a result, we have a map of long exact sequences,

$$\begin{array}{ccccccccc}
\cdots & \longrightarrow & H^n(G, \mathbb{Z}_2) & \xrightarrow{\beta_2} & H^{n+1}(G, \mathbb{Z}_\rho) & \longrightarrow & H^{n+1}(G, \mathbb{Z}_\rho) & \longrightarrow & \cdots \\
& & \big\downarrow{\tilde{i}} & & \big\downarrow{\cong} & & \big\downarrow & & \\
\cdots & \longrightarrow & H^n(G, U(1)_\rho) & \xrightarrow{\beta} & H^{n+1}(G, \mathbb{Z}_\rho) & \longrightarrow & H^{n+1}(G, \mathbb{R}_\rho) & \longrightarrow & \cdots
\end{array} \,. \tag{66}$$

Here we distinguish the first Bockstein homomorphism by denoting it by $\beta_2$, and $\tilde{i}$ denotes the map induced by $i : \mathbb{Z}_2 \to U(1)$ specifically. Hence, we have $\beta_2 = \beta \circ \tilde{i}$. When $H^n(G, \mathbb{Z}_\rho) = (\mathbb{Z}_2)^k$, since $\beta$ is an isomorphism while $\beta_2$ is surjective , $\tilde{i}$ is surjective as well. It suggests that in this case every element $\Omega \in H^n(G, U(1)_\rho)$ can be written as $\tilde{i}(L)$ or $e^{i\pi L}$ for some $L \in H^n(G, \mathbb{Z}_2)$. In fact, every element $\Omega \in H^n(G, U(1)_\rho)$ whose inverse is itself can be written as $e^{i\pi L}$ for some $L \in H^n(G, \mathbb{Z}_2)$. We use this fact throughout the paper.

## A.3 Cup product and $\mathbb{Z}_2$ cohomology ring

In this sub-appendix, we review cup product and $\mathbb{Z}_2$ cohomology ring in group cohomology that we will use [99, 100, 102]. We will specialize to the case where the module is $\mathbb{Z}_2 = \{0, 1\}$ and the group action $\rho$ is trivial. The special feature of $\mathbb{Z}_2$, countrary to e.g. $U(1)$, is the fact that $\mathbb{Z}_2$ is a ring. Note that here addition in $\mathbb{Z}_2$ is regarded as the group multiplication used in Eq. (46), and we will use $+$ to denote this addition in this sub-appendix. There is another ring multiplication that will be important later, which should be distinguished with the group multiplication used in Appendix A.1.

The cross product is defined as the following operation on group cohomology,

$$\times: \quad H^m(G, \mathbb{Z}_2) \otimes H^n(H, \mathbb{Z}_2) \to H^{m+n}(G \times H, \mathbb{Z}_2), \tag{67}$$

such that for $x \in H^m(G, \mathbb{Z}_2)$ and $y \in H^n(H, \mathbb{Z}_2)$, after choosing cochain representatives $\tilde{x}$ and $\tilde{y}$, we have the cochain representative of $x \times y$ as follows,

$$\widetilde{x \times y}((g_1, h_1), \ldots, (g_{m+n}, h_{m+n})) \equiv \tilde{x}(g_1, \ldots, g_m) \cdot \tilde{y}(h_{m+1}, \ldots, h_{m+n}), \tag{68}$$

where $g_i \in G, h_i \in H, i = 1, \ldots, m + n$.

The cup product is defined as the following operation on group cohomology,

$$\cup: \quad H^m(G, \mathbb{Z}_2) \otimes H^n(G, \mathbb{Z}_2) \xrightarrow{\times} H^{m+n}(G \times G, \mathbb{Z}_2) \xrightarrow{\Delta^*} H^{m+n}(G, \mathbb{Z}_2) \;, \tag{69}$$

where $\Delta : G \to G \times G$ is the diagonal embedding $g \to (g, g)$. We can also define it at the cochain level, i.e., for $x \in H^m(G, \mathbb{Z}_2)$ and $y \in H^n(G, \mathbb{Z}_2)$, after choosing cochain representatives $\tilde{x}$ and $\tilde{y}$, we have the cochain representative of $x \cup y$ as follows,

$$\widetilde{x \cup y}(g_1, \ldots, g_{m+n}) \equiv \tilde{x}(g_1, \ldots, g_m) \cdot \tilde{y}(g_{m+1}, \ldots, g_{m+n}). \tag{70}$$

We can prove that cup product is commutative, i.e., $x \cup y = y \cup x$.

The cup product $\cup$ gives a multiplication on the direct sum of cohomology groups

$$H^*(G, \mathbb{Z}_2) = \bigoplus_{k \in \mathbb{N}} H^k(G, \mathbb{Z}_2). \tag{71}$$

Together with the fact that $1 \cup x = x$ where $x$ is any element in $H^*(G, \mathbb{Z}_2)$ and 1 here denotes the nontrivial element in $H^0(G, \mathbb{Z}_2) = \mathbb{Z}_2$, the cup product $\cup$ turns $H^*(G, \mathbb{Z}_2)$ into a ring that is naturally $\mathbb{N}$ graded and commutative. We call this ring *the $\mathbb{Z}_2$ cohomology ring* of $G$.

Moreover, $H^*(G, \mathbb{Z}_2)$ is also a $\mathbb{Z}_2$ algebra, and therefore can be presented by generators and relations, i.e., all elements in $H^n(G, \mathbb{Z}_2)$ for any $n > 0$ are either generators or can be expressed as sum of (cup) products of generators, and generators satisfy some relations which dictate that certain sums of (cup) products actually yield a trivial cohomology element. We will call a generator in $H^n(G, \mathbb{Z}_2)$ a degree $n$ generator. Hence, the $\mathbb{Z}_2$ cohomology ring of $G$, i.e., $H^*(G, \mathbb{Z}_2)$, can be written as follows,

$$H^*(G, \mathbb{Z}_2) = \mathbb{Z}_2[A_\bullet, \cdots, B_\bullet, \cdots]/\text{relations}, \tag{72}$$

with $A_\bullet (B_\bullet)$ generators in degree 1(2) belonging to $H^1(G, \mathbb{Z}_2)(H^2(G, \mathbb{Z}_2))$, and $\bullet$ the name of the generator. Together with potential higher order generators, e.g., $C_\bullet$ in degree 3, they are supposed to form a complete list of generators of the entire cohomology ring.

For example, the $\mathbb{Z}_2$ cohomology ring of the group $\mathbb{Z}_2$ is

$$\mathbb{Z}_2[A_c], \tag{73}$$

where $A_c$ is the nontrivial element in $H^1(\mathbb{Z}_2, \mathbb{Z}_2)$ and can be thought of as nothing but the gauge field of e.g., $C_2$ rotation when pulled back to the spacetime manifold. In other words, for the $\mathbb{Z}_2$ cohomology ring of $\mathbb{Z}_2$, there is a single generator $A_c$ in degree 1 and no relation. Accordingly, we can see that $H^n(G, \mathbb{Z}_2) = \mathbb{Z}_2$ for $n \in \mathbb{N}$, with the nontrivial element given by $A_c^n \equiv A_c \cup A_c \cup \cdots \cup A_c$, the cup product of $n$ $A_c$'s.

As another example, the $\mathbb{Z}_2$ cohomology ring of $\mathbb{Z}_4$ is

$$\mathbb{Z}_2[A_c, B_{c^2}]/\left(A_c^2 = 0\right), \tag{74}$$

where here $A_c$ is the nontrivial element in $H^1(\mathbb{Z}_4, \mathbb{Z}_2)$ and can be thought of as (the $\mathbb{Z}_2$ reduction of) the gauge field of $C_4$ rotation when pulled back to the spacetime manifold, while $B_{c^2}$ is the nontrivial element in $H^2(\mathbb{Z}_4, \mathbb{Z}_2)$, which corresponds to the fractionalization pattern of the $\mathbb{Z}_4$ symmetry on an $SO(3)$ monopole, with $C_4^4 = C_2^2 = -1$. That is to say, for the $\mathbb{Z}_2$ cohomology ring of $\mathbb{Z}_4$, there are two generators at degree 1 and 2 respectively, with the square of degree 1 generator $A_c$ equal to 0. Then we see that $H^n(\mathbb{Z}_4, \mathbb{Z}_2) = \mathbb{Z}_2$ for $n \in \mathbb{N}$ as well, and the nontrivial element is given by $B_{c^2}^k$ when $n = 2k$ and $A_c B_{c^2}^k$ when $n = 2k + 1$ ($k \in \mathbb{N}$). Note that for both $G = \mathbb{Z}_2$ and $G = \mathbb{Z}_4$, $H^n(G, \mathbb{Z}_2) = \mathbb{Z}_2$ for any $n \in \mathbb{N}$, but the $\mathbb{Z}_2$ cohomology rings give more information that differentiates the two groups.

For any two groups $G_1$ and $G_2$, we have $H^*(G_1 \times G_2, \mathbb{Z}_2) = H^*(G_1, \mathbb{Z}_2) \otimes H^*(G_2, \mathbb{Z}_2)$. Moreover, if $G$ can be written as $G_1 \rtimes G_2$, where $G_1$ is a normal subgroup of $G$ and $G_2$ acts on $G_1$ by conjugation, the calculation of the $\mathbb{Z}_2$ cohomology ring of $G$ can be achieved with the help of Lyndon–Hochschild–Serre spectral sequence [99, 100] that also connects $H^*(G, \mathbb{Z}_2)$ with $H^*(G_1, \mathbb{Z}_2)$ and $H^*(G_2, \mathbb{Z}_2)$ which we possibly already know. The general strategy for calculating the $\mathbb{Z}_2$ cohomology ring of wallpaper groups $G$ is as follows:

1. Identify all generators and elements in the $\mathbb{Z}_2$ cohomology ring of $G$ through Lyndon–Hochschild–Serre spectral sequence.

2. If there is no relation, given generators $A_1, A_2, B_1, C_1 \ldots$, all elements of the form $A_1^m A_2^n B_1^p C_1^q \ldots, m, n, p, q \cdots \in \mathbb{N}$ will appear explicitly as different elements in the $\mathbb{Z}_2$ cohomology ring. Therefore, when e.g., some $A_1^2$ is missing, we should identify some relation that relates $A_1^2$ to elements that appear explicitly, which can be achieved through pulling back to (enough) subgroups of $G$.

To illustrate the strategy, in the following we calculate the $\mathbb{Z}_2$ cohomology ring of three space groups in one or two spatial dimensions, including the generators and relations.

- $p1$: $\mathbb{Z}_2[x]/(x^2 = 0)$.

  Consider the line group $p1$, generated by a single translation $T$. The cohomology of $p1$ is $H^1(p1, \mathbb{Z}_2) \cong \mathbb{Z}_2$ while $H^n(p1, \mathbb{Z}_2) \cong 0, \; n > 1$. Denote the nontrivial element in $H^1(p1, \mathbb{Z}_2)$ as $x$, which corresponds to (the $\mathbb{Z}_2$ reduction of) the gauge field of translation, the $\mathbb{Z}_2$ cohomology ring of $p1$ is given by $\mathbb{Z}_2[x]/(x^2 = 0)$.

- $p1m$: $\mathbb{Z}_2[x, m]/(x^2 = xm)$.

  Consider the line group $p1m$, generated by translation $T$ and mirror symmetry $M$ with relation $MTM = T^{-1}$. The cohomology of $p1m$ is $H^n(p1m, \mathbb{Z}_2) \cong \mathbb{Z}_2^2, \; n \geqslant 1$. Since $p1m \cong \mathbb{Z} \rtimes \mathbb{Z}_2$, with the help of the corresponding Serre spectral sequence, we know that $H^n(p1m, \mathbb{Z}_2), \; n \geqslant 1$ is spanned by 2 elements, i.e., $m^n$ and $m^{n-1}x$. where $m, x \in H^1(p1m, \mathbb{Z}_2)$ are two generators that correspond to the gauge field of mirror symmetry and (the $\mathbb{Z}_2$ reduction of) the gauge field of translation, respectively.

  The next thing to do is to identify $x^2$, which does not explicitly appear as elements of the $\mathbb{Z}_2$ cohomology ring. Write $x^2$ as $a_1 xm + a_2 m^2, a_{1,2} \in \{0, 1\}$. By restricting to $\mathbb{Z}_2$ subgroup generated by $M$, whose $\mathbb{Z}_2$ cohomology ring can be denoted by $\mathbb{Z}_2[m']$, we see that $x$ becomes 0 while $m$ becomes $m'$, and thus $a_2 = 0$. By restricting to the $\mathbb{Z}_2$ subgroup generated by $TM$, whose $\mathbb{Z}_2$ cohomology ring can be denoted by $\mathbb{Z}_2[m'']$, we see that both $x$ and $m$ become $m''$, and thus $a_1 = 1$. Therefore, we have $x^2 = xm$.

  Therefore, the $\mathbb{Z}_2$ cohomology ring of $p1m$ is $\mathbb{Z}_2[x, m]/(x^2 = xm)$.

- $cm$: $\mathbb{Z}_2[A_{x+y}, A_m, B_{xy}]/(A_{x+y} A_m = 0, A_{x+y}^2 = 0, B_{xy} A_{x+y} = 0, B_{xy}^2 = 0)$.

  Consider 2$d$ wallpaper group $cm$, generated by two translation symmetries $T_1, T_2$ as well as mirror symmetry $M$ that interchanges the two translations, i.e., $MT_1 M = T_2$ and $MT_2 M = T_1$. The cohomology of $cm$ is $H^n(cm, \mathbb{Z}_2) \cong (\mathbb{Z}_2)^2, \; n \geqslant 1$. Since $cm \cong (\mathbb{Z} \times \mathbb{Z}) \rtimes \mathbb{Z}_2$, with the help of the corresponding Serre spectral sequence, we know that $H^1(cm, \mathbb{Z}_2)$ is spanned by $A_{x+y}$ and $A_m$, while $H^n(cm, \mathbb{Z}_2), n \geqslant 2$ is spanned by $B_{xy} A_m^{n-2}$ and $A_m^n$. Here $A_m, A_{x+y} \in H^1(cm, \mathbb{Z}_2)$ correspond to the gauge field of mirror symmetry and (the $\mathbb{Z}_2$ reduction of) the sum of gauge fields of $T_1$ and $T_2$, respectively. Note that since $T_1$ and $T_2$ map to each other under conjugation by $M$, the gauge field of the two translations $x$ and $y$ individually is not invariant under conjugation by $M$, yet their sum that we denote by $A_{x+y}$ is invariant under conjugation by $M$, which is

a necessary condition for it to be a cohomology element, as required by Eq. (52). To conform to the notation, we also denote the gauge field of mirror symmetry by $A_m$ when considering wallpaper groups. Moreover, there is an extra degree-2 generator $B_{xy}$, i.e., an element belonging to $H^2(cm, \mathbb{Z}_2)$ that cannot be written as sum of cup product of elements in $H^1(cm, \mathbb{Z}_2)$. The name $xy$ comes from the fact that its restriction to subgroup $p1$ generated by $T_1, T_2$ is $A_x A_y$ (see Appendix E).

To identify the relations, we note that there are now 4 missing elements: $A_{x+y}A_m, A^2_{x+y}, B_{xy}A_{x+y}, B^2_{xy}$. By restricting to the subgroup $p1$ generated by $T_1, T_2$ as well as the subgroup $\mathbb{Z}_2$ generated by $M$, we see that $A_{x+y}A_m = A^2_{x+y} = 0$. By restricting to the subgroup $pm$ generated by $T_1 T_2^{-1}, T_1 T_2, M$, we see that $B_{xy}A_{x+y} = 0$ as well as $B^2_{xy} = 0$. Note that the pullback of $A_{x+y}, A_m$ and $B_{xy}$ to the subgroup $pm$ is 0, $A_m$ and $A_y A_m$, respectively.

Therefore, the $\mathbb{Z}_2$ cohomology ring of $cm$ is

$$\mathbb{Z}_2[A_{x+y}, A_m, B_{xy}]/\left(A_{x+y}A_m = 0, A^2_{x+y} = 0, B_{xy}A_{x+y} = 0, B^2_{xy} = 0\right).$$

## A.4 $\mathcal{SQ}^1$

In this sub-appendix, we define a new map we call $\mathcal{SQ}^1$, reminiscent of $Sq^1$ in regular Steenrod algebra, as follows

$$\mathcal{SQ}^1 : H^n(G, \mathbb{Z}_2) \xrightarrow{\tilde{i}} H^n(G, U(1)_\rho) \xrightarrow{\beta} H^{n+1}(G, \mathbb{Z}_\rho) \xrightarrow{\tilde{p}} H^{n+1}(G, \mathbb{Z}_2), \qquad (75)$$

$$\mathcal{SQ}^1 \equiv \tilde{p} \circ \beta \circ \tilde{i}, \qquad (76)$$

where $\tilde{i}$ and $\tilde{p}$ are the map of group cohomology induced by the homomorphism of modules $i : \mathbb{Z}_2 \to U(1)$ and $p : \mathbb{Z} \to \mathbb{Z}_2$, and $\beta$ is the Bockstein homomorphism associated with the short exact sequence $1 \to \mathbb{Z} \to \mathbb{R} \to U(1) \to 1$. Note that $\beta \circ \tilde{i}$ is the Bockstein homomorphism $\beta_2$ associated with the short exact sequence $1 \to \mathbb{Z} \to \mathbb{Z} \to \mathbb{Z}_2 \to 1$, and therefore when the action $\rho$ is trivial, $\mathcal{SQ}^1$ is exactly $Sq^1$ in regular Steenrod algebra.

Moreover, $\mathcal{SQ}^1$ is related to $Sq^1$ via the following simple fact

**Lemma A.1.** For $x \in H^n(G, \mathbb{Z}_2)$, we have

$$\mathcal{SQ}^1(x) = \mathcal{SQ}^1(1) \cup x + Sq^1(x). \qquad (77)$$

*Proof.* According to Eq. (61), choosing a cochain $\tilde{x} \in C^n(G, \mathbb{Z})$ such that the $\mathbb{Z}_2$ reduction of $\tilde{x}$ is $x$, we have

$$\begin{aligned}
\mathcal{SQ}^1(x) = \frac{1}{2}\Bigg( &(-1)^{\tilde{\rho}(g_1)} \tilde{x}(g_2, \ldots, g_{n+1}) \\
&+ \sum_{j=1}^{n}(-1)^j \tilde{x}(g_1, \ldots, g_j g_{j+1}, \ldots, g_{n+1}) + (-1)^{n+1}\tilde{x}(g_1, \ldots, g_n)\Bigg) \\
= &\frac{1}{2}\left((-1)^{\tilde{\rho}(g_1)} - 1\right)\tilde{x}(g_2, \ldots, g_{n+1}) + Sq^1(x) \\
= &\,\mathcal{SQ}^1(1) \cup x + Sq^1(x) \mod 2.
\end{aligned} \qquad (78)$$

$\blacksquare$

For example, for $\mathbb{Z}_2^T$ with nontrivial action on $U(1)$ or $\mathbb{Z}$, we have,

$$\mathcal{SQ}^1(t^{2n+1}) = 0, \qquad \mathcal{SQ}^1(t^{2n}) = t^{2n+1}, \qquad (79)$$

where $t \in H^1(\mathbb{Z}_2^T, \mathbb{Z}_2)$ is the generator of the $\mathbb{Z}_2$ cohomology ring of $\mathbb{Z}_2^T$. We see that in the presence of nontrivial $\rho$, the operation $\mathcal{SQ}^1$ is not distributive with respect to the cup product. Note that $\mathcal{SQ}^1(1)$ is nonzero and equals $t$, which when pulled back to the spacetime manifold $\mathcal{M}$ equals $w_1$ as well, i.e., the first Stiefel-Whitney class of $\mathcal{M}$. In contrast, for $\mathbb{Z}_2$ with trivial action on $U(1)$ or $\mathbb{Z}$, we have

$$\mathcal{SQ}^1(A_c^{2n}) = 0, \qquad \mathcal{SQ}^1(A_c^{2n+1}) = A_c^{2n+2}, \tag{80}$$

where $A_c \in H^1(\mathbb{Z}_2, \mathbb{Z}_2)$ is the generator of the $\mathbb{Z}_2$ cohomology ring of $\mathbb{Z}_2$ as well.

As another example, consider $O(5)$ with $\tilde{\rho} : O(5) \to \mathbb{Z}_2$ the determinant, i.e., an $O(5)$ element complex conjugates an $U(1)$ element or multiplies a $\mathbb{Z}$ element by $-1$ if and only if the determinant of the $O(5)$ element is $-1$. From Lemma A.1 we immediately have,

$$\mathcal{SQ}^1\left(w_4^{O(5)}\right) = w_5^{O(5)}, \tag{81}$$

as suggested by the calculation in the context of DQCP in Refs. [19, 103].

Moreover, even if $\mathcal{SQ}^1$ is not distriutive with respect to the cup product, from Lemma A.1 $\mathcal{SQ}^1$ is still distributive with respect to the cross product involving two different groups, i.e., we have

**Lemma A.2.** *For $x \in H^m(G, \mathbb{Z}_2)$ and $y \in H^n(H, \mathbb{Z}_2)$, we have $x \times y \in H^{m+n}(G \times H, \mathbb{Z}_2)$ and*

$$\mathcal{SQ}^1(x \times y) = \mathcal{SQ}^1(x) \times y + x \times \mathcal{SQ}^1(y). \tag{82}$$

This lemma is also important when calculating $\mathcal{SQ}^1$ because it decomposes the calculation into different pieces corresponding to different groups. For example, with the help of Eqs. (79) and (80), the lemma tells us how to calculate $\mathcal{SQ}^1$ for the group $(\mathbb{Z}_2)^k$ with every $\mathbb{Z}_2$ piece acting trivially or nontrivially on $U(1)$ or $\mathbb{Z}$.

Finally, from the fact that

$$A_{\mathrm{LH}} \cong \ker\left[\tilde{i} : H^2(G_s, \mathbb{Z}_2) \to H^2(G_s, U(1)_\rho)\right], \tag{83}$$

as argued in Section 2.2, for LSM anomaly written as $\exp(i\pi\lambda\eta)$ where $\lambda \in H^2(G_s, \mathbb{Z}_2)$ and $\eta \in H^2(G_{int}, \mathbb{Z}_2)$, we have

$$\mathcal{SQ}^1(\lambda) = 0. \tag{84}$$

This can also be mathematically checked by considering different representations $\rho : G_s \to O(n)$. For example, consider $G_s = p4m$. Then $A_{\mathrm{LH}}$ is spanned by $\lambda_1 = B_{xy} + A_{x+y}(A_{x+y} + A_m) + B_{c^2}$, $\lambda_2 = B_{xy}$, $\lambda_3 = A_{x+y}(A_{x+y} + A_m)$ in $H^2(p4m, \mathbb{Z}_2)$ (see Appendix E), corresponding to LSM anomaly associated to DOF at the site $a$, plaquette center $b$, and bond center $c$ as in Fig. 3, repectively. Consider the following three representations of $p4m$. The first one is

$$T_x \to \begin{pmatrix} -1 & 0 & 0 \\ 0 & -1 & 0 \\ 0 & 0 & 1 \end{pmatrix}, \quad T_y \to \begin{pmatrix} -1 & 0 & 0 \\ 0 & 1 & 0 \\ 0 & 0 & -1 \end{pmatrix}, \quad C_4 \to \begin{pmatrix} 1 & 0 & 0 \\ 0 & 0 & 1 \\ 0 & -1 & 0 \end{pmatrix},$$
$$M \to \begin{pmatrix} 1 & 0 & 0 \\ 0 & 1 & 0 \\ 0 & 0 & -1 \end{pmatrix}. \tag{85}$$

The pullback of $w_2^{O(3)}$ equals $B_{xy} + B_{c^2}$ while the pullback of $w_3^{O(3)}$ is zero (see sub-Section 3.2 and especially Eq. (34)). From $\mathcal{SQ}^1\left(w_2^{O(3)}\right) = w_3^{O(3)}$, we establish that

$$\mathcal{SQ}^1(B_{xy} + B_{c^2}) = 0. \tag{86}$$

The second representation is

$$T_x \to \begin{pmatrix} 1 & 0 \\ 0 & 1 \end{pmatrix}, \quad T_y \to \begin{pmatrix} 1 & 0 \\ 0 & 1 \end{pmatrix}, \quad C_4 \to \begin{pmatrix} 0 & 1 \\ -1 & 0 \end{pmatrix}, \quad M \to \begin{pmatrix} 1 & 0 \\ 0 & -1 \end{pmatrix}. \tag{87}$$

The pullback of $w_2^{O(2)}$ equals $B_{c^2}$, and from $\mathcal{SQ}^1\left(w_2^{O(2)}\right) = 0$, we establish that

$$\mathcal{SQ}^1(B_{c^2}) = 0. \tag{88}$$

The third representation is

$$
\begin{aligned}
T_x \to \begin{pmatrix} -1 & 0 \\ 0 & -1 \end{pmatrix}, \quad T_y \to \begin{pmatrix} -1 & 0 \\ 0 & -1 \end{pmatrix}, \quad C_4 \to \begin{pmatrix} 1 & 0 \\ 0 & 1 \end{pmatrix}, \\
M \to \begin{pmatrix} 1 & 0 \\ 0 & -1 \end{pmatrix},
\end{aligned}
\tag{89}
$$

and we have

$$\mathcal{SQ}^1(A_{x+y}(A_{x+y} + A_m)) = 0. \tag{90}$$

Since $B_{xy} + B_{c^2}, B_{c^2}, A_{x+y}(A_{x+y} + A_m)$ span $A_{\text{LH}}$ as well, indeed we mathematically show that $\mathcal{SQ}^1(\lambda) = 0$ for $\lambda \in A_{\text{LH}}$ in $p4m$. Then according to Lemma A.2 we also have

$$\mathcal{SQ}^1(\lambda\eta) = \mathcal{SQ}^1(\lambda) \times \eta + \lambda \times \mathcal{SQ}^1(\eta) = \lambda \times \mathcal{SQ}^1(\eta). \tag{91}$$

This equation will be very useful in the analysis of anomaly-matching. Note that this equation holds for LSM constraints on lattices with any wallpaper group.

# B  Topological partition function corresponding to LSM

In this appendix, we provide a more rigorous argument that the cocycle corresponding to the topological partition function (TPF) of the $(3 + 1)$-d $G_s \times G_{int}$ SPT, whose boundary has some LSM constraint, can indeed be written in the form of Eq. (1). [15]

To start, first recall that the lattice homotopy picture indicates that all LSM constraints for a given wallpaper group $G_s$ are classified by a group $A_{\text{LH}} = \mathbb{Z}_2^k$ with some integer $k$. This means that the sought-for cocycle in $H^4(G_s \times G_{int}, U(1)_\rho)$ can be written as

$$\Omega(g_1, g_2, g_3, g_4) = e^{i\pi\kappa(g_1, g_2, g_3, g_4)}, \tag{92}$$

---

[15] Since the $(3+1)$-d $G_s \times G_{int}$ SPT is captured by an element in $H^4(G_s \times G_{int}, U(1)_\rho)$, one may attempt to show the validity of Eq. (1) by combining the Kunneth decomposition $H^4(G_s \times G_{int}, U(1)_\rho) \cong \oplus_{i=0}^4 H^i(G_s, H^{4-i}(G_{int}, U(1)_\rho))$ and the fact that the relevant $(1 + 1)$-d $G_{int}$ SPT is captured by $H^2(G_{int}, U(1)_\rho)$, which suggests that in the Kunneth decomposition only the term $H^2(G_s, H^2(G_{int}, U(1)_\rho))$ is relevant to the LSM constraints. Although intuitively appealing, this argument is flawed, because there is generically no unambiguous way to determine whether an element in $H^4(G_s \times G_{int}, U(1)_\rho)$ is in $H^2(G_s, H^2(G_{int}, U(1)_\rho))$. Our argument below does not suffer from this ambiguity. Furthermore, even if we know that the relevant cocycle is in $H^2(G_s, H^2(G_{int}, U(1)_\rho))$, it requires an explanation why its representative cochain can necessarily be written as Eq. (1).

with $\kappa$ taking values in $\{0,1\}$. This allows us to view $\kappa(g_1, g_2, g_3, g_4)$ as a representative cochain in $H^4(G_s \times G_{int}, \mathbb{Z}_2)$, where the multiplication between two elements is implemented by the mod 2 addition of their corresponding representative cochains. Since $H^4(G_s \times G_{int}, \mathbb{Z}_2) \simeq \oplus_{i=0}^4 H^i(G_s, \mathbb{Z}_2) \otimes H^{4-i}(G_{int}, \mathbb{Z}_2)$, we can always write $\Omega$ as

$$\Omega(g_1, g_2, g_3, g_4) = \prod_{i=0}^4 e^{i\pi\lambda_i(l_1, \cdots, l_i)\eta_{4-i}(a_{i+1}, \cdots, a_4)}, \tag{93}$$

where each $g_i \in G_s \times G_{int}$ is again written as $g_i = l_i \otimes a_i$, with $l_i \in G_s$ and $a_i \in G_{int}$. Both $\lambda_i$ and $\eta_{4-i}$ take values in $\{0,1\}$, and they can be viewed as representative cochains in $H^i(G_s, \mathbb{Z}_2)$ and $H^{4-i}(G_{int}, \mathbb{Z}_2)$, respectively. Furthermore, we can view $e^{i\pi\eta_{4-i}(a_{i+1}, \cdots, a_4)}$ as a representative cochain in $H^{4-i}(G_{int}, U(1)_\rho)$, which can be physically interpreted as a $G_{int}$ SPT living in $3-i$ spatial dimensions.

Previous studies of $G_s \times G_{int}$ SPTs indicate that all these SPTs have a real-space construction, in which various lower dimensional SPTs (or invertible states) are decorated into various submanifolds of the entire crystal [9, 104, 105]. Indeed, the SPT relevant to LSM constraints can be constructed by putting copies of $(1+1)$-d $G_{int}$ SPTs at various IWP of the wallpaper group $G_s$. Combining these two observations together, we conclude that in Eq. (93) only the factor with $i = 2$ can possibly be related to LSM constraints, because only that factor can possibly be related to putting $(1+1)$-d $G_{int}$ SPTs at various positions, while other factors involve SPTs living in the wrong dimension (e.g., the $i = 1$ term means that some $(2+1)$-d $G_{int}$ SPT is decorated into the system in some way). Moreover, for a given PR type of the system, $e^{i\pi\eta_2(a_3, a_4)}$ should be the cocycle corresponding to the $(1+1)$-d $G_{int}$ SPT whose boundary hosts this particular PR.

Therefore, the cocycle related to LSM constraints can always be written in a form given by Eq. (1), and $\lambda(l_1, l_2)$, which is written as $\lambda_2(l_1, l_2)$ in Eq. (93), can be viewed as a representative cochain in $H^2(G_s, \mathbb{Z}_2)$. Furthermore, according to the lattice homotopy picture, $\lambda$ or $\lambda_2$ should just encode the information of which IWP host $(1+1)$-d $G_{int}$ SPTs, so it should be completely determined by $G_s$ and the lattice homotopy class corresponding to each LSM constraint, and be the same for all $G_{int}$ and all PR types of the system.

We remark that the above argument does not show that all cocycles in the form of Eq. (1) must be related to LSM constraints. In fact, in Sec. 2.2 we have found that some of them are not. Those SPTs can be constructed by inserting a $(2+1)$-d $Z_2 \times G_{int}$ SPT on the mirror plane, such that the $Z_2$ domain wall is decorated with a $(1+1)$-d $G_{int}$ SPT. See Appendix D for more detail.

We also remark that although we have assumed that the projective representations of $G_{int}$ are $\mathbb{Z}_2^k$-classified with $k$ some integer in the above argument, we expect that the topological partition functions corresponding to LSM constraints can always be written in a form similar to Eq. (1), for any $G_{int}$. Specially, if a PR type of $G_{int}$ has order $n$, then the LSM-related cocycle takes the form

$$\Omega(g_1, g_2, g_3, g_4) = e^{i\frac{2\pi}{n}\lambda(l_1, l_2)\eta(a_3, a_4)}, \tag{94}$$

where $\lambda$ and $\eta$ take integral values, and $e^{\frac{2\pi i}{n}\eta(a_3, a_4)}$ is the cocycle corresponding to the relevant $(1+1)$-d $G_{int}$ SPT. Moreover, this statement, including its special form Eq. (1), has been derived in the special cases where $G_s$ contains only translation or only point group, using equivariant homology [10], and we expect that the method in Ref. [10] can be generalized to an arbitrary lattice symmetry group $G_s$. A systematic proof of this statement is beyond the scope of this paper and we leave it for future work.

# C  Fractionalization pattern involving both translation and glide symmetries

Among all 17 wallpaper groups, there is only one group, $pg$, in which the fractionalization pattern has to be specified in a way that necessarily invokes the glide symmetry. In this appendix, we present its corresponding physical picture.

The group $pg$ is generated by $T_1$ and $G$, a translation and a glide reflection. The translation vector of $T_1$ is flipped under $G$, and $G^2$ is another translation along a direction perpendicular to the translation vector of $T_1$. These generators satisfy $G^{-1}T_1GT_1 = 1$.

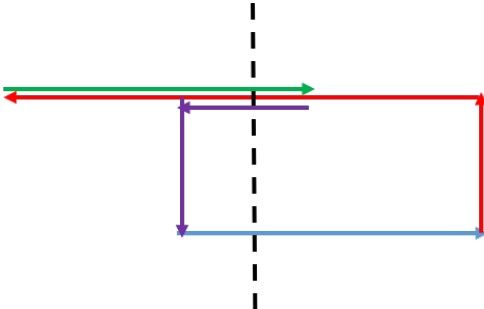

Figure 4: Acting $G^{-1}T_1GT_1$ on an $SO(3)$ monopole. The first $T_1$ action is marked in blue, the following $G$ action is marked in red, the next $T_1$ action is marked in green, and the last $G^{-1}$ action is marked in purple. The dashed line is the reflection axis of $G$. This figure shows that the operation $G^{-1}T_1GT_1$ moves an $SO((3)$ monopole along a trajectory that encloses a fundamental domain.

Consider the case where $G_{int} = SO(3)$. Just as in the main text, we gauge the $SO(3)$ symmetry and examine the fractionalization pattern of $pg$ on the $SO(3)$ monopole by applying the operation $G^{-1}T_1GT_1$ to an $SO(3)$ monopole, which moves the monopole around the fundamental domain (see Fig. 4). If the fundamental domain contains an odd (even) number of Haldane chains, this process results in a $-1$ (1) phase factor, which is a signature of nontrivial (trivial) symmetry fractionalization pattern of the $pg$ symmetry carried by the $SO(3)$ monopole. Because $H^2(pg, \mathbb{Z}_2) = \mathbb{Z}_2$, there is only one nontrivial symmetry fractionalization pattern. A topological invariant detecting the nontrivial element in $H^2(pg, \mathbb{Z}_2)$ is given in Eq. (192), so this must be the topological invariant that diagnoses the fractionalization pattern of the $pg$ symmetry on an $SO(3)$ monopole.

# D  Non-LSM fractionalization patterns

In this appendix, we discuss in more detail the $M \times SO(3)$ SPT corresponding to $\alpha_{\text{non-LSM}} = \frac{\omega(M,M)}{\omega(1,1)} = -1$, which has TPF $\exp(i\pi \int (w_1^{TM})^2 w_2^{SO(3)})$. In particular, we will show that this SPT can be constructed by inserting into the mirror plane of $M$ a $(2+1)$-d $\mathbb{Z}_2 \times SO(3)$ SPT, whose $\mathbb{Z}_2$ domain walls are decorated with Haldane chains. Moreover, we will show that for any $G_{int}$ with $\mathbb{Z}_2^k$-classified PR, the $(3+1)$-d $M \times G_{int}$ SPTs with $\alpha_{\text{non-LSM}} = -1$ can always be constructed by inserting into the mirror plane of $M$ a $(2+1)$-d $\mathbb{Z}_2 \times G_{int}$ SPT, whose $\mathbb{Z}_2$ domain walls are decorated with the relevant nontrivial $(1+1)$-d $G_{int}$ SPT.

Focusing on the case with $G_{int} = SO(3)$, let us enumerate all $(3+1)$-d $M \times SO(3)$ SPTs. According to the crystalline equivalence principle [41], the classification of these SPTs is the same as the classification of $(3+1)$-d $\mathbb{Z}_2^T \times SO(3)$ SPTs, where $\mathbb{Z}_2^T$ is a time reversal symmetry. It is known that the latter are classified by $\mathbb{Z}_2^4$ (e.g., see Appendix F of Ref. [45] for the descrip-

tions of the physical properties of these SPTs). So $(3+1)$-d $M \times SO(3)$ SPTs are $\mathbb{Z}_2^4$-classified. According to Ref. [106], these SPTs can all be constructed by putting on the mirror plane of $M$ some $(2+1)$-d invertible states that have at most a $\mathbb{Z}_2 \times SO(3)$ symmetry (note that this $\mathbb{Z}_2$ symmetry does not reverse the spacetime orientation). The $(2+1)$-d $\mathbb{Z}_2 \times SO(3)$ SPTs are classified by $H^3(\mathbb{Z}_2 \times SO(3), U(1)) = \mathbb{Z}_2 \times \mathbb{Z} \times \mathbb{Z}_2$, where the Kunneth formula is used in calculating this classification. It is easy to read off the physical meaning of the root states of these $(2+1)$-d SPTs: one $\mathbb{Z}_2$ factor represents SPTs protected purely by the $Z_2$ symmetry, the $\mathbb{Z}$ factor represents spin quantum Hall states [107], which are SPTs protected purely by the $SO(3)$ symmetry and has a TPF given by $SO(3)$ Chern-Simons theories, and the other $\mathbb{Z}_2$ factor must represent an SPT protected by both $Z_2$ and $SO(3)$. The decorated-domain-wall method [108] allows us to construct the SPT by decorating the $\mathbb{Z}_2$ domain walls with a Haldane chain.

All the $(2+1)$-d $\mathbb{Z}_2 \times SO(3)$ SPTs can be inserted into the mirror plane of $M$ to construct a $(3+1)$-d $M \times SO(3)$ SPT, and one can also insert an $E_8$ state to the mirror plane. In total, these give the $\mathbb{Z}_2^4$ classification of $(3+1)$-d $M \times SO(3)$ SPTs. Note inserting a $\mathbb{Z}_2$ SPT or an $E_8$ state to the mirror plane results in a $(3+1)$-d SPT protected by $M$ only, so these states will not have a TPF $\exp(i\pi \int (w_1^{TM})^2 w_2^{SO(3)})$, which shows that this SPT requires both $M$ and $SO(3)$ for protection. Now we are only left with the cases where the bosonic spin quantum Hall state and/or the state constructed from decorated domain wall is inserted into the mirror plane. To understand the physical properties of these states, we can refer to the corresponding $(3+1)$-d $\mathbb{Z}_2^T \times SO(3)$ SPTs. If a spin quantum Hall state is decorated into the time reversal domain wall, the resulting state will have fermionic $SO(3)$ monopoles. Using the correspondence between $\mathbb{Z}_2^T \times SO(3)$ SPTs and $M \times SO(3)$ SPTs, this indicates that if the spin quantum Hall state is inserted into the mirror plane, the $SO(3)$ monopole will also be fermionic. However, the TPF $\exp(i\pi \int (w_1^{TM})^2 w_2^{SO(3)})$ means that the $SO(3)$ monopole is a boson (but carries nontrivial fractionalization pattern of the $M$ symmetry). This means that the $(3+1)$-d SPT of interest must be obtained from inserting to the mirror plane the $(2+1)$-d SPT constructed from decorated domain wall.

In fact, one can explicitly demonstrate that a $(3+1)$-d $M \times SO(3)$ SPT constructed in this way indeed has an $SO(3)$ monopole carring the nontrivial fractionalization pattern of the $M$ symmetry. To this end, it suffices to show a simpler version of this statement: suppose we break the $SO(3)$ symmetry in this SPT to $U(1)$, the $U(1)$ monopole in the resulting state will carry the nontrivial fractionalization pattern of $M$. This statement can be explicitly shown using the method in Ref. [46] (see Appendix B therein). This also means that upon this symmetry breaking, the resulting $M \times U(1)$ symmetric state is a nontrivial SPT. According to the general discussion in Sec. 2.2, this implies that $\alpha_{\text{non-LSM}}$ is unrelated to LSM constraints of interest.

The above discussion concerns about the case where $G_{int} = SO(3)$. Now we argue that for any $G_{int}$ with $\mathbb{Z}_2^k$-classified PR, $\alpha_{\text{non-LSM}}$ can be triggered in a $(3+1)$-d $M \times G_{int}$ SPT constructed in a way similar to the above, and all we need to modify is to replace the Haldane chain decorated into the $\mathbb{Z}_2$ domain wall by a $(1+1)$-d $G_{int}$ SPT. To this end, it suffices to show that the TPF of this $(3+1)$-d $M \times G_{int}$ SPT is $e^{i\pi \int (w_1^{TM})^2 \eta}$, where $\eta \in H^2(G_{int}, \mathbb{Z}_2)$ and $e^{i\pi \int \eta}$ is the TPF of the $(1+1)$-d $G_{int}$ SPT. This can be shown by noting i) this SPT is its own inverse, and ii) this construction works for all such $G_{int}$. Then an argument very similar to that in Appendix B suggests that the TPF of this SPT can indeed be written as $e^{i\pi \int (w_1^{TM})^2 \eta}$.

# E Group Cohomology and $\mathbb{Z}_2$ Cohomology ring of wallpaper groups

In this appendix, we list the $\mathbb{Z}_2$ cohomology rings of all 17 wallpaper groups. The calculation is done with the help of spectral sequence. See Appendix A.3 for some brief mathematical introduction of the relevant concepts, and Refs. [99–101] for more details. It turns out that for all wallpaper groups $G_s$ except $p4g$, the cohomology ring can be written as

$$H^*(G_s, \mathbb{Z}_2) = \mathbb{Z}_2[A_\bullet, \cdots, B_\bullet, \cdots]/\text{relations}, \tag{95}$$

with $A_\bullet$ and $B_\bullet$ the generators belonging to $H^1(G_s, \mathbb{Z}_2)$ and $H^2(G_s, \mathbb{Z}_2)$, respectively. Subscripts "$\bullet$" are the names of generators which differ for different $G_s$, and their meanings will often be clear in the context. As a result, all elements of the cohomology ring $H^*(G_s, \mathbb{Z}_2)$ can be written as cup product of the generators $A_\bullet$ and $B_\bullet$, but there are some relations that dictate that certain sums of cup products actually yield a trivial cohomology element. We will present all elements of $H^1(G_s, \mathbb{Z}_2)$ and $H^2(G_s, \mathbb{Z}_2)$ together with their representative cochains, as well as the complete set of the relations. This encodes the full information of the cohomology ring $H^*(G_s, \mathbb{Z}_2)$. The situation for $p4g$ is similar, but we need an extra degree-3 generator $C \in H^3(p4g, \mathbb{Z}_2)$, which, together with the generators in $H^1(p4g, \mathbb{Z}_2)$ and $H^2(p4g, \mathbb{Z}_2)$, forms a complete set of generators of $H^*(p4g, \mathbb{Z}_2)$.

For later usage, we define a set of functions that take integers as their arguments:

$$P(x) = \begin{cases} 1, & x \text{ is odd} \\ 0, & x \text{ is even} \end{cases}, \ P_c(x) = 1 - P(x), \ Q(x) = (-1)^x,$$
$$[x]_a = \{y = x \ (\text{mod } a) | 0 \leqslant y < a\}, \ P_{ab}(x) = \begin{cases} 1, & x = b \ (\text{mod } a), \\ 0, & \text{otherwise}. \end{cases} \tag{96}$$

- Wallpaper group 1: $p1$

  This group is generated by $T_1$ and $T_2$, two independent translations which are commutative,

  $$T_1 T_2 = T_2 T_1. \tag{97}$$

  An arbitrary group element in $p1$ can be written as $g = T_1^x T_2^y$, with $x, y \in \mathbb{Z}$. For $g_1 = T_1^{x_1} T_2^{y_1}$ and $g_2 = T_1^{x_2} T_2^{y_2}$, the group multiplication rule gives

  $$g_1 g_2 = T_1^{x_1 + x_2} T_2^{y_1 + y_2}. \tag{98}$$

  The $\mathbb{Z}_2$ cohomology ring of $p1$ is

  $$\mathbb{Z}_2[A_x, A_y]/(A_x^2 = 0, \ A_y^2 = 0). \tag{99}$$

  Here $H^1(p1, \mathbb{Z}_2) = \mathbb{Z}_2^2$, with generators $\xi_1 = A_x$, $\xi_2 = A_y$, which have representative cochains,

  $$\xi_1(g) = x, \qquad \xi_2(g) = y. \tag{100}$$

  $H^2(p1, \mathbb{Z}_2) = \mathbb{Z}_2$, with generators $\lambda_1 = A_x A_y$, which have representative cochains,

  $$\lambda_1(g_1, g_2) = y_1 x_2. \tag{101}$$

  Indeed $\lambda_1$ generates the LSM constraint.

- Wallpaper group 2: $p2$

  This group is generated by $T_1$, $T_2$ and $C_2$, two independent translations and a $C_2$ rotational symmetry, with the following relations among generators

$$C_2^2 = 1, \qquad C_2 T_1 C_2 = T_1^{-1}, \qquad C_2 T_2 C_2 = T_2^{-1}, \qquad T_1 T_2 = T_2 T_1. \tag{102}$$

An arbitrary group element in $p2$ can be written as $g = T_1^x T_2^y C_2^c$, with $x, y \in \mathbb{Z}$ and $c \in \{0, 1\}$. For $g_1 = T_1^{x_1} T_2^{y_1} C_2^{c_1}$ and $g_2 = T_1^{x_2} T_2^{y_2} C_2^{c_2}$, the group multiplication rule gives

$$g_1 g_2 = T_1^{x_1 + Q(c_1)x_2} T_2^{y_1 + Q(c_1)y_2} C_2^{P(c_1 + c_2)}. \tag{103}$$

The $\mathbb{Z}_2$ cohomology ring of $p2$ is

$$\mathbb{Z}_2[A_x, A_y, A_c]/(A_x^2 = A_x A_c, \; A_y^2 = A_y A_c). \tag{104}$$

Here $H^1(p2, \mathbb{Z}_2) = \mathbb{Z}_2^3$, with generators $\xi_1 = A_x$, $\xi_2 = A_y$, $\xi_3 = A_c$, which have representative cochains,

$$\xi_1(g) = x, \qquad \xi_2(g) = y, \qquad \xi_3(g) = c. \tag{105}$$

$H^2(p2, \mathbb{Z}_2) = \mathbb{Z}_2^4$, with generators $\lambda_1 = (A_x + A_c)(A_y + A_c)$, $\lambda_2 = A_x(A_y + A_c)$, $\lambda_3 = (A_x + A_c)A_y$, $\lambda_4 = A_x A_y$, which have representative cochains,

$$\begin{aligned}
\lambda_1(g_1, g_2) &= y_1 x_2 + c_1(x_2 + y_2 + c_2), \\
\lambda_2(g_1, g_2) &= (y_1 + c_1)x_2, \\
\lambda_3(g_1, g_2) &= y_1 x_2 + c_1 y_2, \\
\lambda_4(g_1, g_2) &= y_1 x_2.
\end{aligned} \tag{106}$$

The generators are chosen so that they have a 1-1 correspondence with topological invariants presented in Appendix F. There we will also see that all of them, $\lambda_1, \lambda_2, \lambda_3, \lambda_4$, generate LSM constraints.

- Wallpaper group 3: $pm$

  This group is generated by $T_1$, $T_2$ and $M$, where $T_1$ and $T_2$ are translations with perpendicular translation vectors, and $M$ is a mirror symmetry such that

$$M^2 = 1, \qquad M T_1 M = T_1^{-1}, \qquad M T_2 M = T_2, \qquad T_1 T_2 = T_2 T_1. \tag{107}$$

An arbitrary element in $pm$ can be written as $g = T_1^x T_2^y M^m$, with $x, y \in \mathbb{Z}$ and $m \in \{0, 1\}$. For $g_1 = T_1^{x_1} T_2^{y_1} M^{m_1}$ and $g_2 = T_1^{x_2} T_2^{y_2} M^{m_2}$, the group multiplication rule gives

$$g_1 g_2 = T_1^{x_1 + Q(m_1)x_2} T_2^{y_1 + y_2} M^{P(m_1 + m_2)}. \tag{108}$$

The $\mathbb{Z}_2$ cohomology ring of $pm$ is

$$\mathbb{Z}_2[A_x, A_y, A_m]/(A_x^2 = A_x A_m, \; A_y^2 = 0). \tag{109}$$

Here $H^1(pm, \mathbb{Z}_2) = \mathbb{Z}_2^3$, with generators $\xi_1 = A_x$, $\xi_2 = A_y$, $\xi_3 = A_m$, which have representative cochains,

$$\xi_1(g) = x, \qquad \xi_2(g) = y, \qquad \xi_3(g) = m. \tag{110}$$

$H^2(pm, \mathbb{Z}_2) = \mathbb{Z}_2^4$, with generators $\lambda_1 = (A_x + A_m)A_y$, $\quad \lambda_2 = A_xA_y$, $\quad \lambda_3 = (A_x + A_m)A_m$, $\lambda_4 = A_xA_m$, which have representative cochains,

$$
\begin{aligned}
\lambda_1(g_1, g_2) &= y_1x_2 + m_1y_2, \\
\lambda_2(g_1, g_2) &= y_1x_2, \\
\lambda_3(g_1, g_2) &= m_1(x_2 + m_2), \\
\lambda_4(g_1, g_2) &= m_1x_2.
\end{aligned}
\tag{111}
$$

In Appendix F, we will see that $\lambda_1, \lambda_2$ generate LSM constraints, while $\lambda_3, \lambda_4$ correspond to non-LSM fractionalization patterns.

- Wallpaper group 4: $pg$

  This group is generated by $T_1$ and $G$, where $T_1$ is a translation and $G$ is a glide reflection, such that

  $$
  G^{-1}T_1G = T_1^{-1}.
  \tag{112}
  $$

  Note that $G^2$ is a translation along the direction perpendicular to the translation vector of $T_1$. An arbitrary element in $pg$ can be written as $g = T_1^x G^s$, with $x, s \in \mathbb{Z}$. For $g_1 = T_1^{x_1}G^{s_1}$ and $g_2 = T_1^{x_2}G^{s_2}$, the group multiplication rule gives

  $$
  g_1g_2 = T_1^{x_1 + Q(s_1)x_2}G^{s_1+s_2}.
  \tag{113}
  $$

  The $\mathbb{Z}_2$ cohomology ring of $pg$ is

  $$
  \mathbb{Z}_2[A_x, A_s]/(A_x^2 = A_xA_s, \ A_s^2 = 0).
  \tag{114}
  $$

  Here $H^1(pg, \mathbb{Z}_2) = \mathbb{Z}_2^2$, with generators $\xi_1 = A_x$, $\quad \xi_2 = A_s$, which have representative cochains,

  $$
  \xi_1(g) = x, \qquad \xi_2(g) = s.
  \tag{115}
  $$

  $H^2(pg, \mathbb{Z}_2) = \mathbb{Z}_2$, with generators $\lambda_1 = A_xA_s$, which have representative cochains,

  $$
  \lambda_1(g_1, g_2) = s_1x_2.
  \tag{116}
  $$

  In Appendix F, we will see that $\lambda_1$ generates LSM constraints.

- Wallpaper group 5: $cm$

  This group is generated by $T_1$, $T_2$ and $M$, two independent translations and a mirror symmetry whose mirror axis bisects the translation vectors of $T_1$ and $T_2$. They satisfy

  $$
  M^2 = 1, \qquad MT_1M = T_2, \qquad MT_2M = T_1, \qquad T_1T_2 = T_2T_1.
  \tag{117}
  $$

  An arbitrary element of $cm$ can be written as $g = T_1^xT_2^yM^m$, with $x, y \in \mathbb{Z}$ and $m \in \{0, 1\}$. For $g_1 = T_1^{x_1}T_2^{y_1}M^{m_1}$ and $g_2 = T_1^{x_2}T_2^{y_2}M^{m_2}$, the group multiplication rule gives

  $$
  g_1g_2 = T_1^{x_1 + P_c(m_1)x_2 + P(m_1)y_2}T_2^{y_1 + P_c(m_1)y_2 + P(m_1)x_2}M^{P(m_1+m_2)}.
  \tag{118}
  $$

  The $\mathbb{Z}_2$ cohomology ring of $cm$ is

  $$
  \mathbb{Z}_2[A_{x+y}, A_m, B_{xy}]/(A_{x+y}^2 = 0, \ A_{x+y}A_m = 0, \ B_{xy}A_{x+y} = 0, \ B_{xy}^2 = 0).
  \tag{119}
  $$

Here $H^1(cm, \mathbb{Z}_2) = \mathbb{Z}_2^2$, with two generators $\xi_1 = A_{x+y}$, $\xi_2 = A_m$, which have representative cochains

$$\xi_1(g) = x + y, \qquad \xi_2(g) = m. \tag{120}$$

$H^2(cm, \mathbb{Z}_2) = \mathbb{Z}_2^2$, with generators $\lambda_1 = B_{xy}$, $\lambda_2 = A_m^2$, which have representative cochains

$$
\begin{aligned}
\lambda_1(g_1, g_2) &= P_c(m_1) y_1 x_2 + P(m_1) y_2 (x_2 + y_1), \\
\lambda_2(g_1, g_2) &= m_1 m_2.
\end{aligned}
\tag{121}
$$

In Appendix F, we will see that $\lambda_1$ generates LSM constraints, while $\lambda_2$ corresponds to non-LSM fractionalization patterns.

- Wallpaper group 6: *pmm*

  This group is generated by $T_1$, $T_2$, $C_2$ and $M$, two translations with perpendicular translation vectors, a $C_2$ rotation and a mirror symmetry such that

$$
\begin{aligned}
M^2 &= 1, & M C_2 M &= C_2, & M T_1 M &= T_1^{-1}, & M T_2 M &= T_2, \\
C_2^2 &= 1, & C_2 T_1 C_2 &= T_1^{-1}, & C_2 T_2 C_2 &= T_2^{-1}, & T_1 T_2 &= T_2 T_1.
\end{aligned}
\tag{122}
$$

  Note that $C_2 M$ is another mirror symmetry that flips the translation vector of $T_2$. An arbitrary element in *pmm* can be written as $g = T_1^x T_2^y C_2^c M^m$, with $x, y \in \mathbb{Z}$ and $c, m \in \{0, 1\}$. For $g_1 = T_1^{x_1} T_2^{y_1} C_2^{c_2} M^{m_1}$ and $g_2 = T_1^{x_2} T_2^{y_2} C_2^{c_2} M^{m_2}$, the group multiplication rule gives

$$g_1 g_2 = T_1^{x_1 + Q(c_1 + m_1) x_2} T_2^{y_1 + Q(c_1) y_2} C_2^{P(c_1 + c_2)} M^{P(m_1 + m_2)}. \tag{123}$$

  The $\mathbb{Z}_2$ cohomology ring of *pmm* is

$$\mathbb{Z}_2[A_x, A_y, A_c, A_m]/(A_x^2 = A_x(A_m + A_c), \ A_y^2 = A_y A_c). \tag{124}$$

  Here $H^1(pmm, \mathbb{Z}_2) = \mathbb{Z}_2^4$, with generators $\xi_1 = A_x$, $\xi_2 = A_y$, $\xi_3 = A_c$, $\xi_4 = A_m$, which have representative cochains,

$$\xi_1(g) = x, \qquad \xi_2(g) = y, \qquad \xi_3(g) = c, \qquad \xi_4(g) = m. \tag{125}$$

  $H^2(pmm, \mathbb{Z}_2) = \mathbb{Z}_2^8$, with generators $\lambda_1 = (A_x + A_c + A_m)(A_y + A_c)$, $\lambda_2 = A_x A_y$, $\lambda_3 = A_x(A_y + A_c)$, $\lambda_4 = (A_x + A_c + A_m)A_y$, $\lambda_5 = (A_x + A_c + A_m)A_m$, $\lambda_6 = (A_y + A_c)A_m$, $\lambda_7 = A_x A_m$, $\lambda_8 = A_y A_m$, which have representative cochains,

$$
\begin{aligned}
\lambda_1(g_1, g_2) &= (y_1 + c_1) x_2 + (c_1 + m_1)(y_2 + c_2), \\
\lambda_2(g_1, g_2) &= y_1 x_2, \\
\lambda_3(g_1, g_2) &= (y_1 + c_1) x_2, \\
\lambda_4(g_1, g_2) &= y_1 x_2 + (c_1 + m_1) y_2, \\
\lambda_5(g_1, g_2) &= m_1 (x_2 + c_2 + m_2), \\
\lambda_6(g_1, g_2) &= m_1 (y_2 + c_2), \\
\lambda_7(g_1, g_2) &= m_1 x_2, \\
\lambda_8(g_1, g_2) &= m_1 y_2.
\end{aligned}
\tag{126}
$$

In Appendix F, we will see that $\lambda_1, \lambda_2, \lambda_3, \lambda_4$ generate LSM constraints, while $\lambda_5, \lambda_6, \lambda_7, \lambda_8$ correspond to non-LSM fractionalization patterns.

- Wallpaper group 7: *pmg*

  This group is generated by $T_1$, $T_2$, $C_2$ and $M$, two translations with perpendicular translation vectors, a 2-fold rotation and a mirror symmetry with mirror axis parallel to the translation vector of $T_2$, and displaced from the $C_2$ rotation center by a quarter of the unit translation vector of $T_1$. They satisfy

  $$\begin{aligned} M^2 &= 1, & MC_2M &= T_1C_2, & MT_1M &= T_1^{-1}, & MT_2M &= T_2, \\ C_2^2 &= 1, & C_2T_1C_2 &= T_1^{-1}, & C_2T_2C_2 &= T_2^{-1}, & T_1T_2 &= T_2T_1. \end{aligned} \tag{127}$$

  An arbitrary element in $pmg$ can be written as $g = T_1^x T_2^y C_2^c M^m$, with $x, y \in \mathbb{Z}$ and $c, m \in \{0, 1\}$. For $g_1 = T_1^{x_1} T_2^{y_1} C_2^{c_1} M^{m_1}$ and $g_2 = T_1^{x_2} T_2^{y_2} C_2^{c_2} M^{m_2}$, the group multiplication rule gives

  $$g_1 g_2 = T_1^{x_1 + Q(c_1+m_1)x_2 + Q(c_1)c_2m_1} T_2^{y_1 + Q(c_1)y_2} C_2^{P(c_1+c_2)} M^{P(m_1+m_2)}. \tag{128}$$

  The $\mathbb{Z}_2$ cohomology ring of $pmg$ is

  $$\mathbb{Z}_2[A_y, A_c, A_m]/(A_y^2 = A_cA_y, \ A_cA_m = 0). \tag{129}$$

  Here $H^1(pmg, \mathbb{Z}_2) = \mathbb{Z}_2^3$, with generators $\xi_1 = A_y$, $\xi_2 = A_c$, $\xi_3 = A_m$, which have representative cochains,

  $$\xi_1(g) = y, \qquad \xi_2(g) = c, \qquad \xi_3(g) = m. \tag{130}$$

  $H^2(pmg, \mathbb{Z}_2) = \mathbb{Z}_2^4$, with generators $\lambda_1 = A_c(A_y + A_c)$, $\lambda_2 = A_cA_y$, $\lambda_3 = A_yA_m$, $\lambda_4 = A_m^2$, which have representative cochains,

  $$\begin{aligned} \lambda_1(g_1, g_2) &= c_1(y_2 + c_2), \\ \lambda_2(g_1, g_2) &= c_1y_2, \\ \lambda_3(g_1, g_2) &= m_1y_2, \\ \lambda_4(g_1, g_2) &= m_1m_2. \end{aligned} \tag{131}$$

  In Appendix F, we will see that $\lambda_1, \lambda_2$ generate LSM constraints, while $\lambda_3, \lambda_4$ correspond to non-LSM fractionalization patterns.

- Wallpaper group 8: *pgg*

  This group is generated by $T_1$, $T_2$, $C_2$ and $G_1$, two translations with perpendicular translation vectors, a $C_2$ rotation, and a glide reflection whose reflection axis is parallel to the translation vector of $T_2$, and displaced from the $C_2$ center by a quarter of the unit translation vector of $T_1$. They satisfy

  $$C_2^2 = 1, \quad G_1C_2G_1^{-1} = T_1T_2C_2, \quad C_2T_1C_2 = T_1^{-1}, \quad G_1T_1G_1^{-1} = T_1^{-1}, \quad G_1^2 = T_2. \tag{132}$$

  An arbitrary element in $pgg$ can be written as $g = T_1^x T_2^y C_2^c G_1^s$, with $x, y \in \mathbb{Z}$ and $c, s \in \{0, 1\}$. For $g_1 = T_1^{x_1} T_2^{y_1} C_2^{c_1} G_1^{s_1}$ and $g_2 = T_1^{x_2} T_2^{y_2} C_2^{c_2} G_1^{s_2}$, the group multiplication rule gives

  $$g_1 g_2 = T_1^{x_1 + Q(c_1+s_1)x_2 + Q(c_1)c_2s_1} T_2^{y_1 + Q(c_1)y_2 + Q(c_1)c_2s_1 + Q(c_1+c_2)s_1s_2} C_2^{P(c_1+c_2)} G_1^{P(s_1+s_2)}. \tag{133}$$

  The $\mathbb{Z}_2$ cohomology ring of $pgg$ is

  $$\mathbb{Z}_2[A_c, A_s, B_{c(x+y)}]/(A_s^2 = 0, \ A_sA_c = 0, \ A_sB_{c(x+y)} = 0, \ B_{c(x+y)}^2 = A_c^2B_{c(x+y)}). \tag{134}$$

Here $H^1(pgg, \mathbb{Z}_2) = \mathbb{Z}_2^2$, with generators $\xi_1 = A_c$, $\xi_2 = A_s$, which have representative cochains,

$$\xi_1(g) = c, \qquad \xi_2(g) = s. \tag{135}$$

$H^2(pgg, \mathbb{Z}_2) = \mathbb{Z}_2^2$, with generators $\lambda_1 = B_{c(x+y)} + A_c^2$, $\omega_2 = B_{c(x+y)}$, which have representative cochains,

$$
\begin{aligned}
\lambda_1(g_1, g_2) &= c_1(x_2 + y_2 + c_2) + (c_1 + c_2)s_1 s_2 + s_1 x_2, \\
\lambda_2(g_1, g_2) &= c_1(x_2 + y_2) + (c_1 + c_2)s_1 s_2 + s_1 x_2.
\end{aligned}
\tag{136}
$$

In Appendix F, we will see that $\lambda_1, \lambda_2$ generate LSM constraints.

- Wallpaper group 9: $cmm$

This group is generated by $T_1$, $T_2$, $C_2$ and $M$, two translations with translation vectors not perpendicular to each other, a $C_2$ rotation, and a mirror symmetry whose mirror axis bisects the translation vectors of $T_1$ and $T_2$. They satisfy

$$
\begin{aligned}
M^2 = 1, \qquad M C_2 M &= C_2, \qquad M T_1 M = T_2, \qquad M T_2 M = T_1, \\
C_2^2 = 1, \qquad C_2 T_1 C_2 &= T_1^{-1}, \qquad C_2 T_2 C_2 = T_2^{-1}, \qquad T_1 T_2 = T_2 T_1.
\end{aligned}
\tag{137}
$$

Note that $C_2 M$ is another mirror symmetry whose mirror axis bisects the translation vectors of $T_1$ and $T_2^{-1}$. An arbitrary element in $cmm$ can be written as $g = T_1^x T_2^y C_2^c M^m$, with $x, y \in \mathbb{Z}$ and $c, m \in \{0, 1\}$. For $g_1 = T_1^{x_1} T_2^{y_1} C_2^{c_1} M^{m_1}$ and $g_2 = T_1^{x_2} T_2^{y_2} C_2^{c_2} M^{m_2}$, the group multiplication rule gives

$$g_1 g_2 = T_1^{x_1 + Q(c_1)X} T_2^{y_1 + Q(c_1)Y} C_2^{P(c_1 + c_2)} M^{P(m_1 + m_2)}, \tag{138}$$

where $X$ and $Y$ are defined as

$$
\begin{aligned}
X &= P_c(m_1)x_2 + P(m_1)y_2, \\
Y &= P_c(m_1)y_2 + P(m_1)x_2.
\end{aligned}
\tag{139}
$$

The $\mathbb{Z}_2$ cohomology ring of $cmm$ is

$$
\begin{aligned}
\mathbb{Z}_2[A_{x+y}, A_c, A_m, B_{xy}]/(A_{x+y}^2 &= A_c A_{x+y}, \ A_{x+y} A_m = 0, \\
B_{xy} A_{x+y} &= 0, \ B_{xy}^2 = (A_c^2 + A_c A_m)B_{xy}).
\end{aligned}
\tag{140}
$$

Here $H^1(cmm, \mathbb{Z}_2) = \mathbb{Z}_2^3$, with generators $\xi_1 = A_{x+y}$, $\xi_2 = A_c$, $\xi_3 = A_m$, which have representative cochains,

$$\xi_1(g) = x + y, \qquad \xi_2(g) = c, \qquad \xi_3(g) = m. \tag{141}$$

$H^2(cmm, \mathbb{Z}_2) = \mathbb{Z}_2^5$, with generators $\lambda_1 = B_{xy} + A_c A_{x+y} + A_c^2 + A_m A_c$, $\lambda_2 = B_{xy}$, $\lambda_3 = A_c A_{x+y}$, $\lambda_4 = (A_c + A_m)A_m$, $\lambda_5 = A_c A_m$, which have representative cochains,

$$
\begin{aligned}
\lambda_1(g_1, g_2) &= P_c(m_1)y_1 x_2 + P(m_1)y_2(x_2 + y_1) + c_1(x_2 + y_2 + c_2 + m_2), \\
\lambda_2(g_1, g_2) &= P_c(m_1)y_1 x_2 + P(m_1)y_2(x_2 + y_1), \\
\lambda_3(g_1, g_2) &= c_1(x_2 + y_2), \\
\lambda_4(g_1, g_2) &= m_1(c_2 + m_2), \\
\lambda_5(g_1, g_2) &= m_1 c_2.
\end{aligned}
\tag{142}
$$

In Appendix F, we will that $\lambda_1, \lambda_2, \lambda_3$ generate LSM constraints, while $\lambda_4, \lambda_5$ correspond to non-LSM fractionalization pattern.

- Wallpaper group 10: $p4$

  This group is generated by $T_1$, $T_2$ and $C_4$, two translations with perpendicular translation vectors that have equal length, and a 4-fold rotational symmetry, such that

  $$C_4^4 = 1, \qquad C_4 T_1 C_4^{-1} = T_2, \qquad C_4 T_2 C_4^{-1} = T_1^{-1}, \qquad T_1 T_2 = T_2 T_1. \tag{143}$$

  An arbitrary element in $p4$ can be written as $g = T_1^x T_2^y C_4^c$, with $x, y \in \mathbb{Z}$ and $c \in \{0, 1, 2, 3\}$. For $g_1 = T_1^{x_1} T_2^{y_1} C_4^{c_1}$ and $g_2 = T_1^{x_2} T_2^{y_2} C_4^{c_2}$, the group multiplication rule gives

  $$g_1 g_2 = T_1^{x_1 + \Delta x(x_2, y_2, c_1)} T_2^{y_1 + \Delta y(x_2, y_2, c_1)} C_4^{[c_1 + c_2]_4}, \tag{144}$$

  where

  $$\Delta x(x, y, c) = \begin{cases} x, & c = 0 \\ -y, & c = 1 \\ -x, & c = 2 \\ y, & c = 3 \end{cases}, \quad \Delta y(x, y, c) = \begin{cases} y, & c = 0 \\ x, & c = 1 \\ -y, & c = 2 \\ -x, & c = 3 \end{cases}. \tag{145}$$

  The $\mathbb{Z}_2$ cohomology ring of $p4$ is

  $$\begin{aligned} \mathbb{Z}_2[A_c, A_{x+y}, B_{c^2}, B_{xy}] / \big( & A_c^2 = 0, \ A_c A_{x+y} = 0, \\ & B_{xy} A_{x+y} = B_{xy} A_c, \ B_{c^2} A_{x+y} = A_{x+y}^3 + B_{xy} A_{x+y}, \\ & B_{xy}^2 = B_{c^2} B_{xy} \big). \end{aligned} \tag{146}$$

  Here $H^1(p4, \mathbb{Z}_2) = \mathbb{Z}_2^2$, with generators $\xi_1 = A_{x+y}, \quad \xi_2 = A_c$, which have representative cochains,

  $$\xi_1(g) = x + y, \qquad \xi_2(g) = c. \tag{147}$$

  $H^2(p4, \mathbb{Z}_2) = \mathbb{Z}_2^3$, with generators $\lambda_1 = B_{xy} + A_{x+y}^2 + B_{c^2}, \quad \lambda_2 = B_{xy}, \quad \lambda_3 = A_{x+y}^2$, which have representative cochains,

  $$\begin{aligned} \lambda_1(g_1, g_2) &= \lambda_2(g_1, g_2) + \lambda_3(g_1, g_2) + \frac{[c_1]_4 + [c_2]_4 - [c_1 + c_2]_4}{4}, \\ \lambda_2(g_1, g_2) &= P_c(c_1) y_1 x_2 + P(c_1) y_2 (x_2 + y_1), \\ \lambda_3(g_1, g_2) &= P_{41}(c_1) x_2 + P_{42}(c_1)(x_2 + y_2) + P_{43}(c_1) y_2. \end{aligned} \tag{148}$$

  In Appendix F, we will see that $\lambda_1, \lambda_2, \lambda_3$ generate LSM constraints.

- Wallpaper group 11: $p4m$

  This group is generated by $T_1$, $T_2$, $C_4$ and $M$, where the first three generators have the same properties as those in $p4$, and the last generator $M$ is a mirror symmetry that flips the translation vector of $T_1$, such that

  $$\begin{aligned} M^2 = 1, \qquad & M C_4 M = C_4^{-1}, \qquad M T_1 M = T_1^{-1}, \qquad M T_2 M = T_2, \\ C_4^4 = 1, \qquad & C_4 T_1 C_4^{-1} = T_2, \qquad C_4 T_2 C_4^{-1} = T_1^{-1}, \qquad T_1 T_2 = T_2 T_1. \end{aligned} \tag{149}$$

  An arbitrary element in $p4m$ can be written as $g = T_1^x T_2^y C_4^c M^m$, with $x, y \in \mathbb{Z}$, $c \in \{0, 1, 2, 3\}$ and $m \in \{0, 1\}$. For $g_1 = T_1^{x_1} T_2^{y_1} C_4^{c_1} M^{m_1}$ and $g_2 = T_1^{x_2} T_2^{y_2} C_4^{c_2} M^{m_2}$, the group multiplication rule gives

  $$g_1 g_2 = T_1^{x_1 + \Delta x(Q(m_1) x_2, y_2, c_1)} T_2^{y_1 + \Delta y(Q(m_1) x_2, y_2, c_1)} C_4^{[c_1 + Q(m_1) c_2]_4} M^{P(m_1 + m_2)}, \tag{150}$$

where $\Delta x(x,y,c)$ and $\Delta y(x,y,c)$ are defined in Eq. (145).

The $\mathbb{Z}_2$ cohomology ring of $p4m$ is

$$
\mathbb{Z}_2[A_c, A_{x+y}, A_m, B_{c^2}, B_{xy}]/\big(A_c(A_c + A_m) = 0,\ A_c A_{x+y} = 0,
$$
$$
B_{xy}A_{x+y} = B_{xy}(A_c + A_m),
$$
$$
B_{c^2}A_{x+y} = A_{x+y}^3 + A_m A_{x+y}^2 + B_{xy}A_{x+y},  \tag{151}
$$
$$
B_{xy}^2 = B_{c^2}B_{xy}\big).
$$

Here $H^1(p4m, \mathbb{Z}_2) = \mathbb{Z}_2^3$, with generators $\xi_1 = A_{x+y}$, $\quad \xi_2 = A_c$, $\quad \xi_3 = A_m$, which have representative cochains,

$$
\xi_1(g) = x + y, \qquad \xi_2(g) = c, \qquad \xi_3(g) = m.  \tag{152}
$$

$H^2(p4m, \mathbb{Z}_2) = \mathbb{Z}_2^6$, with generators $\lambda_1 = B_{xy} + A_{x+y}(A_{x+y} + A_m) + B_{c^2}$, $\quad \lambda_2 = B_{xy}$, $\lambda_3 = A_{x+y}(A_{x+y} + A_m)$, $\quad \lambda_4 = A_m(A_m + A_{x+y} + A_c)$, $\quad \lambda_5 = A_m A_{x+y}$, $\quad \lambda_6 = A_m A_c$, which have representative cochains,

$$
\lambda_1(g_1,g_2) = \lambda_2(g_1,g_2) + \lambda_3(g_1,g_2) + \frac{[c_1]_4 + Q(m_1)[c_2]_4 - [c_1 + Q(m_1)c_2]_4}{4},
$$
$$
\lambda_2(g_1,g_2) = P_c(c_1)y_1 x_2 + P(c_1)y_2(x_2 + y_1),
$$
$$
\lambda_3(g_1,g_2) = P_{41}(c_1)x_2 + P_{42}(c_1)(x_2 + y_2) + P_{43}(c_1)y_2 + m_1 y_2,  \tag{153}
$$
$$
\lambda_4(g_1,g_2) = m_1(x_2 + y_2 + c_2 + m_2),
$$
$$
\lambda_5(g_1,g_2) = m_1(x_2 + y_2),
$$
$$
\lambda_6(g_1,g_2) = m_1 c_2.
$$

In Appendix F, we will see that $\lambda_1, \lambda_2, \lambda_3$ generate LSM constraints, while $\lambda_4, \lambda_5, \lambda_6$ correspond to non-LSM fractionalization patterns.

- Wallpaper group 12: $p4g$

  This group is generated by $T_1$, $T_2$, $C_4$ and $G$. The first three generators have the same properties as those in $p4$, and the last generator $G$ is a glide reflection whose reflection axis passes through the rotation center of $C_4$ and bisects the translation vectors of $T_1$ and $T_2$, such that

$$
C_4^4 = 1, \qquad C_4 T_1 C_4^{-1} = T_2, \qquad C_4 T_2 C_4^{-1} = T_1^{-1}, \qquad T_1 T_2 = T_2 T_1,  \tag{154}
$$
$$
G^2 = T_1 T_2, \qquad G T_1 G^{-1} = T_2, \qquad G T_2 G^{-1} = T_1, \qquad G C_4 G^{-1} = T_2 C_4^{-1}.  \tag{155}
$$

Note that there is also a mirror symmetry $M = T_1^{-1}G$. An arbitrary element in $p4g$ can be written as $g = T_1^x T_2^y C_4^c G^s$, with $x,y \in \mathbb{Z}$, $c \in \{0,1,2,3\}$ and $s \in \{0,1\}$. For $g_1 = T_1^{x_1} T_2^{y_1} C_4^{c_1} G^{s_1}$ and $g_2 = T_1^{x_2} T_2^{y_2} C_4^{c_2} G^{s_2}$, the group multiplication rule gives

$$
g_1 g_2 = T_1^{x_1 + \Delta x(X,Y,c_1)} T_2^{y_1 + \Delta y(X,Y,c_1)} C_4^{[c_1 + Q(s_1)c_2]_4} G^{P(s_1 + s_2)},  \tag{156}
$$

where $\Delta x(x,y,c)$ and $\Delta y(x,y,c)$ are defined in Eq. (145), and

$$
X = P_c(s_1)x_2 + P(s_1)y_2 + \big(P_{42}(c_2) + P_{43}(c_2)\big)s_1 + \Delta x(s_1 s_2, s_1 s_2, [Q(s_1)c_2]_4),
$$
$$
Y = P_c(s_1)y_2 + P(s_1)x_2 + \big(P_{41}(c_2) + P_{42}(c_2)\big)s_1 + \Delta y(s_1 s_2, s_1 s_2, [Q(s_1)c_2]_4).  \tag{157}
$$

The $\mathbb{Z}_2$ cohomology ring of $p4g$ is

$$
\begin{aligned}
\mathbb{Z}_2[A_c, A_s, B_{c^2}, B_{c(x+y)}, C_{c^2(x+y)}]/\big(A_c^2 = 0\,, & A_c A_s = 0\,, \\
& A_c B_{c(x+y)} = 0,\ A_s B_{c(x+y)} = A_s B_{c^2}\,, \\
& B_{c(x+y)}^2 = B_{c^2} B_{c(x+y)},\ A_c C_{c^2(x+y)} = 0\,, \\
& B_{c(x+y)} C_{c^2(x+y)} = B_{c^2} C_{c^2(x+y)}\,, \\
& C_{c^2(x+y)}^2 = B_{c(x+y)}^3 + A_s B_{c^2} C_{c^2(x+y)}\big).
\end{aligned}
\tag{158}
$$

Here $H^1(p4g, \mathbb{Z}_2) = \mathbb{Z}_2^2$, with generators $\xi_1 = A_c, \quad \xi_2 = A_s$, which have representative cochains,

$$
\xi_1(g) = c\,, \qquad \xi_2(g) = s\,.
\tag{159}
$$

$H^2(p4g, \mathbb{Z}_2) = \mathbb{Z}_2^3$, with generators $\lambda_1 = B_{c(x+y)} + B_{c^2}, \quad \lambda_2 = B_{c(x+y)}, \quad \lambda_3 = A_s^2$, which have representative cochains,

$$
\begin{aligned}
\lambda_1(g_1, g_2) &= \lambda_2(g_1, g_2) + \frac{[c_1]_4 + Q(s_1)[c_2]_4 - [c_1 + Q(s_1)c_2]_4}{4}\,, \\
\lambda_2(g_1, g_2) &= P_{40}(c_1)s_1(P_{43}(c_2) + (c_1 - c_2)s_2) \\
&\quad + P_{41}(c_1)[P_c(s_1)y_2 + s_1(x_2 + 1 - P_{40}(c_2) + (c_1 - c_2)s_2)] \\
&\quad + P_{42}(c_1)[P_c(s_1)(x_2 + y_2) + s_1(x_2 + y_2 + P_{41}(c_2) + (c_1 - c_2)s_2)] \\
&\quad + P_{43}(c_1)[P_c(s_1)x_2 + s_1(y_2 + P_{42}(c_2) + (c_1 - c_2)s_2)], \\
\lambda_3(g_1, g_2) &= s_1 s_2\,.
\end{aligned}
\tag{160}
$$

In Appendix F, we will see that $\lambda_1, \lambda_2$ generate LSM constraints, while $\lambda_3$ corresponds to non-LSM fractionalization patterns.

Pay attention that there is a degree-3 generator $C_{c^2(x+y)} \in H^3(p4g, \mathbb{Z}_2)$. We do not have the explicit form of its representative cochain, but it can be determined by its pullback to the subgroup $p4$ generated by $T_1, T_2, C_4$, which is $A_{x+y}^3$, as well as its pullback to the subgroup $cmm$ generated by $T_1, T_2, T_2 C_2, T_1^{-1} G$, which is $A_{x+y}^3 + A_c^3 + B_{xy} A_m + A_c^2 A_m$.

- Wallpaper group 13: $p3$

  This group is generated by $T_1, T_2$ and $C_3$, two translations with translation vectors that have the same length and an angle of $2\pi/3$, and a 3-fold rotational symmetry, such that

  $$
  C_3^3 = 1, \qquad C_3 T_1 C_3^{-1} = T_2, \qquad C_3 T_2 C_3^{-1} = T_1^{-1} T_2^{-1}, \qquad T_1 T_2 = T_2 T_1.
  \tag{161}
  $$

  An arbitrary element in $p3$ can be written as $g = T_1^x T_2^y C_3^c$, with $x, y \in \mathbb{Z}$ and $c \in \{0, 1, 2\}$. For $g_1 = T_1^{x_1} T_2^{y_1} C_3^{c_1}$ and $g_2 = T_1^{x_2} T_2^{y_2} C_3^{c_2}$, the group multiplication rule gives

  $$
  g_1 g_2 = T_1^{x_1 + \Delta x(x_2, y_2, c_1)} T_2^{y_1 + \Delta y(x_2, y_2, c_1)} C_3^{[c_1 + c_2]_3}\,,
  \tag{162}
  $$

  where

  $$
  \Delta x(x, y, c) = \begin{cases} x, & c = 0 \\ -y, & c = 1 \\ -x + y, & c = 2 \end{cases}, \quad \Delta y(x, y, c) = \begin{cases} y, & c = 0 \\ x - y, & c = 1 \\ -x, & c = 2 \end{cases}.
  \tag{163}
  $$

  The $\mathbb{Z}_2$ cohomology ring of $p3$ is

  $$
  \mathbb{Z}_2[B_{xy}]/(B_{xy}^2 = 0)\,.
  \tag{164}
  $$

Here $H^1(p3, \mathbb{Z}_2) = 0$, while $H^2(p3, \mathbb{Z}_2) = \mathbb{Z}_2$, with generator $\lambda_1 = B_{xy}$, which have representative cochain,

$$
\begin{aligned}
\lambda_1(g_1, g_2) =& P_{30}(c_1)y_1 x_2 + P_{31}(c_1)\left(\frac{y_2(y_2-1)}{2} + y_2(x_2+y_1)\right) \\
&+ P_{32}(c_1)\left(\frac{x_2(x_2-1)}{2} + y_1 x_2 + y_2(x_2+y_1)\right).
\end{aligned}
\tag{165}
$$

In Appendix F, we will see that $\lambda_1$ generates LSM constraints.

- Wallpaper group 14: $p3m1$

  This group is generated by $T_1$, $T_2$, $C_3$ and $M$, where the first three generators have the same properties as those in $p3$, and the last one is a mirror symmetry whose mirror axis passes through the $C_3$ center, and is perpendicular to the angle that bisects the two translation vectors of $T_1$ and $T_2$, such that

  $$
  \begin{aligned}
  M^2 &= 1, & MC_3 M &= C_3^{-1}, & MT_1 M &= T_2^{-1}, & MT_2 M &= T_1^{-1}, \\
  C_3^3 &= 1, & C_3 T_1 C_3^{-1} &= T_2, & C_3 T_2 C_3^{-1} &= T_1^{-1}T_2^{-1}, & T_1 T_2 &= T_2 T_1.
  \end{aligned}
  \tag{166}
  $$

  An arbitrary element in $p3m1$ can be written as $g = T_1^x T_2^y C_3^c M^m$, with $x, y \in \mathbb{Z}$, $c \in \{0,1,2\}$ and $M \in \{0,1\}$. For $g_1 = T_1^{x_1} T_2^{y_1} C_3^{c_1} M^{m_1}$ and $g_2 = T_1^{x_2} T_2^{y_2} C_3^{c_2} M^{m_2}$, the group multiplication rule gives

  $$
  g_1 g_2 = T_1^{x_1 + \Delta x(X,Y,c_1)} T_2^{y_1 + \Delta y(X,Y,c_1)} C_3^{[c_1 + Q(m_1)c_2]_3} M^{P(m_1+m_2)},
  \tag{167}
  $$

  where $\Delta x(x,y,c)$ and $\Delta y(x,y,c)$ are defined in Eq. (163), and

  $$
  \begin{aligned}
  X &= P_c(m_1)x_2 - P(m_1)y_2, \\
  Y &= P_c(m_1)y_2 - P(m_1)x_2.
  \end{aligned}
  \tag{168}
  $$

  The $\mathbb{Z}_2$ cohomology ring of $p3m1$ is

  $$
  \mathbb{Z}_2[A_m, B_{xy}]/(B_{xy}^2 = 0).
  \tag{169}
  $$

  Here $H^1(p3m1, \mathbb{Z}_2) = 0$, with generator $\xi_1 = A_m$, which have representative cochain,

  $$
  \xi_1(g) = m.
  \tag{170}
  $$

$H^2(p3m1, \mathbb{Z}_2) = \mathbb{Z}_2^2$, with generators $\lambda_1 = B_{xy}$, $\lambda_2 = A_m^2$, which have representative cochains,

$$
\begin{aligned}
\lambda_1(g_1, g_2) =& P_{30}(c_1)[P_c(m_1)y_1 x_2 + m_1 y_2(x_2+y_1)] \\
&+ P_{31}(c_1)\Bigg[P_c(m_1)\left(\frac{y_2(y_2-1)}{2} + y_2(x_2+y_1)\right) \\
&\qquad + m_1\left(\frac{x_2(x_2+1)}{2} + y_1 x_2\right)\Bigg] \\
&+ P_{32}(c_1)\Bigg[P_c(m_1)\left(\frac{x_2(x_2-1)}{2} + y_1 x_2 + y_2(x_2+y_1)\right) \\
&\qquad + m_1\left(\frac{y_2(y_2+1)}{2} + y_1(x_2+y_2)\right)\Bigg], \\
\lambda_2(g_1, g_2) =& m_1 m_2.
\end{aligned}
\tag{171}
$$

In Appendix F, we will see that $\lambda_1$ generates LSM constraints, while $\lambda_2$ corresponds to non-LSM fractionalization pattern.

- Wallpaper group 15: $p31m$

  This group is generated by $T_1$, $T_2$, $C_3$ and $M$, where the first three generators have the same properties as those in $p3$ and $p3m1$, and the last one is a mirror symmetry whose mirror axis passes through the $C_3$ center and bisects the translation vectors of $T_1$ and $T_2$, such that

$$
\begin{aligned}
M^2 &= 1, & MC_3M &= C_3^{-1}, & MT_1M &= T_2, & MT_2M &= T_1, \\
C_3^3 &= 1, & C_3T_1C_3^{-1} &= T_2, & C_3T_2C_3^{-1} &= T_1^{-1}T_2^{-1}, & T_1T_2 &= T_2T_1.
\end{aligned}
\tag{172}
$$

An arbitrary element in $p31m$ can be written as $g = T_1^x T_2^y C_3^c M^m$, with $x, y \in \mathbb{Z}$, $c \in \{0,1,2\}$ and $M \in \{0,1\}$. For $g_1 = T_1^{x_1} T_2^{y_1} C_3^{c_1} M^{m_1}$ and $g_2 = T_1^{x_2} T_2^{y_2} C_3^{c_2} M^{m_2}$, the group multiplication rule gives

$$
g_1 g_2 = T_1^{x_1 + \Delta x(X,Y,c_1)} T_2^{y_1 + \Delta y(X,Y,c_1)} C_3^{[c_1 + Q(m_1)c_2]_3} M^{P(m_1+m_2)},
\tag{173}
$$

where $\Delta x(x,y,c)$ and $\Delta y(x,y,c)$ are defined in Eq. (163), and

$$
\begin{aligned}
X &= P_c(m_1)x_2 + P(m_1)y_2, \\
Y &= P_c(m_1)y_2 + P(m_1)x_2.
\end{aligned}
\tag{174}
$$

The $\mathbb{Z}_2$ cohomology ring of $p31m$ is

$$
\mathbb{Z}_2[A_m, B_{xy}]/(B_{xy}^2 = 0).
\tag{175}
$$

Here $H^1(p31m, \mathbb{Z}_2) = 0$, with generator $\xi_1 = A_m$, which have representative cochain,

$$
\xi_1(g) = m.
\tag{176}
$$

$H^2(p31m, \mathbb{Z}_2) = \mathbb{Z}_2^2$, with generator $\lambda_1 = B_{xy}$, $\lambda_2 = A_m^2$, which have representative cochains,

$$
\begin{aligned}
\lambda_1(g_1, g_2) &= P_{30}(c_1)\left[ P_c(m_1)y_1 x_2 + m_1 y_2(x_2 + y_1) \right] \\
&+ P_{31}(c_1)\left[ P_c(m_1)\left( \frac{y_2(y_2+1)}{2} + x_2 + y_2(x_2 + y_1) \right) \right. \\
&\left. + m_1\left( \frac{x_2(x_2+1)}{2} + y_2 + y_1 x_2 \right) \right] \\
&+ P_{32}(c_1)\left[ P_c(m_1)\left( \frac{x_2(x_2-1)}{2} + y_2 + y_1 x_2 + y_2(x_2 + y_1) \right) \right. \\
&\left. + m_1\left( \frac{y_2(y_2-1)}{2} + x_2 + y_1(x_2 + y_2) \right) \right] \\
\lambda_2(g_1, g_2) &= m_1 m_2.
\end{aligned}
\tag{177}
$$

In Appendix F, we will see that $\lambda_1$ generates LSM constraints while $\lambda_2$ corresponds to non-LSM fractionalization pattern.

- Wallpaper group 16: $p6$

  This group is generated by $T_1$, $T_2$ and $C_6$, two translations with translation vectors that have the same length and an angle of $2\pi/3$, and a 6-fold rotational symmetry, such that

$$
C_6^6 = 1, \quad C_6 T_1 C_6^{-1} = T_1 T_2, \quad C_6 T_2 C_6^{-1} = T_1^{-1}, \quad T_1 T_2 = T_2 T_1.
\tag{178}
$$

An arbitrary element in $p6$ can be written as $g = T_1^x T_2^y C_6^c$, with $x, y \in \mathbb{Z}$ and $c \in \{0, 1, 2, 3, 4, 5\}$. For $g_1 = T_1^{x_1} T_2^{y_1} C_6^{c_1}$ and $g_2 = T_1^{x_2} T_2^{y_2} C_6^{c_2}$, the group multiplication rule gives

$$g_1 g_2 = T_1^{x_1 + \Delta x(x_2, y_2, c_1)} T_2^{y_1 + \Delta y(x_2, y_2, c_1)} C_6^{[c_1 + c_2]_6}, \tag{179}$$

where

$$\Delta x(x, y, c) = \begin{cases} x, & c = 0 \\ x - y, & c = 1 \\ -y, & c = 2 \\ -x, & c = 3 \\ -x + y, & c = 4 \\ y, & c = 5 \end{cases}, \quad \Delta y(x, y, c) = \begin{cases} y, & c = 0 \\ x, & c = 1 \\ x - y, & c = 2 \\ -y, & c = 3 \\ -x, & c = 4 \\ -x + y, & c = 5 \end{cases}. \tag{180}$$

The $\mathbb{Z}_2$ cohomology ring of $p6$ is

$$\mathbb{Z}_2[A_c, B_{xy}]/(B_{xy}^2 = A_c^2 B_{xy}). \tag{181}$$

Here $H^1(p6, \mathbb{Z}_2) = \mathbb{Z}_2$, with generator $\xi_1 = A_c$, which have representative cochain,

$$\xi_1(g) = c. \tag{182}$$

$H^2(p6, \mathbb{Z}_2) = \mathbb{Z}_2^2$, with generators $\lambda_1 = B_{xy} + A_c^2$, $\lambda_2 = B_{xy}$, which have representative cochains,

$$\begin{aligned}
\lambda_1(g_1, g_2) &= \lambda_2(g_1, g_2) + + \frac{[c_1]_6 + [c_2]_6 - [c_1 + c_2]_6}{6}, \\
\lambda_2(g_1, g_2) &= P_{60}(c_1) y_1 x_2 + P_{61}(c_1) \left( \frac{x_2(x_2 - 1)}{2} + y_1 x_2 + y_2(x_2 + y_1) \right) \\
&\quad + P_{62}(c_1) \left( \frac{y_2(y_2 + 1)}{2} + x_2 + y_2(x_2 + y_1) \right) + P_{63}(c_1)(x_2 + y_2 + y_1 x_2) \\
&\quad + P_{64}(c_1) \left( \frac{x_2(x_2 - 1)}{2} + y_2 + y_1 x_2 + y_2(x_2 + y_1) \right) \\
&\quad + P_{65}(c_1) \left( \frac{y_2(y_2 + 1)}{2} + y_2(x_2 + y_1) \right).
\end{aligned} \tag{183}$$

In Appendix F, we will see that $\lambda_1, \lambda_2$ generate LSM constraints.

- Wallpaper group 17: $p6m$

  This group is generated by $T_1$, $T_2$, $C_6$ and $M$, where the first three generators have the same properties as those in $p6$, and the last one is a mirror symmetry whose mirror axis passes through the $C_6$ center and bisects $T_1$ and $T_2$, such that

  $$\begin{aligned}
  M^2 &= 1, & MC_6 M &= C_6^{-1}, & MT_1 M &= T_2, & MT_2 M &= T_1, \\
  C_6^6 &= 1, & C_6 T_1 C_6^{-1} &= T_1 T_2, & C_6 T_2 C_6^{-1} &= T_1^{-1}, & T_1 T_2 &= T_2 T_1.
  \end{aligned} \tag{184}$$

  An arbitrary element in $p6m$ can be written as $g = T_1^x T_2^y C_6^c M^m$, with $x, y \in \mathbb{Z}$, $c \in \{0, 1, 2, 3, 4, 5\}$ and $m \in \{0, 1\}$. For $g_1 = T_1^{x_1} T_2^{y_1} C_6^{c_1} M^{m_1}$ and $g_2 = T_1^{x_2} T_2^{y_2} C_6^{c_2} M^{m_2}$, the group multiplication rule gives

  $$g_1 g_2 = T_1^{x_1 + \Delta x(X, Y, c_1)} T_2^{y_1 + \Delta y(X, Y, c_1)} C_6^{[c_1 + Q(m_1) c_2]_6} M^{P(m_1 + m_2)}, \tag{185}$$

where $\Delta x(x,y,c)$ and $\Delta y(x,y,c)$ are defined in Eq. (180), and $X$ and $Y$ are defined in Eq. (174).

The $\mathbb{Z}_2$ cohomology ring of $p6m$ is

$$\mathbb{Z}_2[A_c, A_m, B_{xy}]/\left( B_{xy}^2 = (A_c^2 + A_c A_m)B_{xy} \right). \tag{186}$$

Here $H^1(p6m, \mathbb{Z}_2) = \mathbb{Z}_2^2$, with generator $\xi_1 = A_c, \quad \xi_2 = A_m$, which have representative cochains,

$$\xi_1(g) = c, \qquad \xi_2(g) = m. \tag{187}$$

$H^2(p6m, \mathbb{Z}_2) = \mathbb{Z}_2^4$, with generators $\lambda_1 = B_{xy} + A_c^2 + A_c A_m$, $\lambda_2 = B_{xy}$, $\lambda_3 = (A_c + A_m)A_m$, $\lambda_4 = A_c A_m$, which have representative cochains,

$$
\begin{aligned}
\lambda_1(g_1,g_2) &= \lambda_2(g_1,g_2) + \frac{[c_1]_6 + Q(m_1)[c_2]_6 - [c_1 + Q(m_1)c_2]_6}{6}, \\
\lambda_2(g_1,g_2) &= P_{60}(c_1)[P_c(m_1)y_1 x_2 + m_1 y_2(x_2 + y_1)] \\
&\quad + P_{61}(c_1)\left[ P_c(m_1)\left( \frac{x_2(x_2-1)}{2} + y_1 x_2 + y_2(x_2 + y_1) \right) \right. \\
&\quad \left. + m_1\left( \frac{y_2(y_2-1)}{2} + y_1(x_2 + y_2) \right) \right] \\
&\quad + P_{62}(c_1)\left[ P_c(m_1)\left( \frac{y_2(y_2+1)}{2} + x_2 + y_2(x_2 + y_1) \right) \right. \\
&\quad \left. + m_1\left( \frac{x_2(x_2+1)}{2} + y_2 + y_1 x_2 \right) \right] \\
&\quad + P_{63}(c_1)[P_c(m_1)(x_2 + y_2 + y_1 x_2) + m_1(x_2 + y_2 + y_2(x_2 + y_1))] \\
&\quad + P_{64}(c_1)\left[ P_c(m_1)\left( \frac{x_2(x_2-1)}{2} + y_2 + y_1 x_2 + y_2(x_2 + y_1) \right) \right. \\
&\quad \left. + m_1\left( \frac{y_2(y_2-1)}{2} + x_2 + y_1(x_2 + y_2) \right) \right] \\
&\quad + P_{65}(c_1)\left[ P_c(m_1)\left( \frac{y_2(y_2+1)}{2} + y_2(x_2 + y_1) \right) \right. \\
&\quad \left. + m_1\left( \frac{x_2(x_2+1)}{2} + y_1 x_2 \right) \right], \\
\lambda_3(g_1,g_2) &= m_1(c_2 + m_2), \\
\lambda_4(g_1,g_2) &= m_1 c_2.
\end{aligned}
\tag{188}
$$

In Appendix F, we will see that $\lambda_1, \lambda_2$ generate LSM constraints, while $\lambda_3, \lambda_4$ correspond to non-LSM fractionalization patterns.

# F  Topological invariants for all LSM constraints

In this appendix, for each of the 17 wallpaper groups, we present the topological invariants for all LSM constraints and topological invariants for all non-LSM fractionalization patterns. These topological invariants can all be written down by simply inspecting the IWP and/or the

mirror axes of the relevant wallpaper groups, and they correspond to various (3+1)-d $G_s \times G_{int}$ SPTs that can be constructed in a manner described in Sec. 2.2. This physics-based reasoning implies that the topological invariants we present here are complete and independent. In Appendix E, we also provide explicit expressions of the representative cochains that correspond to each of the topological invariants, which show mathematically that the topological invariants here are indeed complete and independent.

- Wallpaper group 1: $p1$

  All points in space correspond to the same IWP for $p1$. The fractionalization patterns of $p1$ are classified by $H^2(p1, \mathbb{Z}_2) = \mathbb{Z}_2$. There is only one nontrivial fractionalization pattern $T_1 T_2 = -T_2 T_1$, detected by the topological invariant

  $$\alpha_1[\omega] = \frac{\omega(T_1, T_2)}{\omega(T_2, T_1)}. \tag{189}$$

  This fractionalization pattern is related to the first of the 3 basic no-go theorems in Sec. 2.1, so it corresponds to an LSM constraint, and the classification of LSM constraints is $\mathbb{Z}_2$. It is straightforward to check that $\alpha_1[(-1)^{\lambda_1}] = -1$, where $\lambda_1$ is defined in Appendix E.

- Wallpaper group 2: $p2$

  There are 4 different IWP for $p2$, which are rotation centers for $C_2$, $T_1 C_2$, $T_2 C_2$ and $T_1 T_2 C_2$, respectively. The fractionalization patterns of $p2$ are classified by $H^2(p2, \mathbb{Z}_2) = \mathbb{Z}_2^4$. All fractionalization patterns are generated by 4 root patterns, detected by the topological invariants,

  $$\begin{aligned}
  \alpha_1[\omega] &= \frac{\omega(C_2, C_2)}{\omega(1, 1)}, \\
  \alpha_2[\omega] &= \frac{\omega(T_1 C_2, T_1 C_2)}{\omega(1, 1)}, \\
  \alpha_3[\omega] &= \frac{\omega(T_2 C_2, T_2 C_2)}{\omega(1, 1)}, \\
  \alpha_4[\omega] &= \frac{\omega(T_1 T_2 C_2, T_1 T_2 C_2)}{\omega(1, 1)},
  \end{aligned} \tag{190}$$

  corresponding to $C_2^2 = -1$, $(T_1 C_2)^2 = -1$, $(T_2 C_2)^2 = -1$ and $(T_1 T_2 C_2)^2 = -1$ respectively. All these topological invariants are related to the third of the 3 basic no-go theorems in Sec. 2.1, so they all correspond to LSM constraints, and the classification of LSM constraints is $\mathbb{Z}_2^4$. It is straightforward to check that $\alpha_i[(-1)^{\lambda_j}] = (-1)^{\delta_{ij}}$ for $i, j = 1, \ldots, 4$, where $\lambda_i$ is defined in Appendix E.

- Wallpaper group 3: $pm$

  There are 2 different IWP for $pm$, which are the mirror axes for $M$ and $T_1 M$, respectively. The fractionalization patterns of $pm$ are classified by $H^2(pm, \mathbb{Z}_2) = \mathbb{Z}_2^4$. All fractionalization patterns are generated by 4 root patterns, detected by the topological invariants

  $$\begin{aligned}
  \alpha_1[\omega] &= \frac{\omega(T_2, M)}{\omega(M, T_2)}, \\
  \alpha_2[\omega] &= \frac{\omega(T_2, T_1 M)}{\omega(T_1 M, T_2)}, \\
  \alpha_3[\omega] &= \frac{\omega(M, M)}{\omega(1, 1)}, \\
  \alpha_4[\omega] &= \frac{\omega(T_1 M, T_1 M)}{\omega(1, 1)}.
  \end{aligned} \tag{191}$$

  SciPost Phys. 13, 066 (2022)

The first two topological invariants are related to the second of the 3 basic no-go theorems, so they correspond to LSM constraints, and the classification of LSM constraints is $\mathbb{Z}_2^2$. The last two are non-LSM fractionalization patterns. It is straightforward to check that $\alpha_i[(-1)^{\lambda_j}] = (-1)^{\delta_{ij}}$ for $i, j = 1, \ldots, 4$, where $\lambda_i$ is defined in Appendix E.

- Wallpaper group 4: $pg$

  All points in space belong to one IWP of $pg$. The fractionalization patterns of $pg$ are classified by $H^2(pg, \mathbb{Z}_2) = \mathbb{Z}_2$. There is only one nontrivial fractionalization pattern, detected by the topological invariant

$$\alpha_1[\omega] = \frac{\omega(T_1 G^{-1}, T_1 G)\omega(T_1, G)}{\omega(G^{-1}, G)\omega(T_1, G^{-1})}, \tag{192}$$

  corresponding to the symmetry fractionalization pattern $G^{-1}T_1 G T_1 = -1$. This topological invariant is related to the first of the 3 basic no-go theorems, so it corresponds to an LSM constraint, and the classification of LSM constraints is $\mathbb{Z}_2$. It is straightforward to check that $\alpha_1[(-1)^{\lambda_1}] = -1$, where $\lambda_1$ is defined in Appendix E.

- Wallpaper group 5: $cm$

  The group $cm$ has one IWP, which includes points along the mirror axis of $M$. The fractionalization patterns of $cm$ are classified by $H^2(cm, \mathbb{Z}_2) = \mathbb{Z}_2^2$. All fractionalization patterns are generated by 2 root patterns, detected by the topological invariants

$$\begin{aligned}
\alpha_1[\omega] &= \frac{\omega(T_1 T_2, M)}{\omega(M, T_1 T_2)}, \\
\alpha_2[\omega] &= \frac{\omega(M, M)}{\omega(1, 1)}.
\end{aligned} \tag{193}$$

  The first topological invariant is related to the second of the 3 basic no-go theorems, so it corresponds to an LSM constraint, and the classification of LSM constraints is $\mathbb{Z}_2$. The second one is a non-LSM fractionalization pattern. It is straightforward to check that $\alpha_i[(-1)^{\lambda_j}] = (-1)^{\delta_{ij}}$ for $i, j = 1, 2$, where $\lambda_i$ is defined in Appendix E.

- Wallpaper group 6: $pmm$

  There are 4 different IWP for $pmm$, which are the intersecting point of $M$ and $C_2 M$, the intersecting point of $T_1 M$ and $C_2 M$, the intersecting point of $M$ and $T_2 C_2 M$, and the intersecting point of $T_1 M$ and $T_2 C_2 M$. Note that these 4 IWP can also be respectively viewed as the rotation centers for the following four $C_2$ rotations: $C_2$, $T_1 C_2$, $T_2 C_2$ and $T_1 T_2 C_2$. The fractionalization patterns of $pmm$ are classified by $H^2(pmm, \mathbb{Z}_2) = \mathbb{Z}_2^8$. All fractionalization patterns are generated by 8 root patterns, detected by the topological

invariants

$$\alpha_1[\omega] = \frac{\omega(C_2, C_2)}{\omega(1,1)},$$

$$\alpha_2[\omega] = \frac{\omega(T_1 T_2 C_2, T_1 T_2 C_2)}{\omega(1,1)},$$

$$\alpha_3[\omega] = \frac{\omega(T_1 C_2, T_1 C_2)}{\omega(1,1)},$$

$$\alpha_4[\omega] = \frac{\omega(T_2 C_2, T_2 C_2)}{\omega(1,1)},$$

$$\alpha_5[\omega] = \frac{\omega(M, M)}{\omega(1,1)},$$

$$\alpha_6[\omega] = \frac{\omega(C_2 M, C_2 M)}{\omega(1,1)},$$

$$\alpha_7[\omega] = \frac{\omega(T_1 M, T_1 M)}{\omega(1,1)},$$

$$\alpha_8[\omega] = \frac{\omega(T_2 C_2 M, T_2 C_2 M)}{\omega(1,1)}.$$

(194)

The first four topological invariants are related to the third of the 3 basic no-go theorems, so they correspond to LSM constraints, and the classification of LSM constraints is $\mathbb{Z}_2^4$. The last four are non-LSM fractionalization patterns. It is straightforward to check that $\alpha_i[(-1)^{\lambda_j}] = (-1)^{\delta_{ij}}$ for $i, j = 1, \dots, 8$, where $\lambda_i$ is defined in Appendix E.

- Wallpaper group 7: $pmg$

  There are 3 different IWP for $pmg$. The first includes the rotation centers of $C_2$ and $T_1 C_2$, the second includes the rotation centers of $T_2 C_2$ and $T_1 T_2 C_2$, and the third includes the mirror axes of $M$ and $T_1 M$. The fractionalization patterns of $pmg$ are classified by $H^2(pmg, \mathbb{Z}_2) = \mathbb{Z}_2^4$. All fractionalization patterns are generated by 4 root patterns, detected by the topological invariants

$$\alpha_1[\omega] = \frac{\omega(C_2, C_2)}{\omega(1,1)},$$

$$\alpha_2[\omega] = \frac{\omega(T_1 T_2 C_2, T_1 T_2 C_2)}{\omega(1,1)},$$

$$\alpha_3[\omega] = \frac{\omega(T_2, M)}{\omega(M, T_2)},$$

$$\alpha_4[\omega] = \frac{\omega(M, M)}{\omega(1,1)}.$$

(195)

The first two topological invariants are related to the third of the 3 basic no-go theorems, and the third is related to the second of the 3 basic no-go theorems, so they correspond to LSM constraints, and the classification of LSM constraints is $\mathbb{Z}_2^3$. The fourth topological invariant is a non-LSM fractionalization pattern. It is straightforward to check that $\alpha_i[(-1)^{\lambda_j}] = (-1)^{\delta_{ij}}$ for $i, j = 1, \dots, 4$, where $\lambda_i$ is defined in Appendix E.

- Wallpaper group 8: $pgg$

  There are 2 different IWP for $pgg$. The first includes the rotation centers of $C_2$ and $T_1 T_2 C_2$, and the second includes the rotation centers of $T_1 C_2$ and $T_2 C_2$. The fractionalization patterns of $pgg$ are classified by $H^2(pgg, \mathbb{Z}_2) = \mathbb{Z}_2^2$. All fractionalization patterns

are generated by 2 root patterns, detected by the topological invariants

$$\alpha_1[\omega] = \frac{\omega(C_2, C_2)}{\omega(1, 1)},$$
$$\alpha_2[\omega] = \frac{\omega(T_1 C_2, T_1 C_2)}{\omega(1, 1)}. \tag{196}$$

Both topological invariants are related to the third of the 3 basic no-go theorems, so they both correspond to LSM constraints, and the classification of LSM constraints is $\mathbb{Z}_2^2$. It is straightforward to check that $\alpha_i[(-1)^{\lambda_j}] = (-1)^{\delta_{ij}}$ for $i, j = 1, 2$, where $\lambda_i$ is defined in Appendix E.

- Wallpaper group 9: $cmm$

  There are 3 different IWP for $cmm$. The first is the 2-fold rotation center of $C_2$, the second is the 2-fold rotation center of $T_1 T_2 C_2$, and the third inlcudes the 2-fold rotation centers of $T_1 C_2$ and $T_2 C_2$.

  The fractionalization patterns of $cmm$ are classified by $H^2(cmm, \mathbb{Z}_2) = \mathbb{Z}_2^5$. All fractionalization patterns are generated by 5 root patterns, detected by the topological invariants

  $$\alpha_1[\omega] = \frac{\omega(C_2, C_2)}{\omega(1, 1)},$$
  $$\alpha_2[\omega] = \frac{\omega(T_1 T_2 C_2, T_1 T_2 C_2)}{\omega(1, 1)},$$
  $$\alpha_3[\omega] = \frac{\omega(T_1 C_2, T_1 C_2)}{\omega(1, 1)}, \tag{197}$$
  $$\alpha_4[\omega] = \frac{\omega(M, M)}{\omega(1, 1)},$$
  $$\alpha_5[\omega] = \frac{\omega(C_2 M, C_2 M)}{\omega(1, 1)}.$$

  The first three topological invariants are related to the third of the 3 basic no-go theorems, so they correspond to LSM constraints, and the classification of LSM constraints is $\mathbb{Z}_2^3$. The last two are non-LSM fractionalization patterns. It is straightforward to check that $\alpha_i[(-1)^{\lambda_j}] = (-1)^{\delta_{ij}}$ for $i, j = 1, \ldots, 5$, where $\lambda_i$ is defined in Appendix E.

- Wallpaper group 10: $p4$

  There are 3 different IWP for $p4$. The first is the 2-fold rotation center of $C_4^2$, the second is the 2-fold rotation center of $T_1 T_2 C_4^2$, and the third includes the 2-fold rotation centers of $T_1 C_4^2$ and $T_2 C_4^2$. Note that the first two IWP are also 4-fold rotation centers. The fractionalization patterns of $p4$ are classfied by $H^2(p4, \mathbb{Z}_2) = \mathbb{Z}_2^3$. All fractionalization patterns are generated by 3 root patterns, detected by the topological invariants

  $$\alpha_1[\omega] = \frac{\omega(C_4^2, C_4^2)}{\omega(1, 1)},$$
  $$\alpha_2[\omega] = \frac{\omega(T_1 T_2 C_4^2, T_1 T_2 C_4^2)}{\omega(1, 1)}, \tag{198}$$
  $$\alpha_3[\omega] = \frac{\omega(T_1 C_4^2, T_1 C_4^2)}{\omega(1, 1)}.$$

  All these topological invariants are related to the third of the 3 basic no-go theorems, so they all correspond to LSM constraints, and the classification of LSM constraints is $\mathbb{Z}_2^3$. It is straightforward to check that $\alpha_i[(-1)^{\lambda_j}] = (-1)^{\delta_{ij}}$ for $i, j = 1, \ldots, 3$, where $\lambda_i$ is defined in Appendix E.

- Wallpaper group 11: $p4m$

  There are 3 different IWP for $p4m$, just like $p4$. The first is the 2-fold rotation center of $C_4^2$, the second is the 2-fold rotation center of $T_1T_2C_4^2$, and the third includes the 2-fold rotation centers for $T_1C_4^2$ and $T_2C_4^2$. All these IWP are also on some mirror axes, and the first two are also 4-fold rotation centers. The fractionalization patterns of $p4m$ are classified by $H^2(p4m, \mathbb{Z}_2) = \mathbb{Z}_2^6$. All fractionalization patterns are genereated by 6 root patterns, detected by the topological invariants

$$
\begin{aligned}
\alpha_1[\omega] &= \frac{\omega(C_4^2, C_4^2)}{\omega(1,1)}, \\
\alpha_2[\omega] &= \frac{\omega(T_1T_2C_4^2, T_1T_2C_4^2)}{\omega(1,1)}, \\
\alpha_3[\omega] &= \frac{\omega(T_1C_4^2, T_1C_4^2)}{\omega(1,1)}, \\
\alpha_4[\omega] &= \frac{\omega(M,M)}{\omega(1,1)}, \\
\alpha_5[\omega] &= \frac{\omega(T_1M, T_1M)}{\omega(1,1)}, \\
\alpha_6[\omega] &= \frac{\omega(C_4M, C_4M)}{\omega(1,1)}.
\end{aligned}
\tag{199}
$$

  The first three topological invariants are related to the third of the 3 basic no-go theorems, so they correspond to LSM constraints, and the classification of LSM constraints is $\mathbb{Z}_2^3$. The last three are non-LSM constraints. It is straightforward to check that $\alpha_i[(-1)^{\lambda_j}] = (-1)^{\delta_{ij}}$ for $i, j = 1, \ldots, 6$, where $\lambda_i$ is defined in Appendix E.

- Wallpaper group 12: $p4g$

  There are 2 different IWP for $p4g$. The first includes the 2-fold rotation centers of $C_4^2$ and $T_1T_2C_4^2$, and the second includes the 2-fold rotation centers of $T_1C_4^2$ and $T_2C_4^2$. Note that the first IWP are also 4-fold rotation centers, and they do not lie on any mirror axis. The second IWP lies on some mirror axes. The fractionalization patterns of $p4g$ are classified by $H^2(p4g, \mathbb{Z}_2) = \mathbb{Z}_2^3$. All fractionalization patterns are generated by 3 root patterns, detected by the topological invariants

$$
\begin{aligned}
\alpha_1[\omega] &= \frac{\omega(C_4^2, C_4^2)}{\omega(1,1)}, \\
\alpha_2[\omega] &= \frac{\omega(T_1C_4^2, T_1C_4^2)}{\omega(1,1)}, \\
\alpha_3[\omega] &= \frac{\omega(T_1^{-1}G, T_1^{-1}G)}{\omega(1,1)}.
\end{aligned}
\tag{200}
$$

  The first two topological invariants are related to the third of the 3 basic no-go theorems, so they correspond to LSM constraints, and the classification of LSM constraints is $\mathbb{Z}_2^2$. The last one is a non-LSM fractionalization pattern. It is straightforward to check that $\alpha_i[(-1)^{\lambda_j}] = (-1)^{\delta_{ij}}$ for $i, j = 1, \ldots, 3$, where $\lambda_i$ is defined in Appendix E.

- Wallpaper group 13: $p3$

  There are 3 IWP for $p3$, and they are all 3-fold rotation centers. The fractionalization patterns of $p3$ are classified by $H^2(p3, \mathbb{Z}_2) = \mathbb{Z}_2$. All fractionalization patterns are gen-

erated by a root pattern, detected by the topological invariant

$$\alpha[\omega] = \frac{\omega(T_1, T_2)}{\omega(T_2, T_1)}. \tag{201}$$

This topological invariant is related to the first of 3 basic no-go theorems, so it corresponds to an LSM constraint, and the classification of LSM constraints is $\mathbb{Z}_2$. It is straightforward to check that $\alpha_1[(-1)^{\lambda_1}] = -1$ for $i, j = 1, \ldots, 8$, where $\lambda_1$ is defined in Appendix E.

- Wallpaper group 14: $p3m1$

  There are 3 different IWP for $p3m1$, and they are all 3-fold rotation centers, just as in $p3$, but they also lie on the mirror axes. The fractionalization patterns of $p3m1$ are classified by $H^2(p3m1, \mathbb{Z}_2) = \mathbb{Z}_2^2$. All fractionalization patterns are generated by 2 root patterns, detected by the topological invariants

$$\begin{aligned}\alpha_1[\omega] &= \frac{\omega(T_1, T_2)}{\omega(T_2, T_1)}, \\ \alpha_2[\omega] &= \frac{\omega(M, M)}{\omega(1, 1)}.\end{aligned} \tag{202}$$

  The first topological invariant is related to the first of the 3 basic no-go theorems, so it corresponds to an LSM constraint, and the classification of LSM constraints is $\mathbb{Z}_2$. The second one is a non-LSM fractionalization pattern. It is straightforward to check that $\alpha_i[(-1)^{\lambda_j}] = (-1)^{\delta_{ij}}$ for $i, j = 1, 2$, where $\lambda_i$ is defined in Appendix E.

- Wallpaper group 15: $p31m$

  There are 3 different IWP for $p31m$, and they are all 3-fold rotation centers, just as in $p3$, but only one of them also lies on the mirror axes. The fractionalization patterns of $p31m$ are classified by $H^2(p31m, \mathbb{Z}_2) = \mathbb{Z}_2^2$. All fractionalization patterns are generated by 2 root patterns, detected by the topological invariants

$$\begin{aligned}\alpha_1[\omega] &= \frac{\omega(T_1 T_2, M)}{\omega(M, T_1 T_2)}, \\ \alpha_2[\omega] &= \frac{\omega(M, M)}{\omega(1, 1)}.\end{aligned} \tag{203}$$

  The first topological invariant is related to the second of the 3 basic no-go theorems, so it corresponds to an LSM constraint, and the classification of LSM constraints is $\mathbb{Z}_2$. The second is a non-LSM fractionalization pattern. It is straightforward to check that $\alpha_i[(-1)^{\lambda_j}] = (-1)^{\delta_{ij}}$ for $i, j = 1, 2$, where $\lambda_i$ is defined in Appendix E.

- Wallpaper group 16: $p6$

  There are 3 different IWP for $p6$, and they are centers of 6-fold, 3-fold and 2-fold rotations, respectively. The fractionalization patterns of $p6$ are classified by $H^2(p6, \mathbb{Z}_2) = \mathbb{Z}_2^2$. All fractionalization patterns are generated by 2 root patterns, detected by the topological invariants

$$\begin{aligned}\alpha_1[\omega] &= \frac{\omega(C_6^3, C_6^3)}{\omega(1, 1)}, \\ \alpha_2[\omega] &= \frac{\omega(T_1 C_6^3, T_1 C_6^3)}{\omega(1, 1)}.\end{aligned} \tag{204}$$

Both topological invariants are related to the third of the 3 basic no-go theorems, so they both correspond to LSM constraints, and the classification of LSM constraints is $\mathbb{Z}_2^2$. It is straightforward to check that $\alpha_i[(-1)^{\lambda_j}] = (-1)^{\delta_{ij}}$ for $i, j = 1, 2$, where $\lambda_i$ is defined in Appendix E.

- Wallpaper group 17: $p6m$

  There are 3 different IWP for $p6m$. Just as $p6$, they are 6-fold, 3-fold and 2-fold rotation centers, respectively. Here all IWP also lie on some mirror axes. The fractionalization patterns of $p6m$ are classified by $H^2(p6m, \mathbb{Z}_2) = \mathbb{Z}_2^4$. All fractionalization patterns are generated by 4 root patterns, detected by topological invariants

$$
\begin{aligned}
\alpha_1[\omega] &= \frac{\omega(C_6^3, C_6^3)}{\omega(1, 1)}, \\
\alpha_2[\omega] &= \frac{\omega(T_1 C_6^3, T_1 C_6^3)}{\omega(1, 1)}, \\
\alpha_3[\omega] &= \frac{\omega(M, M)}{\omega(1, 1)}, \\
\alpha_4[\omega] &= \frac{\omega(C_6^3 M, C_6^3 M)}{\omega(1, 1)}.
\end{aligned}
\tag{205}
$$

  The first two topological invariants are related to the third of the 3 basic no-go theorems, and they correspond to LSM constraints, and the classification of LSM constraints is $\mathbb{Z}_2^2$. The last two are non-LSM fractionalization patterns. It is straightforward to check that $\alpha_i[(-1)^{\lambda_j}] = (-1)^{\delta_{ij}}$ for $i, j = 1, \ldots, 4$, where $\lambda_i$ is defined in Appendix E.

# G  Topological characterization of LSM constraints in $(1+1)$-d

In this appendix, we present the derivation of the topological characterization of the LSM constraints for $(1+1)$-d $G_s \times G_{int}$ symmetric spin systems, where the results are already given in Sec. 2.2.3.

First, we note that an argument similar to the one in Appendix B for the $(2+1)$-d case shows that in this case the relevant cocycle can be written as

$$
\Omega(g_1, g_2, g_3) = e^{i\pi\lambda(l_1)\eta(a_2, a_3)}, \tag{206}
$$

where $g_i \in G_s \times G_{int}$ is written as $g_i = l_i \otimes a_i$, with $l_i \in G_s$ and $a_i \in G_{int}$. The cocycle for the nontrivial $(1+1)$-d $G_{int}$ SPT is precisely $e^{i\pi\eta(a_1, a_2)}$, and $\lambda$ can be viewed as a cocycle in $H^1(G_s, \mathbb{Z}_2)$. Furthermore, $\lambda$ is determined completely by $G_s$ and the lattice homotopy class, and it is the same for all $G_{int}$ with $\mathbb{Z}_2^k$-classified PR and for all PR type of the system.

When $G_s = p1$, the line group that only contains translation generated by $T$, the lattice homotopy picture implies that the LSM constraints in this case are classified by $\mathbb{Z}_2$, and the only nontrivial LSM constraint corresponds to the case where the total PR inside each translation unit cell is nontrivial. On the other hand, $H^1(p1, \mathbb{Z}_2) = \mathbb{Z}_2$, so there is also only one nontrivial cocycle. Writing an element in $p1$ as $T^x$ with $x \in \mathbb{Z}$, $\lambda(T^x) = [x]_2$ is a representative cochain of the nontrivial element in $H^1(p1, \mathbb{Z}_2)$. So we can identify the cocycle corresponding to the nontrivial LSM constraint as

$$
\Omega(g_1, g_2, g_3) = e^{i\pi x_1 \eta(a_2, a_3)}. \tag{207}
$$

When $G_s = p1m$, the line group that contains a translation generated by $T$ and a mirror symmetry generated by $M$, with commutation relation $MTM = T^{-1}$, there are two IWP in

each translation unit cell, which are the mirror centers of $M$ and $TM$, respectively. The lattice homotopy picture implies that the LSM constraints in this case are classified by $\mathbb{Z}_2^2$, and the two root LSM constraints can be taken to correspond to the cases where the total PR at one of the two IWP is nontrivial. On the other hand, $H^1(p1m, \mathbb{Z}_2) = \mathbb{Z}_2^2$, so all nontrivial cocycles in $H^1(p1m, \mathbb{Z}_2)$ must correspond to some nontrivial LSM constraint. These cocycles can be generated by two roots represented by $\lambda_1(T^x M^m) = x + m$ and $\lambda_2(T^x M^m) = x$, with $x \in \mathbb{Z}$ and $m \in \{0, 1\}$. So the cocycles corresponding to the LSM constraints can also be generated by

$$
\begin{aligned}
\Omega_1(g_1, g_2, g_3) &= e^{i\pi(x_1 + m_1)\eta(a_2, a_3)}, \\
\Omega_2(g_1, g_2, g_3) &= e^{i\pi x_1 \eta(a_2, a_3)},
\end{aligned}
\tag{208}
$$

where $g_i \in G_s \times G_{int}$ is written as $g_i = T^{x_i} M^{m_i} \otimes a_i$, with $a_i \in G_{int}$.

Now our task is just to identify $\Omega_1$ and $\Omega_2$ with the distributions of DOF that trigger the LSM constraint. To this end, first note that if $M$ is broken while $T$ is preserved, both $\Omega_1$ and $\Omega_2$ reduces to Eq. (207), which implies that both of them correspond to a distribution of DOF with a net nontrivial PR inside each translation unit cell. So one of them must correspond to the case where the mirror center of $M$ hosts a nontrivial PR, while the other corresponds to the case where the mirror center of $TM$ hosts a nontrivial PR. Suppose the mirror center of $TM$ hosts a nontrivial PR, then after breaking the translation symmetry while keeping $M$ unbroken, the system should have no LSM constraint. Only $\Omega_2$ satisfies this condition, so this distribution of DOF is identified with $\Omega_2$, and $\Omega_1$ corresponds to the case where the mirror center of $M$ hosts a nontrivial PR.

## H   More details of the Stiefel liquids

In this appendix, we discuss more details of Stiefel liquids, some of which do not appear in Ref. [15].

First we present some suggestive argument, but not rigorous proof, supporting that $SL^{(N>6)}$ are non-Lagrangian, i.e., they cannot be described by any weakly-coupled renormalizable Lagrangian in the UV. The key observation is that it appears unlikely for such Lagrangians to realize the $SO(N)$, $SO(N-4)$ and reflection symmetries of $SL^{(N)}$. To see it, let us start with even $N$. Usually in such a Lagrangian, symmetries like $SO(N)$ and $SO(N-4)$ are flavor symmetries, and there is a reflection symmetry that commutes with flavor symmetries. However, due to the locking between spacetime orientation reversals and improper rotations of $O(N)$ and $O(N-4)$, $SL^{(N)}$ has no such a reflection symmetry. This suggests that $SO(N)$ and $SO(N-4)$ cannot be simultaneously flavor symmetries. In the special case of $N = 6$, which does have a renormalizable Lagrangian description, indeed only $SO(6)$ but not $SO(2)$ can be identified as a flavor symmetry. In this example, the $SO(2)$ is realized as the flux conservation symmetry in the gauge theoretic formulation. For $N > 6$, there is no known generalization of the flux conservation symmetry that can give rise to symmetries like $SO(N-4)$. This indicates $SL^{(N>6)}$ with an even $N$ may be non-Lagrangian. Due to the cascade structure of SLs [15], it also suggests all $SL^{(N>6)}$ are non-Lagrangian.

We emphasize that the above is just a suggestive argument, but not a rigorous proof. There can be ways to get around the above obstruction, by, e.g., implementing some symmetries via dualities, considering Lagrangians in very complicated forms, showing that Lagrangians with smaller symmetries can have emergent symmetries of the SLs, etc. After finding a Lagrangian that can realize the symmetries of a SL, one still needs to make sure that its anomaly and low-energy dynamics match with the SL, which appears also challenging. If all these nontrivial challenges can be overcome and a renormalizable Lagrangian can be found to describe the

SL at the end, we believe this process can generate new insights and teach us some valuable general lessons of quantum field theories.

Next we discuss the anomalies of SLs, which should be captured by $\Omega_{\text{IR}}$, an element in $H^4(G_{\text{IR}}, U(1)_\rho)$, where $G_{\text{IR}} = (O(N)^T \times O(N-4)^T)/\mathbb{Z}_2$. Consider the projection: $p_{\text{SL}} : \tilde{G}_{\text{IR}} \equiv O(N)^T \times O(N-4)^T \to G_{\text{IR}}$, which induces a pullback $p^*_{\text{SL}} : H^4(G_{\text{IR}}, U(1)_\rho) \to H^4(\tilde{G}_{\text{IR}}, U(1)_\rho)$. The pullback of $\Omega_{\text{IR}}$, $\tilde{\Omega}_{\text{IR}} \equiv p^*_{\text{SL}} \Omega_{\text{IR}}$, is given by Eq. (21), in a form $\tilde{\Omega}_{\text{IR}} = e^{i\pi\tilde{L}_{\text{IR}}}$, with $\tilde{L}_{\text{IR}} \in H^4(\tilde{G}_{\text{IR}}, \mathbb{Z}_2)$ [15]. For even $N$, the structure of $\Omega_{\text{IR}}$ is still not completely understood, but it is known that $\tilde{\Omega}_{\text{IR}}$ misses some important information. In particular, the form of $\tilde{\Omega}_{\text{IR}}$ suggests that two copies of $\text{SL}^{(N)}$ would be anomaly-free. However, only four copies of $\text{SL}^{(N)}$ is anomaly-free, while two copies is still anomalous [15].

For odd $N$, because $O(N)^T = SO(N) \times \mathbb{Z}_2^T$, $H^4(G_{\text{IR}}, U(1)_\rho)$ has the structure of $\mathbb{Z}_2^k$ with some $k \in \mathbb{N}$, and there exists $L_{\text{IR}} \in H^4(G_{\text{IR}}, \mathbb{Z}_2)$ such that $\Omega_{\text{IR}} = e^{i\pi L_{\text{IR}}}$. Now notice that the pullback from $H^4(G_{\text{IR}}, \mathbb{Z}_2)$ to $H^4(\tilde{G}_{\text{IR}}, \mathbb{Z}_2)$ induced by $p_{\text{SL}}$ is injective, and hence we can uniquely identify $L_{\text{IR}}$ from $\tilde{L}_{\text{IR}}$. The result is

$$L_{\text{IR}} = w_4^{SO(N)} + w_4^{SO(N-4)} + \left(w_2^{SO(N)} + w_2^{SO(N-4)}\right) w_2^{SO(N-4)}$$
$$+ \begin{cases} w_1^2 w_2^{SO(N)}, & N = 1 \ (\text{mod } 8) \\ w_1^2 w_2^{SO(N-4)}, & N = 3 \ (\text{mod } 8) \\ w_1^2 (w_2^{SO(N)} + w_1^2), & N = 5 \ (\text{mod } 8) \\ w_1^2 (w_2^{SO(N-4)} + w_1^2), & N = 7 \ (\text{mod } 8) \end{cases} , \tag{209}$$

where $w_i^{SO(N)}$ and $w_i^{SO(N-4)}$ are the $i$-th Stiefel-Whitney class of the $SO(N)$ and $SO(N-4)$ gauge bundles. Considering enlarging $SO(N)$ and $SO(N-4)$ to $O(N)$ and $O(N-4)$, $w_1$ is sum of the first Stiefel-Whitney classes of the $O(N)$ and $O(N-4)$ gauge bundles. Due to the locking between spacetime orientation reversals and improper rotations of $O(N)$ and $O(N-4)$, $w_1$ can also be viewed as the first Stiefel-Whitney class of the tangent bundle of the spacetime manifold.

Finally, we discuss the effects of relevant operators on the DQCP ($\text{SL}^{(5)}$), DSL ($\text{SL}^{(6)}$) and $\text{SL}^{(7)}$. Because the low-energy dynamics of these states are not fully settled down, this discussion is also conjectural, and it is important to study these issues in a more rigorous manner in the future. However, given our understanding of these states, we believe the expectations below are reasonable.

For all SLs, the $(V_L, V_R)$ operator (or the $SO(5)$ vector for DQCP) should change the emergent order of the state. Due to the cascade structure among SLs [15], it is natural that this operator will just drive $\text{SL}^{(N)}$ to $\text{SL}^{(N-1)}$ (for DQCP, it simply gaps out the state). The time-reversal breaking operator that is a flavor singlet is likely to drive the state into a semion topological order, and this expectation is supported by the gauge-theoretic formulations of DQCP and DSL, as well as the fact that the semion topological order can match the anomaly of $\text{SL}^{(N)}$ if time reversal is broken (for all $N \geqslant 5$) [15]. The $(A_L, A_R)$ operator (for all $N \geqslant 6$) is expected to convert $\text{SL}^{(N)}$ into certain spontaneous-symmetry-breaking state, as supported from the gauge-theoretic formulation of DSL [18, 57, 58]. For DQCP, the traceless symmetric rank-2 tensor of $SO(5)$ drives the state into a spontaneous-symmetry-breaking state, and this operator is the tuning operator of the Neel-VBS transition in the standard realization of DQCP [16, 17]. All these operators change the emergent order of the states.

The remaining relevant operators to be discussed are the conserved current operators, whose effects on various states are complicated. It turns out that some of them can change the emergent order of the states, while others only shift the "zero momenta".

The simplest way to discuss it is perhaps to start from DSL, which has a relatively simple

gauge-theoretic formulation in terms of $N_f = 4$ QED$_3$:

$$\mathcal{L} = \sum_{i=1}^{4} \bar{\psi}_i i\slashed{D}_a \psi_i - \frac{1}{4e^2} f_{\mu\nu} f^{\mu\nu}. \tag{210}$$

There are conserved currents due to the $SO(6)$ and $SO(2)$ symmetries, where a natural basis of the $SO(6)$ currents is $\bar{\psi}\gamma_\mu T^{su(4)}\psi$, with $\gamma_\mu$ the Dirac matrices and $T^{(su(4))}$ the generators of $su(4)$ in its fundamental representation, and the $SO(2)$ currents are $\epsilon_{\mu\nu\lambda}\partial^\nu a^\lambda/(2\pi)$, with $a$ the emergent $U(1)$ gauge field. If the time component of the $SO(6)$ currents is added as a pertubation to the DSL, the Dirac fermions will be doped and acquire a finite Fermi surface, so the emergent order of the state changes. If the time component of the $SO(2)$ currents is added, the Dirac fermions will experience magnetic fields, Landau levels will form, and the emergent order of the state also changes. Below we discuss the effect of the spatial components of the currents.

For the $SO(6)$ spatial currents, depending on the choice of $T^{(su(4))}$ and $\gamma_\mu$, various effects can be triggered. For example, the current $\bar{\psi}\gamma_x \sigma_{30}\psi$ merely shifts the positions of the Dirac cones in the momentum space in a flavor-dependent way, which does not really change the emergent order of DSL (here $\sigma_{ij} \equiv \sigma_i \otimes \sigma_j$, where $\sigma_{i=0,1,2,3}$ are the identity and standard Pauli matrices). The same is true for $\bar{\psi}(\gamma_x \sigma_{30} + \gamma_y \sigma_{03})\psi$. However, as another example, $\bar{\psi}(\gamma_x \sigma_{23} + \gamma_y \sigma_{33})\psi$ actually converts the 4 Dirac cones into 2 pairs of quadratic band touching (and another 2 pairs of gapped bands), which does change the emergent order of the state. By examining the effect of different spatial currents, one can see more complicated patterns. Although a systematic description of the effects of these spatial currents is lacking, it can be analyzed in a case-by-case manner. These spatial currents can all be converted into the language of SL$^{(6)}$, in terms of the $6 \times 2$ matrix $n$. For example, using Appendix E of Ref. [15], we see that $\bar{\psi}\gamma_x \sigma_{30}\psi \sim n_{3i}\partial_x n_{4i}$, $\bar{\psi}(\gamma_x \sigma_{30} + \gamma_y \sigma_{03})\psi \sim n_{3i}\partial_x n_{4i} + n_{1i}\partial_y n_{2i}$, and $\bar{\psi}(\gamma_x \sigma_{23} + \gamma_y \sigma_{33})\psi \sim n_{4i}\partial_x n_{6i} + n_{5i}\partial_y n_{6i}$.

Next we turn to the $SO(2)$ spatial current, which in the language of SL$^{(6)}$ is $n_{i1}\partial_{x,y} n_{i2}$, and in the gauge theory is the electric field of the emergent $U(1)$ gauge field. It is not obvious what this perturbation does to the DSL. However, we argue that its effect is also to shift the zero momenta. To see it, we consider $N_f = 2$ QED$_3$, with Lagrangian

$$\mathcal{L} = \sum_{i=1}^{2} \bar{\psi}_i i\slashed{D}_a \psi_i - \frac{1}{4e^2} f_{\mu\nu} f^{\mu\nu}. \tag{211}$$

This theory is argued to describe the easy-plane DQCP, which has an emergent $O(4)$ unitary symmetry (not to be confused with the DQCP we have been discussing, which has an emergent $SO(5)$ unitary symmetry) [19]. In this theory, $\bar{\psi}\gamma_{x,y}\sigma_3\psi$ clearly only shifts the zero momenta without changing the emergent order of the state. On the other hand, the improper $\mathbb{Z}_2$ rotation of the $O(4)$ symmetry maps this operator into the electric fields of the emergent $U(1)$ gauge fields [19], which means that the electric fields also play the role of shifting the zero momenta without changing the emergent order. So we propose that in DSL (i.e., SL$^{(6)}$), the $SO(2)$ spatial currents also only shift the zero momenta, but maintain the emergent order.

Now we turn to DQCP (SL$^{(5)}$), which has a couple of gauge-theoretic formulations [19]. From any of these formulations, one can see that the time component of the $SO(5)$ currents changes the emergent order of the state. The formulation that has a manifest $O(5)^T$ symmetry is an $SU(2)$ gauge theory with 2 flavors of Dirac fermions, where the $SO(5)$ symmetry is the flavor symmetry of these Dirac fermions. Under similar consideration of the $SO(6)$ spatial currents in DSL, we see that the effects of the $SO(5)$ spatial currents in DQCP are also complicated and need to be analysed in a case-by-case manner: some of them changes the emergent order of the states, while others only shift the zero momenta without changing the emergent order.

We remark that the effects of the spatial currents actually impose very strong constraints on the possible results of our anomaly-based framework of emergibility. Within this framework, it is easy to see that all realizations of DQCP and DSL on $p6m \times O(3)^T$ and $p4m \times O(3)^T$ symmetric lattice spin systems must have all entries of $n$ locating at some high-symmetry momenta in the Brillouin zone, because all possile symmetry embedding patterns satisfy this condition. This means that in all realizations, it is impossible to have a spatial current operator that is allowed by the microscopic symmetries and can shift the zero momenta. As we have explicitly checked, this is indeed true for all realizations obtained in our anomaly-based framework, which can be viewed as a highly nontrivial sanity check of this framework – It nicely corroborates the validity of the hypothesis of emergibility, the proposal that DSL can indeed be described by SL$^{(6)}$, and the dynamics of DSL.

Finally, we turn to SL$^{(7)}$, whose low-energy dynamics is poorly understood so far. It is still likely that the time component of the $SO(7)$ and $SO(3)$ currents will change the emergent order. For the spatial currents, we propose the following rule. Writing an $SO(7)$ spatial current operator as a sum of terms of the form $n_{i_1 j} \partial_{x,y} n_{i_2 j}$, then we consider the effect of the same operator in DSL (it turns out that all such operators allowed by our microscopic symmetries only involve at most 4 rows of $n$, so its corresponding operator in DSL can always be found). If this operator changes the emergent order of DSL, then it also changes the emergent order of SL$^{(7)}$, and if it only shifts the zero momenta of DSL, it also only shifts the zero momenta of SL$^{(7)}$. For the $SO(3)$ spatial currents, it can be expanded as a sum as $\sim a_1 n_{i1}\partial_x n_{i2} + a_2 n_{i1}\partial_x n_{i3} + a_3 n_{i2}\partial_x n_{i3} + b_1 n_{i1}\partial_y n_{i2} + b_2 n_{i1}\partial_y n_{i3} + b_3 n_{i2}\partial_y n_{i3}$. We propose to first convert it into an $SO(7)$ spatial current $\sim a_1 n_{1i}\partial_x n_{2i} + a_2 n_{1i}\partial_x n_{3i} + a_3 n_{2i}\partial_x n_{3i} + b_1 n_{1i}\partial_y n_{2i} + b_2 n_{1i}\partial_y n_{3i} + b_3 n_{2i}\partial_y n_{3i}$. If this $SO(7)$ spatial current changes the emergent order (only shifts zero momenta) using the the above criterion, then the original $SO(3)$ spatial current also changes the emergent order (only shifts zero momenta).

The above proposal is of course conjectural, and more rigorous work is needed to fully settle it down. However, this proposal is supported by our results of anomaly-matching. We have checked all realizations of SL$^{(7)}$ obtained from the anomaly-based framework of emergibility, and found that the current operators that can shift zero momenta (according to the above proposal) are allowed by microscopic symmetries in a realization if and only if this realization belongs to a family where the momenta of some entries of $n$ can change continuously.

# I  More examples of the calculation of pullback

In this appendix, we give three more examples of the analysis of anomaly matching, for $SU(2)_1$, DSL and SL$^{(7)}$. In Appendix I.4, we also provide relevant formula for the calculation of pullback involving 5-dimensional representation of $SO(3)$.

## I.1  $SU(2)_1$ and emergent anomaly

First let us consider a representative $(1+1)$-d quantum critical state, i.e., the $(1+1)$-d $SU(2)_1$ conformal field theory, which describes the spin-1/2 antiferromagnetic Heisenberg chain at low energies [109–111]. The IR symmetry of the theory is $\frac{SU(2) \times SU(2)}{\mathbb{Z}_2} \rtimes \mathbb{Z}_2^T \cong O(4)$.

Ref. [103] works out the anomaly term of $SU(2)_1$ after gauging the $SO(4)$ part of $G_{\text{IR}} = O(4)$, which corresponds to the interger Euler class of $SO(4)$, $e \in H^4(SO(4), \mathbb{Z})$. The bulk topological partition function capturing this anomaly is the Chern-Simons theory at level $(+1, -1)$ for the two $su(2)$ factors of $so(4) \cong su(2) \times su(2)$, which can be written in terms of two $su(2)$ gauge

fields $A^{(1)}, A^{(2)}$ as follows

$$S = \frac{i}{4\pi} \int \mathrm{tr}\left( A^1 \wedge dA^{(1)} + \frac{2}{3} A^{(1)} \wedge A^{(1)} \wedge A^{(1)} \right) - \mathrm{tr}\left( A^{(2)} \wedge dA^{(2)} + \frac{2}{3} A^{(2)} \wedge A^{(2)} \wedge A^{(2)} \right). \quad (212)$$

It is straightforward to inspect that after gauging the $\mathbb{Z}_2^T$ part of $G_{\mathrm{IR}}$, the anomaly term should correspond to the twisted Euler class of $O(4)$, and we denote it by $\tilde{e} \in H^4(O(4), \mathbb{Z}_\rho)$. Note that this anomaly does not correspond to any element in $H^3(G_{\mathrm{IR}}, U(1)_\rho)$, i.e., the group cohomology (not the Borel cohomology in Ref. [39]) of $G_{\mathrm{IR}}$ acting nontrivially on the $U(1)$ coeffficient – this is the only example in this paper where the Bockstein homomorphism in Eq. (62) is not an isomorphism. Hence we need some special care to write down the TPF of the bulk SPT theory. [16]

Consider the following homomorphism $\varphi$ from $G_{\mathrm{UV}} = p1m \times O(3)$ to $G_{\mathrm{IR}} = O(4)$,

$$T \to \begin{pmatrix} -I_3 & \\ & -1 \end{pmatrix}, \quad M \to \begin{pmatrix} I_3 & \\ & -1 \end{pmatrix}, \quad O(3) \to \begin{pmatrix} O(3)^T & \\ & 1 \end{pmatrix}. \quad (216)$$

The LSM anomaly of a $1D$ chain has been worked out in Appendix G, i.e.,

$$\Omega_{\mathrm{UV}} \equiv \exp(i\pi L_{\mathrm{UV}}) = \exp\left( i\pi (x+m) w_2^{O(3)^T} \right). \quad (217)$$

We aim to prove that under the homomorphism $\varphi$, the pullback of the IR theory is the UV theory. Specifically, we need to prove that [17]

$$\beta(\Omega_{\mathrm{UV}}) = \varphi^*(\tilde{e}), \quad (218)$$

where $\beta$ is the Bockstein homomorphism associated to the short exact sequence $1 \to \mathbb{Z} \to \mathbb{R} \to U(1) \to 1$.

From the commutativity of the square in the diagram below

$$
\begin{array}{ccc}
H^4(G_{\mathrm{IR}}, \mathbb{Z}_\rho) & \xrightarrow{\tilde{p}} & H^4(G_{\mathrm{IR}}, \mathbb{Z}_2) \\
\downarrow{\varphi^*} & & \downarrow{\varphi^*} \\
\end{array}
, \quad (219)
$$

$$H^3(G_{\mathrm{UV}}, \mathbb{Z}_2) \xrightarrow{\tilde{i}} H^3(G_{\mathrm{UV}}, U(1)_\rho) \xrightarrow{\beta} H^4(G_{\mathrm{UV}}, \mathbb{Z}_\rho) \xrightarrow{\tilde{p}} H^4(G_{\mathrm{UV}}, \mathbb{Z}_2)$$

---

[16] In this footnote we briefly review how to write down the TPF worked out in Ref. [38]. Suppose a (2+1)-d IR theory has gauge symmmetry $G$ and is defined on the manifold $\mathcal{M}_3$, which serves as the base space of some principal bundle of $G$. Given an element $\omega \in H^4(G, \mathbb{Z})$, it is possible to define a 3d topological gauge theory of $G$ as follows

$$S = \frac{1}{n} \left( \int_{\mathcal{B}_4} \Omega - \langle \gamma^* \omega, [\mathcal{B}_4] \rangle \right) \mod 1, \quad (213)$$

where $\Omega$ is the de Rham representative of the image of $\omega$ in $H^4(BG, \mathbb{R})$, $[\mathcal{B}_4] \in H_4(\mathcal{B}_4, \mathbb{Z})$ is the fundamental class of the manifold $\mathcal{B}_4$ that bounds $n$ copies of the manifold $\mathcal{M}_3$ with some extension of the principle bundle of $G$, and $\gamma$ is the classifying map of the extension. When $\omega$ is a torsion element, $\Omega = 0$, and we retrieve the more familiar form of TPF

$$S = \langle \gamma^*(\beta^{-1}(\omega)), [\mathcal{M}_3] \rangle, \quad (214)$$

where $\beta$ is the Bockstein homomorphism associated to the short exact sequence $1 \to \mathbb{Z} \to \mathbb{R} \to U(1) \to 1$. In particular, when $G = SO(4)$ and $\omega$ corresponds to the Euler class $e$, $\Omega$ can be explicitly written as follows,

$$\Omega = \frac{1}{8\pi^2} \left( \mathrm{tr}\left( F^{(1)} \wedge F^{(1)} \right) - \mathrm{tr}\left( F^{(2)} \wedge F^{(2)} \right) \right). \quad (215)$$

In the presence of anti-unitary symmetries, the manifold $\mathcal{M}_3$ is assumed to be non-orientable. Then we have to choose $\mathcal{B}_4$ to be non-orientable as well, and demand $[\mathcal{B}_4] \in H^4(\mathcal{B}_4, \mathbb{Z}_w)$ to be the fundamental class of the non-orientable manifold $\mathcal{B}_4$ twisted by the orientation character $w$ [101].

[17] There are two terms in Eq. (213). The first term will become 0 when pulled back to $G_{\mathrm{UV}}$, which can be explicitly checked by considering the diagonal embedding of the Lie-algebra of $so(3) \cong su(2)$ into $so(4) \cong su(2) \times su(2)$. Then we just need to consider the pullback of the second term.

we just need to prove that

$$\mathcal{SQ}^1(L_{\text{UV}}) = \varphi^*(\tilde{p}(\tilde{e})). \tag{220}$$

In particular, on the left hand side we have

$$\mathcal{SQ}^1(L_{\text{UV}}) = (x + m)w_3^{O(3)^T}, \tag{221}$$

according to Appendix A.4, where $w_3^{O(3)^T} = w_3^{SO(3)} + tw_2^{SO(3)} + t^3$ and $t \in H^1(\mathbb{Z}_2^T, \mathbb{Z}_2)$ corresponds to the gauge field of time-reversal symmetry when pulled back to the spacetime manifold $\mathcal{M}_3$. On the right hand side we have $\tilde{p}(\tilde{e}) = w_4^{O(4)}$, and

$$\varphi^*\left(w_4^{O(4)}\right) = (x + m)\left(w_3^{SO(3)} + (t + x)w_2^{SO(3)} + (t + x)^3\right). \tag{222}$$

Finally, using the cohomology relation $x^2 = xm$, we see that both sides are equal to each other. Hence we establish that the pullback of the anomaly of IR CFT $SU(2)_1$ under the homomorphism $\varphi$ as in Eq. (216) is the LSM anomaly of $(1+1)$-d spin chain.

Below we discuss the phenomenon of emergent anomalies. Following Ref. [8], by imposing an extra constraint $T^2 = 1$, we can factorize $\varphi$ acting on $p1m$ into two pieces, i.e., a projection $p$ on $\mathbb{Z}_2 \times \mathbb{Z}_2$ generated by $\tilde{T}$ or $M$, where $\tilde{T}$ acts trivially on $U(1)$ or $\mathbb{Z}$ while $M$ acts nontrivially on $U(1)$ or $\mathbb{Z}$, composed with an embedding $\tilde{\varphi}$ of the $\mathbb{Z}_2 \times \mathbb{Z}_2$ into $O(4)$.

$$\varphi = \tilde{\varphi} \circ p: \quad p1m = \mathbb{Z} \rtimes \mathbb{Z}_2 \xrightarrow{p} \mathbb{Z}_2 \times \mathbb{Z}_2 \xrightarrow{\tilde{\varphi}} O(4) . \tag{223}$$

With slight abuse of notation, we denote the gauge field of $\tilde{T}$ as $x$ as well. Then we have

$$\begin{aligned}
\tilde{\varphi}^*\left(w_4^{O(4)}\right) &= (x + m)\left(w_3^{SO(3)} + (t + x)w_2^{SO(3)} + (t + x)^3\right) \\
&= \mathcal{SQ}^1\left((x + m)w_2^{O(3)^T} + (x + m)x^2\right),
\end{aligned} \tag{224}$$

in $H^4(\mathbb{Z}_2 \times \mathbb{Z}_2 \times O(3)^T, \mathbb{Z}_2)$. According to the terminology in Ref. [8], the first term $(x + m)w_2^{O(3)^T}$ as in Eq. (217) is the intrinsic anomaly, while the second term $(x + m)x^2$ is identified as the emergent anomaly. The emergent anomaly should be absent when pulled back to $p1m$, which is guaranteed by the relation $(x + m)x = 0$ present in $p1m$. As a sanity check, in the absence of mirror symmetry, i.e., in the line group $p1$, the intrinsic anomaly becomes $xw_2^{O(3)^T}$ and the emergent anomaly becomes $x^3$, consistent with the example in Ref. [8].

We envision that similar emergent anomaly will be present in IR theories emerging from a 2d lattice system with wallpaper group $G_s$, because a lot of cohomology relations of $G_s$ will be absent when projected to a finite group by imposing $T_1^n = T_2^n = 1$ for some integer $n$. More precisely, write $G_s = (\mathbb{Z} \times \mathbb{Z}) \rtimes O_s$, if we can find an integer $n$ such that $\varphi: G_s \to G_{\text{IR}}$ factorizes as the composition of projection and another embedding

$$\varphi = \tilde{\varphi} \circ p: \quad G_s = (\mathbb{Z} \times \mathbb{Z}) \rtimes O_s \xrightarrow{p} \tilde{G}_s \equiv (\mathbb{Z}_n \times \mathbb{Z}_n) \rtimes O_s \xrightarrow{\tilde{\varphi}} G_{\text{IR}}, \tag{225}$$

then $\tilde{\varphi}^*(\Omega_{\text{IR}}) \in H^4(\tilde{G}_s \times G_{\text{int}}, U(1)_\rho)$ will generically not be in the form of $\exp(i\pi\lambda\eta)$ with $\lambda \in H^2(\tilde{G}_s, \mathbb{Z}_2)$ and $\eta \in H^2(G_{\text{int}}, \mathbb{Z}_2)$, but contains a nonzero piece that nevertheless vanishes when pulled back to $G_s$, using certain cohomology relations of $G_s$ that is not present in $\tilde{G}_s$. Specifically, when $n = 2$, of the 3 important relations displayed in Appendix E, the first two relations, i.e., $x^2 = 0$ in $p1$ and $x^2 = xm$ in $p1m$, will be absent when projected to $\tilde{G}_s$, while the third relation, i.e., $A_{x+y}A_m = 0$ in $cm$, will still be present when projected to $\tilde{G}_s$.

For example, when the IR effective theory is the DQCP emergent from a square lattice spin-1/2 system with wallpaper group $p4m$, we can choose $n = 2$ and $\tilde{G}_s = (\mathbb{Z}_2 \times \mathbb{Z}_2) \rtimes D_4$. The $\mathbb{Z}_2$ cohomology ring of $\tilde{G}_s$ is

$$\mathbb{Z}_2[A_{x+y}, A_m, A_c, B_{xy}, B_{c^2}, B_{c(x+y)}]/\big(A_{x+y}A_c = 0, \ (A_m + A_c)A_c = 0, \ B_{c(x+y)}A_c = 0,$$
$$B_{c(x+y)}\big(B_{c(x+y)} + A_{x+y}(A_m + A_c)\big) = (A_m^2 + A_c^2)B_{xy} + A_{x+y}^2 B_{c^2}\big), \tag{226}$$

with the pullback of $B_{c(x+y)}$ equal to $A_{x+y}(A_{x+y} + A_m)$ in $H^*(p4m, \mathbb{Z}_2)$, and the pullback of $A_{x+y}, A_m, A_c, B_{xy}, B_{c^2}$ their namesake. Then from the fact that the IR anomaly of DQCP corresponds to $w_5^{O(5)} \in H^5(O(5), \mathbb{Z}_2)$, we have

$$\tilde{\varphi}^*\big(w_5^{O(5)}\big) = \big(B_{xy} + B_{c(x+y)} + B_{c^2}\big)\big(w_3^{SO(3)} + (t + A_{x+y})w_2^{SO(3)} + (t + A_{x+y})^3\big)$$
$$= \mathcal{SQ}^1\big(\big(B_{xy} + B_{c(x+y)} + B_{c^2}\big)w_2^{O(3)^T} + \big(B_{xy} + B_{c(x+y)} + B_{c^2}\big)A_{x+y}^2\big). \tag{227}$$

The first term $\big(B_{xy} + B_{c(x+y)} + B_{c^2}\big)w_2^{O(3)^T}$ is again the intrinsic anomaly, while the second term $\big(B_{xy} + B_{c(x+y)} + B_{c^2}\big)A_{x+y}^2$ is the emergent anomaly that vanishes when pulled back to $G_s = p4m$. This is a slight generalization of the result in Ref. [8] to the whole group $p4m$.

## I.2 DSL

Next consider DSL [15, 57, 112], whose IR symmetry $G_{\text{IR}}$ is $\frac{O(6) \times O(2)}{\mathbb{Z}_2}$, where an improper rotation of either $O(6)$ or $O(2)$ complex conjugates the $U(1)$ coefficient of $H^4(G_{\text{IR}}, U(1)_\rho)$. The precise form of the anomaly term for $G_{\text{IR}}$ is unknown, yet it is possible to write down its pullback to $O(6) \times O(2)$ under the projection $p : O(6) \times O(2) \to \frac{O(6) \times O(2)}{\mathbb{Z}_2}$ [15]

$$\tilde{\Omega}_{\text{IR}} \equiv \exp(i\pi \tilde{L}_{\text{IR}})$$
$$= \exp\Big[i\pi\Big(w_4^{O(6)} + w_2^{O(6)}\big(w_2^{O(2)} + (w_1^{O(2)})^2\big)$$
$$+ \big((w_2^{O(2)})^2 + w_2^{O(2)}(w_1^{O(2)})^2 + (w_1^{O(2)})^4\big)\Big)\Big], \tag{228}$$

where $\tilde{L}_{\text{IR}} \in H^4(O(6) \times O(2), \mathbb{Z}_2)$. On a triangular lattice, we consider the following example embedding $\varphi$ of $G_{\text{UV}} = p6m \times O(3)^T$ into $G_{\text{IR}}$,

$$T_1 : n \to \begin{pmatrix} I_3 & & & \\ & 1 & & \\ & & -1 & \\ & & & -1 \end{pmatrix} n \begin{pmatrix} -\frac{1}{2} & -\frac{\sqrt{3}}{2} \\ \frac{\sqrt{3}}{2} & -\frac{1}{2} \end{pmatrix},$$

$$T_2 : n \to \begin{pmatrix} I_3 & & & \\ & -1 & & \\ & & -1 & \\ & & & -1 \end{pmatrix} n \begin{pmatrix} -\frac{1}{2} & -\frac{\sqrt{3}}{2} \\ \frac{\sqrt{3}}{2} & -\frac{1}{2} \end{pmatrix},$$

$$C_6 : n \to \begin{pmatrix} I_3 & & & \\ & 1 & & \\ & & 1 & \\ & & & -1 \end{pmatrix} n \begin{pmatrix} 1 & \\ & -1 \end{pmatrix}, \tag{229}$$

$$M : n \to \begin{pmatrix} I_3 & & & \\ & -1 & & \\ & & -1 & \\ & & & 1 \end{pmatrix} n,$$

$$O(3)^T : n \to \begin{pmatrix} O(3)^T & \\ & I_3 \end{pmatrix} n.$$

Note that $\varphi$ factorizes into an embedding $\tilde{\varphi}$ into $O(6) \times O(2)$ composed with the projection $p$, i.e., $\varphi = p \circ \tilde{\varphi}$. In fact, for $G_{UV} = G_s \times O(3)^T$ with any $G_s$, if $\varphi$ satisfies the condition that some but not all entries of $n$ are left invariant under $SO(3)$, then $\varphi$ can always factorize into $p \circ \tilde{\varphi}$, where $\tilde{\varphi}$ is a homomorphism from $G_{UV}$ to $O(6) \times O(2)$. Therefore, we can think of the IR symmetry as $O(6) \times O(2)$ for simplicity in the calculation of pullback. Moreover, we can always choose $\tilde{\varphi}$ such that $G_s$ acts as identity and $\mathbb{Z}_2^T$ acts as minus identity in the block where $SO(3)$ acts.

The LSM anomaly of a triangular lattice spin-1/2 system has been obtained in Appendix E, and we repeat it here

$$\Omega_{UV} \equiv \exp(i\pi L_{UV}) = \exp\left(i\pi\left(B_{xy} + A_c(A_c + A_m)\right)w_2^{O(3)^T}\right), \tag{230}$$

where $L_{UV} \in H^4(G_{UV}, \mathbb{Z}_2)$. We wish to prove that $\Omega_{UV} = \varphi^*\Omega_{IR}$, which amounts to proving $\Omega_{UV} = \tilde{\varphi}^*\tilde{\Omega}_{IR}$. Again, from the commuting diagram Eq. (24) (with $G_{IR}$ changed to $O(6) \times O(2)$ and $\varphi$ changed to $\tilde{\varphi}$), we just need to prove that

$$\mathcal{SQ}^1(L_{UV}) = \tilde{\varphi}^*\left(\mathcal{SQ}^1(\tilde{L}_{IR})\right). \tag{231}$$

According to Lemma A.1, we have

$$\begin{aligned}\mathcal{SQ}^1(\tilde{L}_{IR}) =\; & w_5^{O(6)} + w_4^{O(6)}w_1^{O(2)} + w_3^{O(6)}\left(w_2^{O(2)} + (w_1^{O(2)})^2\right) + w_2^{O(6)}(w_1^{O(2)})^3 \\ & + w_1^{O(6)}\left((w_2^{O(2)})^2 + w_2^{O(2)}(w_1^{O(2)})^2 + (w_1^{O(2)})^4\right) + \left((w_2^{O(2)})^2 w_1^{O(2)} + (w_1^{O(2)})^5\right).\end{aligned} \tag{232}$$

On the other hand,

$$\mathcal{SQ}^1(L_{UV}) = \left((B_{xy} + A_c(A_c + A_m))\right)w_3^{O(3)^T}, \tag{233}$$

where $w_3^{O(3)^T} = w_3^{SO(3)} + tw_2^{SO(3)} + t^3$ and $t \in H^1(\mathbb{Z}_2^T, \mathbb{Z}_2)$ corresponds to the gauge field of time-reversal symmetry when pulled back to the spacetime manifold $\mathcal{M}_4$.

What remains is the calculation of the pullback $\tilde{\varphi}^*\left(\mathcal{SQ}^1(\tilde{L}_{IR})\right)$, which is a straightforward application of the Whitney product formula. In particular, considering the $O(2)$ block, the pullback gives

$$\begin{aligned}\tilde{\varphi}^*\left(w_1^{O(2)}\right) &= A_c, \\ \tilde{\varphi}^*\left(w_2^{O(2)}\right) &= 0.\end{aligned} \tag{234}$$

On the other hand, $O(6)$ factorizes into two $3 \times 3$ blocks, and for the lower $3 \times 3$ block we have

$$\begin{aligned}\tilde{\varphi}^*\left(w_1^{O(3)}\right) &= A_c + A_m, \\ \tilde{\varphi}^*\left(w_2^{O(3)}\right) &= B_{xy} + A_c^2, \\ \tilde{\varphi}^*\left(w_3^{O(3)}\right) &= A_c B_{xy} + A_c^2(A_c + A_m).\end{aligned} \tag{235}$$

Assembling the Stiefel-Whitney class of the lower $O(3)$ and upper $O(3)^T$ into the Stiefel-Whitney class of $O(6)$, we have

$$\begin{aligned}\tilde{\varphi}^*\left(w_5^{O(6)}\right) &= w_3^{O(3)^T}(B_{xy} + A_c^2) + w_2^{O(3)^T}(A_c B_{xy} + A_c^2(A_c + A_m)), \\ \tilde{\varphi}^*\left(w_4^{O(6)}\right) &= w_3^{O(3)^T}(A_c + A_m) + w_2^{O(3)^T}(B_{xy} + A_c^2) + t(A_c B_{xy} + A_c^2(A_c + A_m)), \\ \tilde{\varphi}^*\left(w_3^{O(6)}\right) &= w_3^{O(3)^T} + w_2^{O(3)^T}(A_c + A_m) + t(B_{xy} + A_c^2) + (A_c B_{xy} + A_c^2(A_c + A_m)), \\ \tilde{\varphi}^*\left(w_2^{O(6)}\right) &= w_2^{O(3)^T} + t(A_c + A_m) + (A_c B_{xy} + A_c^2(A_c + A_m)), \\ \tilde{\varphi}^*\left(w_1^{O(6)}\right) &= t + (A_c + A_m).\end{aligned} \tag{236}$$

Combining Eqs. (232), (233), (234) and (236), indeed we get Eq. (231). Hence we establish that $\Omega_{UV} = \varphi^*\Omega_{IR}$.

## I.3 $\text{SL}^{(7)}$

The next examples we want to consider are two realizations of $N = 7$ Stiefel liquid, i.e. $\text{SL}^{(7)}$, proposed in Ref. [15] (see Sec. VII D therein). The IR symmetry $G_{\text{IR}}$ of the theory is $\frac{O(7) \times O(3)}{\mathbb{Z}_2}$, and the precise form of the anomaly is given in Eq. (209) for $N = 7$. However, following the example in Appendix I.2, for the sake of the analysis of anomaly-matching, we can again think of the IR symmetry as $O(7) \times O(3)$ and consider the pullback of the anomaly under the projection $p : O(7) \times O(3) \to \frac{O(7) \times O(3)}{\mathbb{Z}_2}$,

$$
\begin{aligned}
\tilde{\Omega}_{\text{IR}} &\equiv \exp(i\pi \tilde{L}_{\text{IR}}) \\
&= \exp\Big(i\pi\Big(w_4^{O(7)} + w_2^{O(7)}\big(w_2^{O(3)} + (w_1^{O(3)})^2\big) \\
&\qquad + \big((w_2^{O(3)})^2 + w_2^{O(3)}(w_1^{O(3)})^2 + (w_1^{O(3)})^4\big)\Big)\Big),
\end{aligned}
\tag{237}
$$

where $\tilde{L}_{\text{IR}} \in H^4(O(7) \times O(3), \mathbb{Z}_2)$. We will omit the tilde symbol in the following calculation.

On a triangular lattice, we consider the following embedding $\varphi$ of $G_{\text{UV}} = p6m \times O(3)^T$ into $O(7) \times O(3)$,

$$
\begin{aligned}
T_1 : n &\to \begin{pmatrix} I_3 & & & \\ & -\frac{1}{2} & \frac{\sqrt{3}}{2} & \\ & -\frac{\sqrt{3}}{2} & -\frac{1}{2} & \\ & & & -\frac{1}{2} & \frac{\sqrt{3}}{2} \\ & & & -\frac{\sqrt{3}}{2} & -\frac{1}{2} \end{pmatrix} n \begin{pmatrix} 1 & & \\ & -1 & \\ & & -1 \end{pmatrix}, \\[2mm]
T_2 : n &\to \begin{pmatrix} I_3 & & & \\ & -\frac{1}{2} & \frac{\sqrt{3}}{2} & \\ & -\frac{\sqrt{3}}{2} & -\frac{1}{2} & \\ & & & -\frac{1}{2} & \frac{\sqrt{3}}{2} \\ & & & -\frac{\sqrt{3}}{2} & -\frac{1}{2} \end{pmatrix} n \begin{pmatrix} -1 & & \\ & 1 & \\ & & -1 \end{pmatrix}, \\[2mm]
C_6 : n &\to \begin{pmatrix} I_3 & & & \\ & 1 & & \\ & & -1 & \\ & & & 1 & \\ & & & & -1 \end{pmatrix} n \begin{pmatrix} & & 1 \\ 1 & & \\ & 1 & \end{pmatrix}, \\[2mm]
M : n &\to \begin{pmatrix} I_3 & & & \\ & -1 & & \\ & & -1 & \\ & & & 1 & \\ & & & & 1 \end{pmatrix} n \begin{pmatrix} & & 1 \\ 1 & & \\ & 1 & \end{pmatrix}, \\[2mm]
O(3)^T : n &\to \begin{pmatrix} O(3)^T & \\ & I_4 \end{pmatrix} n.
\end{aligned}
\tag{238}
$$

Again, the LSM anomaly of a triangular lattice spin-1/2 system is

$$
\Omega_{\text{UV}} \equiv \exp(i\pi L_{\text{UV}}) = \exp\Big(i\pi\big(B_{xy} + A_c(A_c + A_m)\big) w_2^{O(3)^T}\Big),
\tag{239}
$$

where $L_{\text{UV}} \in H^4(G_{\text{UV}}, \mathbb{Z}_2)$. We wish to prove that $\Omega_{\text{UV}} = \varphi^* \Omega_{\text{IR}}$. From the commuting diagram Eq. (24), we just need to prove that

$$
\mathcal{SQ}^1(L_{\text{UV}}) = \varphi^*\big(\mathcal{SQ}^1(L_{\text{IR}})\big).
\tag{240}
$$

According to Lemma A.1, we have

$$
\begin{aligned}
\mathcal{SQ}^1(L_{\text{IR}}) =& w_5^{O(7)} + w_4^{O(7)} w_1^{O(3)} + w_3^{O(7)}\left(w_2^{O(3)} + (w_1^{O(3)})^2\right) + w_2^{O(7)}\left(w_3^{O(3)} + (w_1^{O(3)})^3\right) \\
& + w_1^{O(7)}\left((w_2^{O(3)})^2 + w_2^{O(3)}(w_1^{O(3)})^2 + (w_1^{O(3)})^4\right) \\
& + \left(w_3^{O(3)}(w_1^{O(3)})^2 + (w_2^{O(3)})^2 w_1^{O(3)} + (w_1^{O(3)})^5\right).
\end{aligned}
\tag{241}
$$

Also,

$$
\mathcal{SQ}^1(L_{\text{UV}}) = \left((B_{xy} + A_c(A_c + A_m))\right) w_3^{O(3)^T},
\tag{242}
$$

where $w_3^{O(3)^T} = w_3^{SO(3)} + tw_2^{SO(3)} + t^3$ and $t \in H^1(\mathbb{Z}_2^T, \mathbb{Z}_2)$ corresponds to the gauge field of time-reversal symmetry when pulled back to the spacetime manifold $\mathcal{M}_4$.

What remains is the calculation of the pullback $\varphi^*\left(\mathcal{SQ}^1(L_{\text{IR}})\right)$, which is a straightforward application of the Whitney product formula. In particular, $O(7)$ factorizes into one $3 \times 3$ block and two $2 \times 2$ block, and for the $O(3)$ part and the $O(7)$ part seperately, the pullback gives

$$
\begin{aligned}
\varphi^*\left(w_1^{O(3)}\right) &= A_m, \\
\varphi^*\left(w_2^{O(3)}\right) &= B_{xy}, \\
\varphi^*\left(w_3^{O(3)}\right) &= 0, \\
\varphi^*\left(w_1^{O(7)}\right) &= t, \\
\varphi^*\left(w_2^{O(7)}\right) &= w_2^{SO(3)} + t^2 + A_c^2 + A_m^2 + A_m A_c, \\
\varphi^*\left(w_3^{O(7)}\right) &= \left(w_3^{SO(3)} + tw_2^{SO(3)} + t^3\right) + t(A_c^2 + A_m^2 + A_m A_c) + A_c A_m(A_c + A_m), \\
\varphi^*\left(w_4^{O(7)}\right) &= (w_2^{SO(3)} + t^2)(A_c^2 + A_m^2 + A_m A_c) + tA_c A_m(A_c + A_m), \\
\varphi^*\left(w_5^{O(7)}\right) &= \left(w_3^{SO(3)} + tw_2^{SO(3)} + t^3\right)(A_c^2 + A_m^2 + A_m A_c) + (w_2^{SO(3)} + t^2)A_c A_m(A_c + A_m).
\end{aligned}
\tag{243}
$$

Substituting them back into Eq. (241), and using the cohomology relation $B_{xy}^2 = B_{c^2} B_{xy}$, indeed we get Eq. (242) as promised. Hence we establish that $\Omega_{\text{UV}} = \varphi^* \Omega_{\text{IR}}$.

On a Kagome lattice spin-$1/2$ system, we consider the following embedding $\varphi$ of $G_{\text{UV}} = p6m \times O(3)^T$ into $O(7) \times O(3)$,

$$
T_1 : n \to n \begin{pmatrix} 1 & & \\ & -1 & \\ & & -1 \end{pmatrix},
$$

$$
T_2 : n \to n \begin{pmatrix} -1 & & \\ & 1 & \\ & & -1 \end{pmatrix},
$$

$$
C_6 : n \to \begin{pmatrix} I_3 & & & \\ & -1 & & \\ & & 1 & \\ & & & -1 \\ & & & & -1 \end{pmatrix} n \begin{pmatrix} & & -1 \\ 1 & & \\ & 1 & \end{pmatrix},
$$

$$
M : n \to \begin{pmatrix} I_3 & & & \\ & -1 & & \\ & & -1 & \\ & & & 1 \\ & & & & 1 \end{pmatrix} n \begin{pmatrix} & & -1 \\ & -1 & \\ 1 & & \end{pmatrix},
$$

$$
\tag{244}
$$

$$O(3)^T : n \to \begin{pmatrix} O(3)^T & \\ & I_4 \end{pmatrix} n \,.$$

The LSM anomaly of a Kagome lattice spin-1/2 system is

$$\Omega_{\mathrm{UV}} \equiv \exp(i\pi L_{\mathrm{UV}}) = \exp\left(i\pi B_{xy} w_2^{O(3)^T}\right) \,. \tag{245}$$

Again, we wish to prove that $\Omega_{\mathrm{UV}} = \varphi^* \Omega_{\mathrm{IR}}$ by proving $\mathcal{SQ}^1(L_{\mathrm{UV}}) = \varphi^*\left(\mathcal{SQ}^1(L_{\mathrm{IR}})\right)$. $\mathcal{SQ}^1(L_{\mathrm{IR}})$ is given in Eq. (241), while for $\mathcal{SQ}^1(L_{\mathrm{UV}})$ we have

$$\mathcal{SQ}^1(L_{\mathrm{UV}}) = B_{xy} w_3^{O(3)^T} \,. \tag{246}$$

It is now straightforward to calculate the pullback of various Stiefel-Whitney classes in Eq. (241),

$$\begin{aligned}
\varphi^*\left(w_1^{O(3)}\right) &= A_m + A_c \,, \\
\varphi^*\left(w_2^{O(3)}\right) &= B_{xy} + A_c^2 \,, \\
\varphi^*\left(w_3^{O(3)}\right) &= A_c^3 + A_c^2 A_m + A_c B_{xy} \,, \\
\varphi^*\left(w_1^{O(7)}\right) &= t + A_c \,, \\
\varphi^*\left(w_2^{O(7)}\right) &= w_2^{SO(3)} + t^2 + tA_c + A_c^2 + A_c A_m + A_m^2 \,, \\
\varphi^*\left(w_3^{O(7)}\right) &= \left(w_3^{SO(3)} + tw_2^{SO(3)} + t^3\right) + \left(w_2^{SO(3)} + t^2\right)A_c + t\left(A_c^2 + A_c A_m + A_m^2\right) + A_c^3 \,, \\
\varphi^*\left(w_4^{O(7)}\right) &= \left(w_3^{SO(3)} + tw_2^{SO(3)} + t^3\right)A_c + \left(w_2^{SO(3)} + t^2\right)\left(A_c^2 + A_c A_m + A_m^2\right) \\
&\quad + tA_c^3 + A_c^2 A_m(A_c + A_m) \,, \\
\varphi^*\left(w_5^{O(7)}\right) &= \left(w_3^{SO(3)} + tw_2^{SO(3)} + t^3\right)\left(A_c^2 + A_c A_m + A_m^2\right) + \left(w_2^{SO(3)} + t^2\right)A_c^3 \\
&\quad + tA_c^2 A_m(A_c + A_m) \,.
\end{aligned} \tag{247}$$

Substituting them into (241) and using the cohomology relation $B_{xy}^2 = B_{c^2}B_{xy}$, indeed we get Eq. (246), and thus establish that $\mathcal{SQ}^1(L_{\mathrm{UV}}) = \varphi^*\left(\mathcal{SQ}^1(L_{\mathrm{IR}})\right)$.

## I.4 Five dimensional representation of $SO(3)$

In all previous examples presented in this appendix, the $SO(3)$ spin rotation symmetry is embedded into the IR symmetry $G_{\mathrm{IR}}$ as a 3 dimensional representation. It is natural to consider embedding involving other representations of $SO(3)$, whose physical relevance is illustrated in Section 5. In this sub-appendix, we present formula relevant to mapping $SO(3)$ into $G_{\mathrm{IR}}$ as a 5 dimensional representation of $SO(3)$.

First consider the 5 dimensional representation $\varphi_5 : SO(3) \to O(5)$ of $SO(3)$ alone, which can be thought of as a symmetric traceless tensor $V_5$, whose 5 basis are

$$\begin{aligned}
&\frac{1}{\sqrt{2}}(n_1 \otimes n_2 + n_2 \otimes n_1) \,, \quad \frac{1}{\sqrt{2}}(n_2 \otimes n_3 + n_3 \otimes n_2) \,, \quad \frac{1}{\sqrt{2}}(n_3 \otimes n_1 + n_1 \otimes n_3) \,, \\
&\frac{1}{\sqrt{2}}(n_2 \otimes n_2 - n_3 \otimes n_3) \,, \quad \frac{1}{\sqrt{6}}(2n_1 \otimes n_1 - n_2 \otimes n_2 - n_3 \otimes n_3) \,,
\end{aligned} \tag{248}$$

where $n_{1,2,3}$ form an $SO(3)$ vector. Consider the $\mathbb{Z}_2^2$ subgroup of $SO(3)$, generated by $\pi$-rotations around the $x$- and $y$-axes, respectively. Using the above 5 basis, these two $\pi$-rotations

are mapped into $\text{diag}(-1, 1, -1, 1, 1)$ and $\text{diag}(-1, -1, 1, 1, 1)$, respectively, from which (or from the splitting principle [113]) we see that

$$\varphi_5^*\left(w_2^{O(5)}\right) = w_2^{SO(3)}, \quad \varphi_5^*\left(w_3^{O(5)}\right) = w_3^{SO(3)}, \quad \varphi_5^*\left(w_1^{O(5)}\right) = 0, \quad \varphi_5^*\left(w_4^{O(5)}\right) = 0,$$
$$\varphi_5^*\left(w_5^{O(5)}\right) = 0. \tag{249}$$

Now go back to $G_{\text{UV}} = SO(3) \times \tilde{G}$ and consider a 5 dimensional representation $\varphi_5 : G_{\text{UV}} \to O(5)$ that can be written as $V_5 \otimes V_1$, where $V_5$ denotes the 5 dimensional representation of $SO(3)$ while $V_1$ denotes a 1 dimensional real representation of $\tilde{G}$ corresponding to $x \in H^1(\tilde{G}, \mathbb{Z}_2)$. Again from inspecting the action of the diagonal $\mathbb{Z}_2^2$ subgroup, we have

$$\varphi_5^*\left(w_1^{O(5)}\right) = x,$$
$$\varphi_5^*\left(w_2^{O(5)}\right) = w_2^{SO(3)},$$
$$\varphi_5^*\left(w_3^{O(5)}\right) = w_3^{SO(3)} + x w_2^{SO(3)}, \tag{250}$$
$$\varphi_5^*\left(w_4^{O(5)}\right) = x^2 w_2^{SO(3)} + x^4,$$
$$\varphi_5^*\left(w_5^{O(5)}\right) = x^2 w_3^{SO(3)} + x^3 w_2^{SO(3)} + x^5.$$

## J   Strategy of exhaustive search of SEP and results

In this appendix, we briefly review our strategy of the exhaustive search of SEP. We also illustrate how to check all the SEPs from the *csv* data files we provide in the *Data_and_Codes* folder and the mathematica file *embedding.m*, which transforms the data in *csv* files into matrices representing generators $C_6/C_4$, $M$, $T_1$, $T_2$ and $\mathcal{T}$. Some interesting realizations have been shown in Sections 4 and 5.

In order to enumerate all SEPs that match LSM constraints with IR anomaly, we just need to enumerate all embeddings from $G_{\text{UV}}$ to $G_{\text{IR}}$ and, following Section 3 and Appendix I, calculate the pullback $\varphi^*(\Omega_{\text{IR}})$ to see if it is identical to $\Omega_{\text{UV}}$ corresponding to a particular LSM constraint. Motivated by quantum magnetism, we assume that the IR theory will emerge as a consequence of the competition between a magnetic state and a non-magnetic state. Therefore, we only consider embeddings such that, in terms of the $N \times (N - 4)$ matrix $n$ for $\text{SL}^{(N)}$, some but not all entries of $n$ transform under the $SO(3)$ symmetry.

For DQCP, since the IR symmetry is $O(5)$, all embeddings are just composed of representations of $G_{\text{UV}}$. For DSL and $\text{SL}^{(7)}$, even though the IR symmetry is $\frac{O(6) \times O(2)}{\mathbb{Z}_2}$ and $\frac{O(7) \times O(3)}{\mathbb{Z}_2}$, respectively, because of the constraints on the embeddings, it suffices to only consider embeddings into $\frac{O(6) \times O(2)}{\mathbb{Z}_2}$ and $\frac{O(7) \times O(3)}{\mathbb{Z}_2}$ which can be respectively lifted to an embedding into $O(6) \times O(2)$ or $O(7) \times O(3)$, as discussed below Eq. (229). Therefore, all embeddings we consider are just composed of real representations of $G_{\text{UV}}$. In other words, our task to specify an embedding becomes finding appropriate irreducible representations of $G_{\text{UV}}$, and fill them into the $O(N)$ and $O(N - 4)$ matrices in a block diagonal form.

Hence let us make a detour and discuss representations of $G_{\text{UV}} = G_s \times SO(3) \times \mathbb{Z}_2^T$. For any two groups $G_{1,2}$, an irreducible representation $V$ of $G_1 \times G_2$ is $V_1 \otimes V_2$, where $V_{1,2}$ is an irreducible representation of $G_{1,2}$, respectively. So any irreducible representation $V$ of $G_{\text{UV}}$ takes the form of $V = V_{SO(3)}^{2n+1} \otimes V_s \otimes V_T$, where $V_{SO(3)}^{2n+1}$ is a $(2n + 1)$-dimensional irreducible representation of $SO(3)$ with $n \in \mathbb{N}$, $V_s$ is an irreducible representation of $G_s$, and $V_T = \pm 1$ is an irreducible representation of $\mathbb{Z}_2^T$. The complete list of irreducible representations $V_s$ of $G_s$ can be found using the method of induced representations [114–116], and we provide complete lists of irreducible representations of $p4m$ and $p6m$ in the Mathematica file *Representation.nb*.

To figure out which representations of $G_{\text{UV}}$ are relevant to our discussions, it is useful to analyze in which blocks the $SO(3)$ can act nontrivially, with the assumption that some but not all entries of $n$ transform under the $SO(3)$ symmetry. For DQCP, $SO(3)$ must act nontrivially in a 3-d block, while the rest 2-d block should be a reducible or irreducible representation of $G_s \times \mathbb{Z}_2^T$. That is, the relevant representation $V$ of $G_{\text{UV}}$ schematically takes the form

$$V_{\text{DQCP}} = \begin{pmatrix} \left(V_{SO(3)}^3 \otimes V_s^1 \otimes V_T^1\right)^{3\times3} & \\ & \left(V_{SO(3)}^1 \otimes V_{G_s\times\mathbb{Z}_2^T}^2\right)^{2\times2} \end{pmatrix}, \tag{251}$$

where $V_s^1$ and $V_T^1$ are 1-d representations of $G_s$ and $\mathbb{Z}_2^T$ respectively, and $V_{G_s\times\mathbb{Z}_2^T}^2$ is a 2-d (reducible or irreducible) representation of $G_s \times \mathbb{Z}_2^T$.

For DSL, the block involving nontrivial $SO(3)$ actions can be 3-d or 5-d, and it has to lie in $O(6)$. For $\text{SL}^{(7)}$, the block involving $SO(3)$ should embed into $O(7)$ and can be 3-d, 5-d or 6-d. The 6-d representation takes the form of $V_{SO(3)} \otimes V_{G_s\times\mathbb{Z}_2^T}^2$, where $V_{SO(3)}$ is the 3-d representation of $SO(3)$, and $V_{G_s\times\mathbb{Z}_2^T}^2$ involves either two 1-d representations of $G_s \times \mathbb{Z}_2^T$ or one irreducible 2-d representation of $G_s \times \mathbb{Z}_2^T$. However, it turns out that it is impossible to match the anomaly with any LSM constraint in the presence of some 6-d block involving $SO(3)$. Therefore, for DSL and $\text{SL}^{(7)}$, we consider two cases, i.e., either $SO(3)$ embeds as a 3-d representation, corresponding to deconfined quantum criticle points or quantum critical spin liquids in Section 4, or as a 5-d representation, corresponding to quantum critical spin-quadrupolar liquids in Section 5.

For DSL and $\text{SL}^{(7)}$, we still have freedom to choose the lifting to $O(6)\times O(2)$ or $O(7)\times O(3)$, and different embeddings into $O(6) \times O(2)$ or $O(7) \times O(3)$ may correspond to the same embedding into $\frac{O(6)\times O(2)}{\mathbb{Z}_2}$ or $\frac{O(7)\times O(3)}{\mathbb{Z}_2}$. For embeddings involving 3-d representation of $SO(3)$, we choose such that only $\mathcal{T}$ acts in the $3 \times 3$ block as $-I_3$, while $G_s$ acts trivially in that block. That is, for DSL and $\text{SL}^{(7)}$, the relevant representations $V$ of $G_{\text{UV}}$ schematically take the form as follows

$$V_{\text{DSL}} = \begin{pmatrix} \left(V_{SO(3)}^3 \otimes 1_s \otimes (-1_T)\right)^{3\times3} & \\ & \left(V_{SO(3)}^1 \otimes V_{G_s\times\mathbb{Z}_2^T}^3\right)^{3\times3} \end{pmatrix} \times \left(V_{SO(3)}^1 \otimes V_{G_s\times\mathbb{Z}_2^T}^2\right)^{2\times2}, \tag{252}$$

and

$$V_{\text{SL}^{(7)}} = \begin{pmatrix} \left(V_{SO(3)}^3 \otimes 1_s \otimes (-1_T)\right)^{3\times3} & \\ & \left(V_{SO(3)}^1 \otimes V_{G_s\times\mathbb{Z}_2^T}^4\right)^{4\times4} \end{pmatrix} \times \left(V_{SO(3)}^1 \otimes V_{G_s\times\mathbb{Z}_2^T}^3\right)^{3\times3}, \tag{253}$$

where $1_s$ is the 1-d trivial representation of $G_s$, and $-1_T$ is the 1-d non-trivial representation of $\mathbb{Z}_2^T$. For embeddings involving 5-d representation of $SO(3)$, we choose such that both $\mathbb{Z}_2^T$ and $G_s$ act trivially in the $5 \times 5$ block. That is, the relevant representations $V$ of $G_{\text{UV}}$ schematically take the form

$$V_{\text{DSL}} = \begin{pmatrix} \left(V_{SO(3)}^5 \otimes 1_s \otimes 1_T\right)^{5\times5} & \\ & \left(V_{SO(3)}^1 \otimes V_{G_s\times\mathbb{Z}_2^T}^1\right)^{1\times1} \end{pmatrix} \times \left(V_{SO(3)}^1 \otimes V_{G_s\times\mathbb{Z}_2^T}^2\right)^{2\times2}, \tag{254}$$

and

$$V_{\text{SL}^{(7)}} = \begin{pmatrix} \left(V_{SO(3)}^5 \otimes 1_s \otimes 1_T\right)^{5\times5} & \\ & \left(V_{SO(3)}^1 \otimes V_{G_s\times\mathbb{Z}_2^T}^2\right)^{2\times2} \end{pmatrix} \times \left(V_{SO(3)}^1 \otimes V_{G_s\times\mathbb{Z}_2^T}^3\right)^{3\times3}, \tag{255}$$

where $1_s$ and $1_T$ are 1-d trivial representations of $G_s$ and $\mathbb{Z}_2^T$, respectively.

Having identified all possible embeddings, it is a striaghtforward exercise to calculate the pullback in each case following Section 3 and Appendix I. We use *Mathematica* to automate the computation and store results in *csv* files in the ancillary folder [59]. For example, *data.csv* contains data for matching LSM constraints with IR anomaly of $\mathrm{SL}^{(N=5,6,7)}$ when $SO(3)$ embeds into $O(6)$ as a 3-d representaion, while *dataSL5Rep.csv* contains data for matching LSM constraints of $p4m$ with IR anomaly of $\mathrm{SL}^{(7)}$ when $SO(3)$ embeds into $O(7)$ as a 5-d representaion. Moreover, for both $p4m$ and $p6m$, there is a single embedding involving 5-d representation of $SO(3)$ that can match IR anomaly of DSL, shown in Eq. (40), which actually matches IR anomaly with zero LSM constraint.

To read the embeddings, i.e., to read the explicit image of the generators $C_4/C_6$, $M$, $T_1$, $T_2$ and $\mathcal{T}$ in $G_{\mathrm{IR}}$, we provide a wrapper file *Embedding.m*. When $SO(3)$ embeds into $G_{\mathrm{IR}}$ as a 3-d representation, it provides two functions

```
p4mPrintEmbedding[n_Integer, lsm_Integer, p_Integer]

p6mPrintEmbedding[n_Integer, lsm_Integer, p_Integer]
```

The arguments are $n = 5, 6, 7$ corresponding to DQCP, DSL and $\mathrm{SL}^{(7)}$ respectively, $lsm = 1, \ldots, 8$ in $p4m$ or $lsm = 1, \ldots, 4$ corresponding to a particular LSM constraint with the order shown in Table 1, and $p$ corresponding to a position in the array for a particular embedding/realization. When $SO(3)$ embeds into $G_{\mathrm{IR}}$ as a 5-d representation and the IR theory is $\mathrm{SL}^{(7)}$, it also provides two functions

```
p4m5dPrintEmbedding[lsm_Integer, p_Integer]

p6m5dPrintEmbedding[lsm_Integer, p_Integer]
```

with similar arguments and output. Note that in this scenario for DQCP there is no realization, and for DSL there is a single realization in $p4m$ or $p6m$ shown in Eq. (40). For $p4m$ and $\mathrm{SL}^{(7)}$, it also provides a function

```
IncommensuratePrintEmbedding[lsm_Integer, p_Integer]
```

to check whether some embedding corresponds to an incommensurate order, and if it does, output the corresponding incommensurate embedding. An illustration of how to use these functions is provided in *ReadMe.nb*.

# K    Stable realizations on various lattice spin systems

In this appendix, we list all stable realizations of DQCP, DSL and $\mathrm{SL}^{(7)}$ on triangular, kagome, and square lattice half-integer spin systems, as well as those on $p6m$-anomaly-free systems (including honeycomb lattice half-integer spin systems and all integer-spin systems with $p6m$ lattice symmetry) and $p4m$-anomaly-free systems (including all integer-spin systems with $p4m$ lattice symmetry). For square lattice, we only list the realizations in lattice homotopy class with PR at the type-a IWP, from which the ones with PR at the type-b IWP can be obtained by redefining the $C_4$ center. As in the main text, here a stable DQCP means a realization with only a single relevant perturbation allowed by microscopic symmetries, so that it can be realized as a generic (pseudo-)critical point. A stable DSL means a realization with no relevant perturbation allowed by microscopic symmetries, so that it can be realized as a stable phase. A stable $\mathrm{SL}^{(7)}$

means a realization with either no relevant perturbation allowed by microscopic symmetries, or a single symmetry-allowed relevant perturbation that does not change the emergent order but only shifts the "zero momenta", so that this realization can still be viewed as a stable phase. All stable realizations of these states, including those on lattice systems discussed here and also those on other lattice systems, are explicitly documented in *ReadMe.nb*.

### K.1 Stable realizations of DQCP

On all these systems, there is a single new stable realization of DQCP, given by Eq. (35), adjacent to ferromagnetic order on triangular lattice, kagome lattice integer spin systems or honeycomb lattice half-integer/integer spin systems. There is a known stable realization of DQCP on the square lattice half-integer spin system, given by Eq. (29), adjacent to anti-ferromagnetic (Neel) order. There is another known stable realization of DQCP on $p6m$-anomaly-free system [17], adjacent to anti-ferromagnetic order on honeycomb lattice half-integer/integer spin systems, given by

$$
T_{1,2} : n \to \begin{pmatrix} I_3 & & \\ & -\frac{1}{2} & \frac{\sqrt{3}}{2} \\ & -\frac{\sqrt{3}}{2} & -\frac{1}{2} \end{pmatrix} n, \quad C_6 : n \to \begin{pmatrix} -I_3 & & \\ & 1 & \\ & & -1 \end{pmatrix} n,
$$
$$
M : n \to \begin{pmatrix} -I_3 & & \\ & 1 & \\ & & 1 \end{pmatrix} n, \quad O(3)^T : n \to \begin{pmatrix} O(3)^T & \\ & I_2 \end{pmatrix} n.
\tag{256}
$$

These three are all stable realizations of DQCP.

### K.2 Stable realizations of DSL

On $p6m$-anomaly-free systems, there is a single stable realization of DSL where the most relevant spin fluctuations carry spin-1, given by Eq. (36). On both $p6m$-anomaly-free systems and $p4m$-anomaly-free systems, there is also a single stable realization of DSL where the most relevant spinful fluctuations carry spin-2, given by Eq. (40). Below we discuss the other systems.

**Triangular lattice half-integer spin systems**

On triangular lattice half-integer spin systems, there are 3 stable realizations of DSL. One of them is known [15, 57, 58, 69–73], given by Eq. (42), adjacent to 120° order. The other two have identical actions of $T_{1,2}$, $C_6$ and $O(3)^T$:

$$
T_1 : n \to \begin{pmatrix} I_3 & & & \\ & 1 & & \\ & & -1 & \\ & & & -1 \end{pmatrix} n, \quad T_2 : n \to \begin{pmatrix} I_3 & & & \\ & -1 & & \\ & & 1 & \\ & & & -1 \end{pmatrix} n,
$$
$$
C_6 : n \to \begin{pmatrix} I_3 & & & \\ & & 1 & \\ & & & 1 \\ & -1 & & \end{pmatrix} n \begin{pmatrix} -1 & \\ & 1 \end{pmatrix}, \quad O(3)^T : n \to \begin{pmatrix} O(3)^T & \\ & I_3 \end{pmatrix} n.
\tag{257}
$$

The action of the mirror symmetry $M$ in these two realizations are respectively

$$
M : n \to \begin{pmatrix} I_3 & & & \\ & & -1 & \\ & -1 & & \\ & & & 1 \end{pmatrix} n,
\tag{258}
$$

and

$$
M : n \to \begin{pmatrix} I_3 & & \\ & 1 & \\ & & 1 & \\ & & & -1 \end{pmatrix} n \begin{pmatrix} -1 & \\ & 1 \end{pmatrix}. \tag{259}
$$

**Kagome lattice half-integer spin systems**

On kagome lattice half-integer spin systems, there are 3 stable realizations of DSL. One of them is known [15, 22, 57, 58, 65–68], adjacent to $q = 0$ order, given by

$$
T_1 : n \to \begin{pmatrix} I_3 & & \\ & 1 & \\ & & -1 & \\ & & & -1 \end{pmatrix} n, \quad T_2 : n \to \begin{pmatrix} I_3 & & \\ & -1 & \\ & & 1 & \\ & & & -1 \end{pmatrix} n,
$$

$$
C_6 : n \to \begin{pmatrix} I_3 & & \\ & 1 & \\ & & 1 & \\ & & & 1 \end{pmatrix} n \begin{pmatrix} -\frac{1}{2} & -\frac{\sqrt{3}}{2} \\ \frac{\sqrt{3}}{2} & -\frac{1}{2} \end{pmatrix},
$$

$$
M : n \to \begin{pmatrix} I_3 & & \\ & -1 & \\ & & -1 & \\ & & & -1 \end{pmatrix} n \begin{pmatrix} -1 & \\ & 1 \end{pmatrix}, \tag{260}
$$

$$
O(3)^T : n \to \begin{pmatrix} O(3)^T & \\ & I_3 \end{pmatrix} n.
$$

The other two have the same actions of $T_{1,2}$, $C_6$ and $O(3)^T$:

$$
C_6 : n \to \begin{pmatrix} I_3 & & \\ & 1 & \\ & & 1 & \\ & & & 1 \end{pmatrix} n, \quad T_1 : n \to \begin{pmatrix} I_3 & & \\ & 1 & \\ & & -1 & \\ & & & -1 \end{pmatrix} n,
$$

$$
T_2 : n \to \begin{pmatrix} I_3 & & \\ & -1 & \\ & & 1 & \\ & & & -1 \end{pmatrix} n, \quad O(3)^T : n \to \begin{pmatrix} O(3)^T & \\ & I_3 \end{pmatrix} n. \tag{261}
$$

And the action of $M$ in the two realizations are respectively

$$
M : n \to \begin{pmatrix} I_3 & & \\ & 1 & \\ & & 1 & \\ & & & 1 \end{pmatrix} n, \tag{262}
$$

and

$$
M : n \to \begin{pmatrix} I_3 & & \\ & -1 & \\ & & -1 & \\ & & & -1 \end{pmatrix} n \begin{pmatrix} -1 & \\ & 1 \end{pmatrix}. \tag{263}
$$

**Square lattice half-integer spin systems**

On square lattice half-integer spin systems, there are 3 stable realizations of DSLs. One of them is given by Eq. (37). The other two have the same actions of $T_{1,2}$, $C_4$ and $O(3)^T$:

$$
\begin{aligned}
T_1 : n &\to \begin{pmatrix} I_3 & & & \\ & -1 & & \\ & & 1 & \\ & & & 1 \end{pmatrix} n \begin{pmatrix} -1 & \\ & 1 \end{pmatrix}, \\[1em]
T_2 : n &\to \begin{pmatrix} I_3 & & & \\ & 1 & & \\ & & -1 & \\ & & & 1 \end{pmatrix} n \begin{pmatrix} -1 & \\ & 1 \end{pmatrix}, \\[1em]
C_4 : n &\to \begin{pmatrix} I_3 & & & \\ & & 1 & \\ & -1 & & \\ & & & -1 \end{pmatrix} n \begin{pmatrix} -1 & \\ & 1 \end{pmatrix}, \\[1em]
O(3)^T : n &\to \begin{pmatrix} O(3)^T & \\ & I_3 \end{pmatrix} n.
\end{aligned}
\tag{264}
$$

The action of $M$ on these two realizations are respectively

$$
M : n \to \begin{pmatrix} I_3 & & & \\ & 1 & & \\ & & -1 & \\ & & & 1 \end{pmatrix} n,
\tag{265}
$$

and

$$
M : n \to \begin{pmatrix} I_3 & & & \\ & -1 & & \\ & & 1 & \\ & & & -1 \end{pmatrix} n \begin{pmatrix} -1 & \\ & 1 \end{pmatrix}.
\tag{266}
$$

### K.3 Stable realizations of SL$^{(7)}$

Below we list the stable realizations of SL$^{(7)}$ on various systems.

**$p6m$-anomaly-free systems**

On $p6m$-anomaly-free systems, there are two stable realizations of SL$^{(7)}$, both of which have the most relevant spinful fluctuations carrying spin-2. The symmetry actions of one of them is given by Eq. (41). The other one has symmetry actions:

$$
\begin{aligned}
SO(3) : n &\to \begin{pmatrix} \varphi_5(SO(3)) & \\ & I_2 \end{pmatrix} n, \quad \mathcal{T} : n \to \begin{pmatrix} I_5 & & \\ & -1 & \\ & & -1 \end{pmatrix} n \begin{pmatrix} -1 & & \\ & 1 & \\ & & 1 \end{pmatrix}, \\[1em]
T_{1,2} : n &\to \begin{pmatrix} I_5 & & \\ & -\frac{1}{2} & \frac{\sqrt{3}}{2} \\ & -\frac{\sqrt{3}}{2} & -\frac{1}{2} \end{pmatrix} n, \quad C_6 : n \to \begin{pmatrix} I_5 & & \\ & 1 & \\ & & -1 \end{pmatrix} n \begin{pmatrix} -1 & & \\ & 1 & \\ & & 1 \end{pmatrix}, \\[1em]
M : n &\to \begin{pmatrix} I_5 & & \\ & -1 & \\ & & -1 \end{pmatrix} n \begin{pmatrix} -1 & & \\ & 1 & \\ & & 1 \end{pmatrix}.
\end{aligned}
\tag{267}
$$

**Triangular lattice half-integer spin systems**

On triangular lattice half-integer spin systems, there are 8 stable realizations of $SL^{(7)}$. The first has appeared in Ref. [15], given by Eq. (238).

The second has symmetry actions:

$$T_1 : n \to n \begin{pmatrix} 1 & & \\ & -1 & \\ & & -1 \end{pmatrix}, \quad T_2 : n \to n \begin{pmatrix} -1 & & \\ & 1 & \\ & & -1 \end{pmatrix},$$

$$C_6 : n \to \begin{pmatrix} I_3 & & & & \\ & -1 & & & \\ & & 1 & & \\ & & & -1 & \\ & & & & 1 \end{pmatrix} n \begin{pmatrix} & & 1 \\ 1 & & \\ & 1 & \end{pmatrix},$$

$$M : n \to \begin{pmatrix} I_3 & & & & \\ & -1 & & & \\ & & -1 & & \\ & & & 1 & \\ & & & & 1 \end{pmatrix} n \begin{pmatrix} & & 1 \\ 1 & & \\ & 1 & \end{pmatrix},$$

$$O(3)^T : n \to \begin{pmatrix} O(3)^T & \\ & I_4 \end{pmatrix} n.$$

(268)

The third has symmetry actions:

$$T_1 : n \to \begin{pmatrix} I_3 & & & & \\ & -\frac{1}{2} & \frac{\sqrt{3}}{2} & & \\ & -\frac{\sqrt{3}}{2} & -\frac{1}{2} & & \\ & & & 1 & \\ & & & & 1 \end{pmatrix} n \begin{pmatrix} 1 & & \\ & -1 & \\ & & -1 \end{pmatrix},$$

$$T_2 : n \to \begin{pmatrix} I_3 & & & & \\ & -\frac{1}{2} & \frac{\sqrt{3}}{2} & & \\ & -\frac{\sqrt{3}}{2} & -\frac{1}{2} & & \\ & & & 1 & \\ & & & & 1 \end{pmatrix} n \begin{pmatrix} -1 & & \\ & 1 & \\ & & -1 \end{pmatrix},$$

$$C_6 : n \to \begin{pmatrix} I_3 & & & & \\ & 1 & & & \\ & & -1 & & \\ & & & -1 & \\ & & & & 1 \end{pmatrix} n \begin{pmatrix} & & 1 \\ 1 & & \\ & 1 & \end{pmatrix},$$

$$M : n \to \begin{pmatrix} I_3 & & & & \\ & -1 & & & \\ & & -1 & & \\ & & & 1 & \\ & & & & 1 \end{pmatrix} n \begin{pmatrix} & & 1 \\ 1 & & \\ & 1 & \end{pmatrix},$$

$$O(3)^T : n \to \begin{pmatrix} O(3)^T & \\ & I_4 \end{pmatrix} n.$$

(269)

The fourth has symmetry actions:

$$
T_1 : n \rightarrow \begin{pmatrix} I_3 & & & & \\ & -\frac{1}{2} & \frac{\sqrt{3}}{2} & & \\ & -\frac{\sqrt{3}}{2} & -\frac{1}{2} & & \\ & & & 1 & \\ & & & & 1 \end{pmatrix} n \begin{pmatrix} 1 & & \\ & -1 & \\ & & -1 \end{pmatrix},
$$

$$
T_2 : n \rightarrow \begin{pmatrix} I_3 & & & & \\ & -\frac{1}{2} & \frac{\sqrt{3}}{2} & & \\ & -\frac{\sqrt{3}}{2} & -\frac{1}{2} & & \\ & & & 1 & \\ & & & & 1 \end{pmatrix} n \begin{pmatrix} -1 & & \\ & 1 & \\ & & -1 \end{pmatrix},
$$

$$
C_6 : n \rightarrow \begin{pmatrix} I_3 & & & & \\ & 1 & & & \\ & & -1 & & \\ & & & -1 & \\ & & & & 1 \end{pmatrix} n \begin{pmatrix} & & 1 \\ 1 & & \\ & 1 & \end{pmatrix},
$$

$$
M : n \rightarrow \begin{pmatrix} I_3 & & & & \\ & 1 & & & \\ & & 1 & & \\ & & & -1 & \\ & & & & -1 \end{pmatrix} n \begin{pmatrix} & & 1 \\ 1 & & \\ & 1 & \end{pmatrix},
$$

$$
O(3)^T : n \rightarrow \begin{pmatrix} O(3)^T & \\ & I_4 \end{pmatrix} n.
$$

(270)

The first to the fourth realization are all adjacent to tetrahedral order.

The fifth has symmetry actions:

$$
T_1 : n \rightarrow n \begin{pmatrix} 1 & & \\ & -1 & \\ & & -1 \end{pmatrix}, \quad T_2 : n \rightarrow n \begin{pmatrix} -1 & & \\ & 1 & \\ & & -1 \end{pmatrix},
$$

$$
C_6 : n \rightarrow \begin{pmatrix} I_3 & & & & \\ & -1 & & & \\ & & 1 & & \\ & & & 1 & \\ & & & & -1 \end{pmatrix} n \begin{pmatrix} & & 1 \\ 1 & & \\ & 1 & \end{pmatrix},
$$

$$
M : n \rightarrow \begin{pmatrix} I_3 & & & & \\ & -1 & & & \\ & & -1 & & \\ & & & -1 & \\ & & & & 1 \end{pmatrix} n \begin{pmatrix} & & -1 \\ -1 & & \\ & -1 & \end{pmatrix},
$$

$$
O(3)^T : n \rightarrow \begin{pmatrix} O(3)^T & \\ & I_4 \end{pmatrix} n.
$$

(271)

The sixth has symmetry actions:

$$
\begin{aligned}
T_1 : n &\to \begin{pmatrix} I_3 & & & \\ & -\frac{1}{2} & \frac{\sqrt{3}}{2} & \\ & -\frac{\sqrt{3}}{2} & -\frac{1}{2} & \\ & & & 1 \\ & & & & 1 \end{pmatrix} n \begin{pmatrix} 1 & & \\ & -1 & \\ & & -1 \end{pmatrix}, \\[2mm]
T_2 : n &\to \begin{pmatrix} I_3 & & & \\ & -\frac{1}{2} & \frac{\sqrt{3}}{2} & \\ & -\frac{\sqrt{3}}{2} & -\frac{1}{2} & \\ & & & 1 \\ & & & & 1 \end{pmatrix} n \begin{pmatrix} -1 & & \\ & 1 & \\ & & -1 \end{pmatrix}, \\[2mm]
C_6 : n &\to \begin{pmatrix} I_3 & & & \\ & 1 & & \\ & & -1 & \\ & & & 1 \\ & & & & -1 \end{pmatrix} n \begin{pmatrix} & & 1 \\ 1 & & \\ & 1 & \end{pmatrix}, \\[2mm]
M : n &\to \begin{pmatrix} I_3 & & & \\ & -1 & & \\ & & -1 & \\ & & & -1 \\ & & & & 1 \end{pmatrix} n \begin{pmatrix} & & -1 \\ -1 & & \\ & & -1 \end{pmatrix}, \\[2mm]
O(3)^T : n &\to \begin{pmatrix} O(3)^T & \\ & I_4 \end{pmatrix} n.
\end{aligned}
\tag{272}
$$

The seventh has symmetry actions:

$$
\begin{aligned}
T_1 : n &\to \begin{pmatrix} I_3 & & & \\ & 1 & & \\ & & 1 & \\ & & & -1 \\ & & & & -1 \end{pmatrix} n \begin{pmatrix} -\frac{1}{2} & -\frac{\sqrt{3}}{2} & \\ \frac{\sqrt{3}}{2} & -\frac{1}{2} & \\ & & 1 \end{pmatrix}, \\[2mm]
T_2 : n &\to \begin{pmatrix} I_3 & & & \\ & 1 & & \\ & & -1 & \\ & & & 1 \\ & & & & -1 \end{pmatrix} n \begin{pmatrix} -\frac{1}{2} & -\frac{\sqrt{3}}{2} & \\ \frac{\sqrt{3}}{2} & -\frac{1}{2} & \\ & & 1 \end{pmatrix}, \\[2mm]
C_6 : n &\to \begin{pmatrix} I_3 & & & \\ & -1 & & \\ & & 1 & \\ & & & 1 \\ & & & & -1 \end{pmatrix} n \begin{pmatrix} 1 & & \\ & -1 & \\ & & -1 \end{pmatrix}, \\[2mm]
M : n &\to \begin{pmatrix} I_3 & & & \\ & 1 & & \\ & & 1 & \\ & & & 1 \\ & & & & -1 \end{pmatrix} n \begin{pmatrix} 1 & & \\ & 1 & \\ & & -1 \end{pmatrix}, \\[2mm]
O(3)^T : n &\to \begin{pmatrix} O(3)^T & \\ & I_4 \end{pmatrix} n.
\end{aligned}
\tag{273}
$$

The eighth has symmetry actions:

$$T_1 : n \to \begin{pmatrix} I_3 & & & & \\ & 1 & & & \\ & & 1 & & \\ & & & -1 & \\ & & & & -1 \end{pmatrix} n \begin{pmatrix} -\frac{1}{2} & -\frac{\sqrt{3}}{2} & \\ \frac{\sqrt{3}}{2} & -\frac{1}{2} & \\ & & 1 \end{pmatrix},$$

$$T_2 : n \to \begin{pmatrix} I_3 & & & & \\ & 1 & & & \\ & & -1 & & \\ & & & 1 & \\ & & & & -1 \end{pmatrix} n \begin{pmatrix} -\frac{1}{2} & -\frac{\sqrt{3}}{2} & \\ \frac{\sqrt{3}}{2} & -\frac{1}{2} & \\ & & 1 \end{pmatrix},$$

$$C_6 : n \to \begin{pmatrix} I_3 & & & & \\ & 1 & & & \\ & & & 1 & \\ & & & & 1 \\ & & -1 & & \end{pmatrix} n \begin{pmatrix} 1 & & \\ & -1 & \\ & & 1 \end{pmatrix}, \qquad (274)$$

$$M : n \to \begin{pmatrix} I_3 & & & & \\ & -1 & & & \\ & & & 1 & \\ & & 1 & & \\ & & & & -1 \end{pmatrix} n \begin{pmatrix} -1 & & \\ & -1 & \\ & & 1 \end{pmatrix},$$

$$O(3)^T : n \to \begin{pmatrix} O(3)^T & \\ & I_4 \end{pmatrix} n.$$

**Kagome lattice half-integer spin systems**

On kagome lattice half-integer spin systems, there are 9 stable realizations of $SL^{(7)}$. The first has appeared in Ref. [15] and is given by Eq. (244). The second is given by Eq. (38). Both realizations are adjacent to cuboc1 order.

The third has symmetry actions

$$T_1 : n \to n \begin{pmatrix} 1 & & \\ & -1 & \\ & & -1 \end{pmatrix}, \quad T_2 : n \to n \begin{pmatrix} -1 & & \\ & 1 & \\ & & -1 \end{pmatrix},$$

$$C_6 : n \to \begin{pmatrix} I_3 & & & & \\ & -1 & & & \\ & & -1 & & \\ & & & 1 & \\ & & & & -1 \end{pmatrix} n \begin{pmatrix} & & -1 \\ 1 & & \\ & 1 & \end{pmatrix}, \qquad (275)$$

$$M : n \to \begin{pmatrix} I_3 & & & & \\ & -1 & & & \\ & & -1 & & \\ & & & -1 & \\ & & & & 1 \end{pmatrix} n \begin{pmatrix} & 1 & \\ 1 & & \\ & & -1 \end{pmatrix},$$

$$O(3)^T : n \to \begin{pmatrix} O(3)^T & \\ & I_4 \end{pmatrix}.$$

The fourth has symmetry actions

$$
T_1 : n \to
\begin{pmatrix}
I_3 & & & & & \\
& -\frac{1}{2} & \frac{\sqrt{3}}{2} & & \\
& -\frac{\sqrt{3}}{2} & -\frac{1}{2} & & \\
& & & 1 & \\
& & & & 1
\end{pmatrix}
n
\begin{pmatrix}
1 & & \\
& -1 & \\
& & -1
\end{pmatrix},
$$

$$
T_2 : n \to
\begin{pmatrix}
I_3 & & & & & \\
& -\frac{1}{2} & \frac{\sqrt{3}}{2} & & \\
& -\frac{\sqrt{3}}{2} & -\frac{1}{2} & & \\
& & & 1 & \\
& & & & 1
\end{pmatrix}
n
\begin{pmatrix}
-1 & & \\
& 1 & \\
& & -1
\end{pmatrix},
$$

$$
C_6 : n \to
\begin{pmatrix}
I_3 & & & & \\
& 1 & & & \\
& & -1 & & \\
& & & -1 & \\
& & & & -1
\end{pmatrix}
n
\begin{pmatrix}
& & -1 \\
1 & & \\
& 1 &
\end{pmatrix},
$$

$$
M : n \to
\begin{pmatrix}
I_3 & & & & \\
& -1 & & & \\
& & -1 & & \\
& & & -1 & \\
& & & & 1
\end{pmatrix}
n
\begin{pmatrix}
& 1 & \\
1 & & \\
& & -1
\end{pmatrix},
$$

$$
O(3)^T : n \to
\begin{pmatrix}
O(3)^T & \\
& I_4
\end{pmatrix}
n.
$$

(276)

The fifth has symmetry actions

$$
T_1 : n \to
\begin{pmatrix}
I_3 & & & & \\
& 1 & & & \\
& & 1 & & \\
& & & -\frac{1}{2} & \frac{\sqrt{3}}{2} \\
& & & -\frac{\sqrt{3}}{2} & -\frac{1}{2}
\end{pmatrix}
n
\begin{pmatrix}
1 & & \\
& -1 & \\
& & -1
\end{pmatrix},
$$

$$
T_2 : n \to
\begin{pmatrix}
I_3 & & & & \\
& 1 & & & \\
& & 1 & & \\
& & & -\frac{1}{2} & \frac{\sqrt{3}}{2} \\
& & & -\frac{\sqrt{3}}{2} & -\frac{1}{2}
\end{pmatrix}
n
\begin{pmatrix}
-1 & & \\
& 1 & \\
& & -1
\end{pmatrix},
$$

$$
C_6 : n \to
\begin{pmatrix}
I_3 & & & & \\
& \frac{1}{2} & \frac{\sqrt{3}}{2} & & \\
& -\frac{\sqrt{3}}{2} & \frac{1}{2} & & \\
& & & 1 & \\
& & & & -1
\end{pmatrix}
n
\begin{pmatrix}
& & -1 \\
1 & & \\
& 1 &
\end{pmatrix},
$$

(277)

$$
M : n \to
\begin{pmatrix}
I_3 & & & & \\
& -1 & & & \\
& & 1 & & \\
& & & -1 & \\
& & & & -1
\end{pmatrix}
n
\begin{pmatrix}
& 1 & \\
1 & & \\
& & -1
\end{pmatrix},
$$

$$O(3)^T : n \rightarrow \begin{pmatrix} O(3)^T & \\ & I_4 \end{pmatrix} n \,.$$

The third to the fifth realization are all adjacent to cuboc2 order.

The sixth has symmetry actions:

$$T_1 : n \rightarrow \begin{pmatrix} I_3 & & & \\ & 1 & & \\ & & 1 & \\ & & & -1 \\ & & & & -1 \end{pmatrix} n \,,$$

$$T_2 : n \rightarrow \begin{pmatrix} I_3 & & & \\ & 1 & & \\ & & -1 & \\ & & & 1 \\ & & & & -1 \end{pmatrix} n \,,$$

$$C_6 : n \rightarrow \begin{pmatrix} I_3 & & & \\ & 1 & & \\ & & 1 & \\ & & & 1 \\ & 1 & & & \end{pmatrix} n \begin{pmatrix} -\frac{1}{2} & -\frac{\sqrt{3}}{2} & \\ \frac{\sqrt{3}}{2} & -\frac{1}{2} & \\ & & 1 \end{pmatrix} \,, \tag{278}$$

$$M : n \rightarrow \begin{pmatrix} I_3 & & & \\ & -1 & & \\ & & 1 & \\ & 1 & & \\ & & & 1 \end{pmatrix} n \begin{pmatrix} -1 & & \\ & 1 & \\ & & 1 \end{pmatrix} \,,$$

$$O(3)^T : n \rightarrow \begin{pmatrix} O(3)^T & \\ & I_4 \end{pmatrix} n \,.$$

The sixth realization is adjacent to $q = 0$ umbrella order.

The seventh has symmetry actions:

$$T_1 : n \rightarrow \begin{pmatrix} I_3 & & & \\ & 1 & & \\ & & 1 & \\ & & & -1 \\ & & & & -1 \end{pmatrix} n \begin{pmatrix} -\frac{1}{2} & -\frac{\sqrt{3}}{2} & \\ \frac{\sqrt{3}}{2} & -\frac{1}{2} & \\ & & 1 \end{pmatrix} \,,$$

$$T_2 : n \rightarrow \begin{pmatrix} I_3 & & & \\ & 1 & & \\ & & -1 & \\ & & & 1 \\ & & & & -1 \end{pmatrix} n \begin{pmatrix} -\frac{1}{2} & -\frac{\sqrt{3}}{2} & \\ \frac{\sqrt{3}}{2} & -\frac{1}{2} & \\ & & 1 \end{pmatrix} \,,$$

$$C_6 : n \rightarrow \begin{pmatrix} I_3 & & & \\ & -1 & & \\ & & 1 & \\ & & & 1 \\ & 1 & & \end{pmatrix} n \begin{pmatrix} 1 & & \\ & -1 & \\ & & 1 \end{pmatrix} \,, \tag{279}$$

$$M : n \to \begin{pmatrix} I_3 & & & & \\ & 1 & & & \\ & & & 1 & \\ & & 1 & & \\ & & & & 1 \end{pmatrix} n \,,$$

$$O(3)^T : n \to \begin{pmatrix} O(3)^T & \\ & I_4 \end{pmatrix} n \,.$$

The seventh realization is adjacent to $q = \sqrt{3} \times \sqrt{3}$ umbrella order.

The eighth has symmetry actions:

$$T_1 : n \to \begin{pmatrix} I_3 & & & \\ & 1 & & \\ & & 1 & \\ & & & -1 \\ & & & & -1 \end{pmatrix} n \begin{pmatrix} -\frac{1}{2} & -\frac{\sqrt{3}}{2} & \\ \frac{\sqrt{3}}{2} & -\frac{1}{2} & \\ & & 1 \end{pmatrix} \,,$$

$$T_2 : n \to \begin{pmatrix} I_3 & & & \\ & 1 & & \\ & & -1 & \\ & & & 1 \\ & & & & -1 \end{pmatrix} n \begin{pmatrix} -\frac{1}{2} & -\frac{\sqrt{3}}{2} & \\ \frac{\sqrt{3}}{2} & -\frac{1}{2} & \\ & & 1 \end{pmatrix} \,,$$

$$C_6 : n \to \begin{pmatrix} I_3 & & & \\ & -1 & & \\ & & & 1 \\ & & & & 1 \\ & 1 & & \end{pmatrix} n \begin{pmatrix} 1 & & \\ & -1 & \\ & & 1 \end{pmatrix} \,,$$

$$M : n \to \begin{pmatrix} I_3 & & & \\ & 1 & & \\ & & & -1 \\ & & -1 & \\ & & & & -1 \end{pmatrix} n \begin{pmatrix} 1 & & \\ & 1 & \\ & & -1 \end{pmatrix} \,,$$

$$O(3)^T : n \to \begin{pmatrix} O(3)^T & \\ & I_4 \end{pmatrix} n \,.$$

The ninth has symmetry actions:

$$T_1 : n \to \begin{pmatrix} I_3 & & & \\ & 1 & & \\ & & 1 & \\ & & & -1 \\ & & & & -1 \end{pmatrix} n \begin{pmatrix} -\frac{1}{2} & -\frac{\sqrt{3}}{2} & \\ \frac{\sqrt{3}}{2} & -\frac{1}{2} & \\ & & 1 \end{pmatrix} \,,$$

$$T_2 : n \to \begin{pmatrix} I_3 & & & \\ & 1 & & \\ & & -1 & \\ & & & 1 \\ & & & & -1 \end{pmatrix} n \begin{pmatrix} -\frac{1}{2} & -\frac{\sqrt{3}}{2} & \\ \frac{\sqrt{3}}{2} & -\frac{1}{2} & \\ & & 1 \end{pmatrix} \,,$$

$$C_6 : n \to \begin{pmatrix} I_3 & & & \\ & -1 & & \\ & & & 1 \\ & & & & 1 \\ & 1 & & \end{pmatrix} n \begin{pmatrix} 1 & & \\ & -1 & \\ & & 1 \end{pmatrix} \,,$$

(280)

(281)

$$M : n \to \begin{pmatrix} I_3 & & & \\ & -1 & & \\ & & -1 & \\ & & & -1 \\ & & & & -1 \end{pmatrix} n \begin{pmatrix} -1 & & \\ & -1 & \\ & & 1 \end{pmatrix},$$

$$O(3)^T : n \to \begin{pmatrix} O(3)^T & \\ & I_4 \end{pmatrix} n.$$

### $p4m$-anomaly-free systems

On $p4m$-anomaly-free systems, there are 4 stable realizations of $\mathrm{SL}^{(7)}$, all of which has the most relevant spinful fluctuations carrying spin-2.

The first has symmetry actions:

$$SO(3) : n \to \begin{pmatrix} \varphi_5(SO(3)) & \\ & I_2 \end{pmatrix} n, \quad \mathcal{T} : n \to \begin{pmatrix} I_5 & & \\ & -1 & \\ & & 1 \end{pmatrix} n,$$

$$T_{1,2} : n \to \begin{pmatrix} I_5 & & \\ & 1 & \\ & & -1 \end{pmatrix} n \begin{pmatrix} -1 & & \\ & 1 & \\ & & 1 \end{pmatrix}, \tag{282}$$

$$C_4 : n \to \begin{pmatrix} I_5 & & \\ & 1 & \\ & & -1 \end{pmatrix} n \begin{pmatrix} 1 & & \\ & -1 & \\ & & 1 \end{pmatrix}, \quad M : n \to \begin{pmatrix} I_5 & & \\ & -1 & \\ & & 1 \end{pmatrix} n.$$

The second has symmetry actions:

$$SO(3) : n \to \begin{pmatrix} \varphi_5(SO(3)) & \\ & I_2 \end{pmatrix} n, \quad \mathcal{T} : n \to \begin{pmatrix} I_5 & & \\ & -1 & \\ & & 1 \end{pmatrix} n,$$

$$T_{1,2} : n \to \begin{pmatrix} I_5 & & \\ & 1 & \\ & & -1 \end{pmatrix} n \begin{pmatrix} -1 & & \\ & 1 & \\ & & 1 \end{pmatrix}, \tag{283}$$

$$C_4 : n \to n \begin{pmatrix} -1 & & \\ & -1 & \\ & & 1 \end{pmatrix}, \quad M : n \to \begin{pmatrix} I_5 & & \\ & -1 & \\ & & -1 \end{pmatrix} n \begin{pmatrix} -1 & & \\ & 1 & \\ & & 1 \end{pmatrix}.$$

The third has symmetry actions:

$$SO(3) : n \to \begin{pmatrix} \varphi_5(SO(3)) & \\ & I_2 \end{pmatrix} n, \quad \mathcal{T} : n \to \begin{pmatrix} I_5 & & \\ & -1 & \\ & & -1 \end{pmatrix} n \begin{pmatrix} -1 & & \\ & 1 & \\ & & 1 \end{pmatrix},$$

$$T_{1,2} : n \to \begin{pmatrix} I_5 & & \\ & -1 & \\ & & 1 \end{pmatrix} n \begin{pmatrix} -1 & & \\ & 1 & \\ & & 1 \end{pmatrix}, \quad C_4 : n \to \begin{pmatrix} I_5 & & \\ & -1 & \\ & & -1 \end{pmatrix} n, \tag{284}$$

$$M : n \to \begin{pmatrix} I_5 & & \\ & 1 & \\ & & -1 \end{pmatrix} n.$$

The fourth has symmetry actions:

$$SO(3): n \to \begin{pmatrix} \varphi_5(SO(3)) & \\ & I_2 \end{pmatrix} n, \quad \mathcal{T}: n \to \begin{pmatrix} I_5 & & \\ & -1 & \\ & & -1 \end{pmatrix} n \begin{pmatrix} -1 & & \\ & 1 & \\ & & 1 \end{pmatrix},$$

$$T_{1,2}: n \to \begin{pmatrix} I_5 & & \\ & -1 & \\ & & 1 \end{pmatrix} n \begin{pmatrix} -1 & & \\ & 1 & \\ & & 1 \end{pmatrix},$$

$$C_4: n \to \begin{pmatrix} I_5 & & \\ & 1 & \\ & & -1 \end{pmatrix} n \begin{pmatrix} -1 & & \\ & 1 & \\ & & 1 \end{pmatrix},$$

$$M: n \to \begin{pmatrix} I_5 & & \\ & -1 & \\ & & -1 \end{pmatrix} n \begin{pmatrix} -1 & & \\ & 1 & \\ & & 1 \end{pmatrix}. \tag{285}$$

**Square lattice half-integer spin systems**

On square lattice half-integer spin systems, there are 2 stable realizations of $SL^{(7)}$ where the most relevant spinful fluctuations have spin-1 and all $n$ modes are at high-symmetry momenta in the Brillouin zone. There are also three realizations where some $n$ modes can have continuously changing momenta, among which two of them have only a single symmetric relevant perturbation that shifts the momenta of the $n$ modes, as long as these momenta are not tuned to high-symmetry point. This is the case for the third family of realizations for most non-high-symmetry momenta, except at two special momentum points (see below). Furthermore, there is also one stable realization where the most relevant spinful fluctuations have spin-2 and all $n$ modes are at high-symmetry momenta.

We start with the 2 realizations with the most relevant spinful fluctuations carrying spin-1 and all $n$ modes locating at high-symmetry momenta. The first has symmetry actions:

$$T_1: n \to \begin{pmatrix} I_3 & & & \\ & -1 & & \\ & & 1 & \\ & & & 1 \\ & & & & 1 \end{pmatrix} n \begin{pmatrix} -1 & & \\ & 1 & \\ & & 1 \end{pmatrix},$$

$$T_2: n \to \begin{pmatrix} I_3 & & & \\ & 1 & & \\ & & -1 & \\ & & & 1 \\ & & & & 1 \end{pmatrix} n \begin{pmatrix} -1 & & \\ & 1 & \\ & & 1 \end{pmatrix},$$

$$C_4: n \to \begin{pmatrix} I_3 & & & \\ & 1 & & \\ & & -1 & \\ & & & 1 \\ & & & & 1 \end{pmatrix} n \begin{pmatrix} 1 & & \\ & -1 & \\ & & -1 \end{pmatrix}, \tag{286}$$

$$M: n \to \begin{pmatrix} I_3 & & & \\ & -1 & & \\ & & 1 & \\ & & & -1 \\ & & & & -1 \end{pmatrix} n \begin{pmatrix} -1 & & \\ & -1 & \\ & & 1 \end{pmatrix},$$

$$O(3)^T : n \to \begin{pmatrix} O(3)^T & \\ & I_4 \end{pmatrix} n.$$

The second has symmetry actions:

$$T_1 : n \to \begin{pmatrix} I_3 & & & \\ & -1 & & \\ & & 1 & \\ & & & -1 \\ & & & & 1 \end{pmatrix} n \begin{pmatrix} -1 & & \\ & -1 & \\ & & 1 \end{pmatrix},$$

$$T_2 : n \to \begin{pmatrix} I_3 & & & \\ & 1 & & \\ & & -1 & \\ & & & -1 \\ & & & & 1 \end{pmatrix} n \begin{pmatrix} -1 & & \\ & -1 & \\ & & 1 \end{pmatrix},$$

$$C_4 : n \to \begin{pmatrix} I_3 & & & \\ & 1 & & \\ & & -1 & \\ & & & 1 \\ & & & & 1 \end{pmatrix} n \begin{pmatrix} -1 & & \\ & -1 & \\ & & 1 \end{pmatrix},$$

$$M : n \to \begin{pmatrix} I_3 & & & \\ & -1 & & \\ & & 1 & \\ & & & 1 \\ & & & & -1 \end{pmatrix} n \begin{pmatrix} -1 & & \\ & 1 & \\ & & 1 \end{pmatrix},$$

$$O(3)^T : n \to \begin{pmatrix} O(3)^T & \\ & I_4 \end{pmatrix} n.$$

(287)

Next, we turn to the three realizations with some $n$ modes at continuously changing momenta. The first has symmetry actions given by Eq. (39), adjacent to tetrahedral umbrella order, and the second has symmetry actions

$$C_4 : n \to \begin{pmatrix} I_3 & & & \\ & & 1 & \\ & & & 1 \\ & 1 & & \\ 1 & & & \end{pmatrix} n \begin{pmatrix} & 1 & \\ 1 & & \\ & & 1 \end{pmatrix},$$

$$M : n \to \begin{pmatrix} I_3 & & & \\ & -1 & & \\ & & -1 & \\ & & & -1 \\ & & & & -1 \end{pmatrix} n \begin{pmatrix} -1 & & \\ & -1 & \\ & & 1 \end{pmatrix},$$

$$T_1 : n \to \begin{pmatrix} I_3 & & \\ & \cos k & \sin k \\ & -\sin k & \cos k \\ & & & I_2 \end{pmatrix} n \begin{pmatrix} -1 & & \\ & 1 & \\ & & -1 \end{pmatrix},$$

(288)

$$T_2 : n \to \begin{pmatrix} I_3 & & \\ & I_2 & \\ & & \cos k & -\sin k \\ & & \sin k & \cos k \end{pmatrix} n \begin{pmatrix} 1 & & \\ & -1 & \\ & & -1 \end{pmatrix},$$

$$O(3)^T : n \to \begin{pmatrix} O(3)^T & \\ & I_4 \end{pmatrix} n,$$

where $k \in (-\pi, \pi)$ is a generic momentum. In both of these two realizations, the only relevant perturbation that is allowed by microscopic symmetries is the one that shifts the momenta of the $n$ modes.

The third has symmetry actions:

$$C_4 : n \to \begin{pmatrix} I_3 & & & & \\ & & 1 & & \\ & & & 1 & \\ & 1 & & & \\ & & & & 1 \end{pmatrix} n \begin{pmatrix} & 1 & \\ 1 & & \\ & & 1 \end{pmatrix},$$

$$M : n \to \begin{pmatrix} I_3 & & & & \\ & & & -1 & \\ & & -1 & & \\ & -1 & & & \\ -1 & & & & \end{pmatrix} n \begin{pmatrix} 1 & & \\ & 1 & \\ & & -1 \end{pmatrix},$$

$$T_1 : n \to \begin{pmatrix} I_3 & & & & \\ & \cos k & \sin k & & \\ & -\sin k & \cos k & & \\ & & & \cos k & \sin k \\ & & & -\sin k & \cos k \end{pmatrix} n \begin{pmatrix} -1 & & \\ & 1 & \\ & & -1 \end{pmatrix},$$

$$T_2 : n \to \begin{pmatrix} I_3 & & & & \\ & \cos k & \sin k & & \\ & -\sin k & \cos k & & \\ & & & \cos k & -\sin k \\ & & & \sin k & \cos k \end{pmatrix} n \begin{pmatrix} 1 & & \\ & -1 & \\ & & -1 \end{pmatrix},$$

$$O(3)^T : n \to \begin{pmatrix} O(3)^T & \\ & I_4 \end{pmatrix} n,$$
(289)

where $k \in (-\pi, \pi)$ is a generic momentum. For this family of realizations, as long as $k \neq \pm \pi/2$, the only symmetric relevant perturbation is the one that shifts the momenta of the $n$ modes. When $k = \pm \pi/2$, besides this symmetric relevant perturbation, there is an additional one that can change the emergent order of SL$^{(7)}$ and make it unstable.

Finally, there is one realization where all $n$ modes are at high-symmetry momentum, and the most spinful fluctuations have spin-2. It has symmetry actions:

$$SO(3) : n \to \begin{pmatrix} \varphi_5(SO(3)) & \\ & I_2 \end{pmatrix} n, \quad \mathcal{T} : n \to \begin{pmatrix} I_5 & & \\ & -1 & \\ & & -1 \end{pmatrix} n \begin{pmatrix} -1 & & \\ & 1 & \\ & & 1 \end{pmatrix},$$

$$T_1 : n \to \begin{pmatrix} I_5 & & \\ & -1 & \\ & & 1 \end{pmatrix} n \begin{pmatrix} -1 & & \\ & 1 & \\ & & 1 \end{pmatrix},$$
(290)

$$T_2 : n \to \begin{pmatrix} I_5 & & \\ & 1 & \\ & & -1 \end{pmatrix} n \begin{pmatrix} -1 & & \\ & 1 & \\ & & 1 \end{pmatrix},$$

$$C_4 : n \to \begin{pmatrix} I_5 & & \\ & & 1 \\ & -1 & \end{pmatrix} n, \quad M : n \to \begin{pmatrix} I_5 & & \\ & 1 & \\ & & -1 \end{pmatrix} n.$$

## L Classical regular magnetic orders

Ref. [60] studied regular magnetic orders, i.e., magnetic orders that respect all the lattice symmetries modulo global $O(3)^T$ spin transformations (rotations and/or spin flips). In particular, on triangular, kagome, honeycomb and square lattices, all *classical* regular magnetic orders are classified. These classical orders can all be realized by a product state, where each spin moment on the lattice can be assigned a definite orientation. In this appendix, we explicitly write down the spin configurations of these classical regular magnetic orders, and the lattice symmetry actions on the order parameters.

In terms of the symmetry breaking pattern of the spin $O(3)^T$ symmetry, there are three types of magnetic orders: collinear, coplanar and non-coplanar. The order parameter of a collinear magnetic order is a three-component vector, $\boldsymbol{n}$, which transforms in the spin-1 representation of the $O(3)^T$ spin symmetry. The order parameters of a coplanar magnetic order consists of two orthonormal three-component vectors, $\boldsymbol{n}_{1,2}$, both transforming in the spin-1 representation of the $O(3)^T$ spin symmetry. The order parameters of a non-coplanar magnetic order consists of three orthonormal three-component vectors, $\boldsymbol{n}_{1,2,3}$, all transforming in the spin-1 representation of the $O(3)^T$ spin symmetry.

We start from the triangular lattice. We will denote the position $\boldsymbol{r}$ of a site on a triangular lattice by its coordinates in the basis of translation vectors of $T_{1,2}$ (see Fig. 2), such that $\boldsymbol{r} = x\boldsymbol{T}_1 + y\boldsymbol{T}_2$, where $\boldsymbol{T}_{1,2}$ is the translation vector of $T_{1,2}$. Under the $p6m$ symmetry,

$$
\begin{aligned}
&T_1 : (x,y) \rightarrow (x+1, y), \\
&T_2 : (x,y) \rightarrow (x, y+1), \\
&C_6 : (x,y) \rightarrow (x-y, x), \\
&M : (x,y) \rightarrow (y,x).
\end{aligned}
\tag{291}
$$

1. There is a single collinear classical regular magnetic order, the ferromagnetic order, where $\boldsymbol{S}(x,y) = \boldsymbol{n}$. Under the $p6m$ symmetry, $\boldsymbol{n}$ is invariant.

2. There is a single coplanar classical regular magnetic order, the 120° order, where $\boldsymbol{S}(x,y) = (-1)^{x+y} \cos\frac{\pi(x+y)}{3} \boldsymbol{n}_1 + (-1)^{x+y} \sin\frac{\pi(x+y)}{3} \boldsymbol{n}_2$. Under the $p6m$ symmetry,

$$
\begin{aligned}
&T_{1,2} : \boldsymbol{n}_1 \rightarrow -\frac{1}{2}\boldsymbol{n}_1 - \frac{\sqrt{3}}{2}\boldsymbol{n}_2, \quad \boldsymbol{n}_2 \rightarrow \frac{\sqrt{3}}{2}\boldsymbol{n}_1 - \frac{1}{2}\boldsymbol{n}_2, \\
&C_6 : \boldsymbol{n}_1 \rightarrow \boldsymbol{n}_1, \quad \boldsymbol{n}_2 \rightarrow -\boldsymbol{n}_2, \\
&M : \boldsymbol{n}_{1,2} \rightarrow \boldsymbol{n}_{1,2}.
\end{aligned}
\tag{292}
$$

3. There are two non-coplanar classical regular magnetic order. The first is the tetrahedral order, where $\boldsymbol{S}(x,y) = (-1)^x \boldsymbol{n}_1 + (-1)^y \boldsymbol{n}_2 + (-1)^{x+y} \boldsymbol{n}_3$. Under the $p6m$ symmetry,

$$
\begin{aligned}
&T_1 : \boldsymbol{n}_1 \rightarrow -\boldsymbol{n}_1, \quad \boldsymbol{n}_2 \rightarrow \boldsymbol{n}_2, \quad \boldsymbol{n}_3 \rightarrow -\boldsymbol{n}_3, \\
&T_2 : \boldsymbol{n}_1 \rightarrow \boldsymbol{n}_1, \quad \boldsymbol{n}_2 \rightarrow -\boldsymbol{n}_2, \quad \boldsymbol{n}_3 \rightarrow -\boldsymbol{n}_3, \\
&C_6 : \boldsymbol{n}_1 \rightarrow \boldsymbol{n}_2, \quad \boldsymbol{n}_2 \rightarrow \boldsymbol{n}_3, \quad \boldsymbol{n}_3 \rightarrow \boldsymbol{n}_1, \\
&M : \boldsymbol{n}_1 \rightarrow \boldsymbol{n}_2, \quad \boldsymbol{n}_2 \rightarrow \boldsymbol{n}_1, \quad \boldsymbol{n}_3 \rightarrow \boldsymbol{n}_3.
\end{aligned}
\tag{293}
$$

4. The second non-coplanar classical regular magnetic order is the F-umbrella order, where $\boldsymbol{S}(x,y) = (-1)^{x+y} \cos\frac{\pi(x+y)}{3} \sin\theta \boldsymbol{n}_1 + (-1)^{x+y} \sin\frac{\pi(x+y)}{3} \sin\theta \boldsymbol{n}_2 + \cos\theta \boldsymbol{n}_3$, with $\theta$ a free parameter. Under the $p6m$ symmetry,

$$
\begin{aligned}
&T_{1,2} : \boldsymbol{n}_1 \rightarrow -\frac{1}{2}\boldsymbol{n}_1 + \frac{\sqrt{3}}{2}\boldsymbol{n}_2, \quad \boldsymbol{n}_2 \rightarrow -\frac{\sqrt{3}}{2}\boldsymbol{n}_1 - \frac{1}{2}\boldsymbol{n}_2, \quad \boldsymbol{n}_3 \rightarrow \boldsymbol{n}_3, \\
&C_6 : \boldsymbol{n}_1 \rightarrow \boldsymbol{n}_1, \quad \boldsymbol{n}_2 \rightarrow -\boldsymbol{n}_2, \quad \boldsymbol{n}_3 \rightarrow \boldsymbol{n}_3, \\
&M : \boldsymbol{n}_{1,2,3} \rightarrow \boldsymbol{n}_{1,2,3}.
\end{aligned}
\tag{294}
$$



Next we turn to the kagome lattice. Each unit cell in a kagome lattice includes three sites, so the spin configuration will be written as $S_i(x, y)$, where $(x, y)$ labels the position of the unit cell in the same way as the triangular lattice, and $i = 1, 2, 3$ represents the site obtained by applying a half translation $T_1/2$, $T_2/2$ and $(T_1 + T_2)/2$ to the $C_6$ center of the unit cell, respectively.

1. There is a single collinear classical regular magnetic order, the ferromagnetic order, where $S_i(x, y) = n$ for $i = 1, 2, 3$. Under the $p6m$ symmetry, $n$ is invariant.

2. There are two coplanar classical regular magnetic orders. The first is the $q = 0$ order, where $S_1(x, y) = n_1$, $S_2(x, y) = -\frac{1}{2}n_1 + \frac{\sqrt{3}}{2}n_2$, and $S_3(x.y) = -\frac{1}{2}n_1 - \frac{\sqrt{3}}{2}n_2$. Under the $p6m$ symmetry,

$$
\begin{aligned}
T_{1,2} &: n_{1,2} \to n_{1,2}, \\
C_6 &: n_1 \to -\frac{1}{2}n_1 - \frac{\sqrt{3}}{2}n_2, \quad n_2 \to \frac{\sqrt{3}}{2}n_1 - \frac{1}{2}n_2, \\
M &: n_1 \to -\frac{1}{2}n_1 + \frac{\sqrt{3}}{2}n_2, \quad n_2 \to \frac{\sqrt{3}}{2}n_1 + \frac{1}{2}n_2.
\end{aligned}
\tag{295}
$$

3. The second coplanar classical regular magnetic order is the $q = \sqrt{3} \times \sqrt{3}$ order, where $S_1(x, y) = (-1)^{x+y} \cos \frac{\pi(x+y)}{3} n_1 + (-1)^{x+y} \sin \frac{\pi(x+y)}{3} n_2$, $S_2(x, y) = S_1(x, y)$, and $S_3(x, y) = (-1)^{x+y} \cos \frac{\pi(x+y+2)}{3} n_1 + (-1)^{x+y} \sin \frac{\pi(x+y+2)}{3} n_2$. Under the $p6m$ symmetry,

$$
\begin{aligned}
T_{1,2} &: n_1 \to -\frac{1}{2}n_1 - \frac{\sqrt{3}}{2}n_2, \quad n_2 \to \frac{\sqrt{3}}{2}n_1 - \frac{1}{2}n_2, \\
C_6 &: n_1 \to -\frac{1}{2}n_1 + \frac{\sqrt{3}}{2}n_2, \quad n_2 \to \frac{\sqrt{3}}{2}n_1 + \frac{1}{2}n_2, \\
M &: n_{1,2} \to n_{1,2}.
\end{aligned}
\tag{296}
$$

4. There are five non-coplanar classical regular magnetic orders. The first is the octahedral order, where $S_1(x, y) = (-1)^y n_1$, $S_2(x, y) = (-1)^x n_2$ and $S_3(x, y) = (-1)^{x+y} n_3$. Under the $p6m$ symmetry,

$$
\begin{aligned}
T_1 &: n_1 \to n_1, \quad n_2 \to -n_2, \quad n_3 \to (-1)^{x+y} n_3, \\
T_2 &: n_1 \to -n_1, \quad n_2 \to n_2, \quad n_3 \to -n_3, \\
C_6 &: n_1 \to n_3, \quad n_2 \to n_1, \quad n_3 \to n_2, \\
M &: n_1 \to n_2, \quad n_2 \to n_1, \quad n_3 \to n_3.
\end{aligned}
\tag{297}
$$

5. The second non-coplanar classical regular magnetic order is the cuboc1 order, where $S_1(x, y) = (-1)^x n_2 + (-1)^{x+y} n_3$, $S_2(x, y) = (-1)^y n_1 + (-1)^{x+y} n_3$ and $S_3(x, y) = -(-1)^x n_2 - (-1)^y n_1$. Under the $p6m$ symmetry,

$$
\begin{aligned}
T_1 &: n_1 \to n_1, \quad n_2 \to -n_2, \quad n_3 \to -n_3, \\
T_2 &: n_1 \to -n_1, \quad n_2 \to n_2, \quad n_3 \to -n_3, \\
C_6 &: n_1 \to -n_3, \quad n_2 \to -n_1, \quad n_3 \to -n_2, \\
M &: n_1 \to n_2, \quad n_2 \to n_1, \quad n_3 \to n_3.
\end{aligned}
\tag{298}
$$

6. The third non-coplanar classical regular magnetic order is the cuboc2 order, where $S_1(x,y) = (-1)^x n_2 - (-1)^{x+y} n_3$, $S_2(x,y) = (-1)^y n_1 + (-1)^{x+y} n_3$ and $S_3(x,y) = (-1)^x n_2 + (-1)^y n_1$. Under the $p6m$ symmetry,

$$
\begin{aligned}
T_1 &: n_1 \to n_1, & n_2 &\to -n_2, & n_3 &\to -n_3, \\
T_2 &: n_1 \to -n_1, & n_2 &\to n_2, & n_3 &\to -n_3, \\
C_6 &: n_1 \to n_3, & n_2 &\to n_1, & n_3 &\to -n_2, \\
M &: n_1 \to n_2, & n_2 &\to n_1, & n_3 &\to -n_3.
\end{aligned}
\tag{299}
$$

7. The fourth non-coplanar is the $q = 0$ umbrella order, where $S_1(x,y) = \sin\theta n_1 + \cos\theta n_3$, $S_2(x,y) = -\frac{1}{2}\sin\theta n_1 + \frac{\sqrt{3}}{2}\sin\theta n_2 + \cos\theta n_3$, and $S_3(x,y) = -\frac{1}{2}\sin\theta n_1 - \frac{\sqrt{3}}{2}\sin\theta n_2 + \cos\theta n_3$, with $\theta$ a free parameter. Under the $p6m$ symmetry,

$$
\begin{aligned}
T_{1,2} &: n_{1,2,3} \to n_{1,2,3}, \\
C_6 &: n_1 \to -\frac{1}{2}n_1 - \frac{\sqrt{3}}{2}n_2, & n_2 &\to \frac{\sqrt{3}}{2}n_1 - \frac{1}{2}n_2, & n_3 &\to n_3, \\
M &: n_1 \to -\frac{1}{2}n_1 + \frac{\sqrt{3}}{2}n_2, & n_2 &\to \frac{\sqrt{3}}{2}n_1 + \frac{1}{2}n_2, & n_3 &\to n_3.
\end{aligned}
\tag{300}
$$

8. The last non-coplanar classical regular magnetic order is the $q = \sqrt{3} \times \sqrt{3}$ umbrella order, where $S_1(x,y) = (-1)^{x+y}\cos\frac{\pi(x+y)}{3}\sin\theta n_1 + (-1)^{x+y}\sin\frac{\pi(x+y)}{3}\sin\theta n_2 + \cos\theta n_3$, $S_2(x,y) = S_1(x,y)$, and $S_3(x,y) = -(-1)^{x+y}\cos\frac{\pi(x+y-1)}{3}\sin\theta n_1 - (-1)^{x+y}\sin\frac{\pi(x+y-1)}{3}\sin\theta n_2 + \cos\theta n_3$. Under the $p6m$ symmetry,

$$
\begin{aligned}
T_{1,2} &: n_1 \to -\frac{1}{2}n_1 - \frac{\sqrt{3}}{2}n_2, & n_2 &\to \frac{\sqrt{3}}{2}n_1 - \frac{1}{2}n_2, & n_3 &\to n_3, \\
C_6 &: n_1 \to -\frac{1}{2}n_1 + \frac{\sqrt{3}}{2}n_2, & n_2 &\to \frac{\sqrt{3}}{2}n_1 + \frac{1}{2}n_2, & n_3 &\to n_3, \\
M &: n_{1,2,3} \to n_{1,2,3}.
\end{aligned}
\tag{301}
$$

Now we turn to the honeycomb lattice. Each unit cell of a honeycomb lattice includes two sites, so the spin configuration will be written in terms of $S_A(x,y)$ and $S_B(x,y)$, where the $A$ and $B$ sublattice can be obtained by translating by $\frac{2T_1+T_2}{3}$ and $\frac{T_1-T_2}{3}$ from the $C_6$ center, respectively.

1. There are two collinear classical regular magnetic orders. The first is the ferromagnetic order, where $S_A(x,y) = S_B(x,y) = n$. Under the $p6m$ symmetry, $n$ is invariant.

2. The second collinear classical regular magnetic order is the anti-ferromagnetic order, where $S_A(x,y) = -S_B(x,y) = n$. Under the $p6m$ symmetry,

$$
\begin{aligned}
T_{1,2} &: n \to n, \\
C_6 &: n \to -n, \\
M &: n \to -n.
\end{aligned}
\tag{302}
$$

3. There is a single coplanar classical regular magnetic order, the $V$ order, where $S_A(x,y) = \cos\theta n_1 - \sin\theta n_2$ and $S_B(x,y) = \cos\theta n_1 + \sin\theta n_2$, with $\theta$ a free parameter. Under the $p6m$ symmetry,

$$
\begin{aligned}
T_{1,2} &: n_{1,2} \to n_{1,2}, \\
C_6 &: n_1 \to n_1, & n_2 &\to -n_2, \\
M &: n_1 \to n_1, & n_2 &\to -n_2.
\end{aligned}
\tag{303}
$$

4. There are two non-coplanar classical regular magnetic orders. The first is the cubic order, where $S_A(x,y) = (-1)^x n_1 + (-1)^y n_2 + (-1)^{x+y} n_3$ and $S_B(x,y) = (-1)^x n_1 - (-1)^y n_2 + (-1)^{x+y} n_3$. Under the $p6m$ symmetry,

$$
\begin{aligned}
T_1 &: n_1 \to -n_1, \quad n_2 \to n_2, \quad n_3 \to -n_3, \\
T_2 &: n_1 \to n_1, \quad n_2 \to -n_2, \quad n_3 \to -n_3, \\
C_6 &: n_1 \to n_2, \quad n_2 \to -n_3, \quad n_3 \to n_1, \\
M &: n_1 \to n_2, \quad n_2 \to n_1, \quad n_3 \to -n_3.
\end{aligned}
\tag{304}
$$

5. The second non-coplanar classical regular magnetic order is the tetrahedral order, where $S_A(x,y) = (-1)^x n_1 + (-1)^y n_2 + (-1)^{x+y} n_3$ and $S_B(x,y) = -(-1)^x n_1 + (-1)^y n_2 - (-1)^{x+y} n_3$. Under the $p6m$ symmetry,

$$
\begin{aligned}
T_1 &: n_1 \to -n_1, \quad n_2 \to n_2, \quad n_3 \to -n_3, \\
T_2 &: n_1 \to n_1, \quad n_2 \to -n_2, \quad n_3 \to -n_3, \\
C_6 &: n_1 \to -n_2, \quad n_2 \to n_3, \quad n_3 \to -n_1, \\
M &: n_1 \to -n_2, \quad n_2 \to -n_1, \quad n_3 \to n_3.
\end{aligned}
\tag{305}
$$

Finally, we discuss the square lattice. We will denote the position of a site by its coordinates in the basis of translation vectors of $T_{1,2}$ (see Fig. 3). such that $r = x T_1 + y T_2$, where $T_{1,2}$ is the translation vector of $T_{1,2}$. Under the $p4m$ symmetry,

$$
\begin{aligned}
T_1 &: (x,y) \to (x+1, y), \\
T_2 &: (x,y) \to (x, y+1), \\
C_4 &: (x,y) \to (-y, x), \\
M &: (x,y) \to (-x, y).
\end{aligned}
\tag{306}
$$

1. There are two collinear classical regular magnetic orders. The first is the ferromagnetic order, where $S(x,y) = n$. Under the $p4m$ symmetry, $n$ is invariant.

2. The second collinear classical regular magnetic order is the anti-ferromagnetic order, where $S(x,y) = (-1)^{x+y} n$. Under the $p4m$ symmetry,

$$
\begin{aligned}
T_{1,2} &: n \to -n, \\
C_4 &: n \to -n, \\
M &: n \to n.
\end{aligned}
\tag{307}
$$

3. There are two coplanar classical regular magnetic orders. The first is the orthogonal order, where $S(x,y) = \frac{(-1)^x + (-1)^y}{2} n_1 + \frac{-(-1)^x + (-1)^y}{2} n_2$. Under the $p4m$ symmetry,

$$
\begin{aligned}
T_1 &: n_1 \to n_2, \quad n_2 \to n_1, \\
T_2 &: n_1 \to -n_2, \quad n_2 \to -n_1, \\
C_4 &: n_1 \to n_1, \quad n_2 \to -n_2, \\
M &: n_{1,2} \to n_{1,2}.
\end{aligned}
\tag{308}
$$

4. The second coplanar classical regular magnetic order is the $V$ order, where $S(x,y) = \cos\theta\, n_1 - (-1)^{x+y} \sin\theta\, n_2$, where $\theta$ a free parameter. Under the $p4m$ symmetry,

$$
\begin{aligned}
T_{1,2} &: n_1 \to n_1, \quad n_2 \to -n_2, \\
C_4 &: n_{1,2} \to n_{1,2}, \\
M &: n_{1,2} \to n_{1,2}.
\end{aligned}
\tag{309}
$$

5. There are two non-coplanar classical regular magnetic orders. The first is the tetrahedral umbrealla order (also known as the AF umbrella order), where $S(x,y) = \frac{(-1)^x \sin\theta}{\sqrt{2}} n_1 - \frac{(-1)^y \sin\theta}{\sqrt{2}} n_2 - (-1)^{x+y} \cos\theta n_3$, with $\theta$ a free parameter. Under the $p4m$ symmetry,

$$
\begin{aligned}
T_1 &: n_1 \to -n_1, & n_2 \to n_2, & \quad n_3 \to -n_3, \\
T_2 &: n_1 \to n_1, & n_2 \to -n_2, & \quad n_3 \to -n_3, \\
C_4 &: n_1 \to -n_2, & n_2 \to -n_1, & \quad n_3 \to n_3, \\
M &: n_{1,2,3} \to n_{1,2,3}.
\end{aligned}
\tag{310}
$$

6. The second non-coplanar classical regular magnetic order is the umbrella order (also known as the F umbrella order), where $S(x,y) = \cos\theta n_1 + \frac{(-1)^x \sin\theta}{\sqrt{2}} n_2 + \frac{(-1)^y \sin\theta}{\sqrt{2}} n_3$. Under the $p4m$ symmetry,

$$
\begin{aligned}
T_1 &: n_1 \to n_1, & n_2 \to -n_2, & \quad n_3 \to n_3, \\
T_2 &: n_1 \to n_1, & n_2 \to n_2, & \quad n_3 \to -n_3, \\
C_4 &: n_1 \to n_1, & n_2 \to n_3, & \quad n_3 \to n_2, \\
M &: n_{1,2,3} \to n_{1,2,3}.
\end{aligned}
\tag{311}
$$

# M  Stability of DSL realizations on NaYbO$_2$ and twisted bilayer WSe$_2$

In this appendix, we discuss the stability of a few more examples of DSL realizations on systems with spin-orbit coupling (SOC). The specific systems we have in mind are NaYbO$_2$ and twisted bilayer WSe$_2$ (tWSe$_2$). Recently, it was pointed out that tWSe$_2$ is a good quantum simulator of triangular lattice Hubbard model, which can be effectively described by a triangular lattice spin-1/2 system in the strong coupling regime [86–88].

The symmetries of NaYbO$_2$ are given in Eq. (43). The symmetries of tWSe$_2$ are

$$
T_{1,2}, \ C_3 \equiv C_6^2, \ SO(2), \ \mathcal{T},
\tag{312}
$$

where $SO(2)$ is a reduced spin rotational symmetry [18].

On triangular lattice spin-1/2 systems with the full $p6m \times O(3)^T$ symmetry, our exhaustive search finds 3 realizations of DSL, given by Eqs. (42), (257) and (258). Using these symmetry actions, it is straightforward to see that for all three realizations, the remaining symmetries of NaYbO$_2$ are sufficient to forbid all relevant operators of DSL listed in Sec. 3.1. However, for the symmetry setting of tWSe$_2$ and for all three realizations, the $(A_L, A_R)$ operator (the fermion mass that transforms in the adjoint representation of the flavor symmetry) is always symmetry-allowed and will destablize the DSL. This means if a DSL is stably realized in tWSe$_2$, that realization cannot be compatible with a full $p6m \times O(3)^T$ symmetry.

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
