# Peer review of "Topological characterization of Lieb-Schultz-Mattis constraints and applications to symmetry-enriched quantum criticality"

_SciPost Physics, doi:SciPost Phys. 13, 066 (2022)_

## Round 1 · Referee Report · Anonymous (Referee 1) · 2022-3-24

Strengths

1- An effective sigma-model framework capable of describing a variety of gapless states in the IR. 2-A framwork to match UV symmetries and anomalies to those of the sigma model to test the possibility of the gapless state emerging from the micriscopic system. 3- Detailed examples, tabulated results and appendices.

Weaknesses

1- Introduction of several new terminology which might create some confusion in the field. 2- Reliance on relatively involved mathematical concepts, although I appreciate that this might have been necessary for the approach of the authors.

Report

The manuscript concerns the possible long-distance fates of two-dimensional quantum many-body systems with internal and lattice symmetries. In particular, the authors focus on how the LSM constraints present in the kinematics of the microscopic systems i.e. the projective class of on-site representations and the group of lattice symmetries, constrain and enrich possible gapless states at long distances. Below, I briefly summarize the key results of the manuscript.

1) A review of LSM constraints via lattice homotopy and a recharacterization of LSM anomalies in terms of higher-dimensional topological partition functions. 2) A review of the effective framework to characterize a variety of gapless states in terms of matrix valued nonlinear sigma models indexed by an integer N, SL(N). Depending on the choice of N, this effective model can describe various 2+1 dimensional gapless states such as the deconfined quantum critical point, the Dirac spin liquid or the various Steifel liquids introduced earlier by the authors. The symmetries and anomalies of SL(N) are specified. 3) A description of the homomorphism $\phi$ between the UV and IR symmetries as well as the relation between the anomalies via pullback. This establishes a means to test the authors' "emergability hypothesis" to see if the anomaly matches, and thus the gapless IR state can be realized starting from the microscopic premise. This is, in my understanding, the main result of the paper. 4) A demonstration of using the emergability hypothesis is presented using various examples. When emergability is established, the framework also allows a way to classify the distinct ways in which the gapless states SL(N) can be "enriched" by determining the distinct anomaly-compatible homomorphisms. The results are tabulated for various symmetry groups of interest. 5) Particular examples of experimental interest are also highlighted. For example, the authors determine that when certain symmetries are explicitly broken, as is expected in practical situations due to spin-orbit coupling, the residual symmetries of NaYbO2 still allow the emergence of DSL.

The results are, to the best of my understanding, scientifically correct and of interest to researchers working on theoretical and experimental aspects of strongly coupled quantum systems. My recommendation is that the manuscript is suitable for publication. My evaluation is on the basis of my interpretation of the acceptance criteria for SciPost [1], specifically expectations (3) and (4) . Although elements of the work i.e. the conceptual framework of using anomaly matching as a tool to determine the emergability of SL(N) and the operational details of going about it were introduced in the authors' previous work [2], the results novel to this work i.e. application of the anomaly matching criterion to all wallpaper groups as well as to experimentally relevant examples sufficiently merit its publication.

[1] https://scipost.org/SciPostPhys/about#criteria [2] Phys. Rev. X 11, 031043 (2021)

Requested changes

As far as presentation is concerned, I would like to offer a suggestion for improvement: the authors appear to coin an unusually large number of new phrases in the manuscript. It might be worthwhile to reconsider this in the general interest of avoiding a confusing proliferation of jargon that is unfortunately endemic to condensed matter theory. Some phrases, such as "symmetry embedding patterns" or "symmetry protected distinction" are superfluous while others such as the phrase in the title- "symmetry enriched criticality"- could be a cause of confusion, especially since the exact same phrase has been used elsewhere (eg: Phys. Rev. X 11, 041059 (2021), referenced as [95] in the manuscript) and it is unclear in what ways the usage is similar or dissimilar (my own understanding is that they are subtly distinct). On the other hand, standard terms such as irreducible Wyckoff points, which are central to the presentation of the work in certain parts, could be better introduced that would greatly help the reader.

However, these are mere suggestions for the authors and irrespective of whether or not they choose to act upon it should not affect publication.

  • validity: good
  • significance: good
  • originality: high
  • clarity: good
  • formatting: acceptable
  • grammar: excellent

Author:  Liujun Zou  on 2022-07-05  [id 2638]

(in reply to Report 1 on 2022-03-24)

We thank the referee for considering our work "scientifically correct and of interest to researchers", for the recommendation for publication and for the constructive comments. Below we address the referee's comments.

The referee mentioned that we reviewed a sigma-model based framework to describe gapless quantum states. In a previous paper (arXiv: 2101.07805), we indeed defined the states considered in this paper in terms of sigma models. However, in the present paper we actually do not mention the sigma-model description. Part of the reason is that although the sigma-model description is very effective in capturing the kinematic properties of the theories, these non-renormalizable theories have an infinite-dimensional parameter space, and they do not fully specify the universal low-energy physics of the system until all their infinitely many parameters are specified. So, given that most of the Stiefel liquids are (conjectured to be) non-Lagrangian, i.e., they cannot be described by renormalizable Lagrangians, it would be good to have a definition of these theories without using any Lagrangian. In Sec. III A, where we review Stiefel liquids, we list their symmetries, anomalies and some dynamical properties. These properties may actually serve as such a more intrinsic definition of them, without explicitly referring to any Lagrangian. In the revised draft, we make this point more explicit.

The referee also mentioned that "the conceptual framework of using anomaly matching as a tool to determine the emergability of SL(N) and the operational details of going about it were introduced in the authors' previous work". Indeed, the conceptual framework was already proposed in the previous work. However, we have to admit that the operational details in that work are actually oversimplified and even misleading, in that both the expressions of the LSM anomalies and the calculations of anomaly-matching are problematic. The current paper provides the correct version. We add a footnote in the introduction to make this point more explicit in the revised draft.

We thank the referee for the suggestion about the presentation. We think that a good research paper in physics should contain not only technical results of specific problems, but also conceptual framework that applies universally. In particular, we think that it is important to spell out some important concepts. We think that the notions like "symmetry embedding patterns" and "symmetry-protected distinction" are important ones that should be the backbones of the framework in the study of symmetry-enriched states, so we think that they deserve being explicitly spelled out. As for "symmetry-enriched quantum criticality", in our introduction we give a clear definition to it, and we think this definition is in accordance with the usual terminology of symmetry-enriched states in the literature. Moreover, as far as we can tell, our definition is actually the same as the one presented in the first paragraph in Phys. Rev. X 11, 041059 (2021). However, the focus of our paper is different from that one, in the sense that we focus on symmetry-protected distinctions reflected in local operators, while that paper focuses more on non-local and/or gapped operators. As commented in our discussion section, that other aspect is also important and interesting. Moreover, that paper mostly focuses on internal symmetries, but we study both internal symmetries and lattice symmetries.

---

## Round 1 · Referee Report · Anonymous (Referee 2) · 2022-6-24

Report

This paper gives a systematic and thorough discussion of a topological perspective on LSM theorems for spin systems in 2+1 dimensions (with certain symmetry groups), and their consequences for stability of exotic states found therein. It represents a proof of principle that this anomaly-based perspective can be used to produce lists of candidate emergible states and to analyze their stability with respect to perturbations with various quantum numbers. This opens the door to a great deal more work in this direction, for example by generalizing to systems with mobile charges.

The timely application to the spin liquid candidate NaYbO${}_2$ is very nice and encouraging.

It should certainly be published, and I expect that it will have a significant impact on the study of exotic states in real materials.

Some more comments, mostly about presentation:

1- The paper is both dense and long. It is quite technical and not always easy to read. As an example, the giant paragraph in the right column of page 8 is quite hard to get through. Clearly the authors have made an effort to make the paper readable, in that the thirteen very complete appendices are already about three times as long as the body of the paper. Also, the restriction to symmetry groups with only $\mathbb{Z}_2$ factors in their group cohomology was a nice way to limit the scope to some extent. I also think there are too many acronyms, but I don't know the solution to that.

2- An important comment: I have to complain about the authors' use of the term "gauging", first in section IIB and in several later places. The authors will agree that coupling to background (non-dynamical) gauge fields is distinct from changing the Hilbert space by coupling to dynamical gauge fields. I feel strongly that the latter process (where the symmetry no longer acts on the new physical Hilbert space) is what should be called "gauging". In this paper, the authors only do the former.

3- It would be nice to make the paper more self-contained by including some more of a reminder about `irreducible Wyckoff positions'.

4- A confusion about the advantage of the 3+1-d description in terms of a crystalline SPT. In the case of internal symmetries, such a description allows the protecting symmetry to be onsite. But here it seems to still be a lattice symmetry in the 3+1-d description.

5- A naive question: What is the difference between the case a and case b locations of the spin-half with $p4m$ symmetry? Apparently, these produce two distinct square-lattice LSM obstructions, but I didn't get a clear sense of what was the difference. Is it just that the position a is invariant under the $C_4$ rotation and $b$ is not?

6- Perhaps for pedagogical reasons the authors might consider putting the 1d example (in II.B.3) first?

7- The authors give an accounting of the symmetry action on their fractionalized states in terms of gauge-invariant quantities. This is to be applauded.
However, given the long history of the subject, I think it would be useful if there were some more comparison to previous work in terms of classification of possible PSGs by Wen and Hermele and others.

8- I was a bit confused about the status of the main statement in section II, i.e. the three basic no-go theorems the authors describe for determining the presence or absence of an LSM constraint. Is all of this (and the absence of any others) proved in reference 5?

9- I have to admit that I didn't understand the broader significance of the "non-LSM SPT" (discussed in appendix D and section IIB), though I think I eventually managed to follow the details. Where is the breakdown of the correspondence between 3+1d crystalline SPTs and 2+1d LSM theorems in general?
The edge theory of such an SPT still realizes the symmetry in an anomalous way; why doesn't this imply some non-triviality condition on the groundstate manifold?

And when does such a breakdown occur? Does it happen whenever the symmetry contains reflections?

Requested changes

-- Fix the use of the term "gauging"

-- Consider the other suggestions and questions above.

  • validity: -
  • significance: -
  • originality: -
  • clarity: -
  • formatting: -
  • grammar: -

Author:  Liujun Zou  on 2022-07-05  [id 2639]

(in reply to Report 2 on 2022-06-24)

We thank the referee for thinking highly of our paper. We are especially grateful to the referee for recognizing that "clearly the authors have made an effort to make the paper readable". We also thank the referee for constructive comments and questions. Below we address the comments and questions of the referee.

  1. We have split the paragraph in the right column of page 8 into two paragraphs.

  2. In this draft, we have clarified that what we previously called "gauging" means coupling to background probe gauge fields, as the referee suggests.

  3. In this draft, we have phrased those special positions in a lattice simply as "high symmetry points", and point out that they are called irreducible Wyckoff positions in crystallography. The precise definition of irreducible Wyckoff positions is actually complicated and not particularly illuminating, and we refer the readers to Ref. [5].

  4. Indeed, because the 2 + 1 dimensional anomalous systems under consideration have anomalies related to lattice symmetries, the corresponding 3+1 bulk also has lattice symmetries. The purpose of using the bulk is its convenience: the anomaly can be compactly encoded in the topological partition function of the bulk.

  5. Yes, that is indeed the difference. This difference is physical in the sense that after fixing what our symmetry setting is, the C4 centers are fixed, and these two cases cannot be smoothly deformed into each other without breaking the symmetry. We have clarified this point in this draft.

  6. We thank the referee for this suggestion. When we wrote the paper, we also considered this option. However, at the end we decided to present the different sections in the current form, because i) The mathematics of the 1+1 dimensional case is different and does not reflect the symmetry fractionalization pattern of Gs on the SO(3) monopole as in 2 + 1 dimension (see Appendix G), and ii) this case is a bit detached from the rest of the paper.

  7. As far as we know, PSG-based classifications of the states considered in this paper are not available in the literature. Part of the reason of this may be our comment in the introduction: the PSG-based approach is not based on the intrinsic properties of the states, so it can be very inconvenient to carry out a classification of complicated quantum critical states within this approach.

  8. The first two were proved in a reference of Ref. [5], and the third one was proved in Ref. [5].

  9. This is related to what LSM anomalies mean: these are specific anomalies that are determined by the symmetry, the projective class of degrees of freedom and locations of the degrees of freedom. The non-LSM anomaly is still a non-trivial anomaly, and it does mean that a system with this anomaly cannot be symmetric and short-range entangled. However, as we discuss in detail in Appendix D, 3 + 1 dimensional bulk corresponding to this anomaly can be constructed by inserting a 2+1 dimensional symmetry-protected topological state on the reflection plane, so it is clearly not associated with the projective class of degrees of freedom and locations of the degrees of freedom, and this anomaly is not an LSM anomaly. A typical lattice system has no such anomaly.

---

## Round 1 · Referee Report · Anonymous (Referee 3) · 2022-6-30

Report

Taking the perspective that LSM constraints are usefully viewed as anomalies, this paper encodes such constraints in the partition function of a corresponding SPT phase. This is then used to do anomaly matching with many examples of symmetry-enriched quantum criticality, showing when such states may emerge in a model with given LSM constraints. The paper is long and technical; the quality of the writing is adequate for experts but might be very hard to follow for readers who do not already know about lattice homotopy, who are not experts on group cohomology as applied to SPT phases, etc. Nonetheless I believe the results are correct as far as I can tell, and a number of extremely interesting results have been obtained. So this paper is an important contribution and I recommend it be published in its current form. I do not think it would be a good use of referees'/editors'/authors' time to ask for extensive revisions to improve the presentation.
  • validity: high
  • significance: high
  • originality: high
  • clarity: ok
  • formatting: excellent
  • grammar: excellent

Author:  Liujun Zou  on 2022-07-05  [id 2640]

(in reply to Report 3 on 2022-06-30)

We thank the referee for thinking highly of our paper.

---

## Round 2 · Referee Report · Anonymous (Referee 3) · 2022-7-14

Report

I am satisfied with the revised manuscript and recommend publication.

---

## Round 2 · Referee Report · Anonymous (Referee 2) · 2022-7-19

Report

The paper should be published now.

---

## Round 2 · Referee Report · Anonymous (Referee 1) · 2022-7-26

Report

The revised manuscript can be published.

---

## Round 2 · Author Response

We thank the editor for dealing with our draft, and all referees for their constructive comments and suggestions. The response to each report is given under the report, and a summary of changes is attached below.

---

## Round 2 · List of Changes

Besides various minor changes, the major changes in the revised draft are summarized as follows.

1. In Sec. III A, we add a sentence to explicitly point out that the symmetries, anomalies and dynamical properties of the Stiefel liquids reviewed there serve as an intrinsic definition of Stiefel liquids, without explicitly referring to any Lagrangian.

2. In the introduction, we add a footnote to clarify the relation between the current paper and arXiv: 2101.07805, and in the present paper we correct some mistakes in arXiv: 2101.07805.

3. We have split the paragraph in the right column of page 8 into two paragraphs.

4. We have changed the last paragraph of the introduction to briefly summarize the main results.

---

## Editorial Decision

published